# Continuous vs. Discrete Optimization of Deep Neural Networks

**Omer Elkabetz**
Tel Aviv University
omer.elkabetz@cs.tau.ac.il

**Nadav Cohen**
Tel Aviv University
cohennadav@cs.tau.ac.il

## Abstract

Existing analyses of optimization in deep learning are either continuous, focusing on (variants of) gradient flow, or discrete, directly treating (variants of) gradient descent. Gradient flow is amenable to theoretical analysis, but is stylized and disregards computational efficiency. The extent to which it represents gradient descent is an open question in the theory of deep learning. The current paper studies this question. Viewing gradient descent as an approximate numerical solution to the initial value problem of gradient flow, we find that the degree of approximation depends on the curvature around the gradient flow trajectory. We then show that over deep neural networks with homogeneous activations, gradient flow trajectories enjoy favorable curvature, suggesting they are well approximated by gradient descent. This finding allows us to translate an analysis of gradient flow over deep linear neural networks into a guarantee that gradient descent efficiently converges to global minimum *almost surely* under random initialization. Experiments suggest that over simple deep neural networks, gradient descent with conventional step size is indeed close to gradient flow. We hypothesize that the theory of gradient flows will unravel mysteries behind deep learning.[1]

## 1 Introduction

The success of deep neural networks is fueled by the mysterious properties of gradient-based optimization, namely, the ability of (variants of) gradient descent to minimize non-convex training objectives while exhibiting tendency towards solutions that generalize well. Vast efforts are being directed at mathematically analyzing this phenomenon, with existing results typically falling into one of two categories: continuous or discrete. Continuous analyses usually focus on gradient flow (or variants thereof), which corresponds to gradient descent (or variants thereof) with infinitesimally small step size. Compared to their discrete (positive step size) counterparts, continuous settings are oftentimes far more amenable to theoretical analysis (*e.g.* they admit use of the theory of differential equations), but on the other hand are stylized, and disregard the critical aspect of computational efficiency (number of steps required for convergence). Works analyzing gradient flow over deep neural networks either accept the latter shortcomings (see for example [49, 4, 46]), or attempt to reproduce part of the results via completely separate analysis of gradient descent (*cf.* [30, 18, 5]). The extent to which gradient flow represents gradient descent is an open question in the theory of deep learning.

The current paper formally studies the foregoing question. Viewing gradient descent as a numerical method for approximately solving the initial value problem corresponding to gradient flow, we turn to the literature on numerical analysis, and invoke a fundamental theorem concerning the approximation error. The theorem implies that in general, the match between gradient descent and gradient flow is determined by the curvature around gradient flow's trajectory. In particular, the "more convex" the trajectory, *i.e.* the larger the (possibly negative) minimal eigenvalue of the Hessian is around the trajectory, the bet-

---

[1] Due to lack of space, essential portions of this paper were deferred to supplementary material. We refer the reader to [21] for a self-contained version of the text.

35th Conference on Neural Information Processing Systems (NeurIPS 2021).

ter the match is guaranteed to be.[2] We show that when applied to deep neural networks (fully connected as well as convolutional) with homogeneous activations (*e.g.* linear, rectified linear or leaky rectified linear), gradient flow emanating from near-zero initialization (as commonly employed in practice) follows trajectories that are "roughly convex," in the sense that the minimal eigenvalue of the Hessian along them is far greater than in arbitrary points in space, particularly towards convergence. This implies that over deep neural networks, gradient descent with moderately small step size may in fact be close to its continuous limit, *i.e.* to gradient flow. We exemplify an application of this finding by translating an analysis of gradient flow over deep linear neural networks into a convergence guarantee for gradient descent. The guarantee we obtain is, to our knowledge, the first to ensure that a conventional gradient-based algorithm optimizing a deep (three or more layer) neural network of fixed (data-independent[3]) size efficiently converges[4] to global minimum *almost surely* under random (data-independent) near-zero initialization.

We corroborate our theoretical analysis through experiments with basic deep learning settings, which demonstrate that reducing the step size of gradient descent often leads to only slight changes in its trajectory. This confirms that, in basic settings, central aspects of deep neural network optimization may indeed be captured by gradient flow. Recent works (*e.g.* [8, 33, 53]) suggest that by appropriately modifying gradient flow it is possible to account for advanced settings as well, including ones with momentum, stochasticity and large step size. Encouraged by these developments, we hypothesize that the vast bodies of knowledge on continuous dynamical systems, and gradient flow in particular (see, *e.g.*, [23, 3]), will pave way to unraveling mysteries behind deep learning.

## 1.1 Contributions

The main contributions of this work are: *(i)* we conduct the first formal study for the discrepancy between continuous and discrete optimization of deep neural networks; *(ii)* we demonstrate the use of *generic* mathematical machinery for translating a continuous non-convex convergence result into a discrete one; *(iii)* to our knowledge, the discrete result we obtain forms the first guarantee of random (data-independent) near-zero initialization *almost surely* leading a conventional gradient-based algorithm optimizing a deep (three or more layer) neural network of fixed (data-independent) size to efficiently converge to global minimum; *(iv)* the fundamental theorem (from numerical analysis) we employ is seldom used in machine learning contexts and may be of independent interest; and *(v)* we provide empirical evidence suggesting that gradient descent over simple deep neural networks is often close to gradient flow.

## 2 Preliminaries: Numerical Solution of Initial Value Problems

Let $d \in \mathbb{N}$. Given a function $\mathbf{g} : [0, \infty) \times \mathbb{R}^d \to \mathbb{R}^d$ (viewed as a time-dependent vector field) and a point $\boldsymbol{\theta}_s \in \mathbb{R}^d$, consider the *initial value problem*:

$$\boldsymbol{\theta}(0) = \boldsymbol{\theta}_s \quad , \quad \tfrac{d}{dt}\boldsymbol{\theta}(t) = \mathbf{g}(t, \boldsymbol{\theta}(t)) \ \text{ for } t \geq 0 \,. \tag{1}$$

The following result — an extension of the well known Picard-Lindelöf Theorem — establishes that local Lipschitz continuity of $\mathbf{g}(\cdot)$ suffices for ensuring existence and uniqueness of a solution $\boldsymbol{\theta}(\cdot)$.

**Theorem 1** (Existence-Uniqueness). *Consider the initial value problem in Equation* (1)*, and suppose* $\mathbf{g}(\cdot)$ *is locally Lipschitz continuous. Then, there exists a solution* $\boldsymbol{\theta} : [0, t_e) \to \mathbb{R}^d$*, where either:* (i) $t_e = \infty$*; or* (ii) $t_e < \infty$ *and* $\lim_{t \nearrow t_e} \|\boldsymbol{\theta}(t)\|_2 = \infty$*. Moreover, the solution is unique in the sense that any other solution* $\boldsymbol{\theta}' : [0, t'_e) \to \mathbb{R}^d$ *must satisfy* $t'_e \leq t_e$ *and* $\forall t \in [0, t'_e) : \boldsymbol{\theta}'(t) = \boldsymbol{\theta}(t)$.

*Proof.* The theorem is a direct consequence of the results in Section 1.5 of [25].[5] □

It is typically the case that the solution to Equation (1) cannot be expressed in closed form, and a numerical approximation is sought after. Various numerical methods for approximately solving initial value problems have been developed over the years (see Chapter 12 in [55] for an introduction). The most basic one, *Euler's method*, is parameterized by a *step size* $\eta > 0$, and when applied to Equation (1) follows the recursive scheme:

---

[2]In addition to the minimal eigenvalue of the Hessian, local smoothness and Lipschitz constants also affect the guaranteed match between gradient descent and gradient flow. However, the impact of these constants is exponentially weaker than that of the Hessian's minimal eigenvalue. For details see Theorem 3.

[3]By data-independence we mean that no assumptions on training data are made beyond it being subject to standard whitening and normalization procedures.

[4]We regard convergence as efficient if its computational complexity is polynomial in training set size and dimensions, as well as the desired level of accuracy.

[5]A minor subtlety is that in [25] the vector field $\mathbf{g}(\cdot)$ is defined over an open domain. To account for this requirement, simply extend $\mathbf{g}(\cdot)$ to the domain $(-\infty, \infty) \times \mathbb{R}^d$ by setting $\mathbf{g}(t, \mathbf{q}) = \mathbf{g}(0, \mathbf{q})$ for all $t < 0$, $\mathbf{q} \in \mathbb{R}^d$.

$$\boldsymbol{\theta}_{k+1} = \boldsymbol{\theta}_k + \eta \mathbf{g}(t_k, \boldsymbol{\theta}_k) \text{ for } k = 0, 1, 2, \ldots, \tag{2}$$

where $t_k := k\eta$ and the initial point $\boldsymbol{\theta}_0$ is typically set to $\boldsymbol{\theta}_s$. The motivation behind Euler's method is straightforward — a first order Taylor expansion of the exact solution $\boldsymbol{\theta}(\cdot)$ around time $t_k$ yields: $\boldsymbol{\theta}(t_{k+1}) = \boldsymbol{\theta}(t_k + \eta) \approx \boldsymbol{\theta}(t_k) + \eta \frac{d}{dt}\boldsymbol{\theta}(t_k) = \boldsymbol{\theta}(t_k) + \eta \mathbf{g}(t_k, \boldsymbol{\theta}(t_k))$, therefore if $\boldsymbol{\theta}(t_k)$ is well approximated by $\boldsymbol{\theta}_k$, we may expect $\boldsymbol{\theta}_{k+1}$ to resemble $\boldsymbol{\theta}(t_{k+1})$. The numerical solution produced by Euler's method may be viewed as a continuous polygonal curve:

$$\bar{\boldsymbol{\theta}} : [0, \infty) \to \mathbb{R}^d \quad, \quad \bar{\boldsymbol{\theta}}(0) = \boldsymbol{\theta}_0 \quad, \quad \frac{d}{dt}\bar{\boldsymbol{\theta}}(t) = \mathbf{g}(t_k, \boldsymbol{\theta}_k) \text{ for } t \in (t_k, t_{k+1}), k = 0, 1, 2, \ldots. \tag{3}$$

The quality of the numerical solution then boils down to the distance between this curve and the exact solution, *i.e.* between $\bar{\boldsymbol{\theta}}(t)$ and $\boldsymbol{\theta}(t)$ for $t \geq 0$. Many efforts have been made to derive tight bounds for this distance. We provide below a modern result known as "Fundamental Theorem."

**Theorem 2** (Fundamental Theorem). *Consider the initial value problem in Equation* (1)*, and suppose* $\mathbf{g}(\cdot)$ *is continuously differentiable. Let* $\boldsymbol{\theta} : [0, t_e) \to \mathbb{R}^d$ *be the solution to this problem (see Theorem 1), and let* $\bar{\boldsymbol{\theta}} : [0, \infty) \to \mathbb{R}^d$ *be a continuous polygonal curve (Equation* (3)*) born from Euler's method (Equation* (2)*). For any* $t \in [0, t_e), \mathbf{q} \in \mathbb{R}^d$*, denote by* $J(t, \mathbf{q}) \in \mathbb{R}^{d,d}$ *the Jacobian of* $\mathbf{g}(\cdot)$ *with respect to its second argument at the point* $(t, \mathbf{q})$*, and by* $\lambda_{max}(t, \mathbf{q})$ *the maximal eigenvalue of* $\frac{1}{2}(J(t, \mathbf{q}) + J(t, \mathbf{q})^\top)$*.[6] Let* $m : [0, t_e) \to \mathbb{R}$ *be an integrable function satisfying:* $\lambda_{max}(t, \mathbf{q}) \leq m(t)$ *for all* $t \in [0, t_e)$ *and* $\mathbf{q} \in [\boldsymbol{\theta}(t), \bar{\boldsymbol{\theta}}(t)]$*, where* $[\boldsymbol{\theta}(t), \bar{\boldsymbol{\theta}}(t)]$ *stands for the line segment (in* $\mathbb{R}^d$*) between* $\boldsymbol{\theta}(t)$ *and* $\bar{\boldsymbol{\theta}}(t)$*. Let* $\delta : [0, t_e) \to \mathbb{R}_{\geq 0}$ *be an integrable function that meets:* $\|\frac{d}{dt}\bar{\boldsymbol{\theta}}(t^+) - \mathbf{g}(t, \bar{\boldsymbol{\theta}}(t))\|_2 \leq \delta(t)$ *for all* $t \in [0, t_e)$*, where* $\frac{d}{dt}\bar{\boldsymbol{\theta}}(t^+)$ *represents the right derivative of* $\bar{\boldsymbol{\theta}}(\cdot)$ *at time* $t$*. Then, for all* $t \in [0, t_e)$*:*

$$\|\boldsymbol{\theta}(t) - \bar{\boldsymbol{\theta}}(t)\|_2 \leq e^{\mu(t)}\left(\|\boldsymbol{\theta}(0) - \bar{\boldsymbol{\theta}}(0)\|_2 + \int_0^t e^{-\mu(t')}\delta(t')dt'\right), \tag{4}$$

*where* $\mu(t) := \int_0^t m(t')dt'$.

*Proof.* The theorem is simply a restatement of Theorem 10.6 in [27]. □

## 3 Continuous vs. Discrete Optimization: Match Determined by Convexity

Let $f : \mathbb{R}^d \to \mathbb{R}$, where $d \in \mathbb{N}$, be a twice continuously differentiable function which we would like to minimize. Consider continuous optimization via *gradient flow* initialized at $\boldsymbol{\theta}_s \in \mathbb{R}^d$:

$$\boldsymbol{\theta}(0) = \boldsymbol{\theta}_s \quad, \quad \frac{d}{dt}\boldsymbol{\theta}(t) = -\nabla f(\boldsymbol{\theta}(t)) \text{ for } t \geq 0. \tag{5}$$

This is a special case of the initial value problem presented in Equation (1).[7] By Theorem 1, it admits a unique solution $\boldsymbol{\theta} : [0, t_e) \to \mathbb{R}^d$, where either: *(i)* $t_e = \infty$; or *(ii)* $t_e < \infty$ and $\lim_{t \nearrow t_e} \|\boldsymbol{\theta}(t)\|_2 = \infty$. Numerically approximating this solution via Euler's method (Equation (2)) yields a discrete optimization algorithm which is no other than *gradient descent*:

$$\boldsymbol{\theta}_{k+1} = \boldsymbol{\theta}_k - \eta \nabla f(\boldsymbol{\theta}_k) \text{ for } k = 0, 1, 2, \ldots, \tag{6}$$

where $\eta > 0$ is the chosen step size. We may thus invoke the Fundamental Theorem (Theorem 2) and obtain a bound on the distance between the trajectories of gradient flow and gradient descent.

**Theorem 3.** *Consider the trajectory of gradient flow (solution to Equation* (5)*)* $\boldsymbol{\theta} : [0, t_e) \to \mathbb{R}^d$*, and let* $\tilde{t} \in (0, t_e)$ *and* $\epsilon > 0$*. Define* $\mathcal{D}_{\tilde{t},\epsilon} := \bigcup_{t \in [0,\tilde{t}]} \mathcal{B}_\epsilon(\boldsymbol{\theta}(t))$*, where* $\mathcal{B}_\epsilon(\boldsymbol{\theta}(t)) \subset \mathbb{R}^d$ *stands for the (closed) Euclidean ball of radius* $\epsilon$ *centered at* $\boldsymbol{\theta}(t)$*. Let* $\beta_{\tilde{t},\epsilon}, \gamma_{\tilde{t},\epsilon} > 0$ *be such that:* $\sup_{\mathbf{q} \in \mathcal{D}_{\tilde{t},\epsilon}} \|\nabla^2 f(\mathbf{q})\|_{spectral} \leq \beta_{\tilde{t},\epsilon}$ *and* $\sup_{\mathbf{q} \in \mathcal{D}_{\tilde{t},\epsilon}} \|\nabla f(\mathbf{q})\|_2 \leq \gamma_{\tilde{t},\epsilon}$*. Let* $m : [0, \tilde{t}] \to \mathbb{R}$ *be an integrable function satisfying:* $-\lambda_{min}(\nabla^2 f(\mathbf{q})) \leq m(t)$ *for all* $t \in [0, \tilde{t}]$ *and* $\mathbf{q} \in \mathcal{B}_\epsilon(\boldsymbol{\theta}(t))$*, where* $\lambda_{min}(\nabla^2 f(\mathbf{q}))$ *stands for the minimal eigenvalue of* $\nabla^2 f(\mathbf{q})$*. Then, if the step size* $\eta > 0$ *chosen for gradient descent (Equation* (6)*) satisfies:*

$$\eta < \inf_{t \in (0,\tilde{t}]} \frac{\epsilon - e^{\int_0^t m(t')dt'}\|\boldsymbol{\theta}_0 - \boldsymbol{\theta}(0)\|_2}{\beta_{\tilde{t},\epsilon}\gamma_{\tilde{t},\epsilon}\int_0^t e^{\int_{t'}^t m(t'')dt''}dt'}, \tag{7}$$

*the first* $\lfloor \tilde{t}/\eta \rfloor$ *iterates of gradient descent will* $\epsilon$*-approximate the trajectory of gradient flow up to time* $\tilde{t}$*, i.e. we will have* $\|\boldsymbol{\theta}_k - \boldsymbol{\theta}(k\eta)\|_2 \leq \epsilon$ *for all* $k \in \{1, 2, \ldots, \lfloor \tilde{t}/\eta \rfloor\}$*.*

*Proof sketch (for complete proof see Subappendix J.2).* The result follows from applying the Fundamental Theorem (Theorem 2) with $\delta(\cdot)$ fixed at $\beta_{\tilde{t},\epsilon}\gamma_{\tilde{t},\epsilon}\eta$. □

---

[6]This maximal eigenvalue is known as the *logarithmic norm* of $J(t, \mathbf{q})$ (*cf.* Section I.10 in [27]).

[7]The vector field in this case is time-independent (given by $\mathbf{g}(t, \mathbf{q}) = -\nabla f(\mathbf{q})$ for all $t \in [0, \infty), \mathbf{q} \in \mathbb{R}^d$). Initial value problems of this type are known as *autonomous*.

Theorem 3 gives a sufficient condition — upper bound on step size $\eta$ (Equation (7)) — for gradient descent to follow gradient flow up to a given time $\tilde{t}$. The bound is inversely proportional to smoothness and Lipschitz constants ($\beta_{\tilde{t},\epsilon}$ and $\gamma_{\tilde{t},\epsilon}$ respectively), and more importantly, depends exponentially on the integral of $m(\cdot)$ along the gradient flow trajectory, where $m(\cdot)$ corresponds to minus the minimal eigenvalue of the Hessian. The smaller the integral of $m(\cdot)$, *i.e.* the larger (less negative or more positive) the minimal eigenvalue of the Hessian around the trajectory is, the more relaxed the bound will be. That is, *the "more convex" the objective function is around the gradient flow trajectory, the better the match between gradient flow and gradient descent* is guaranteed to be.

Corollary 1 below coarsely applies Theorem 3 by fixing $m(\cdot)$ to minus the minimal eigenvalue of the Hessian *across the entire space*. If $m(\cdot) \equiv m$ (now a constant) is negative, *i.e.* the objective function $f(\cdot)$ is strongly convex, the upper bound on the step size $\eta$ becomes constant, meaning it is independent of the time $\tilde{t}$ until which gradient descent is required to follow gradient flow. If $m$ is equal to zero, *i.e.* $f(\cdot)$ is non-strongly convex, the upper bound on $\eta$ mildly decreases with $\tilde{t}$, namely it scales as $1/\tilde{t}$. If on the other hand $m$ is positive, meaning $f(\cdot)$ is non-convex, the bound on $\eta$ shrinks to zero (becoming prohibitively restrictive) exponentially fast as $\tilde{t}$ grows. This suggests that as opposed to (strongly or non-strongly) convex objectives, over which gradient descent can easily be made to follow gradient flow, over non-convex objectives, in the worst case, gradient descent will immediately divert from gradient flow unless its step size is exponentially small. In Appendix B we present a simple example of such a worst case scenario. In this worst case, the minimal eigenvalue of the Hessian is bounded below and away from zero around the gradient flow trajectory. A question is then whether there are non-convex objectives in which the minimal eigenvalue of the Hessian around gradient flow trajectories is large enough for them to be followed by gradient descent. We will see that training losses of deep neural networks can meet this property.

**Corollary 1.** *Assume that the objective function $f(\cdot)$ is non-negative and $\beta$-smooth with $\beta > 0$.[8] Denote $m := -\inf_{\mathbf{q} \in \mathbb{R}^d} \lambda_{min}(\nabla^2 f(\mathbf{q}))$, where $\lambda_{min}(\nabla^2 f(\mathbf{q}))$ stands for the minimal eigenvalue of $\nabla^2 f(\mathbf{q})$. Consider the trajectory of gradient flow (solution to Equation (5)) $\boldsymbol{\theta} : [0, t_e) \to \mathbb{R}^d$,[9] and let $\tilde{t} \in (0, t_e)$ and $\epsilon > 0$. Then, if the step size $\eta > 0$ for gradient descent (Equation (6)) satisfies:*

$$\eta < \begin{cases} c(\epsilon - \|\boldsymbol{\theta}_0 - \boldsymbol{\theta}(0)\|_2)|m| & \text{,if } m < 0 \quad \text{(strong convexity)} \\ c(\epsilon - \|\boldsymbol{\theta}_0 - \boldsymbol{\theta}(0)\|_2)(1/\tilde{t}) & \text{,if } m = 0 \quad \text{(non-strong convexity)} \\ c(\epsilon - \|\boldsymbol{\theta}_0 - \boldsymbol{\theta}(0)\|_2 e^{m\tilde{t}})(e^{m\tilde{t}} - 1)^{-1} m & \text{,if } m > 0 \quad \text{(non-convexity)} \end{cases},$$

*where $c := \left(\sqrt{2\beta^3 f(\boldsymbol{\theta}(0))} + \beta^2 \epsilon\right)^{-1}$, we will have $\|\boldsymbol{\theta}_k - \boldsymbol{\theta}(k\eta)\|_2 \leq \epsilon$ for all $k \in \{1, 2, ..., \lfloor \tilde{t}/\eta \rfloor\}$.*

*Proof sketch (for complete proof see Subappendix J.3).* The result follows from applying Theorem 3 with $\beta_{\tilde{t},\epsilon} = \beta$, $\gamma_{\tilde{t},\epsilon} = \sqrt{2\beta f(\boldsymbol{\theta}(0))} + \beta\epsilon$ and $m(\cdot) \equiv m$. $\qquad\qquad\square$

## 4 Optimization of Deep Neural Networks is Roughly Convex

Section 3 has shown that the extent to which gradient descent matches gradient flow depends on "how convex" the objective function is around the gradient flow trajectory. More precisely, the larger (less negative or more positive) the minimal eigenvalue of the Hessian is around this trajectory, the longer gradient descent (with given step size) is guaranteed to follow it.[2] In this section we establish that over training losses of deep neural networks (fully connected as well as convolutional) with homogeneous activations (*e.g.* linear, rectified linear or leaky rectified linear), when emanating from near-zero initialization (as commonly employed in practice), trajectories of gradient flow are "roughly convex," in the sense that the minimal eigenvalue of the Hessian along them is far greater than in arbitrary points in space, particularly towards convergence. This finding suggests that when optimizing deep neural networks, gradient descent may closely resemble gradient flow. We demonstrate a formal application of the finding in Section 5, translating an analysis of gradient flow over deep linear neural networks into a guarantee of efficient convergence (to global minimum) for gradient descent, which applies *almost surely* with respect to a random near-zero initialization.

---

[8]Namely, $\|\nabla^2 f(\mathbf{q})\|_{spectral} \leq \beta$ for all $\mathbf{q} \in \mathbb{R}^d$.

[9]Lemma 3 in Appendix A shows that in the current context ($\beta$-smoothness of the objective function $f(\cdot)$), it necessarily holds that $t_e = \infty$, *i.e.* the trajectory of gradient flow is defined over $[0, \infty)$. For simplicity, the statement of the corollary does not rely on this fact.

## 4.1 Fully Connected Architectures

Consider the mappings realized by a fully connected neural network with depth $n \in \mathbb{N}_{\geq 2}$, input dimension $d_0 \in \mathbb{N}$, hidden widths $d_1, d_2, ..., d_{n-1} \in \mathbb{N}$, and output dimension $d_n \in \mathbb{N}$:

$$h_{\boldsymbol{\theta}} : \mathbb{R}^{d_0} \to \mathbb{R}^{d_n} \ , \ h_{\boldsymbol{\theta}}(\mathbf{x}) = W_n \sigma(W_{n-1} \sigma(W_{n-2} \cdots \sigma(W_1 \mathbf{x}) \cdots)) \, , \tag{8}$$

where: $W_j \in \mathbb{R}^{d_j, d_{j-1}}$, $j = 1, 2, ..., n$, are learned weight matrices; $\boldsymbol{\theta} \in \mathbb{R}^d$, with $d := \sum_{j=1}^n d_j d_{j-1}$, is their arrangement as a vector;[10] and $\sigma : \mathbb{R} \to \mathbb{R}$ is a predetermined activation function that operates element-wise when applied to a vector.[11] We assume that $\sigma(\cdot)$ is (positively) *homogeneous*, meaning $\sigma(cz) = c\sigma(z)$ for all $c \geq 0, z \in \mathbb{R}$. This allows for linear ($\sigma(z) = z$), as well as the commonly employed rectified linear ($\sigma(z) = \max\{z, 0\}$) and leaky rectified linear ($\sigma(z) = \max\{z, \bar{\alpha}z\}$ for some $0 < \bar{\alpha} < 1$) activations.

Let $\mathcal{Y}$ be a set of possible labels, and let $\mathcal{S} = ((\mathbf{x}_i, y_i))_{i=1}^{|\mathcal{S}|}$, with $\mathbf{x}_i \in \mathbb{R}^{d_0}, y_i \in \mathcal{Y}$ for $i = 1, 2, ..., |\mathcal{S}|$, be a sequence of labeled inputs. Given a loss function $\ell : \mathbb{R}^{d_n} \times \mathcal{Y} \to \mathbb{R}$ convex and twice continuously differentiable in its first argument (common choices include square, logistic and exponential losses), we learn the weights of the neural network by minimizing its *training loss* — average loss over elements of $\mathcal{S}$:

$$f : \mathbb{R}^d \to \mathbb{R} \ , \ f(\boldsymbol{\theta}) = \frac{1}{|\mathcal{S}|} \sum_{i=1}^{|\mathcal{S}|} \ell(h_{\boldsymbol{\theta}}(\mathbf{x}_i), y_i) \, . \tag{9}$$

Subsubsections 4.1.1 and 4.1.2 below show (for linear and non-linear activation functions, respectively) that although the minimal eigenvalue of $\nabla^2 f(\boldsymbol{\theta})$ (Hessian of training loss) — denoted $\lambda_{min}(\nabla^2 f(\boldsymbol{\theta}))$ — can in general be arbitrarily negative, along trajectories of gradient flow (which emanate from near-zero initialization) it is no less than moderately negative, approaching non-negativity towards convergence. In light of Section 3, this suggests that over fully connected deep neural networks, gradient flow may lend itself to approximation by gradient descent — a prospect we confirm (for a case with linear activation) in Section 5.

### 4.1.1 Linear Activation

Assume that the activation function of the fully connected neural network (Equation (8)) is linear, *i.e.* $\sigma(z) = z$, and define the *end-to-end matrix*:

$$W_{n:1} := W_n W_{n-1} \cdots W_1 \in \mathbb{R}^{d_n, d_0} \, . \tag{10}$$

The mappings realized by the network can then be written as $h_{\boldsymbol{\theta}}(\mathbf{x}) = W_{n:1} \mathbf{x}$, and the training loss as $f(\boldsymbol{\theta}) = \phi(W_{n:1})$, where

$$\phi : \mathbb{R}^{d_n, d_0} \to \mathbb{R} \ , \ \phi(W) = \frac{1}{|\mathcal{S}|} \sum_{i=1}^{|\mathcal{S}|} \ell(W \mathbf{x}_i, y_i) \tag{11}$$

is convex and twice continuously differentiable. Lemma 1 below expresses $\nabla^2 f(\boldsymbol{\theta})$ in this case.

**Lemma 1.** *For any $\boldsymbol{\theta} \in \mathbb{R}^d$, regard $\nabla^2 f(\boldsymbol{\theta})$ not only as a (symmetric) matrix in $\mathbb{R}^{d,d}$, but also as a quadratic form $\nabla^2 f(\boldsymbol{\theta})[\cdot]$ that intakes a tuple $(\Delta W_1, \Delta W_2, ..., \Delta W_n) \in \mathbb{R}^{d_1, d_0} \times \mathbb{R}^{d_2, d_1} \times \cdots \times \mathbb{R}^{d_n, d_{n-1}}$, arranges it as a vector $\Delta \boldsymbol{\theta} \in \mathbb{R}^d$ (in correspondence with how weight matrices $W_1, W_2, ..., W_n$ are arranged to create $\boldsymbol{\theta}$), and returns $\Delta \boldsymbol{\theta}^\top \nabla^2 f(\boldsymbol{\theta}) \Delta \boldsymbol{\theta} \in \mathbb{R}$. Similarly, for any $W \in \mathbb{R}^{d_n, d_0}$, regard $\nabla^2 \phi(W)$ as a quadratic form $\nabla^2 \phi(W)[\cdot]$ that intakes a matrix in $\mathbb{R}^{d_n, d_0}$ and returns a scalar (non-negative since $\phi(\cdot)$ is convex). Then, $\nabla^2 f(\boldsymbol{\theta})$ is given by:*

$$\nabla^2 f(\boldsymbol{\theta})[\Delta W_1, \Delta W_2, ..., \Delta W_n] = \nabla^2 \phi(W_{n:1}) \Big[ \sum_{j=1}^n W_{n:j+1}(\Delta W_j) W_{j-1:1} \Big] \tag{12}$$

$$+ 2 \mathrm{Tr} \Big( \nabla \phi(W_{n:1})^\top \sum_{1 \leq j < j' \leq n} W_{n:j'+1}(\Delta W_{j'}) W_{j'-1:j+1}(\Delta W_j) W_{j-1:1} \Big) \, ,$$

*where $W_{j':j}$, for any $j, j' \in \{1, 2, ..., n\}$, is defined as $W_{j'} W_{j'-1} \cdots W_j$ if $j \leq j'$, and as an identity matrix (with size to be inferred by context) otherwise.*

*Proof.* See Subappendix J.4. $\qquad \square$

The following proposition makes use of Lemma 1 to show that (under mild conditions) $\lambda_{min}(\nabla^2 f(\boldsymbol{\theta}))$ can be arbitrarily negative, *i.e.* $\inf_{\boldsymbol{\theta} \in \mathbb{R}^d} \lambda_{min}(\nabla^2 f(\boldsymbol{\theta})) = -\infty$.

**Proposition 1.** *Assume that the network is deep ($n \geq 3$), and that the zero mapping is not a global minimizer of the training loss (meaning $\nabla \phi(0) \neq 0$).[12] Then $\inf_{\boldsymbol{\theta} \in \mathbb{R}^d} \lambda_{min}(\nabla^2 f(\boldsymbol{\theta})) = -\infty$.*

---

[10] The exact order by which the entries of $W_1, W_2, ..., W_n$ are placed in $\boldsymbol{\theta}$ is insignificant for our purposes — all that matters is that the same order be used throughout.

[11] Our analysis can easily be extended to account for different activation functions at different hidden layers. We assume identical activation functions for simplicity of presentation.

[12] Both of these assumptions are necessary, in the sense that removing any of them (without imposing further assumptions) renders the proposition false — see Claim 1 in Appendix F.

*Proof.* See Subappendix J.5. □

Building on Lemma 1, Lemma 2 below provides a lower bound on $\lambda_{min}(\nabla^2 f(\boldsymbol{\theta}))$.

**Lemma 2.** *For any $\boldsymbol{\theta} \in \mathbb{R}^d$:[13]*

$$\lambda_{min}(\nabla^2 f(\boldsymbol{\theta})) \geq -(n-1)\sqrt{\min\{d_0,d_n\}}\|\nabla\phi(W_{n:1})\|_{Frobenius} \max_{\substack{\mathcal{J}\subseteq\{1,2,...,n\}\\|\mathcal{J}|=n-2}}\prod_{j\in\mathcal{J}}\|W_j\|_{spectral}. \quad (13)$$

*Proof.* See Subappendix J.6. □

Assuming the training loss is non-constant and the network is deep ($n \geq 3$), the infimum (over $\boldsymbol{\theta} \in \mathbb{R}^d$) of the lower bound in Equation (13) is minus infinity. In particular, if $\boldsymbol{\theta}$ is not a global minimizer ($\nabla\phi(W_{n:1}) \neq 0$) and at least $n-2$ of its weight matrices $W_1, W_2, ..., W_n$ are non-zero, then by rescaling the latter it is possible to take the lower bound to minus infinity while keeping the end-to-end matrix $W_{n:1}$ (and thus the input-output mapping $h_{\boldsymbol{\theta}}(\cdot)$ and the training loss value $f(\boldsymbol{\theta})$) intact. However, gradient flow over fully connected neural networks (with homogeneous activations) initialized near zero is known to maintain balance between weight matrices — see [18] — and so along its trajectories the lower bound in Equation (13) takes a much tighter form. This is formalized in Proposition 2 below.

**Proposition 2.** *If $\boldsymbol{\theta} \in \mathbb{R}^d$ resides on a trajectory of gradient flow (over $f(\cdot)$) emanating from some point $\boldsymbol{\theta}_s \in \mathbb{R}^d$, with $\|\boldsymbol{\theta}_s\|_2 \leq \epsilon$ for some $\epsilon \in (0, \frac{1}{2n}]$, then:*

$$\lambda_{min}(\nabla^2 f(\boldsymbol{\theta})) \geq -(n-1)\sqrt{\min\{d_0,d_n\}}\|\nabla\phi(W_{n:1})\|_{Frobenius}\|W_{n:1}\|_{spectral}^{1-2/n} - c\epsilon^{1-2/n}, \quad (14)$$

*where $c := \frac{4n(n-1)}{(4n)^{2/n}}\sqrt{\min\{d_0,d_n\}}\|\nabla\phi(W_{n:1})\|_{Frobenius}\max\{1,\max\{\|W_j\|_{spectral}\}_{j=1}^n\}^{2(n-2)}$.*

*Proof.* See Subappendix J.7. □

Assume the network is deep ($n \geq 3$), and consider a trajectory of gradient flow (over $f(\cdot)$) emanating from near-zero initialization. For every point on the trajectory, Proposition 2 may be applied with small $\epsilon$, leading the lower bound in Equation (14) to depend primarily on the sizes (norms) of the end-to-end matrix $W_{n:1}$ and the gradient of the loss with respect to it, *i.e.* $\nabla\phi(W_{n:1})$ (see Equations (10) and (11)). In the course of optimization, $W_{n:1}$ is initially small, and (since the loss $f(\boldsymbol{\theta}) = \phi(W_{n:1})$ is monotonically non-increasing) remains confined to sublevel sets of $\phi(\cdot)$ (which is convex) thereafter. $\nabla\phi(W_{n:1})$ on the other hand tends to zero upon convergence to global minimum. We conclude that the lower bound on $\lambda_{min}(\nabla^2 f(\boldsymbol{\theta}))$ in Equation (14) starts off slightly negative, and approaches non-negativity (if and) as the trajectory converges to global minimum. In light of Section 3, this implies that the gradient flow trajectory may lend itself to approximation by gradient descent. Indeed, the results of the current Subsubsection are used in Section 5 to establish proximity between gradient flow and gradient descent, thereby translating an analysis of gradient flow into a guarantee of efficient convergence (to global minimum) for gradient descent.

#### 4.1.2 Non-Linear Activation

Due to lack of space, we defer our analysis for fully connected neural networks with non-linear activation to Appendix C. This analysis is similar in spirit to the one in Subsubsection 4.1.1 treating linear activation. In particular, it makes use of the fact that gradient flow initialized near zero maintains balance between weight matrices — *cf.* [18]. A key difference brought forth by non-linear activation is that the training loss $f(\cdot)$ (Equation (9)) is no longer differentiable. We circumvent this challenge by excluding from the analysis points of non-differentiability, which form a negligible (closed and zero measure) set.

### 4.2 Convolutional Architectures

We account for convolutional neural networks by allowing for weight sharing and sparsity patterns to be imposed on the layers of the fully connected model analyzed in Subsection 4.1. Namely, we consider the exact same mappings as in Equation (8), but now, rather than being learned directly, the matrices $W_j \in \mathbb{R}^{d_j,d_{j-1}}$, $j = 1,2,...,n$, are determined by learned weight vectors $\mathbf{w}_j \in \mathbb{R}^{d'_j}$, with $d'_j \in \mathbb{N}$, $j=1,2,...,n$, such that each entry of $W_j$ is either fixed at zero or connected to a predetermined coordinate of $\mathbf{w}_j$ (with no repetition of coordinates within the same row). The weight setting $\boldsymbol{\theta} \in \mathbb{R}^d$ is then simply a concatenation of the weight vectors $\mathbf{w}_1,\mathbf{w}_2,...,\mathbf{w}_n$, and its dimension is accordingly $d = \sum_{j=1}^n d'_j$. Our analysis for this model (which includes convolutional neural networks as a special case) is essentially the same as that presented for fully connected neural networks with non-linear activation (Subsubsection 4.1.2). In particular, we use the fact that even with weight sharing and sparsity patterns imposed on the layers of a fully connected neural network (with homogeneous activation), when initialized near zero, gradient flow over the network maintains balance between weights of different layers — *cf.* [18]. For the complete analysis see Appendix D.

---

[13]Note that by convention, an empty product (*i.e.* a product over the elements of the empty set) is equal to one.

# 5 Continuous Proof of Discrete Convergence for Deep Linear Neural Networks

Section 3 invoked the Fundamental Theorem for numerical solution of initial value problems (Theorem 2) to show that, in general, the extent to which gradient descent provably matches gradient flow is determined by how large (less negative or more positive) the minimal eigenvalue of the Hessian is around the gradient flow trajectory.[2] Section 4 established that for training losses of deep neural networks, along trajectories of gradient flow emanating from near-zero initialization (as commonly employed in practice), the minimal eigenvalue of the Hessian is far greater than in arbitrary points in space, particularly towards convergence. In this section we combine the two findings, translating an analysis of gradient flow over deep linear neural networks into a convergence guarantee for gradient descent. The guarantee we obtain is, to our knowledge, the first to ensure that a conventional gradient-based algorithm optimizing a deep (three or more layer) neural network of fixed (data-independent[3]) size efficiently converges[4] to global minimum *almost surely* under random (data-independent) near-zero initialization.

Deep linear neural networks — fully connected neural networks with linear activation (see Subsection 4.1) — are perhaps the most common subject of theoretical study in the context of optimization in deep learning. Though trivial from an expressiveness point of view (realize only linear input-output mappings), they induce highly non-convex training losses, giving rise to highly non-trivial phenomena under gradient-based optimization. In recent years, various results concerning gradient flow over deep linear neural networks have been proven, most notably for the case of *balanced initialization* (see for example [49, 4, 34, 6, 46]). Under the notations of Subsection 4.1 (in particular with $W_1, W_2, ..., W_n$ standing for network weight matrices), balanced initialization means that when optimization commences:

$$W_{j+1}^\top W_{j+1} = W_j W_j^\top \ \text{ for } j = 1, 2, ..., n-1 \,. \tag{15}$$

The condition holds approximately with any near-zero initialization, and exactly when the following procedure (adaptation of Procedure 1 in [5]) is employed.

**Procedure 1** (random balanced initialization). *With a distribution $\mathcal{P}$ over $d_n$-by-$d_0$ matrices of rank at most $\min\{d_0, d_1, ..., d_n\}$, initialize $W_j \in \mathbb{R}^{d_j, d_{j-1}}$, $j = 1, 2, ..., n$, via following steps:* (i) *sample $A \sim \mathcal{P}$;* (ii) *take singular value decomposition $A = U\Sigma V^\top$, where $U \in \mathbb{R}^{d_n, \min\{d_0, d_n\}}$ and $V \in \mathbb{R}^{d_0, \min\{d_0, d_n\}}$ have orthonormal columns, and $\Sigma \in \mathbb{R}^{\min\{d_0, d_n\}, \min\{d_0, d_n\}}$ is diagonal and holds the singular values of $A$; and* (iii) *set $W_n \simeq U\Sigma^{1/n}, W_{n-1} \simeq \Sigma^{1/n}, W_{n-2} \simeq \Sigma^{1/n}, ..., W_2 \simeq \Sigma^{1/n}, W_1 \simeq \Sigma^{1/n}V^\top$, where "$\simeq$" stands for equality up to zero-valued padding.*

Compared to gradient flow, little is known about gradient descent when it comes to optimization of deep (three or more layer) linear neural networks. Indeed, there are relatively few results along this line (*cf.* [9, 30, 5]), and these are typically highly specific, built upon technical proofs that are difficult to generalize. Being able to obtain results via translation of gradient flow analyses is thus of prime interest.

We focus in this section on deep[14] linear neural networks trained for scalar regression per least-squares criterion. In the context of Subsection 4.1, this means that the activation function $\sigma(\cdot)$ is linear ($\sigma(z) = z$), the output dimension $d_n$ is one, and the loss function $\ell(\cdot)$ is the square loss (*i.e.* $\mathcal{Y} = \mathbb{R}$ and $\ell(\hat{y}, y) = \frac{1}{2}(\hat{y} - y)^2$). We assume that training inputs are whitened, *i.e.* have been transformed such that their empirical (uncentered) covariance matrix $\Lambda_{xx} := \frac{1}{|\mathcal{S}|}\sum_{i=1}^{|\mathcal{S}|}\mathbf{x}_i\mathbf{x}_i^\top \in \mathbb{R}^{d_0, d_0}$ is equal to identity. A standard calculation (see Appendix G) shows that in this case the function $\phi(\cdot)$ defined by Equation (11) becomes $\phi(W) = \frac{1}{2}\|W - \Lambda_{yx}\|_{Frobenius}^2 + c$, where $\Lambda_{yx} := \frac{1}{|\mathcal{S}|}\sum_{i=1}^{|\mathcal{S}|}y_i\mathbf{x}_i^\top \in \mathbb{R}^{1, d_0}$ is the empirical (uncentered) cross-covariance matrix between training labels and inputs, and $c \in \mathbb{R}$ is a constant (independent of $W$). We may thus write the training loss $f(\cdot)$ (Equation (9)) as:

$$f(\boldsymbol{\theta}) = \frac{1}{2}\|W_{n:1} - \Lambda_{yx}\|_{Frobenius}^2 + c = \frac{1}{2}\|W_{n:1} - \Lambda_{yx}\|_{Frobenius}^2 + \min_{\mathbf{q} \in \mathbb{R}^d}f(\mathbf{q}) \,, \tag{16}$$

where $W_{n:1} \in \mathbb{R}^{1, d_0}$ is the network's end-to-end matrix (Equation (10)). We disregard the degenerate case where $\Lambda_{yx} = 0$, *i.e.* where the zero mapping attains the global minimum, and assume that training labels are normalized (jointly scaled) such that $\Lambda_{yx}$ has unit length ($\|\Lambda_{yx}\|_{Frobenius} = 1$).

Proposition 3 below analyzes gradient flow over the training loss in Equation (16). Relying on a known characterization for the dynamics of the end-to-end matrix (*cf.* [4]), it establishes convergence to global minimum. Moreover, harnessing the results of Section 4, it derives a lower bound on (the integral of) the minimal eigenvalue of the Hessian around the gradient flow trajectory.

---

[14]Our results apply to shallow (two layer) networks as well. We highlight the deep (three or more layer) setting as it is far less understood (*cf.* [5]), and arguably more central to deep learning.

**Proposition 3.** *Consider minimization of the training loss $f(\cdot)$ in Equation (16) via gradient flow (Equation (5)) starting from initial point $\boldsymbol{\theta}_s \in \mathbb{R}^d$ that meets the balancedness condition (Equation (15)). Denote by $W_{n:1,s}$ the initial value of the end-to-end matrix (Equation (10)), and suppose that $\|W_{n:1,s}\|_{Frobenius} \in (0, 0.2]$ (initialization is small but non-zero). Assume that $W_{n:1,s}$ is not antiparallel to $\Lambda_{yx}$, i.e. $\nu := \mathrm{Tr}(\Lambda_{yx}^\top W_{n:1,s}) / \big(\|\Lambda_{yx}\|_{Frobenius}\|W_{n:1,s}\|_{Frobenius}\big) \neq -1$. Then, the trajectory of gradient flow is defined over infinite time, and with $\boldsymbol{\theta} : [0,\infty) \to \mathbb{R}^d$ representing this trajectory, for any $\bar{\epsilon} > 0$, the following time $\bar{t}$ satisfies $f(\boldsymbol{\theta}(\bar{t})) - \min_{\mathbf{q} \in \mathbb{R}^d} f(\mathbf{q}) \leq \bar{\epsilon}$:*

$$\bar{t} = \frac{2n\big(\max\{1, \frac{3}{2} \cdot \frac{1-\nu}{1+\nu}\}\big)^n}{\|W_{n:1,s}\|_{Frobenius}} \ln\left(\frac{15n\max\{1, \frac{1-\nu}{1+\nu}\}}{\|W_{n:1,s}\|_{Frobenius}\min\{1, 2\bar{\epsilon}\}}\right). \tag{17}$$

*Moreover, under the notations of Theorem 3, for any $t > 0$ and $\epsilon \in \big(0, \frac{1}{2n}\big]$ with corresponding $\mathcal{D}_{t,\epsilon}$ ($\epsilon$-neighborhood of gradient flow trajectory up to time $t$), we have the smoothness and Lipschitz constants $\beta_{t,\epsilon} = 16n$ and $\gamma_{t,\epsilon} = 6\sqrt{n}$ respectively, and the following (upper) bound on the integral of (minus) the minimal eigenvalue of the Hessian:*

$$\int_0^t m(t')dt' \leq \frac{15n^3\big(\max\{1, \frac{3}{2} \cdot \frac{1-\nu}{1+\nu}\}\big)^n t\epsilon}{\|W_{n:1,s}\|_{Frobenius}} + \ln\left(\frac{n^2\big(e^2\max\{1, \frac{1-\nu}{1+\nu}\}\big)^{5(n-1)/2}}{\|W_{n:1,s}\|_{Frobenius}^2}\right), \tag{18}$$

*where the function $m : [0,t] \to \mathbb{R}$ is non-negative.*

*Proof.* See Subappendix J.8. $\qquad\square$

Plugging the gradient flow results of Proposition 3 into the generic Theorem 3 translates them to the following convergence guarantee for gradient descent.

**Theorem 4.** *Assume the same conditions as in Proposition 3, but with minimization via gradient descent (Equation (6)) instead of gradient flow.[15] Then, with $\boldsymbol{\theta}_0, \boldsymbol{\theta}_1, \boldsymbol{\theta}_2, \ldots$ representing the iterates of gradient descent, $W_{n:1,0}$ standing for the end-to-end matrix (Equation (10)) of the initial point $\boldsymbol{\theta}_0$, and $\nu := \mathrm{Tr}(\Lambda_{yx}^\top W_{n:1,0}) / \big(\|\Lambda_{yx}\|_{Frobenius}\|W_{n:1,0}\|_{Frobenius}\big)$, for any $\tilde{\epsilon} > 0$, if the step size $\eta$ meets:*

$$\eta \leq \frac{\|W_{n:1,0}\|_{Frobenius}^5 \min\{1, \tilde{\epsilon}\}}{n^{17/2}e^{7n+6}\big(\max\{1, \frac{1-\nu}{1+\nu}\}\big)^{(11n-5)/2}} \left(\ln\left(\frac{15n\max\{1, \frac{1-\nu}{1+\nu}\}}{\|W_{n:1,0}\|_{Frobenius}\min\{1, \tilde{\epsilon}\}}\right)\right)^{-2} \in \tilde{\Omega}\left(\frac{\|W_{n:1,0}\|_{Frobenius}^5 \tilde{\epsilon}}{n^{17/2}\big(poly\big(\frac{1-\nu}{1+\nu}\big)\big)^n}\right), \tag{19}$$

*it holds that $f(\boldsymbol{\theta}_k) - \min_{\mathbf{q} \in \mathbb{R}^d} f(\mathbf{q}) \leq \tilde{\epsilon}$, where:*

$$k = \left\lfloor \frac{2n\big(\max\{1, \frac{3}{2} \cdot \frac{1-\nu}{1+\nu}\}\big)^n}{\|W_{n:1,0}\|_{Frobenius}\eta} \ln\left(\frac{15n\max\{1, \frac{1-\nu}{1+\nu}\}}{\|W_{n:1,0}\|_{Frobenius}\min\{1, \tilde{\epsilon}\}}\right) + 1 \right\rfloor \in \tilde{\mathcal{O}}\left(\frac{n\big(poly\big(\frac{1-\nu}{1+\nu}\big)\big)^n \ln\big(\frac{1}{\tilde{\epsilon}}\big)}{\|W_{n:1,0}\|_{Frobenius}\eta}\right). \tag{20}$$

*Proof.* See Subappendix J.9. $\qquad\square$

**Remark 1.** *Theorem 3 — our generic tool for translating analyses between gradient flow and gradient descent — allows for the two to be initialized differently. Accordingly, the convergence guarantee of Theorem 4 may be extended to account for initialization which is not perfectly balanced, i.e. which satisfies Equation (15) only approximately. For details see Appendix H.*

**Remark 2.** *The convergence guarantee of Theorem 4 requires a number of iterates that scales exponentially with network depth ($n$). [51] has proven that under mild conditions, for a deep linear neural network whose input, hidden and output dimensions are all equal to one (i.e., in our notations, $d_0 = d_1 = \cdots = d_n = 1$), such exponential dependence on depth is unavoidable. We defer to future work the question of whether this also holds in the context of Theorem 4.*

Combining Theorem 4 with random balanced initialization (Procedure 1) yields what is, to our knowledge, the first guarantee of random (data-independent) near-zero initialization *almost surely* leading a conventional gradient-based algorithm optimizing a deep (three or more layer) neural network of fixed (data-independent) size to efficiently converge to global minimum.

**Corollary 2.** *Consider minimization of the training loss $f(\cdot)$ in Equation (16) via gradient descent (Equation (6)) emanating from a random balanced initialization (Procedure 1) whose underlying distribution $\mathcal{P}$ is continuous and satisfies $\mathrm{Pr}_{A \sim \mathcal{P}}\big[\|A\|_{Frobenius} \leq 0.2\big] = 1$. Assume $d_0$ (network input dimension) is greater than one, and let $W_{n:1,0}$ and $\nu$ be as defined in Theorem 4. Then, almost surely with respect to (i.e. with probability one over) initialization, for any $\tilde{\epsilon} > 0$, if the step size $\eta$ meets Equation (19), the value of $f(\cdot)$ after $k$ iterates will be within $\tilde{\epsilon}$ from global minimum, where $k$ is given by Equation (20).*

*Proof.* See Subappendix J.10. $\qquad\square$

---

[15] The conditions on $\boldsymbol{\theta}_s$ in Proposition 3 are now satisfied by the initialization of gradient descent, *i.e.* by $\boldsymbol{\theta}_0$.

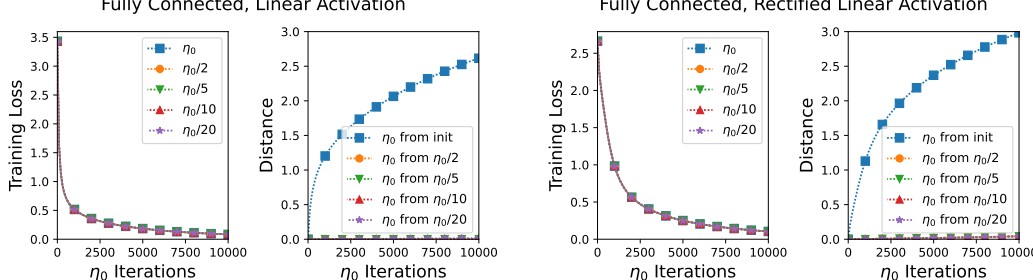

Figure 1: Over deep fully connected neural networks, trajectories of gradient descent with conventional step size barely change when step size is reduced, suggesting they are close to the continuous limit, *i.e.* to trajectories of gradient flow. Presented results were obtained on fully connected neural networks as analyzed in Subsection 4.1, trained to classify MNIST handwritten digits (28-by-28 grayscale images, each labeled as an integer between $0$ and $9$ — *cf.* [35]). Networks had depth $n=3$, input dimension $d_0=784$ (corresponding to $28 \cdot 28 = 784$ pixels), hidden widths $d_1=d_2=50$ and output dimension $d_3=10$ (corresponding to ten possible labels). Training was based on gradient descent applied to cross-entropy loss with no regularization, starting from a near-zero point drawn from Xavier distribution (*cf.* [24]). Separately on each network, we compared runs differing only in the step size $\eta$. Specifically, with $\eta_0=0.001$ (standard choice of step size) and $r$ ranging over $\{2,5,10,20\}$, we compared, in terms of training loss value and location in weight space, every iteration of a run using $\eta=\eta_0$ to every $r$'th iteration of a run in which $\eta=\eta_0/r$. Left pair of plots reports results obtained on a network with linear activation ($\sigma(z)=z$), while right pair corresponds to a network with rectified linear activation ($\sigma(z)=\max\{z,0\}$). In each pair, left plot displays training loss values, and right one shows (Euclidean) distances in weight space, namely, distance between initialization and run with $\eta=\eta_0$, alongside distances between run with $\eta=\eta_0$ and runs having $\eta=\eta_0/r$ for different values of $r$. Horizontal axes represent time in units of $\eta=\eta_0$ iterations (meaning each time unit corresponds to $r$ iterations of a run with $\eta=\eta_0/r$). Notice that the drift between runs with different step sizes is minor compared to the distance traveled. For further implementation details, and results of similar experiments on convolutional neural networks, see Appendix I.

# 6   Experiments

In this section we corroborate our theory by presenting experiments suggesting that over simple deep neural networks, gradient descent with conventional step size is indeed close to the continuous limit, *i.e.* to gradient flow. Our experimental protocol is simple — on several deep neural networks classifying MNIST handwritten digits ([35]), we compare runs of gradient descent differing only in the step size $\eta$. Specifically, separately on each evaluated network, with $\eta_0=0.001$ (standard choice of step size) and $r$ ranging over $\{2,5,10,20\}$, we compare, in terms of training loss value and location in weight space, every iteration of a run using $\eta=\eta_0$ to every $r$'th iteration of a run in which $\eta=\eta_0/r$. Figure 1 reports the results obtained on fully connected neural networks (as analyzed in Subsection 4.1), with both linear and non-linear activation. As can be seen, reducing the step size $\eta$ leads to only slight changes, suggesting that the trajectory of gradient descent with $\eta=\eta_0$ is already close to the continuous limit. Similar results obtained on convolutional neural networks (see Subsection 4.2 for corresponding analysis) are reported by Figure 3 in Subappendix I.1.

Our experimental findings suggest that in practice, proximity between gradient descent and gradient flow may take place even when the step size of gradient descent is larger than permitted by current theory. Indeed, the theoretical machinery developed in this paper brings forth upper bounds on step size that guarantee proximity, and while such upper bounds can be asymptotically tight under worst case conditions (see Appendix B), they are by no means tight in every given scenario, and therefore larger step sizes may also admit proximity. For illustration, a step size of $\eta_0$, which in our experiments was seemingly sufficient for ensuring proximity, is many orders of magnitude greater than the upper bound on step size required by Theorem 4 (Equation (19)).

# 7   Related Work

Theoretical study of gradient-based optimization in deep learning is an extremely active area of research. While far too wide to fully cover here, we note that analyses in this area can broadly be categorized as continuous (see for example [49, 4, 34, 6, 1, 20, 57, 46, 31, 47, 60, 7, 62]) or discrete (*e.g.* [9, 26, 17, 2, 16, 66, 28]). There are works comprising analyses of both types (*cf.* [18, 30, 5, 61, 39, 19, 11, 12]), but with these developed separately, wherein continuous proofs typically serve as inspiration for discrete ones (which are often far more technical and brittle).

When relating continuous and discrete optimization, the algorithms at play are most commonly gradient flow and gradient descent. There are however works that draw analogies between other algorithms, replacing gradient flow on the continuous end and/or gradient descent on the discrete one (see, *e.g.*, [54, 58, 59, 45, 50, 37, 52, 63, 22, 43, 40, 8, 33, 53]). The literature includes works which, similarly to the current paper, provide formal results concerning the accumulated (non-local) discrepancy between continuous and discrete optimization (*cf.* [50, 43]). However, such works typically focus on simple objective functions (for example convex or quadratic), whereas we center on (non-convex and non-smooth) training losses of deep neural networks. Several recent works (*e.g.* [8, 33, 14]) also considered continuous *vs.* discrete optimization of deep neural networks, but they did not provide formal results concerning the accumulated discrepancy. We are not aware of any study (prior to the current) formally quantifying the accumulated discrepancy between continuous and discrete optimization of deep neural networks.

With regards to the convergence guarantee we obtain in Section 5 (via translation of gradient flow analysis to gradient descent) — Theorem 4 and Corollary 2 — relevant results are those that establish efficient convergence[4] to global minimum for a conventional (discrete) gradient-based algorithm optimizing a deep (three or more layer) neural network. Existing results meeting this criterion either: *(i)* apply to neural networks (linear or non-linear) whose size depends on the data (*i.e.* is not data-independent[3]), predominantly in an impractical fashion (*cf.* [65, 17, 2, 19, 64, 42]); or *(ii)* apply to linear neural networks of fixed (data-independent) size, similarly to our guarantee. Results of type *(ii)* often treat the residual setting, which boils down to (possibly scaled) identity initialization, perhaps with input and/or output layers initialized differently (see for example [9, 61, 66]). Exceptions include [5], [16] and [28]. [5] allows for random balanced initialization, as we do. Its results account for networks with multi-dimensional output, and require a number of iterates polynomial in network depth. Our guarantee on the other hand is limited to networks with one-dimensional output, and calls for a number of iterates scaling exponentially with network depth. However, while [5] demands that initialization be sufficiently close to global minimum, thereby excluding the possibility of saddle points being encountered, our guarantee holds *almost surely* (*i.e.* with probability one) under random (data-independent) near-zero initialization. The fact that we account for evasion of saddle points (in particular that at the origin, which is non-strict[16] when network depth is three or more) may be the source of the gap in number of iterates — see Remark 2. As for the results of [16] and [28], these also hold with high probability under random initialization, but they require network size to grow towards infinity in order for the probability to approach one.

## 8 Discussion

Our work puts forth a potential explanation to a puzzling phenomenon in deep learning, namely, the effect of weight decay ($L_2$ regularization). While traditionally viewed as a regularizer, it is known (*cf.* [32]) that in deep learning, weight decay can assist in minimizing the training loss. In light of our findings, a possible reason for this is that weight decay translates to adding a positive constant to Hessian eigenvalues, thereby bringing gradient descent closer to gradient flow, which often enjoys favorable convergence properties. Theoretical and/or empirical investigation of this prospect is a potential avenue for future work.

Emerging evidence (*cf.* [38, 36, 29]) suggests that for (variants of) gradient descent optimizing deep neural networks, large step size is often beneficial in terms of generalization (*i.e.* in terms of test accuracy). While the large step size regime is not necessarily captured by standard (variants of) gradient flow (see [14]), recent works (*e.g.* [8, 33, 53]) argue that it is captured by a certain modified version of (variants of) gradient flow. Formally quantifying the discrepancy between gradient descent with large step size and such modified version of gradient flow is a promising direction for future research.

The demonstration we provided for translation of a gradient flow analysis to gradient descent (Section 5) culminated in a convergence guarantee, but in fact entails much more information. Namely, since the translated gradient flow analysis includes a careful trajectory characterization, not only do we know that gradient descent converges to global minimum (and how fast that happens), but we also have access to information about the trajectory it takes to get there. This allows, for example, shedding light on how saddle points (non-strict ones in particular[16]) are evaded. A nascent belief (*cf.* [5, 6]) is that understanding the trajectories of gradient descent is key to unraveling mysteries behind optimization and generalization (implicit regularization) in deep learning. The machinery developed in the current paper may contribute to this understanding, by translating results from the vast bodies of literature on continuous dynamical systems.

---

[16]A saddle point is said to be non-strict if its Hessian has no negative eigenvalues. Saddle points that are non-strict are generally regarded as more difficult to evade — *cf.* [5].

## Acknowledgments and Disclosure of Funding

We thank Sanjeev Arora, Noah Golowich, Wei Hu, Michael Lee, Zhiyuan Li, Kaifeng Lyu, Govind Menon and Zsolt Veraszto for helpful discussions. This work was supported by a Google Research Scholar Award, a Google Research Gift, the Yandex Initiative in Machine Learning, the Israel Science Foundation (grant 1780/21), Len Blavatnik and the Blavatnik Family Foundation, and Amnon and Anat Shashua.

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
