## A  Infinite Time for Gradient Flow Over Smooth Objective

By Theorem 1, gradient flow over a twice continuously differentiable objective function $f : \mathbb{R}^d \to \mathbb{R}$ (Equation (5)) admits a unique solution $\boldsymbol{\theta} : [0, t_e) \to \mathbb{R}^d$, where either: *(i)* $t_e = \infty$; or *(ii)* $t_e < \infty$ and $\lim_{t \nearrow t_e} \|\boldsymbol{\theta}(t)\|_2 = \infty$. Lemma 3 below shows that if $f(\cdot)$ is $\beta$-smooth then necessarily $t_e = \infty$.

**Lemma 3.** *Let $f : \mathbb{R}^d \to \mathbb{R}$ be twice continuously differentiable and $\beta$-smooth with $\beta > 0$ (meaning $\|\nabla^2 f(\mathbf{q})\|_{spectral} \le \beta$ for all $\mathbf{q} \in \mathbb{R}^d$). Then, for any $\boldsymbol{\theta}_s \in \mathbb{R}^d$, there exists a solution $\boldsymbol{\theta} : [0, \infty) \to \mathbb{R}^d$ to gradient flow over $f(\cdot)$ initialized at $\boldsymbol{\theta}_s$ (Equation (5)).*

*Proof.* In light of Theorem 1, there exists a solution (to gradient flow over $f(\cdot)$ initialized at $\boldsymbol{\theta}_s$) $\boldsymbol{\theta} : [0, t_e) \to \mathbb{R}^d$, where either: *(i)* $t_e = \infty$; or *(ii)* $t_e < \infty$ and $\lim_{t \nearrow t_e} \|\boldsymbol{\theta}(t)\|_2 = \infty$. It suffices to prove that condition *(ii)* is not satisfied. Assume by way of contradiction that it is. Then, there exists $t_0 \in [0, t_e)$ such that for every $t \in [t_0, t_e)$, $\|\boldsymbol{\theta}(t)\|_2 \ne 0$ and we may write:

$$\begin{aligned}
\tfrac{d}{dt} \|\boldsymbol{\theta}(t)\|_2 &= \big(\boldsymbol{\theta}(t)/\|\boldsymbol{\theta}(t)\|_2\big)^\top \tfrac{d}{dt} \boldsymbol{\theta}(t) \\
&= \big(\boldsymbol{\theta}(t)/\|\boldsymbol{\theta}(t)\|_2\big)^\top \big(-\nabla f(\boldsymbol{\theta}(t))\big) \\
&\le \|\nabla f(\boldsymbol{\theta}(t))\|_2 \\
&= \|\nabla f(\mathbf{0}) + \nabla f(\boldsymbol{\theta}(t)) - \nabla f(\mathbf{0})\|_2 \\
&\le \|\nabla f(\mathbf{0})\|_2 + \|\nabla f(\boldsymbol{\theta}(t)) - \nabla f(\mathbf{0})\|_2 \\
&\le \|\nabla f(\mathbf{0})\|_2 + \beta \|\boldsymbol{\theta}(t)\|_2 \,,
\end{aligned}$$

where the first transition follows from the chain rule, the second holds since $\boldsymbol{\theta}(\cdot)$ is a solution to gradient flow over $f(\cdot)$, the third is an application of the Cauchy-Schwartz inequality, the fourth is trivial, the fifth results from the triangle inequality, and the sixth is due to $\beta$-smoothness of $f(\cdot)$. Dividing by the right-hand side above and integrating between $t_0$ and some $t' \in [t_0, t_e)$, we obtain:

$$\beta^{-1} \ln\big(\|\nabla f(\mathbf{0})\|_2 + \beta \|\boldsymbol{\theta}(t')\|_2\big) - \beta^{-1} \ln\big(\|\nabla f(\mathbf{0})\|_2 + \beta \|\boldsymbol{\theta}(t_0)\|_2\big) \le t' - t_0 \,,$$

which in turn implies:

$$\|\boldsymbol{\theta}(t')\|_2 \le \beta^{-1} \Big( \big(\|\nabla f(\mathbf{0})\|_2 + \beta \|\boldsymbol{\theta}(t_0)\|_2\big) \exp\big(\beta(t' - t_0)\big) - \|\nabla f(\mathbf{0})\|_2 \Big) \,.$$

We conclude that for any $t' \in [t_0, t_e)$, it holds that $\|\boldsymbol{\theta}(t')\|_2 \le c$, where:

$$c := \beta^{-1} \Big( \big(\|\nabla f(\mathbf{0})\|_2 + \beta \|\boldsymbol{\theta}(t_0)\|_2\big) \exp\big(\beta(t_e - t_0)\big) - \|\nabla f(\mathbf{0})\|_2 \Big) < \infty \,.$$

This of course contradicts $\lim_{t \nearrow t_e} \|\boldsymbol{\theta}(t)\|_2 = \infty$, affirming that condition *(ii)* above is false. $\qquad\square$

## B  Worst Case Scenario

Theorem 3 in Section 3 established that if gradient descent (Equation (6)) is applied with step size $\eta$ meeting a certain upper bound (Equation (7)), then its trajectory will $\epsilon$-approximate that of gradient flow (Equation (5)) up to a given time $\tilde{t}$. The upper bound on $\eta$ decays exponentially with the integral of $m(\cdot)$ along the gradient flow trajectory up to time $\tilde{t}$, where $m(\cdot)$ corresponds to minus the minimal eigenvalue of the Hessian. Replacing $m(\cdot)$ by a constant $m$ equal to minus the minimal eigenvalue of the Hessian *across the entire space* results in a coarse bound, which for a non-convex objective ($m > 0$) scales as $e^{-m\tilde{t}}$ — see Corollary 1. The current appendix shows that in the worst case, such exponential scaling is necessary. That is, there exist objective functions and initializations with which the location of gradient flow at time $\tilde{t}$ will not be $\epsilon$-approximated by the trajectory of gradient descent (at any iteration) unless the step size of gradient descent is $\mathcal{O}(e^{-m\tilde{t}})$. We prove this via an example, whose crux is that the gradient flow trajectories it entails traverse through regions where Hessian eigenvalues coincide with the minimal one across space.

Let $a > 0$, $b \ge 3$ and $\epsilon \in (0, 1)$. Define the "cut points" $z_c := be^{30} + 1$ and $\bar{z}_c := b + 1$, and the "transition width" $\bar{\rho} := \min\{e^{-12}/2, \epsilon/2b\}$. Consider the functions $\varphi, \bar{\varphi} : \mathbb{R} \to \mathbb{R}$ given by:

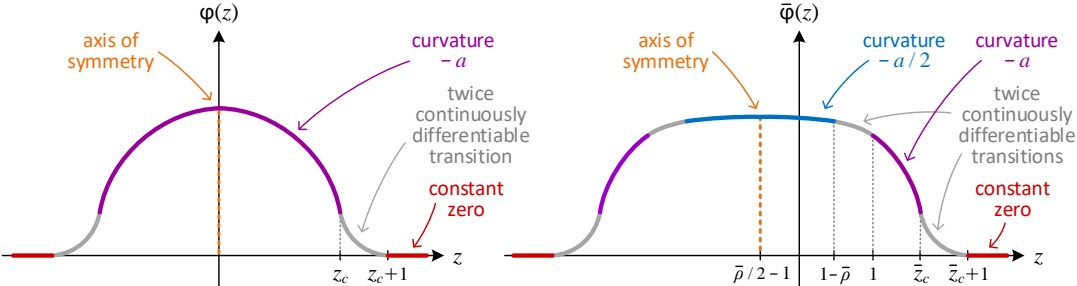

Figure 2: Illustrations of the functions $\varphi(\cdot)$ and $\bar{\varphi}(\cdot)$ defined in Equations (21) and (22) respectively.

$$\varphi(z)=\begin{cases} \frac{1}{2}a(z_c+1)^2-\frac{5}{12}a-\frac{1}{2}az_c & ,z=0 \\ \varphi(0)-\frac{1}{2}az^2 & ,z\in(0,z_c) \\ \varphi(0)-\frac{1}{2}az^2+a\left(\frac{2}{3}+z_c\right)(z-z_c)^3-a\left(\frac{1}{4}+\frac{1}{2}z_c\right)(z-z_c)^4 & ,z\in[z_c,z_c+1] \\ 0 & ,z\in(z_c+1,\infty) \\ \varphi(|z|) & ,z\in(-\infty,0) \end{cases}, \qquad (21)$$

$$\bar{\varphi}(z)=\begin{cases} \frac{1}{2}a(\bar{z}_c+1)^2+\frac{1}{12}a-\frac{1}{2}a\bar{z}_c-a\left(\frac{1}{2}\bar{\rho}-\frac{7}{48}\bar{\rho}^2\right) & ,z=\frac{1}{2}\bar{\rho}-1 \\ \bar{\varphi}\left(\frac{1}{2}\bar{\rho}-1\right)-\frac{1}{4}a\left(z-\left(\frac{1}{2}\bar{\rho}-1\right)\right)^2 & ,z\in\left(\frac{1}{2}\bar{\rho}-1,1-\bar{\rho}\right) \\ \bar{\varphi}\left(\frac{1}{2}\bar{\rho}-1\right)-\frac{1}{2}a+a\left(\frac{1}{2}\bar{\rho}-\frac{7}{48}\bar{\rho}^2\right)-\frac{1}{2}az^2-\frac{1}{12}a\bar{\rho}^{-1}(z-1)^3 & ,z\in[1-\bar{\rho},1] \\ \bar{\varphi}\left(\frac{1}{2}\bar{\rho}-1\right)-\frac{1}{2}a+a\left(\frac{1}{2}\bar{\rho}-\frac{7}{48}\bar{\rho}^2\right)-\frac{1}{2}az^2 & ,z\in(1,\bar{z}_c) \\ \bar{\varphi}\left(\frac{1}{2}\bar{\rho}-1\right)-\frac{1}{2}a+a\left(\frac{1}{2}\bar{\rho}-\frac{7}{48}\bar{\rho}^2\right)-\frac{1}{2}az^2 \\ \qquad +a\left(\frac{2}{3}+\bar{z}_c\right)(z-\bar{z}_c)^3-a\left(\frac{1}{4}+\frac{1}{2}\bar{z}_c\right)(z-\bar{z}_c)^4 & ,z\in[\bar{z}_c,\bar{z}_c+1] \\ 0 & ,z\in(\bar{z}_c+1,\infty) \\ \bar{\varphi}\left(\left|z-\left(\frac{1}{2}\bar{\rho}-1\right)\right|+\frac{1}{2}\bar{\rho}-1\right) & ,z\in\left(-\infty,\frac{1}{2}\bar{\rho}-1\right) \end{cases}. \qquad (22)$$

Both $\varphi(\cdot)$ and $\bar{\varphi}(\cdot)$ are twice continuously differentiable, non-negative and smooth,[17] with minimal curvature (second derivative) equal to $-a$. $\varphi(\cdot)$ comprises three parts — *(i)* constant zero over $(-\infty,-z_c-1)$; *(ii)* quadratic with curvature $-a$ over $(-z_c,z_c)$; and *(iii)* constant zero over $(z_c+1,\infty)$ — with twice continuously differentiable transitions in-between. $\bar{\varphi}(\cdot)$ consists of five parts — *(i)* constant zero over $(-\infty,-\bar{z}_c-3+\bar{\rho})$; *(ii)* quadratic with curvature $-a$ over $(-\bar{z}_c-2+\bar{\rho},-3+\bar{\rho})$; *(iii)* quadratic with curvature $-a/2$ over $(-3+2\bar{\rho},1-\bar{\rho})$; *(iv)* quadratic with curvature $-a$ over $(1,\bar{z}_c)$; and *(v)* constant zero over $(\bar{z}_c+1,\infty)$ — also joined by twice continuously differentiable transitions. Illustrations of $\varphi(\cdot)$ and $\bar{\varphi}(\cdot)$ are presented in Figure 2.

Let $d\in\mathbb{N}_{\geq 3}$, and consider the objective function $f:\mathbb{R}^d\to\mathbb{R}$ defined by:

$$f(\mathbf{q})=\varphi(q_1)+\bar{\varphi}(q_2)+6aq_3^2, \qquad (23)$$

where $q_1$, $q_2$ and $q_3$ stand for the first, second and third coordinates (respectively) of $\mathbf{q}\in\mathbb{R}^d$. $f(\cdot)$ meets the conditions of Corollary 1 — it is twice continuously differentiable, non-negative and smooth.[18] The minimal eigenvalue of its Hessian across space (*i.e.* $\inf_{\mathbf{q}\in\mathbb{R}^d}\lambda_{min}(\nabla^2 f(\mathbf{q}))$, where $\lambda_{min}(\nabla^2 f(\mathbf{q}))$ represents the minimal eigenvalue of $\nabla^2 f(\mathbf{q})$) is $-a$, meaning the constant $m:=-\inf_{\mathbf{q}\in\mathbb{R}^d}\lambda_{min}(\nabla^2 f(\mathbf{q}))$ is equal to $a$. Building on the fact that in the region $(0,z_c)\times(1,\bar{z}_c)\times\mathbb{R}^{d-2}$ the Hessian has eigenvalues coinciding with the minimum (*i.e.* equal to $-a$), Proposition 4 below establishes the sought-after result — over $f(\cdot)$, there exist gradient flow trajectories whose $\epsilon$-approximation at a given time $\tilde{t}$ requires gradient descent to have step size $\mathcal{O}(e^{-m\tilde{t}})$.

**Proposition 4.** *Let $\boldsymbol{\theta}_s=(\theta_{s,1},\theta_{s,2},...,\theta_{s,d})\in\mathbb{R}^d$ be such that $\theta_{s,1}\in(0.5,1)$, $\theta_{s,2}\in(e^{-12}/2-1,$ $e^{-12}-1)$ and $\theta_{s,3}>2$. In the above context (in particular with the objective function $f:\mathbb{R}^d\to\mathbb{R}$ defined*

---

[17]Their second derivatives are bounded.

[18]There exists $\beta>0$ such that $\|\nabla^2 f(\mathbf{q})\|_{spectral}\leq\beta$ for all $\mathbf{q}\in\mathbb{R}^d$.

*by Equation (23), for which* $m := -\inf_{\mathbf{q} \in \mathbb{R}^d} \lambda_{min}(\nabla^2 f(\mathbf{q})) = a)$, *denote by* $\boldsymbol{\theta}(\cdot)$ *the trajectory of gradient flow initialized at* $\boldsymbol{\theta}_s$ *(solution to Equation (5)), and by* $\boldsymbol{\theta}_0, \boldsymbol{\theta}_1, \boldsymbol{\theta}_2, ...$ *the iterates of gradient descent with step size* $\eta > 0$ *(Equation (6)) emanating from the same point (i.e. with* $\boldsymbol{\theta}_0 = \boldsymbol{\theta}_s$*). Then, for any time* $\tilde{t} \in \left[ \frac{2}{a}\ln\left(\frac{2-3\bar{\rho}/2}{\theta_{s,2}-(\bar{\rho}/2-1)}\right) + \frac{1}{a}\ln\left(\frac{2}{1-\bar{\rho}}\right), \frac{2}{a}\ln\left(\frac{2-3\bar{\rho}/2}{\theta_{s,2}-(\bar{\rho}/2-1)}\right) + \frac{1}{a}\ln\left(\frac{1+\bar{\rho}/4}{1-3\bar{\rho}/4}\right) + \frac{1}{a}\ln(b) \right],$ [19] *if* $\eta \geq \frac{10^{14}}{a}e^{-a\tilde{t}}\epsilon$, *it holds that* $\|\boldsymbol{\theta}_k - \boldsymbol{\theta}(\tilde{t})\|_2 > \epsilon$ *for all* $k \in \mathbb{N} \cup \{0\}$.[20]

*Proof sketch (for complete proof see Subappendix J.11).* Since $f(\cdot)$ is additively separable (can be expressed as a sum of terms, each depending on a single input variable), the dynamics in $\mathbb{R}^d$ induced by gradient flow and gradient descent can be analyzed separately for different coordinates. Restricting our attention to the first two coordinates, we observe that gradient flow and gradient descent initially traverse through an "anisotropic" region, where curvature is $-a$ in the first coordinate and $-a/2$ in the second, and from there move to an "isotropic" region, where curvature is $-a$ in both the first and second coordinates. In the isotropic region, if gradient descent is placed along a gradient flow trajectory it will continue down the same path, but otherwise, if there is any discrepancy between gradient descent and gradient flow, this discrepancy will grow exponentially with time, namely will scale as $e^{at}$. Carefully characterizing the dynamics along the anisotropic region reveals that upon entrance to the isotropic one, there is indeed a discrepancy between gradient descent and gradient flow, the magnitude of which is proportional to $\eta$ (step size of gradient descent). Since this magnitude scales as $e^{at}$ thereafter, it will exceed $\epsilon$ at time $\tilde{t}$ if $\eta \notin \mathcal{O}(e^{-a\tilde{t}}\epsilon)$, which is what we set out to prove. The above analysis assumes $\eta$ is no greater than a certain constant. However, larger values for $\eta$ lead to divergence in the third coordinate (due to the term $6aq_3^2$ in the definition of $f(\cdot)$ — Equation (23)), thus these are accounted for as well (they preclude the possibility of gradient descent $\epsilon$-approximating gradient flow at time $\tilde{t}$). $\qquad\square$

## C   Analysis for Fully Connected Architectures with Non-Linear Activation

In this appendix we provide our analysis for fully connected architectures with non-linear activation, outlined in Subsubsection 4.1.2.

When the (homogeneous) activation function of the fully connected neural network defined in Equation (8) (and surrounding text) is non-linear, *i.e.* $\sigma(z) = \alpha\max\{z,0\} - \bar{\alpha}\max\{-z,0\}$ for some $\alpha, \bar{\alpha} \in \mathbb{R}$, $\alpha \neq \bar{\alpha}$, the training loss $f(\cdot)$ (Equation (9)) is (typically) not everywhere differentiable. It is however locally Lipschitz thus differentiable almost everywhere (see Theorem 9.1.2 in [10]). Moreover, as established by Proposition 9 in Appendix E, for almost every $\boldsymbol{\theta}' \in \mathbb{R}^d$ there exist diagonal matrices $D'_{i,j} \in \mathbb{R}^{d_j, d_j}$, $i = 1, 2, ..., |\mathcal{S}|$, $j = 1, 2, ..., n-1$, with diagonal elements in $\{\alpha, \bar{\alpha}\}$, such that $f(\cdot)$ coincides with the function

$$\boldsymbol{\theta} \mapsto \frac{1}{|\mathcal{S}|}\sum_{i=1}^{|\mathcal{S}|} \ell(W_n D'_{i,n-1} W_{n-1} D'_{i,n-2} W_{n-2} \cdots D'_{i,1} W_1 \mathbf{x}_i, y_i) \tag{24}$$

on an open region $\mathcal{D}_{\boldsymbol{\theta}'} \subseteq \mathbb{R}^d$ containing $\boldsymbol{\theta}'$, that is closed under positive rescaling of weight matrices (*i.e.* under $(W_1, W_2, ..., W_n) \mapsto (c_1 W_1, c_2 W_2, ..., c_n W_n)$ with $c_1, c_2, ..., c_n > 0$). The notion of gradient flow over a non-differentiable locally Lipschitz objective function is typically formalized via differential inclusion and Clarke subdifferentials (*cf.* [15, 18]). To our knowledge there exists no analogue of the Fundamental Theorem (Theorem 2) that applies to this formalization, thus we focus on (open) regions of the form $\mathcal{D}_{\boldsymbol{\theta}'}$, where $f(\cdot)$ is given by Equation (24), and in particular is twice continuously differentiable. On such regions the analysis of Section 3 applies, and since they constitute the entire weight space but a negligible (closed and zero measure) set, they can facilitate a "piecewise characterization" of the discrepancy between gradient flow and gradient descent.

Lemma 4 below expresses $\nabla^2 f(\boldsymbol{\theta})$ for $\boldsymbol{\theta} \in \mathcal{D}_{\boldsymbol{\theta}'}$.

**Lemma 4.** *Let* $\boldsymbol{\theta} \in \mathcal{D}_{\boldsymbol{\theta}'}$. *For any* $i \in \{1, 2, ..., |\mathcal{S}|\}$ *and* $j, j' \in \{1, 2, ..., n\}$ *define* $(D'_{i,*} W_*)_{j':j}$ *to be the matrix* $D'_{i,j'} W_{j'} D'_{i,j'-1} W_{j'-1} \cdots D'_{i,j} W_j$ *(where by convention* $D'_{i,n} \in \mathbb{R}^{d_n, d_n}$ *stands for identity) if* $j \leq j'$, *and an identity matrix (with size to be inferred by context) otherwise. For* $i \in \{1, 2, ..., |\mathcal{S}|\}$ *let* $\nabla \ell_i \in \mathbb{R}^{d_n}$ *and* $\nabla^2 \ell_i \in \mathbb{R}^{d_n, d_n}$ *be the gradient and Hessian (respectively) of the loss* $\ell(\cdot)$ *at the point*

---

[19]Note that the upper bound on $\tilde{t}$ can be made arbitrarily large via suitable (sufficiently large) choice of $b$.

[20]Since $f(\cdot)$ is twice continuously differentiable and smooth, $\boldsymbol{\theta}(\tilde{t})$ necessarily exists (see Lemma 3 in Appendix A).

$\big((D'_{i,*}W_*)_{n:1}\mathbf{x}_i,y_i\big)$ *with respect to its first argument. Then, regarding Hessians as quadratic forms (see examples in Lemma 1), it holds that:*

$$\nabla^2 f(\boldsymbol{\theta})[\Delta W_1,\Delta W_2,...,\Delta W_n]=\frac{1}{|\mathcal{S}|}\sum_{i=1}^{|\mathcal{S}|}\nabla^2\ell_i\left[\sum_{j=1}^{n}(D'_{i,*}W_*)_{n:j+1}D'_{i,j}(\Delta W_j)(D'_{i,*}W_*)_{j\text{-}1:1}\mathbf{x}_i\right] \quad (25)$$

$$+\frac{2}{|\mathcal{S}|}\sum_{i=1}^{|\mathcal{S}|}\nabla\ell_i^{\top}\sum_{1\le j<j'\le n}(D'_{i,*}W_*)_{n:j'+1}D'_{i,j'}(\Delta W_{j'})(D'_{i,*}W_*)_{j'\text{-}1:j+1}D'_{i,j}(\Delta W_j)(D'_{i,*}W_*)_{j\text{-}1:1}\mathbf{x}_i.$$

*Proof sketch (for complete proof see Subappendix J.12).* The proof is similar to that of Lemma 1. Namely, it expands the function in Equation (24) and then extracts second order terms. □

The following proposition employs Lemma 4 to show that (under mild conditions) there exists $\boldsymbol{\theta}\in\mathbb{R}^d$ for which $\lambda_{min}(\nabla^2 f(\boldsymbol{\theta}))$ is arbitrarily negative.

**Proposition 5.** *Assume that:* (i) *the network is deep ($n\ge 3$); and* (ii) *the loss function $\ell(\cdot)$ and training set $\mathcal{S}$ are non-degenerate, in the sense that there exists a weight setting $\boldsymbol{\theta}\in\mathbb{R}^d$ for which $\sum_{i=1}^{|\mathcal{S}|}\nabla\ell(\mathbf{0},y_i)^{\top}h_{\boldsymbol{\theta}}(\mathbf{x}_i)\neq 0$, where $\nabla\ell(\cdot)$ stands for the gradient of $\ell(\cdot)$ with respect to its first argument, and $h_{\boldsymbol{\theta}}(\cdot)$ is the input-output mapping realized by the network (Equation (8)).[21] Then, it holds that $\inf_{\boldsymbol{\theta}\in\mathbb{R}^d \text{ s.t.}\nabla^2 f(\boldsymbol{\theta}) \text{ exists}}\lambda_{min}(\nabla^2 f(\boldsymbol{\theta}))=-\infty.*

*Proof sketch (for complete proof see Subappendix J.13).* Let $\boldsymbol{\theta}\in\mathbb{R}^d$ be a weight setting realizing the non-degeneracy condition, *i.e.* for which $\sum_{i=1}^{|\mathcal{S}|}\nabla\ell(\mathbf{0},y_i)^{\top}h_{\boldsymbol{\theta}}(\mathbf{x}_i)\neq 0$. Without loss of generality, we may assume that $\boldsymbol{\theta}$ satisfies the condition $\sum_{i=1}^{|\mathcal{S}|}\nabla\ell(\mathbf{0},y_i)^{\top}h_{\boldsymbol{\theta}}(\mathbf{x}_i)<0$ (if this is not the case then simply flip the signs of the entries in $\boldsymbol{\theta}$ corresponding to the last weight matrix $W_n$). From continuity, there exists a neighborhood of $\boldsymbol{\theta}$ consisting of weight settings that all meet the latter condition. There must exist a region of the form $\mathcal{D}_{\boldsymbol{\theta}'}$ intersecting this neighborhood (since these regions constitute all of $\mathbb{R}^d$ but a zero measure set), so we may assume, without loss of generality, that $\boldsymbol{\theta}\in\mathcal{D}_{\boldsymbol{\theta}'}$. Lemma 4 then applies. Moreover, since $\mathcal{D}_{\boldsymbol{\theta}'}$ is closed under positive rescaling of weight matrices (*i.e.* of $W_1,W_2,...,W_n$), the lemma remains applicable even when $\boldsymbol{\theta}$ is subject to such rescaling. The proof proceeds by fixing $\Delta W_1,\Delta W_2,...,\Delta W_n$ to certain values, and positively rescaling $W_1,W_2,...,W_n$ in a certain way, such that the expression for $\nabla^2 f(\boldsymbol{\theta})[\Delta W_1,\Delta W_2,...,\Delta W_n]$ provided in Lemma 4 becomes arbitrarily negative. □

Relying on Lemma 4, Lemma 5 below provides a lower bound on $\lambda_{min}(\nabla^2 f(\boldsymbol{\theta}))$ for $\boldsymbol{\theta}\in\mathcal{D}_{\boldsymbol{\theta}'}$.

**Lemma 5.** *With the notations of Lemma 4, for any $\boldsymbol{\theta}\in\mathcal{D}_{\boldsymbol{\theta}'}$:[13]*

$$\lambda_{min}(\nabla^2 f(\boldsymbol{\theta}))\ge -\max\{|\alpha|,|\bar{\alpha}|\}^{n-1}\frac{n-1}{|\mathcal{S}|}\sum_{i=1}^{|\mathcal{S}|}\|\nabla\ell_i\|_2\|\mathbf{x}_i\|_2\max_{\substack{\mathcal{J}\subseteq\{1,2,...,n\}\\|\mathcal{J}|=n-2}}\prod_{j\in\mathcal{J}}\|W_j\|_{Frobenius}. \quad (26)$$

*Proof sketch (for complete proof see Subappendix J.14).* The proof is analogous to that of Lemma 2. Namely, it appeals to Lemma 4, and lower bounds the right-hand side of Equation (25). Convexity of $\ell(\cdot)$ (with respect to its first argument) implies that the first summand is non-negative. For the second summand, we use known matrix inequalities (as well as the fact that $\|D'_{i,j}\|_{spectral}$ is no greater than $\max\{|\alpha|,|\bar{\alpha}|\}$ for $j=1,2,...,n-1$, and equal to one for $j=n$) to establish a lower bound of $c\sum_{j=1}^{n}\|\Delta W_j\|^2_{Frobenius}$, with $c$ being the expression on the right-hand side of Equation (26). □

The lower bound in Equation (26) is highly sensitive to the scales of the individual weight matrices. Specifically, assuming the network is deep ($n\ge 3$), if $\boldsymbol{\theta}$ does not perfectly fit all non-zero training inputs (meaning there exists $i\in\{1,2,...,|\mathcal{S}|\}$ for which $\nabla\ell_i\neq\mathbf{0}$ and $\mathbf{x}_i\neq\mathbf{0}$), and if at least $n-2$ of its weight matrices $W_1,W_2,...,W_n$ are non-zero, then it is possible to rescale each $W_j$ by $c_j>0$, with $\prod_{j=1}^{n}c_j=1$, such that the lower bound in Equation (26) becomes arbitrarily negative[22] despite the

---

[21] Assumptions *(i)* and *(ii)* are both necessary, in the sense that removing any of them (without imposing further assumptions) renders the proposition false — see Claim 2 in Appendix F. Assumption *(ii)* in particular is extremely mild, *e.g.* if $\ell(\cdot)$ is the square loss (*i.e.* $\mathcal{Y}=\mathbb{R}^{d_n}$ and $\ell(\hat{\mathbf{y}},\mathbf{y})=\frac{1}{2}\|\hat{\mathbf{y}}-\mathbf{y}\|_2^2$), the slightest change in a single label ($\mathbf{y}_i$) corresponding to a non-zero prediction ($h_{\boldsymbol{\theta}}(\mathbf{x}_i)\neq\mathbf{0}$) can ensure the inequality.

[22] The bound remains applicable since $\mathcal{D}_{\boldsymbol{\theta}'}$ is closed under positive rescaling of weight matrices.

input-output mapping $h_{\boldsymbol{\theta}}(\cdot)$ (and thus the training loss value $f(\boldsymbol{\theta})$) remaining unchanged. Nevertheless, similarly to the case of linear activation (Subsubsection 4.1.1), we may employ the fact that gradient flow over fully connected neural networks (with homogeneous activations) initialized near zero maintains balance between weight matrices — *cf.* [18] — to show that along its trajectories, the lower bound in Equation (26) assumes a tighter form. This is done in Proposition 6 below.

**Proposition 6.** *If $\boldsymbol{\theta} \in \mathcal{D}_{\boldsymbol{\theta}'}$ resides on a trajectory of gradient flow (over $f(\cdot)$)[23] initialized at some point $\boldsymbol{\theta}_s \in \mathbb{R}^d$, with $\|\boldsymbol{\theta}_s\|_2 \leq \epsilon$ for some $\epsilon > 0$, then, using the notations of Lemma 4:*

$$\lambda_{min}(\nabla^2 f(\boldsymbol{\theta})) \geq -\max\{|\alpha|,|\bar{\alpha}|\}^{n-1}\frac{n-1}{|\mathcal{S}|}\sum_{i=1}^{|\mathcal{S}|}\|\nabla\ell_i\|_2\|\mathbf{x}_i\|_2\Big(\min_{j\in\{1,2,...,n\}}\|W_j\|_{Frobenius}+\epsilon\Big)^{n-2}. \quad (27)$$

*Proof sketch (for complete proof see Subappendix J.15).* By the analysis of [18], for any $j,j' \in \{1,2,...,n\}$, the quantity $\|W_{j'}\|_{Frobenius}^2 - \|W_j\|_{Frobenius}^2$ is invariant (constant) along a gradient flow trajectory. This implies that along a trajectory emanating from a point with (Euclidean) norm $\mathcal{O}(\epsilon)$, it holds that $\|W_{j'}\|_{Frobenius}^2 - \|W_j\|_{Frobenius}^2 \in \mathcal{O}(\epsilon^2)$ for all $j,j' \in \{1,2,...,n\}$, which in turn implies $\|W_{j'}\|_{Frobenius} \leq \min_{j\in\{1,2,...,n\}}\|W_j\|_{Frobenius} + \mathcal{O}(\epsilon)$ for all $j' \in \{1,2,...,n\}$. Plugging this into Equation (26) yields the desired result (Equation (27)). $\qquad\square$

Assume the network is deep ($n \geq 3$), and consider a trajectory of gradient flow (over $f(\cdot)$) emanating from near-zero initialization. For every point on the trajectory, Proposition 6 may be applied with small $\epsilon$, leading the lower bound in Equation (27) to depend primarily on the *minimal* size (Frobenius norm) of a weight matrix $W_j$, and on $\nabla\ell_1, \nabla\ell_2, ..., \nabla\ell_{|\mathcal{S}|}$ — gradients of the loss function with respect to the predictions over the training set. In the course of optimization, $W_1, W_2, ..., W_n$ are initially small, and if a perfect fit of the training set is ultimately achieved, $\nabla\ell_1, \nabla\ell_2, ..., \nabla\ell_{|\mathcal{S}|}$ will converge to zero. Therefore, if not *all* weight matrices $W_1, W_2, ..., W_n$ become large during optimization, the lower bound on $\lambda_{min}(\nabla^2 f(\boldsymbol{\theta}))$ in Equation (27) will only be moderately negative before approaching non-negativity (if and) as the trajectory converges to a perfect fit. In light of Section 3, this suggests that the gradient flow trajectory may lend itself to approximation by gradient descent. For a case with linear activation (Subsubsection 4.1.1) such prospect is theoretically verified in Section 5. For non-linear activation we provide empirical corroboration in Section 6, deferring to future work a complete theoretical affirmation.

## D    Analysis for Convolutional Architectures

In this appendix we provide our analysis for convolutional architectures, outlined in Subsection 4.2.

Suppose we modify the fully connected neural network defined in Equation (8) (and surrounding text) by converting each learned weight matrix $W_j \in \mathbb{R}^{d_j, d_{j-1}}$, $j = 1, 2, ..., n$, into a function $W_j : \mathbb{R}^{d'_j} \to \mathbb{R}^{d_j, d_{j-1}}$, with $d'_j \in \mathbb{N}$, that intakes a learned weight vector $\mathbf{w}_j \in \mathbb{R}^{d'_j}$, and returns a matrix where each element is either fixed at zero or connected to a predetermined coordinate of $\mathbf{w}_j$, with no repetition of coordinates within the same row (that is, each row of $W_j(\cdot)$ realizes a function of the form $\mathbf{w}_j \mapsto P\mathbf{w}_j$, where $P \in \mathbb{R}^{d_{j-1}, d'_j}$ is a matrix in which no row or column includes more than a single non-zero element, and all non-zero elements are equal to one). This allows imposing various weight sharing and sparsity patterns on the layers of the model, in particular ones giving rise to convolutional neural networks. The resulting input-output mapping has the form:

$$h_{\boldsymbol{\theta}} : \mathbb{R}^{d_0} \to \mathbb{R}^{d_n} , h_{\boldsymbol{\theta}}(\mathbf{x}) = W_n(\mathbf{w}_n)\sigma\big(W_{n\text{-}1}(\mathbf{w}_{n\text{-}1})\sigma\big(W_{n\text{-}2}(\mathbf{w}_{n\text{-}2})\cdots\sigma\big(W_1(\mathbf{w}_1)\mathbf{x}\big)\cdots\big), \quad (28)$$

where $\boldsymbol{\theta} \in \mathbb{R}^d$, with $d := \sum_{j=1}^{n} d'_j$, is the concatenation of the weight vectors $\mathbf{w}_1, \mathbf{w}_2, ..., \mathbf{w}_n$,[24] and as before, $\sigma : \mathbb{R} \to \mathbb{R}$ is a predetermined activation function (operating element-wise when

---

[23]Recall that in the current context, the optimized objective function $f(\cdot)$ is locally Lipschitz but (typically) non-differentiable. Following a conventional formalization in such settings (*cf.* [15, 18]), we regard a curve in $\mathbb{R}^d$ as a trajectory of gradient flow if it satisfies the differential inclusion $\frac{d}{dt}\boldsymbol{\theta}(t) \in -\partial f(\boldsymbol{\theta}(t))$ for almost every time $t$, where $\partial f(\boldsymbol{\theta}(t)) \subseteq \mathbb{R}^d$ stands for the Clarke subdifferential (see [13]) of $f(\cdot)$ at $\boldsymbol{\theta}(t)$.

[24]The exact order by which $\mathbf{w}_1, \mathbf{w}_2, ..., \mathbf{w}_n$ are concatenated is insignificant for our purposes — all that matters is that the same order be used throughout.

applied to a vector) that is (positively) homogeneous, meaning there exist $\alpha, \bar{\alpha} \in \mathbb{R}$ such that $\sigma(z) = \alpha \max\{z, 0\} - \bar{\alpha} \max\{-z, 0\}$ for all $z \in \mathbb{R}$.[25]

Let $f : \mathbb{R}^d \to \mathbb{R}$ be the training loss defined by applying Equation (9) (and surrounding text) to the above neural network (*i.e.* with $h_{\boldsymbol{\theta}}(\cdot)$ given by Equation (28)). In line with our analysis of fully connected architectures (Subsection 4.1), we will show that although the minimal eigenvalue of $\nabla^2 f(\boldsymbol{\theta})$ (Hessian of training loss) — denoted $\lambda_{min}(\nabla^2 f(\boldsymbol{\theta}))$ — can in general be arbitrarily negative, along trajectories of gradient flow (which emanate from near-zero initialization) it is no less than moderately negative, approaching non-negativity towards convergence. In light of Section 3, this suggests that over deep convolutional neural networks, gradient flow may lend itself to approximation by gradient descent — a prospect we empirically corroborate in Subappendix I.1.

Proposition 10 in Appendix E establishes that for almost every $\boldsymbol{\theta}' \in \mathbb{R}^d$ there exist diagonal matrices $D'_{i,j} \in \mathbb{R}^{d_j, d_j}$, $i = 1, 2, ..., |\mathcal{S}|$, $j = 1, 2, ..., n-1$, with diagonal elements in $\{\alpha, \bar{\alpha}\}$, such that $f(\cdot)$ coincides with the function

$$\boldsymbol{\theta} \mapsto \frac{1}{|\mathcal{S}|} \sum_{i=1}^{|\mathcal{S}|} \ell\big(W_n(\mathbf{w}_n) D'_{i,n\text{-}1} W_{n\text{-}1}(\mathbf{w}_{n\text{-}1}) D'_{i,n\text{-}2} W_{n\text{-}2}(\mathbf{w}_{n\text{-}2}) \cdots D'_{i,1} W_1(\mathbf{w}_1) \mathbf{x}_i; y_i\big) \qquad (29)$$

on an open region $\mathcal{D}_{\boldsymbol{\theta}'} \subseteq \mathbb{R}^d$ containing $\boldsymbol{\theta}'$, that is closed under positive rescaling of weight vectors (*i.e.* under $(\mathbf{w}_1, \mathbf{w}_2, ..., \mathbf{w}_n) \mapsto (c_1 \mathbf{w}_1, c_2 \mathbf{w}_2, ..., c_n \mathbf{w}_n)$ with $c_1, c_2, ..., c_n > 0$). Analogously to the case of fully connected architectures with non-linear activation (*cf.* Appendix C), we will focus on (open) regions of the form $\mathcal{D}_{\boldsymbol{\theta}'}$, where $f(\cdot)$ is given by Equation (29), and in particular is twice continuously differentiable. On such regions the analysis of Section 3 applies, and since they constitute the entire weight space but a negligible (closed and zero measure) set, they can facilitate a "piecewise characterization" of the discrepancy between gradient flow and gradient descent.[26]

Lemma 6 below expresses $\nabla^2 f(\boldsymbol{\theta})$ for $\boldsymbol{\theta} \in \mathcal{D}_{\boldsymbol{\theta}'}$.

**Lemma 6.** *Let $\boldsymbol{\theta} \in \mathcal{D}_{\boldsymbol{\theta}'}$. For any $i \in \{1, 2, ..., |\mathcal{S}|\}$ and $j, j' \in \{1, 2, ..., n\}$ define $(D'_{i,*} W_*(\mathbf{w}_*))_{j':j}$ to be the matrix $D'_{i,j'} W_{j'}(\mathbf{w}_{j'}) D'_{i,j'\text{-}1} W_{j'\text{-}1}(\mathbf{w}_{j'\text{-}1}) \cdots D'_{i,j} W_j(\mathbf{w}_j)$ (where by convention $D'_{i,n} \in \mathbb{R}^{d_n, d_n}$ stands for identity) if $j \leq j'$, and an identity matrix (with size to be inferred by context) otherwise. For $i \in \{1, 2, ..., |\mathcal{S}|\}$ let $\nabla \ell_i \in \mathbb{R}^{d_n}$ and $\nabla^2 \ell_i \in \mathbb{R}^{d_n, d_n}$ be the gradient and Hessian (respectively) of the loss $\ell(\cdot)$ at the point $((D'_{i,*} W_*(\mathbf{w}_*))_{n:1} \mathbf{x}_i; y_i)$ with respect to its first argument. Then, regarding Hessians as quadratic forms (see examples in Lemma 1), it holds that:*

$$\nabla^2 f(\boldsymbol{\theta})[\Delta\mathbf{w}_1, \Delta\mathbf{w}_2, ..., \Delta\mathbf{w}_n] = \qquad (30)$$

$$\frac{1}{|\mathcal{S}|} \sum_{i=1}^{|\mathcal{S}|} \nabla^2 \ell_i \left[ \sum_{j=1}^{n} \big(D'_{i,*} W_*(\mathbf{w}_*)\big)_{n:j+1} D'_{i,j} W_j(\Delta\mathbf{w}_j) \big(D'_{i,*} W_*(\mathbf{w}_*)\big)_{j\text{-}1:1} \mathbf{x}_i \right] +$$

$$\frac{2}{|\mathcal{S}|} \sum_{i=1}^{|\mathcal{S}|} \nabla \ell_i^\top \sum_{1 \leq j < j' \leq n} \big(D'_{i,*} W_*(\mathbf{w}_*)\big)_{n:j'+1} D'_{i,j'} W_{j'}(\Delta\mathbf{w}_{j'}) \big(D'_{i,*} W_*(\mathbf{w}_*)\big)_{j'\text{-}1:j+1} \cdot$$

$$D'_{i,j} W_j(\Delta\mathbf{w}_j) \big(D'_{i,*} W_*(\mathbf{w}_*)\big)_{j\text{-}1:1} \mathbf{x}_i \, .$$

*Proof sketch (for complete proof see Subappendix J.16).* The proof is similar to those of Lemmas 1 and 4. Namely, it expands the function in Equation (29) and then extracts second order terms. $\qquad \square$

The following proposition employs Lemma 6 to show that (under mild conditions) there exists $\boldsymbol{\theta} \in \mathbb{R}^d$ for which $\lambda_{min}(\nabla^2 f(\boldsymbol{\theta}))$ is arbitrarily negative.

**Proposition 7.** *Assume that: (i) the network is deep ($n \geq 3$); and (ii) the network, loss function $\ell(\cdot)$ and training set $\mathcal{S}$ are non-degenerate, in the sense that there exists a weight setting $\boldsymbol{\theta} \in \mathbb{R}^d$ for which $\sum_{i=1}^{|\mathcal{S}|} \nabla\ell(\mathbf{0}; y_i)^\top h_{\boldsymbol{\theta}}(\mathbf{x}_i) \neq 0$, where $\nabla\ell(\cdot)$ stands for the gradient of $\ell(\cdot)$ with respect to its first*

---

[25] Similarly to our analysis of fully connected architectures (Subsection 4.1), that of convolutional architectures (current appendix) readily extends to the case of different (homogeneous) activation functions at different hidden layers.

[26] Such "piecewise characterization" is holistic when the activation function $\sigma(\cdot)$ is linear, *i.e.* $\sigma(z) = z$ (or more generally, $\alpha = \bar{\alpha}$). Indeed, in this case $f(\cdot)$ is twice continuously differentiable throughout, and we may take $\mathcal{D}_{\boldsymbol{\theta}'} = \mathbb{R}^d$.

*argument, and $h_{\boldsymbol{\theta}}(\cdot)$ is the input-output mapping realized by the network (Equation (28)).*[27] *Then, it holds that* $\inf_{\boldsymbol{\theta}\in\mathbb{R}^d \text{ s.t.} \nabla^2 f(\boldsymbol{\theta}) \text{ exists}} \lambda_{min}(\nabla^2 f(\boldsymbol{\theta})) = -\infty$.

*Proof sketch (for complete proof see Subappendix J.17).* The proof is analogous to that of Proposition 5. Specifically, it establishes that there exists $\boldsymbol{\theta}\in\mathcal{D}_{\boldsymbol{\theta}'}$ for which $\sum_{i=1}^{|\mathcal{S}|}\nabla\ell(\mathbf{0},y_i)^\top h_{\boldsymbol{\theta}}(\mathbf{x}_i) < 0$, and then makes use of Lemma 6 to show that fixing $\Delta\mathbf{w}_1, \Delta\mathbf{w}_2, ..., \Delta\mathbf{w}_n$ to certain values, and positively rescaling $\mathbf{w}_1, \mathbf{w}_2, ..., \mathbf{w}_n$ in a certain way, leads $\nabla^2 f(\boldsymbol{\theta})[\Delta\mathbf{w}_1, \Delta\mathbf{w}_2, ..., \Delta\mathbf{w}_n]$ to become arbitrarily negative. $\qquad\square$

Relying on Lemma 6, Lemma 7 below provides a lower bound on $\lambda_{min}(\nabla^2 f(\boldsymbol{\theta}))$ for $\boldsymbol{\theta}\in\mathcal{D}_{\boldsymbol{\theta}'}$.

**Lemma 7.** *With the notations of Lemma 6, for any $\boldsymbol{\theta}\in\mathcal{D}_{\boldsymbol{\theta}'}$:*[13]

$$\lambda_{min}(\nabla^2 f(\boldsymbol{\theta})) \geq -\max\{|\alpha|, |\bar{\alpha}|\}^{n-1}\frac{n-1}{|\mathcal{S}|}\sum_{i=1}^{|\mathcal{S}|}\|\nabla\ell_i\|_2\|\mathbf{x}_i\|_2 \cdot \tag{31}$$
$$\prod_{j=1}^n\|W_j(\cdot)\|_{op}\max_{\substack{\mathcal{J}\subseteq\{1,2,...,n\}\\|\mathcal{J}|=n-2}}\prod_{j\in\mathcal{J}}\|\mathbf{w}_j\|_2,$$

*where $\|W_j(\cdot)\|_{op}$, $j=1,2,...,n$, denotes the operator norm of $W_j(\cdot)$ induced by the Frobenius norm.*[28]

*Proof sketch (for complete proof see Subappendix J.18).* The proof mirrors those of Lemmas 2 and 5 — it establishes that the right-hand side of Equation (30) in Lemma 6 is lower bounded by $c\sum_{j=1}^n\|\Delta\mathbf{w}_j\|_2^2$, with $c$ being the expression on the right-hand side of Equation (31). $\qquad\square$

The lower bound in Equation (31) is highly sensitive to the scales of the individual weight vectors. Specifically, assuming the network is deep ($n \geq 3$) and is non-degenerate, in the sense that all of its layers can realize non-zero mappings (that is, the activation function $\sigma(\cdot)$ is not identically zero, *i.e.* $\alpha$ and $\bar{\alpha}$ are not both equal to zero, and for all $j\in\{1,2,...,n\}$, $W_j(\cdot)$ is not the zero mapping, *i.e.* $\|W_j(\cdot)\|_{op} > 0$), if $\boldsymbol{\theta}$ does not perfectly fit all non-zero training inputs (meaning there exists $i\in\{1,2,...,|\mathcal{S}|\}$ for which $\nabla\ell_i\neq\mathbf{0}$ and $\mathbf{x}_i\neq\mathbf{0}$), and if at least $n-2$ of its weight vectors $\mathbf{w}_1, \mathbf{w}_2, ..., \mathbf{w}_n$ are non-zero, then it is possible to rescale each $\mathbf{w}_j$ by $c_j > 0$, with $\prod_{j=1}^n c_j = 1$, such that the lower bound in Equation (31) becomes arbitrarily negative[29] despite the input-output mapping $h_{\boldsymbol{\theta}}(\cdot)$ (and thus the training loss value $f(\boldsymbol{\theta})$) remaining unchanged. Nevertheless, as with fully connected architectures (see Subsection 4.1), gradient flow over convolutional architectures (*i.e.* over neural networks as defined in Equation (28) and surrounding text) initialized near zero maintains balance between weight vectors — *cf.* [18] — and so along its trajectories the lower bound in Equation (31) assumes a tighter form. This is formalized in Proposition 8 below.

**Proposition 8.** *If $\boldsymbol{\theta}\in\mathcal{D}_{\boldsymbol{\theta}'}$ resides on a trajectory of gradient flow (over $f(\cdot)$)*[23] *initialized at some point $\boldsymbol{\theta}_s\in\mathbb{R}^d$, with $\|\boldsymbol{\theta}_s\|_2\leq\epsilon$ for some $\epsilon>0$, then, using the notations of Lemmas 6 and 7:*

$$\lambda_{min}(\nabla^2 f(\boldsymbol{\theta})) \geq -\max\{|\alpha|, |\bar{\alpha}|\}^{n-1}\frac{n-1}{|\mathcal{S}|}\sum_{i=1}^{|\mathcal{S}|}\|\nabla\ell_i\|_2\|\mathbf{x}_i\|_2 \cdot \tag{32}$$
$$\prod_{j=1}^n\|W_j(\cdot)\|_{op}\Big(\min_{j\in\{1,2,...,n\}}\|\mathbf{w}_j\|_2 + \epsilon\Big)^{n-2}.$$

*Proof sketch (for complete proof see Subappendix J.19).* By the analysis of [18], for any $j, j' \in \{1, 2, ..., n\}$, the quantity $\|\mathbf{w}_{j'}\|_2^2 - \|\mathbf{w}_j\|_2^2$ is invariant (constant) along a gradient flow trajectory. This implies that along a trajectory emanating from a point with (Euclidean) norm $\mathcal{O}(\epsilon)$, it holds that $\|\mathbf{w}_{j'}\|_2^2 - \|\mathbf{w}_j\|_2^2 \in \mathcal{O}(\epsilon^2)$ for all $j, j' \in \{1, 2, ..., n\}$, which in turn implies

---

[27] Assumptions *(i)* and *(ii)* are both necessary, in the sense that removing any of them (without imposing further assumptions) renders the proposition false — see Claim 3 in Appendix F. Assumption *(ii)* in particular is extremely mild, *e.g.* if $\ell(\cdot)$ is the square loss (*i.e.* $\mathcal{Y}=\mathbb{R}^{d_n}$ and $\ell(\hat{\mathbf{y}},\mathbf{y})=\frac{1}{2}\|\hat{\mathbf{y}}-\mathbf{y}\|_2^2$), the slightest change in a single label ($\mathbf{y}_i$) corresponding to a non-zero prediction ($h_{\boldsymbol{\theta}}(\mathbf{x}_i)\neq\mathbf{0}$) can ensure the inequality.

[28] From the structure of $W_j(\cdot)$ (see beginning of this appendix) it follows that $\|W_j(\cdot)\|_{op}$ is equal to square root of the maximal number of elements in $W_j(\mathbf{w}_j)$ connected to the same coordinate of $\mathbf{w}_j$.

[29] The bound remains applicable since $\mathcal{D}_{\boldsymbol{\theta}'}$ is closed under positive rescaling of weight vectors.

$\|\mathbf{w}_{j'}\|_2 \leq \min_{j \in \{1,2,...,n\}} \|\mathbf{w}_j\|_2 + \mathcal{O}(\epsilon)$ for all $j' \in \{1,2,...,n\}$. Plugging this into Equation (31) yields the desired result (Equation (32)). $\qquad\square$

Assume the network is deep ($n \geq 3$) and non-degenerate ($\alpha$ and $\bar{\alpha}$ are not both equal to zero, and $\|W_j(\cdot)\|_{op} > 0$ for all $j \in \{1,2,...,n\}$), and consider a trajectory of gradient flow (over $f(\cdot)$) emanating from near-zero initialization. For every point on the trajectory, Proposition 8 may be applied with small $\epsilon$, leading the lower bound in Equation (32) to depend primarily on the *minimal* size (Euclidean norm) of a weight vector $\mathbf{w}_j$, and on $\nabla\ell_1, \nabla\ell_2, ..., \nabla\ell_{|\mathcal{S}|}$ — gradients of the loss function with respect to the predictions over the training set. In the course of optimization, $\mathbf{w}_1, \mathbf{w}_2, ..., \mathbf{w}_n$ are initially small, and if a perfect fit of the training set is ultimately achieved, $\nabla\ell_1, \nabla\ell_2, ..., \nabla\ell_{|\mathcal{S}|}$ will converge to zero. Therefore, if not *all* weight vectors $\mathbf{w}_1, \mathbf{w}_2, ..., \mathbf{w}_n$ become large during optimization, the lower bound on $\lambda_{min}(\nabla^2 f(\boldsymbol{\theta}))$ in Equation (32) will only be moderately negative before approaching non-negativity (if and) as the trajectory converges to a perfect fit. In light of Section 3, this suggests that the gradient flow trajectory may lend itself to approximation by gradient descent. For a case of fully connected neural networks with linear activation (analyzed in Subsubsection 4.1.1), such prospect is theoretically verified in Section 5. For convolutional architectures (subject of the current appendix) we provide empirical corroboration in Subappendix I.1, deferring to future work a complete theoretical affirmation.

## E    Regions of Differentiability

In this appendix we prove that for fully connected and convolutional architectures with non-linear activation, there exist regions of differentiability $\mathcal{D}_{\boldsymbol{\theta}'}$ as described in Appendixes C and D respectively.

**Proposition 9** (regions of differentiability for fully connected architectures)**.** *Consider a fully connected neural network as defined in Equation* (8) *(and surrounding text), and assume that its (homogeneous) activation function is non-linear, i.e.* $\sigma(z) = \alpha\max\{z,0\} - \bar{\alpha}\max\{-z,0\}$ *for some* $\alpha, \bar{\alpha} \in \mathbb{R}$, $\alpha \neq \bar{\alpha}$. *Then, for almost every (in the sense of Lebesgue measure)* $\boldsymbol{\theta}' \in \mathbb{R}^d$, *there exist diagonal matrices* $D'_{i,j} \in \mathbb{R}^{d_j,d_j}$, $i = 1,2,...,|\mathcal{S}|$, $j = 1,2,...,n-1$, *with diagonal elements in* $\{\alpha, \bar{\alpha}\}$, *such that the training loss* $f(\cdot)$ *(Equation* (9)*) coincides with the function defined in Equation* (24) *on an open region* $\mathcal{D}_{\boldsymbol{\theta}'} \subseteq \mathbb{R}^d$ *containing* $\boldsymbol{\theta}'$, *that is closed under positive rescaling of weight matrices (i.e. under* $(W_1, W_2, ..., W_n) \mapsto (c_1 W_1, c_2 W_2, ..., c_n W_n)$ *with* $c_1, c_2, ..., c_n > 0$*).*

*Proof.* If for $\boldsymbol{\theta}' \in \mathbb{R}^d$ there exist diagonal matrices $(D'_{i,j})_{i,j}$ and an open region $\mathcal{D}_{\boldsymbol{\theta}'}$ as above, then we refer to $\boldsymbol{\theta}'$ as an *admissible* weight setting, to $(D'_{i,j})_{i,j}$ as its *activation matrices*, and to $\mathcal{D}_{\boldsymbol{\theta}'}$ as its *differentiability region*.[30]

Without loss of generality, we may assume $|\mathcal{S}| = 1$, *i.e.* that the training set comprises a single labeled input $(\mathbf{x}, y) \in \mathbb{R}^{d_0} \times \mathcal{Y}$, meaning the training loss takes the form $f(\boldsymbol{\theta}) = \ell(h_{\boldsymbol{\theta}}(\mathbf{x}), y)$. To see this, assume the sought-after result holds for a single labeled input, and suppose $|\mathcal{S}| > 1$. We may then apply the result separately for each labeled input $(\mathbf{x}_i, y_i)$, $i = 1,2,...,|\mathcal{S}|$, and obtain, for every admissible $\boldsymbol{\theta}' \in \mathbb{R}^d$, activation matrices $(D'^{(\mathbf{x}_i,y_i)}_j)_{j=1}^{n-1}$ and a differentiability region $\mathcal{D}^{(\mathbf{x}_i,y_i)}_{\boldsymbol{\theta}'}$. Since the weight settings not admissible for a certain labeled input $(\mathbf{x}_i, y_i)$ form a set of zero (Lebesgue) measure, those not admissible for any of the $|\mathcal{S}|$ labeled inputs also constitute a zero measure set. That is, almost every $\boldsymbol{\theta}' \in \mathbb{R}^d$ is jointly admissible for all $\left((\mathbf{x}_i,y_i)\right)_{i=1}^{|\mathcal{S}|}$. Given such $\boldsymbol{\theta}'$, consider the activation matrices and differentiability regions obtained for the different labeled inputs — $(D'^{(\mathbf{x}_i,y_i)}_j)_{j=1}^{n-1}$ and $\mathcal{D}^{(\mathbf{x}_i,y_i)}_{\boldsymbol{\theta}'}$, $i = 1,2,...,|\mathcal{S}|$. Defining $D'_{i,j} := D'^{(\mathbf{x}_i,y_i)}_j$, $i = 1,2,...,|\mathcal{S}|$, $j = 1,2,...,n-1$, and $\mathcal{D}_{\boldsymbol{\theta}'} := \cap_{i=1}^{|\mathcal{S}|} \mathcal{D}^{(\mathbf{x}_i,y_i)}_{\boldsymbol{\theta}'}$, we have that $\boldsymbol{\theta}'$ is admissible for $\mathcal{S}$, with activation matrices $(D'_{i,j})_{i,j}$ and differentiability region $\mathcal{D}_{\boldsymbol{\theta}'}$. The sought-after result thus holds for $\mathcal{S}$.

In light of the above, we assume hereafter that $\mathcal{S} = \left((\mathbf{x},y)\right)$. Recursively define the functions $\mathbf{f}^{(j)} : \mathbb{R}^d \to \mathbb{R}^{d_j}$, $j = 0,1,...,n-1$:

$$\mathbf{f}^{(0)}(\boldsymbol{\theta}) \equiv \mathbf{x} \quad , \quad \mathbf{f}^{(j)}(\boldsymbol{\theta}) = \sigma\left(W_j \mathbf{f}^{(j-1)}(\boldsymbol{\theta})\right) \text{ for } j = 1,2,...,n-1 \,.$$

---

[30]Note that given an admissible weight setting, activation matrices and differentiability region are not necessarily determined uniquely.

We will prove by induction that given $j' \in \{0,1,...,n-1\}$, for almost every $\boldsymbol{\theta}' \in \mathbb{R}^d$, there exist diagonal matrices $D'_j \in \mathbb{R}^{d_j,d_j}$, $j=1,2,...,j'$, with diagonal elements in $\{\alpha,\bar{\alpha}\}$, such that $\mathbf{f}^{(j')}(\cdot)$ meets the following conditions on an open region $\mathcal{D}_{\boldsymbol{\theta}'} \subseteq \mathbb{R}^d$ containing $\boldsymbol{\theta}'$, that is closed under positive rescaling of weight matrices:

*(i)* $\mathbf{f}^{(j')}(\cdot)$ coincides with the function $\boldsymbol{\theta} \mapsto D'_{j'} W_{j'} D'_{j'-1} W_{j'-1} \cdots D'_1 W_1 \mathbf{x}$; and

*(ii)* each entry of $\mathbf{f}^{(j')}(\cdot)$ is either nowhere zero or identically zero.

Continuing the terminology defined earlier, in the context of $\mathbf{f}^{(j')}(\cdot)$, $j'=0,1,...,n-1$, we refer to $\boldsymbol{\theta}'$, $(D'_j)_j$ and $\mathcal{D}_{\boldsymbol{\theta}'}$ satisfying the above as *admissible*, *activation matrices* and *differentiability region*, respectively. Note that the training loss $f(\cdot)$ can be expressed as $f(\boldsymbol{\theta}) = \ell(W_n \mathbf{f}^{(n-1)}(\boldsymbol{\theta}),y)$, and therefore proving the inductive hypothesis for $j'=n-1$ yields the desired result. The base case for the induction ($j'=0$) is trivial, so all that remains is to establish the induction step.

Given $j' \in \{1,2,...,n-1\}$, assume that the inductive hypothesis holds for $j'-1$, and in the context of $\mathbf{f}^{(j'-1)}(\cdot)$, let $\boldsymbol{\theta}'$ be an admissible weight setting, with corresponding activation matrices $(D'_j)_{j=1}^{j'-1}$ and differentiability region $\mathcal{D}_{\boldsymbol{\theta}'}$. We refer to $\boldsymbol{\theta}'$ as *nullifying* if $\mathbf{f}^{(j'-1)}(\boldsymbol{\theta}') = \mathbf{0}$, which implies $\mathbf{f}^{(j'-1)}(\boldsymbol{\theta}) = \mathbf{0}$ for all $\boldsymbol{\theta} \in \mathcal{D}_{\boldsymbol{\theta}'}$. In this case $\boldsymbol{\theta}'$ is clearly admissible in the context of $\mathbf{f}^{(j')}(\cdot)$ (as activation matrices we may take $(D'_j)_{j=1}^{j'-1}$ along with any diagonal matrix $D'_{j'} \in \mathbb{R}^{d_{j'},d_{j'}}$ whose diagonal elements are in $\{\alpha,\bar{\alpha}\}$, and as differentiability region we can simply use $\mathcal{D}_{\boldsymbol{\theta}'}$). Consider now the case where $\boldsymbol{\theta}'$ is non-nullifying, *i.e.* where $\mathbf{f}^{(j'-1)}(\boldsymbol{\theta}') \neq \mathbf{0}$. We refer to $\boldsymbol{\theta}'$ as *regular* if all entries of $W'_{j'} \mathbf{f}^{(j'-1)}(\boldsymbol{\theta}')$ are non-zero, with $W'_{j'} \in \mathbb{R}^{d_{j'},d_{j'-1}}$ denoting the value of weight matrix $j'$ held in $\boldsymbol{\theta}'$. If $\boldsymbol{\theta}'$ is regular then it is admissible in the context of $\mathbf{f}^{(j')}(\cdot)$. To see this, note that a valid choice of activation matrices is $(D'_j)_{j=1}^{j'-1}$ along with the diagonal matrix $D'_{j'} \in \mathbb{R}^{d_{j'},d_{j'}}$ whose diagonal elements corresponding to positive entries of $W'_{j'} \mathbf{f}^{(j'-1)}(\boldsymbol{\theta}')$ hold $\alpha$, and those corresponding to negative entries hold $\bar{\alpha}$. From continuity, and homogeneity with slopes $\alpha$ and $\bar{\alpha}$ of the activation function $\sigma(\cdot)$, there exists an open neighborhood of $\boldsymbol{\theta}'$ (subset of $\mathcal{D}_{\boldsymbol{\theta}'}$) on which conditions *(i)* and *(ii)* hold. Extending this neighborhood to include, for each of its weight settings $\boldsymbol{\theta}$, all positive rescalings of weight matrices $W_1,W_2,...,W_n$, yields a valid differentiability region for $\boldsymbol{\theta}'$ in the context of $\mathbf{f}^{(j')}(\cdot)$, thereby confirming admissibility.

We conclude the proof by showing that almost every $\boldsymbol{\theta}' \in \mathbb{R}^d$ is admissible in the context of $\mathbf{f}^{(j')}(\cdot)$. Per the above, if $\boldsymbol{\theta}' \in \mathbb{R}^d$ does not meet this condition then it must either be inadmissible in the context of $\mathbf{f}^{(j'-1)}(\cdot)$, or be non-nullifying and irregular. By our inductive hypothesis, weight settings inadmissible in the context of $\mathbf{f}^{(j'-1)}(\cdot)$ form a set of measure zero, so it suffices to show that the collection of non-nullifying and irregular weight settings, denoted $\mathcal{C}$, is also of measure zero. Note that whether a weight setting $\boldsymbol{\theta}$ is nullifying (*i.e.* $\mathbf{f}^{(j'-1)}(\boldsymbol{\theta}) = \mathbf{0}$) or not depends only on the weight matrices $W_1,W_2,...,W_{j'-1}$, and given these matrices, whether it is regular (*i.e.* all entries of $W'_{j'} \mathbf{f}^{(j'-1)}(\boldsymbol{\theta}')$ are non-zero) or not depends only on $W_{j'}$. We may thus apply Fubini's Theorem (*cf.* [48]), and compute the measure of $\mathcal{C}$ by integrating over non-nullifying configurations of $W_1,W_2,...,W_{j'-1}$, where for each, the measure of values for $W_{j'},W_{j'+1},...,W_n$ leading to irregularity is integrated. The latter measure is zero, since for any $\mathbf{0} \neq \mathbf{q} \in \mathbb{R}^{d_{j'-1}}$, the set $\{W \in \mathbb{R}^{d_{j'},d_{j'-1}} : \text{there exists a coordinate of } W\mathbf{q} \text{ equal to zero}\}$ has measure zero, thus its Cartesian product with $\mathbb{R}^{d_{j'+1},d_{j'}} \times \mathbb{R}^{d_{j'+2},d_{j'+1}} \times \cdots \times \mathbb{R}^{d_n,d_{n-1}}$ is also of measure zero. This implies that $\mathcal{C}$ has measure zero, thereby completing the proof. $\square$

**Proposition 10** (regions of differentiability for convolutional architectures). *Consider a neural network with weight sharing and sparsity as defined in Equation* (28) *(and surrounding text), and assume that its (homogeneous) activation function is non-linear, i.e.* $\sigma(z) = \alpha\max\{z,0\} - \bar{\alpha}\max\{-z,0\}$ *for some* $\alpha,\bar{\alpha} \in \mathbb{R}$, $\alpha \neq \bar{\alpha}$. *Then, for almost every (in the sense of Lebesgue measure)* $\boldsymbol{\theta}' \in \mathbb{R}^d$, *there exist diagonal matrices* $D'_{i,j} \in \mathbb{R}^{d_j,d_j}$, $i=1,2,...,|\mathcal{S}|$, $j=1,2,...,n-1$, *with diagonal elements in* $\{\alpha,\bar{\alpha}\}$, *such that the training loss* $f(\cdot)$ *(Equation* (9)*) coincides with the function defined in Equation* (29) *on an open region* $\mathcal{D}_{\boldsymbol{\theta}'} \subseteq \mathbb{R}^d$ *containing* $\boldsymbol{\theta}'$, *that is closed under positive rescaling of weight vectors (i.e. under* $(\mathbf{w}_1,\mathbf{w}_2,...,\mathbf{w}_n) \mapsto (c_1\mathbf{w}_1,c_2\mathbf{w}_2,...,c_n\mathbf{w}_n)$ *with* $c_1,c_2,...,c_n > 0$*).*

*Proof.* The proof begins similarly to that of Proposition 9, and then takes a slightly different (more involved) route. We provide a self-contained presentation, repeating details from the proof of Proposition 9 as needed.

If for $\boldsymbol{\theta}' \in \mathbb{R}^d$ there exist diagonal matrices $(D'_{i,j})_{i,j}$ and an open region $\mathcal{D}_{\boldsymbol{\theta}'}$ as in proposition statement, then we refer to $\boldsymbol{\theta}'$ as an *admissible* weight setting, to $(D'_{i,j})_{i,j}$ as its *activation matrices*, and to $\mathcal{D}_{\boldsymbol{\theta}'}$ as its *differentiability region*.[30]

Without loss of generality, we may assume $|\mathcal{S}| = 1$, *i.e.* that the training set comprises a single labeled input $(\mathbf{x}, y) \in \mathbb{R}^{d_0} \times \mathcal{Y}$, meaning the training loss takes the form $f(\boldsymbol{\theta}) = \ell(h_{\boldsymbol{\theta}}(\mathbf{x}), y)$. To see this, assume the sought-after result holds for a single labeled input, and suppose $|\mathcal{S}| > 1$. We may then apply the result separately for each labeled input $(\mathbf{x}_i, y_i)$, $i = 1, 2, ..., |\mathcal{S}|$, and obtain, for every admissible $\boldsymbol{\theta}' \in \mathbb{R}^d$, activation matrices $(D'^{(\mathbf{x}_i, y_i)}_j)^{n-1}_{j=1}$ and a differentiability region $\mathcal{D}^{(\mathbf{x}_i, y_i)}_{\boldsymbol{\theta}'}$. Since the weight settings not admissible for a certain labeled input $(\mathbf{x}_i, y_i)$ form a set of zero (Lebesgue) measure, those not admissible for any of the $|\mathcal{S}|$ labeled inputs also constitute a zero measure set. That is, almost every $\boldsymbol{\theta}' \in \mathbb{R}^d$ is jointly admissible for all $((\mathbf{x}_i, y_i))^{|\mathcal{S}|}_{i=1}$. Given such $\boldsymbol{\theta}'$, consider the activation matrices and differentiability regions obtained for the different labeled inputs — $(D'^{(\mathbf{x}_i, y_i)}_j)^{n-1}_{j=1}$ and $\mathcal{D}^{(\mathbf{x}_i, y_i)}_{\boldsymbol{\theta}'}$, $i = 1, 2, ..., |\mathcal{S}|$. Defining $D'_{i,j} := D'^{(\mathbf{x}_i, y_i)}_j$, $i = 1, 2, ..., |\mathcal{S}|$, $j = 1, 2, ..., n-1$, and $\mathcal{D}_{\boldsymbol{\theta}'} := \cap^{|\mathcal{S}|}_{i=1} \mathcal{D}^{(\mathbf{x}_i, y_i)}_{\boldsymbol{\theta}'}$, we have that $\boldsymbol{\theta}'$ is admissible for $\mathcal{S}$, with activation matrices $(D'_{i,j})_{i,j}$ and differentiability region $\mathcal{D}_{\boldsymbol{\theta}'}$. The sought-after result thus holds for $\mathcal{S}$.

In light of the above, we assume hereafter that $\mathcal{S} = ((\mathbf{x}, y))$. Recursively define the functions $\mathbf{f}^{(j)} : \mathbb{R}^d \to \mathbb{R}^{d_j}$, $j = 0, 1, ..., n-1$:

$$\mathbf{f}^{(0)}(\boldsymbol{\theta}) \equiv \mathbf{x} \quad , \quad \mathbf{f}^{(j)}(\boldsymbol{\theta}) = \sigma(W_j(\mathbf{w}_j)\mathbf{f}^{(j-1)}(\boldsymbol{\theta})) \text{ for } j = 1, 2, ..., n-1 .$$

We will prove by induction that given $j' \in \{0, 1, ..., n-1\}$, for almost every $\boldsymbol{\theta}' \in \mathbb{R}^d$, there exist diagonal matrices $D'_j \in \mathbb{R}^{d_j, d_j}$, $j = 1, 2, ..., j'$, with diagonal elements in $\{\alpha, \bar{\alpha}\}$, such that $\mathbf{f}^{(j')}(\cdot)$ meets the following conditions on an open region $\mathcal{D}_{\boldsymbol{\theta}'} \subseteq \mathbb{R}^d$ containing $\boldsymbol{\theta}'$, that is closed under positive rescaling of weight vectors:

*(i)* $\mathbf{f}^{(j')}(\cdot)$ coincides with the function $\boldsymbol{\theta} \mapsto D'_{j'} W_{j'}(\mathbf{w}_{j'}) D'_{j'-1} W_{j'-1}(\mathbf{w}_{j'-1}) \cdots D'_1 W_1(\mathbf{w}_1)\mathbf{x}$; and

*(ii)* each entry of $\mathbf{f}^{(j')}(\cdot)$ is either nowhere zero or identically zero.

Continuing the terminology defined earlier, in the context of $\mathbf{f}^{(j')}(\cdot)$, $j' = 0, 1, ..., n-1$, we refer to $\boldsymbol{\theta}'$, $(D'_j)_j$ and $\mathcal{D}_{\boldsymbol{\theta}'}$ satisfying the above as *admissible*, *activation matrices* and *differentiability region*, respectively. Note that the training loss $f(\cdot)$ can be expressed as $f(\boldsymbol{\theta}) = \ell(W_n(\mathbf{w}_n)\mathbf{f}^{(n-1)}(\boldsymbol{\theta}), y)$, and therefore proving the inductive hypothesis for $j' = n-1$ yields the desired result. The base case for the induction ($j' = 0$) is trivial, so all that remains is to establish the induction step.

Given $j' \in \{1, 2, ..., n-1\}$, assume that the inductive hypothesis holds for $j'-1$, and in the context of $\mathbf{f}^{(j'-1)}(\cdot)$, let $\boldsymbol{\theta}'$ be an admissible weight setting, with corresponding activation matrices $(D'_j)^{j'-1}_{j=1}$ and differentiability region $\mathcal{D}_{\boldsymbol{\theta}'}$. We define the *nullity pattern* of $\boldsymbol{\theta}'$ to be the vector $\mathbf{e} \in \mathbb{R}^{d_{j'-1}}$ holding zero in the coordinates where $\mathbf{f}^{(j'-1)}(\boldsymbol{\theta}')$ holds zero, and one elsewhere (that is, $\mathbf{e}$ is the vector obtained by setting to one all non-zero entries of $\mathbf{f}^{(j'-1)}(\boldsymbol{\theta}')$). With $\mathbf{1} \in \mathbb{R}^{d_{j'}}$ standing for an all-ones vector, we refer to the coordinates of $\mathbb{R}^{d_{j'}}$ where $W_{j'}(\mathbf{1})\mathbf{e}$ holds zero as *infeasible*, and to the rest as *feasible*. Note that a coordinate of $\mathbb{R}^{d_{j'}}$ is infeasible if and only if $W_{j'}(\mathbf{q})\mathbf{f}^{(j'-1)}(\boldsymbol{\theta}')$ holds zero in that coordinate for all $\mathbf{q} \in \mathbb{R}^{d_{j'}}$. We shall say that $\boldsymbol{\theta}'$ is *regular* if $W_{j'}(\mathbf{w}'_{j'})\mathbf{f}^{(j'-1)}(\boldsymbol{\theta}')$ is non-zero in all feasible coordinates, where $\mathbf{w}'_{j'} \in \mathbb{R}^{d_{j'}}$ denotes the value of weight vector $j'$ in $\boldsymbol{\theta}'$. Hereafter we show that regularity of $\boldsymbol{\theta}'$ implies that it is admissible in the context of $\mathbf{f}^{(j')}(\cdot)$. By admissibility in the context of $\mathbf{f}^{(j'-1)}(\cdot)$ we have that across $\mathcal{D}_{\boldsymbol{\theta}'}$, each entry of $\mathbf{f}^{(j'-1)}(\cdot)$ is either nowhere zero or identically zero. This implies the nullity pattern is constant across $\mathcal{D}_{\boldsymbol{\theta}'}$, which in turn means the same for the set of infeasible coordinates. The coordinates where $W_{j'}(\mathbf{w}'_{j'})\mathbf{f}^{(j'-1)}(\boldsymbol{\theta}')$ holds zero thus vanish in $W_{j'}(\mathbf{w}_{j'})\mathbf{f}^{(j'-1)}(\boldsymbol{\theta})$ for all $\boldsymbol{\theta} \in \mathcal{D}_{\boldsymbol{\theta}'}$. From continuity, and the fact that around any $z \neq 0$, the activation function $\sigma(\cdot)$ is either nowhere zero or identically zero,[31] it follows that there exists an open neighborhood $\mathcal{N} \subseteq \mathcal{D}_{\boldsymbol{\theta}'}$ of $\boldsymbol{\theta}'$ on which condition *(ii)* holds. Let $D'_{j'} \in \mathbb{R}^{d_{j'}, d_{j'}}$ be a diagonal matrix whose diagonal elements corresponding to positive entries in $W_{j'}(\mathbf{w}'_{j'})\mathbf{f}^{(j'-1)}(\boldsymbol{\theta}')$

---

[31]The latter is possible only if $\alpha = 0$ or $\bar{\alpha} = 0$.

hold $\alpha$, those corresponding to negative entries hold $\bar{\alpha}$, and the rest hold either $\alpha$ or $\bar{\alpha}$. Since $\mathbf{f}^{(j'-1)}(\cdot)$ coincides with the function $\boldsymbol{\theta} \mapsto D'_{j'-1} W_{j'-1}(\mathbf{w}_{j'-1}) D'_{j'-2} W_{j'-2}(\mathbf{w}_{j'-2}) \cdots D'_1 W_1(\mathbf{w}_1)\mathbf{x}$ on $\mathcal{D}_{\boldsymbol{\theta}'}$, and since $\sigma(\cdot)$ is homogeneous with slopes $\alpha$ and $\bar{\alpha}$, condition *(i)* holds across $\mathcal{N}$. Consider the extension of $\mathcal{N}$ comprising, for each of its weight settings, all positive rescalings of weight vectors. Along with $(D'_j)_{j=1}^{j'}$ as activation matrices, this extension serves as a valid differentiability region for $\boldsymbol{\theta}'$ in the context of $\mathbf{f}^{(j')}(\cdot)$. The sought-after admissibility is thus established.

We conclude the proof by showing that almost every $\boldsymbol{\theta}' \in \mathbb{R}^d$ is admissible in the context of $\mathbf{f}^{(j')}(\cdot)$. Per the above, if $\boldsymbol{\theta}' \in \mathbb{R}^d$ does not meet this condition then either it is inadmissible in the context of $\mathbf{f}^{(j'-1)}(\cdot)$, or it is irregular. By our inductive hypothesis, weight settings inadmissible in the context of $\mathbf{f}^{(j'-1)}(\cdot)$ form a set of measure zero, so it suffices to show that the collection of irregular weight settings, denoted $\mathcal{C}$, is also of measure zero. We first establish that $\mathcal{C}$ is measurable. Let $\mathbf{e} \in \mathbb{R}^{d_{j'-1}}$ be an arbitrary nullity pattern (vector with entries in $\{0,1\}$), and consider the feasible coordinates it induces. The following two sets are measurable: weight settings with nullity pattern $\mathbf{e}$; and weight settings $\boldsymbol{\theta}$ for which $W_{j'}(\mathbf{w}_{j'})\mathbf{f}^{(j'-1)}(\boldsymbol{\theta})$ holds zero in at least one of the feasible coordinates induced by $\mathbf{e}$. The collection of irregular weight settings with nullity pattern $\mathbf{e}$, denoted $\mathcal{C}_{\mathbf{e}}$, is equal to the intersection of these two sets, and therefore is measurable. Taking union of $\mathcal{C}_{\mathbf{e}}$ with $\mathbf{e}$ ranging over all (finitely many) possible nullity patterns yields $\mathcal{C}$, from which it follows that the latter is indeed measurable. Given weight vectors $\mathbf{w}_1, \mathbf{w}_2, ..., \mathbf{w}_{j'-1}$, whether or not a weight setting $\boldsymbol{\theta}$ is regular depends only on $\mathbf{w}_{j'}$. We may thus apply Fubini's Theorem (*cf.* [48]), and compute the measure of $\mathcal{C}$ by integrating over configurations of $\mathbf{w}_1, \mathbf{w}_2, ..., \mathbf{w}_{j'-1}$, where for each, the measure of values for $\mathbf{w}_{j'}, \mathbf{w}_{j'+1}, ..., \mathbf{w}_n$ leading to irregularity is integrated. We now establish that the latter measure is zero, which in turn implies that $\mathcal{C}$ has measure zero (thereby completing the proof). Since the Cartesian product of a zero measure subset of $\mathbb{R}^{d_{j'}}$ with $\mathbb{R}^{d_{j'+1}} \times \mathbb{R}^{d_{j'+2}} \times \cdots \times \mathbb{R}^{d_n}$ has zero measure, it suffices to show that given any configuration of $\mathbf{w}_1, \mathbf{w}_2, ..., \mathbf{w}_{j'-1}$, the measure of values for $\mathbf{w}_{j'}$ leading to irregularity is zero. $\mathbf{w}_1, \mathbf{w}_2, ..., \mathbf{w}_{j'-1}$ fully determine $\mathbf{f}^{(j'-1)}(\boldsymbol{\theta})$, and as a consequence, the nullity pattern of $\boldsymbol{\theta}$. Consider the feasible coordinates induced by this nullity pattern. On each of these, the linear function $\mathbf{w}_{j'} \mapsto W_{j'}(\mathbf{w}_{j'})\mathbf{f}^{(j'-1)}(\boldsymbol{\theta})$ is not identically zero. The measure of values for $\mathbf{w}_{j'}$ leading $W_{j'}(\mathbf{w}_{j'})\mathbf{f}^{(j'-1)}(\boldsymbol{\theta})$ to vanish in a feasible coordinate, *i.e.* leading $\boldsymbol{\theta}$ to be irregular, is thus zero. This completes the proof. $\square$

## F   Necessity of Assumptions in Propositions 1, 5 and 7

In this appendix we prove that the assumptions in Propositions 1, 5 and 7 are necessary, in the sense that each of the latter becomes false if any of its assumptions are removed (and no further assumptions are imposed).

**Claim 1** (necessity of assumptions in Proposition 1). *In the context of Proposition 1, if the network is shallow ($n = 2$) or the zero mapping is a global minimizer of the training loss (meaning $\nabla\phi(0) = 0$), then the stated result may not hold, i.e. it may be that $\inf_{\boldsymbol{\theta} \in \mathbb{R}^d} \lambda_{min}(\nabla^2 f(\boldsymbol{\theta})) > -\infty$.*

*Proof.* Suppose the network is shallow ($n = 2$). With the notations of Lemma 1, for any $\boldsymbol{\theta} \in \mathbb{R}^d$, $(\Delta W_1, \Delta W_2) \in \mathbb{R}^{d_1, d_0} \times \mathbb{R}^{d_2, d_1}$:

$$\nabla^2 f(\boldsymbol{\theta})[\Delta W_1, \Delta W_2] = \nabla^2\phi(W_{2:1})[W_2(\Delta W_1) + (\Delta W_2)W_1] + 2\mathrm{Tr}\big(\nabla\phi(W_{2:1})^\top(\Delta W_2)(\Delta W_1)\big)$$
$$\geq 2\mathrm{Tr}\big(\nabla\phi(W_{2:1})^\top(\Delta W_2)(\Delta W_1)\big)$$
$$\geq -2\|\nabla\phi(W_{2:1})\|_{Frobenius}\|(\Delta W_2)(\Delta W_1)\|_{Frobenius}$$
$$\geq -2\|\nabla\phi(W_{2:1})\|_{Frobenius}\|\Delta W_2\|_{Frobenius}\|\Delta W_1\|_{Frobenius}$$
$$\geq -\|\nabla\phi(W_{2:1})\|_{Frobenius}\big(\|\Delta W_2\|_{Frobenius}^2 + \|\Delta W_1\|_{Frobenius}^2\big)$$
$$= -\|\nabla\phi(W_{2:1})\|_{Frobenius}\|(\Delta W_1, \Delta W_2)\|_{Frobenius}^2,$$

where the first transition follows from Lemma 1, the second holds since $\phi(\cdot)$ is convex, the third is an application of the Cauchy-Schwarz inequality, the fourth follows from submultiplicativity of the Frobenius norm, and the latter two are based on simple arithmetics. It follows from the above that $\lambda_{min}(\nabla^2 f(\boldsymbol{\theta})) \geq -\|\nabla\phi(W_{2:1})\|_{Frobenius}$. Therefore if $\nabla\phi(\cdot)$ is bounded (*e.g.* if $\ell(\cdot)$ is the logistic loss — see Equation (11)) we will have $\inf_{\boldsymbol{\theta} \in \mathbb{R}^d} \lambda_{min}(\nabla^2 f(\boldsymbol{\theta})) > -\infty$, as required.

It remains to show that if the zero mapping is a global minimizer of the training loss (meaning $\nabla\phi(0) = 0$), then, regardless of network depth (*i.e.* with either $n \geq 3$ or $n = 2$), it may be that

$\inf_{\boldsymbol{\theta}\in\mathbb{R}^d}\lambda_{min}(\nabla^2 f(\boldsymbol{\theta})) > -\infty$. This is trivial — simply consider the case where the training set $\mathcal{S}$ is such that $\mathbf{x}_i = \mathbf{0}$ for all $i = 1, 2, ..., |\mathcal{S}|$. The training loss in this case is constant (see Equations (8) and (9)), implying $\inf_{\boldsymbol{\theta}\in\mathbb{R}^d}\lambda_{min}(\nabla^2 f(\boldsymbol{\theta})) = 0$. $\qquad\square$

**Claim 2** (necessity of assumptions in Proposition 5). *In the context of Proposition 5, if assumptions* (i) *or* (ii) *are not satisfied, then the stated result may not hold, i.e. it may be that* $\inf_{\boldsymbol{\theta}\in\mathbb{R}^d \text{ s.t.} \nabla^2 f(\boldsymbol{\theta}) \text{ exists}}\lambda_{min}(\nabla^2 f(\boldsymbol{\theta})) > -\infty$.

*Proof.* Suppose that assumption *(i)* is not satisfied, *i.e.* that the network is shallow ($n = 2$). With the notations of Lemma 4, for any $\boldsymbol{\theta}\in\mathcal{D}_{\boldsymbol{\theta}'}$, $(\Delta W_1, \Delta W_2)\in\mathbb{R}^{d_1,d_0}\times\mathbb{R}^{d_2,d_1}$:

$$\nabla^2 f(\boldsymbol{\theta})[\Delta W_1, \Delta W_2] = \frac{1}{|\mathcal{S}|}\sum_{i=1}^{|\mathcal{S}|}\nabla^2\ell_i\big[W_2 D'_{i,1}(\Delta W_1)\mathbf{x}_i + (\Delta W_2)D'_{i,1}W_1\mathbf{x}_i\big]$$
$$+ \frac{2}{|\mathcal{S}|}\sum_{i=1}^{|\mathcal{S}|}\nabla\ell_i^\top(\Delta W_2)D'_{i,1}(\Delta W_1)\mathbf{x}_i$$
$$\geq \frac{2}{|\mathcal{S}|}\sum_{i=1}^{|\mathcal{S}|}\nabla\ell_i^\top(\Delta W_2)D'_{i,1}(\Delta W_1)\mathbf{x}_i$$
$$\geq -\frac{2}{|\mathcal{S}|}\sum_{i=1}^{|\mathcal{S}|}\|\nabla\ell_i\|_2\|(\Delta W_2)D'_{i,1}(\Delta W_1)\mathbf{x}_i\|_2$$
$$\geq -\frac{2}{|\mathcal{S}|}\sum_{i=1}^{|\mathcal{S}|}\|\nabla\ell_i\|_2\|\mathbf{x}_i\|_2\|(\Delta W_2)D'_{i,1}(\Delta W_1)\|_{spectral}$$
$$\geq -\frac{2}{|\mathcal{S}|}\sum_{i=1}^{|\mathcal{S}|}\|\nabla\ell_i\|_2\|\mathbf{x}_i\|_2\|\Delta W_2\|_{spectral}\|D'_{i,1}\|_{spectral}\|\Delta W_1\|_{spectral}$$
$$\geq -\max\{|\alpha|,|\bar{\alpha}|\}\frac{2}{|\mathcal{S}|}\sum_{i=1}^{|\mathcal{S}|}\|\nabla\ell_i\|_2\|\mathbf{x}_i\|_2\|\Delta W_2\|_{spectral}\|\Delta W_1\|_{spectral}$$
$$\geq -\max\{|\alpha|,|\bar{\alpha}|\}\frac{2}{|\mathcal{S}|}\sum_{i=1}^{|\mathcal{S}|}\|\nabla\ell_i\|_2\|\mathbf{x}_i\|_2\|\Delta W_2\|_{Frobenius}\|\Delta W_1\|_{Frobenius}$$
$$\geq -\max\{|\alpha|,|\bar{\alpha}|\}\frac{1}{|\mathcal{S}|}\sum_{i=1}^{|\mathcal{S}|}\|\nabla\ell_i\|_2\|\mathbf{x}_i\|_2\big(\|\Delta W_2\|^2_{Frobenius} + \|\Delta W_1\|^2_{Frobenius}\big)$$
$$= -\max\{|\alpha|,|\bar{\alpha}|\}\frac{1}{|\mathcal{S}|}\sum_{i=1}^{|\mathcal{S}|}\|\nabla\ell_i\|_2\|\mathbf{x}_i\|_2\|(\Delta W_1, \Delta W_2)\|^2_{Frobenius},$$

where the first transition follows from Lemma 4, the second holds since $\ell(\cdot)$ is convex with respect to its first argument (recall from Lemma 4 that $\nabla^2\ell_i$ is defined to be the Hessian of $\ell(\cdot)$ at the point $(W_2 D'_{i,1}W_1\mathbf{x}_i, y_i)$ with respect to its first argument), the third is an application of the Cauchy-Schwarz inequality, the fourth follows from the spectral norm being the operator norm induced by the Euclidean norm, the fifth is due to submultiplicativity of the spectral norm, the sixth results from $D'_{i,1}$ being diagonal with diagonal elements in $\{\alpha, \bar{\alpha}\}$, the seventh holds since spectral norm is upper bounded by Frobenius norm, and the latter two are based on simple arithmetics. It follows from the above that $\lambda_{min}(\nabla^2 f(\boldsymbol{\theta})) \geq -\max\{|\alpha|,|\bar{\alpha}|\}\frac{1}{|\mathcal{S}|}\sum_{i=1}^{|\mathcal{S}|}\|\nabla\ell_i\|_2\|\mathbf{x}_i\|_2$. Consider the case where the gradient of $\ell(\cdot)$ with respect to its first argument has Euclidean norm bounded by some constant $c > 0$ (this holds, for example, if $\ell(\cdot)$ is the logistic loss). Recalling (from Lemma 4) that $\nabla\ell_i$ stands for this gradient at the point $(W_2 D'_{i,1}W_1\mathbf{x}_i, y_i)$, we obtain $\lambda_{min}(\nabla^2 f(\boldsymbol{\theta})) \geq -c\max\{|\alpha|,|\bar{\alpha}|\}\frac{1}{|\mathcal{S}|}\sum_{i=1}^{|\mathcal{S}|}\|\mathbf{x}_i\|_2$. The latter holds for any $\boldsymbol{\theta}$ belonging to any region of the form $\mathcal{D}_{\boldsymbol{\theta}'}$. Since these regions constitute the entire weight space but a zero measure set, and since by definition existence of $\nabla^2 f(\boldsymbol{\theta})$ for some $\boldsymbol{\theta}\in\mathbb{R}^d$ implies that $f(\cdot)$ is twice continuously differentiable (and therefore $\lambda_{min}(\nabla^2 f(\cdot))$ is continuous) on a neighborhood of $\boldsymbol{\theta}$, it necessarily holds that $\inf_{\boldsymbol{\theta}\in\mathbb{R}^d \text{ s.t.}\nabla^2 f(\boldsymbol{\theta})\text{ exists}}\lambda_{min}(\nabla^2 f(\boldsymbol{\theta})) \geq -c\max\{|\alpha|,|\bar{\alpha}|\}\frac{1}{|\mathcal{S}|}\sum_{i=1}^{|\mathcal{S}|}\|\mathbf{x}_i\|_2 > -\infty$. This establishes necessity of assumption *(i)*.

It remains to show that if assumption *(ii)* is not satisfied, *i.e.* if $\sum_{i=1}^{|\mathcal{S}|}\nabla\ell(\mathbf{0}, y_i)^\top h_{\boldsymbol{\theta}}(\mathbf{x}_i) = 0$ for all $\boldsymbol{\theta}\in\mathbb{R}^d$, then, regardless of whether or not assumption *(i)* holds (*i.e.* of whether $n \geq 3$ or $n = 2$), it may be that $\inf_{\boldsymbol{\theta}\in\mathbb{R}^d \text{ s.t.}\nabla^2 f(\boldsymbol{\theta})\text{ exists}}\lambda_{min}(\nabla^2 f(\boldsymbol{\theta})) > -\infty$. This is trivial — simply consider the case where the training set $\mathcal{S}$ is such that $\mathbf{x}_i = \mathbf{0}$ for all $i = 1, 2, ..., |\mathcal{S}|$. The training loss in this case is constant (see Equations (8) and (9)), implying $\inf_{\boldsymbol{\theta}\in\mathbb{R}^d \text{ s.t.}\nabla^2 f(\boldsymbol{\theta})\text{ exists}}\lambda_{min}(\nabla^2 f(\boldsymbol{\theta})) = 0$. $\qquad\square$

**Claim 3** (necessity of assumptions in Proposition 7). *In the context of Proposition 7, if assumptions* (i) *or* (ii) *are not satisfied, then the stated result may not hold, i.e. it may be that* $\inf_{\boldsymbol{\theta}\in\mathbb{R}^d \text{ s.t.}\nabla^2 f(\boldsymbol{\theta})\text{ exists}}\lambda_{min}(\nabla^2 f(\boldsymbol{\theta})) > -\infty$.

*Proof.* Suppose that assumption *(i)* is not satisfied, *i.e.* that the network is shallow ($n = 2$). With the notations of Lemmas 6 and 7, for any $\boldsymbol{\theta} \in \mathcal{D}_{\boldsymbol{\theta}'}$, $(\Delta\mathbf{w}_1, \Delta\mathbf{w}_2) \in \mathbb{R}^{d'_1} \times \mathbb{R}^{d'_2}$:

$$\nabla^2 f(\boldsymbol{\theta})[\Delta\mathbf{w}_1, \Delta\mathbf{w}_2] = \frac{1}{|\mathcal{S}|} \sum\nolimits_{i=1}^{|\mathcal{S}|} \nabla^2 \ell_i \big[ W_2(\mathbf{w}_2) D'_{i,1} W_1(\Delta\mathbf{w}_1) \mathbf{x}_i + W_2(\Delta\mathbf{w}_2) D'_{i,1} W_1(\mathbf{w}_1) \mathbf{x}_i \big]$$
$$+ \frac{2}{|\mathcal{S}|} \sum\nolimits_{i=1}^{|\mathcal{S}|} \nabla \ell_i^\top W_2(\Delta\mathbf{w}_2) D'_{i,1} W_1(\Delta\mathbf{w}_1) \mathbf{x}_i$$
$$\geq \frac{2}{|\mathcal{S}|} \sum\nolimits_{i=1}^{|\mathcal{S}|} \nabla \ell_i^\top W_2(\Delta\mathbf{w}_2) D'_{i,1} W_1(\Delta\mathbf{w}_1) \mathbf{x}_i$$
$$\geq -\frac{2}{|\mathcal{S}|} \sum\nolimits_{i=1}^{|\mathcal{S}|} \|\nabla \ell_i\|_2 \|W_2(\Delta\mathbf{w}_2) D'_{i,1} W_1(\Delta\mathbf{w}_1) \mathbf{x}_i\|_2$$
$$\geq -\frac{2}{|\mathcal{S}|} \sum\nolimits_{i=1}^{|\mathcal{S}|} \|\nabla \ell_i\|_2 \|\mathbf{x}_i\|_2 \|W_2(\Delta\mathbf{w}_2) D'_{i,1} W_1(\Delta\mathbf{w}_1)\|_{spectral}$$
$$\geq -\frac{2}{|\mathcal{S}|} \sum\nolimits_{i=1}^{|\mathcal{S}|} \|\nabla \ell_i\|_2 \|\mathbf{x}_i\|_2 \|W_2(\Delta\mathbf{w}_2)\|_{spectral} \|D'_{i,1}\|_{spectral} \|W_1(\Delta\mathbf{w}_1)\|_{spectral}$$
$$\geq -\max\{|\alpha|, |\bar{\alpha}|\} \frac{2}{|\mathcal{S}|} \sum\nolimits_{i=1}^{|\mathcal{S}|} \|\nabla \ell_i\|_2 \|\mathbf{x}_i\|_2 \|W_2(\Delta\mathbf{w}_2)\|_{spectral} \|W_1(\Delta\mathbf{w}_1)\|_{spectral}$$
$$\geq -\max\{|\alpha|, |\bar{\alpha}|\} \frac{2}{|\mathcal{S}|} \sum\nolimits_{i=1}^{|\mathcal{S}|} \|\nabla \ell_i\|_2 \|\mathbf{x}_i\|_2 \|W_2(\Delta\mathbf{w}_2)\|_{Frobenius} \|W_1(\Delta\mathbf{w}_1)\|_{Frobenius}$$
$$\geq -\max\{|\alpha|, |\bar{\alpha}|\} \frac{2}{|\mathcal{S}|} \sum\nolimits_{i=1}^{|\mathcal{S}|} \|\nabla \ell_i\|_2 \|\mathbf{x}_i\|_2 \|W_2(\cdot)\|_{op} \|\Delta\mathbf{w}_2\|_2 \|W_1(\cdot)\|_{op} \|\Delta\mathbf{w}_1\|_2$$
$$\geq -\max\{|\alpha|, |\bar{\alpha}|\} \frac{1}{|\mathcal{S}|} \sum\nolimits_{i=1}^{|\mathcal{S}|} \|\nabla \ell_i\|_2 \|\mathbf{x}_i\|_2 \|W_2(\cdot)\|_{op} \|W_1(\cdot)\|_{op} \big( \|\Delta\mathbf{w}_2\|_2^2 + \|\Delta\mathbf{w}_1\|_2^2 \big)$$
$$= -\max\{|\alpha|, |\bar{\alpha}|\} \frac{1}{|\mathcal{S}|} \sum\nolimits_{i=1}^{|\mathcal{S}|} \|\nabla \ell_i\|_2 \|\mathbf{x}_i\|_2 \prod\nolimits_{j=1}^{2} \|W_j(\cdot)\|_{op} \|(\Delta\mathbf{w}_1, \Delta\mathbf{w}_2)\|_{Frobenius}^2 \,,$$

where the first transition follows from Lemma 6, the second holds since $\ell(\cdot)$ is convex with respect to its first argument (recall from Lemma 6 that $\nabla^2 \ell_i$ is defined to be the Hessian of $\ell(\cdot)$ at the point $(W_2(\mathbf{w}_1) D'_{i,1} W_1(\mathbf{w}_1) \mathbf{x}_i, y_i)$ with respect to its first argument), the third is an application of the Cauchy-Schwarz inequality, the fourth follows from the spectral norm being the operator norm induced by the Euclidean norm, the fifth is due to submultiplicativity of the spectral norm, the sixth results from $D'_{i,1}$ being diagonal with diagonal elements in $\{\alpha, \bar{\alpha}\}$, the seventh holds since spectral norm is upper bounded by Frobenius norm, the eighth is due to the definition of $\|W_j(\cdot)\|_{op}$ (operator norm of $W_j(\cdot)$ induced by the Frobenius norm), and the latter two are based on simple arithmetics. The above implies that $\lambda_{min}(\nabla^2 f(\boldsymbol{\theta})) \geq -\max\{|\alpha|, |\bar{\alpha}|\} \frac{1}{|\mathcal{S}|} \sum_{i=1}^{|\mathcal{S}|} \|\nabla \ell_i\|_2 \|\mathbf{x}_i\|_2 \prod_{j=1}^{2} \|W_j(\cdot)\|_{op}$. Consider the case where the gradient of $\ell(\cdot)$ with respect to its first argument has Euclidean norm bounded by some constant $c > 0$ (this holds, for example, if $\ell(\cdot)$ is the logistic loss). Recalling (from Lemma 6) that $\nabla \ell_i$ stands for this gradient at the point $(W_2(\mathbf{w}_2) D'_{i,1} W_1(\mathbf{w}_1) \mathbf{x}_i, y_i)$, we obtain $\lambda_{min}(\nabla^2 f(\boldsymbol{\theta})) \geq -c\max\{|\alpha|, |\bar{\alpha}|\} \frac{1}{|\mathcal{S}|} \sum_{i=1}^{|\mathcal{S}|} \|\mathbf{x}_i\|_2 \prod_{j=1}^{2} \|W_j(\cdot)\|_{op}$. The latter holds for any $\boldsymbol{\theta}$ belonging to any region of the form $\mathcal{D}_{\boldsymbol{\theta}'}$. Since these regions constitute the entire weight space but a zero measure set, and since by definition existence of $\nabla^2 f(\boldsymbol{\theta})$ for some $\boldsymbol{\theta} \in \mathbb{R}^d$ implies that $f(\cdot)$ is twice continuously differentiable (and therefore $\lambda_{min}(\nabla^2 f(\cdot))$ is continuous) on a neighborhood of $\boldsymbol{\theta}$, it necessarily holds that:

$$\inf_{\boldsymbol{\theta} \in \mathbb{R}^d \text{ s.t.} \nabla^2 f(\boldsymbol{\theta}) \text{ exists}} \lambda_{min}(\nabla^2 f(\boldsymbol{\theta})) \geq -c\max\{|\alpha|, |\bar{\alpha}|\} \frac{1}{|\mathcal{S}|} \sum_{i=1}^{|\mathcal{S}|} \|\mathbf{x}_i\|_2 \prod_{j=1}^{2} \|W_j(\cdot)\|_{op} > -\infty \,.$$

This establishes necessity of assumption *(i)*.

It remains to show that if assumption *(ii)* is not satisfied, *i.e.* if $\sum_{i=1}^{|\mathcal{S}|} \nabla \ell(\mathbf{0}, y_i)^\top h_{\boldsymbol{\theta}}(\mathbf{x}_i) = 0$ for all $\boldsymbol{\theta} \in \mathbb{R}^d$, then, regardless of whether or not assumption *(i)* holds (*i.e.* of whether $n \geq 3$ or $n = 2$), it may be that $\inf_{\boldsymbol{\theta} \in \mathbb{R}^d \text{ s.t.} \nabla^2 f(\boldsymbol{\theta}) \text{ exists}} \lambda_{min}(\nabla^2 f(\boldsymbol{\theta})) > -\infty$. This is trivial — simply consider the case where the training set $\mathcal{S}$ is such that $\mathbf{x}_i = \mathbf{0}$ for all $i = 1, 2, ..., |\mathcal{S}|$. The training loss in this case is constant (see Equations (28) and (9)), implying $\inf_{\boldsymbol{\theta} \in \mathbb{R}^d \text{ s.t.} \nabla^2 f(\boldsymbol{\theta}) \text{ exists}} \lambda_{min}(\nabla^2 f(\boldsymbol{\theta})) = 0$. $\qquad \square$

## G  Training Loss for Least-Squares Linear Regression on Whitened Data

In this appendix we derive a simplified expression for the training loss corresponding to scalar linear regression on whitened data per least-squares criterion. Concretely, we simplify the function

$\phi : \mathbb{R}^{d_n,d_0} \to \mathbb{R}$ defined by Equation (11) in the special case where: $d_n = 1$; the empirical (uncentered) covariance matrix of the training inputs — $\Lambda_{xx} := \frac{1}{|\mathcal{S}|} \sum_{i=1}^{|\mathcal{S}|} \mathbf{x}_i \mathbf{x}_i^\top \in \mathbb{R}^{d_0,d_0}$ — is equal to identity; and the loss function $\ell : \mathbb{R}^{d_n} \times \mathcal{Y} \to \mathbb{R}$ is the square loss, *i.e.* $\mathcal{Y} = \mathbb{R}$ and $\ell(\hat{y}, y) = \frac{1}{2}(\hat{y} - y)^2$.

Let $X \in \mathbb{R}^{d_0,|\mathcal{S}|}$ and $Y \in \mathbb{R}^{1,|\mathcal{S}|}$ be the matrices whose $i$'th columns hold, respectively, the training input $\mathbf{x}_i$ and its label $y_i$, $i = 1, 2, ..., |\mathcal{S}|$. Denote by $\Lambda_{yx}$ the empirical (uncentered) cross-covariance matrix between training labels and inputs, *i.e.* $\Lambda_{yx} := \frac{1}{|\mathcal{S}|} Y X^\top \in \mathbb{R}^{1,d_0}$. In the special case under consideration, for any $W \in \mathbb{R}^{1,d_0}$:

$$
\begin{aligned}
\phi(W) &= \tfrac{1}{2|\mathcal{S}|} \sum_{i=1}^{|\mathcal{S}|} (W\mathbf{x}_i - y_i)^2 \\
&= \tfrac{1}{2|\mathcal{S}|} \| WX - Y \|_{Frobenius}^2 \\
&= \tfrac{1}{2|\mathcal{S}|} \mathrm{Tr}\big( (WX - Y)(WX - Y)^\top \big) \\
&= \tfrac{1}{2|\mathcal{S}|} \mathrm{Tr}\big( WXX^\top W^\top \big) - \tfrac{1}{|\mathcal{S}|} \mathrm{Tr}\big( YX^\top W^\top \big) + \tfrac{1}{2|\mathcal{S}|} \mathrm{Tr}\big( YY^\top \big) \\
&= \tfrac{1}{2} \mathrm{Tr}\big( W\Lambda_{xx} W^\top \big) - \mathrm{Tr}\big( \Lambda_{yx} W^\top \big) + \tfrac{1}{2|\mathcal{S}|} \mathrm{Tr}\big( YY^\top \big).
\end{aligned}
$$

Since $\Lambda_{xx}$ is equal to identity, we have:

$$
\begin{aligned}
\phi(W) &= \tfrac{1}{2} \mathrm{Tr}\big( WW^\top \big) - \mathrm{Tr}\big( \Lambda_{yx} W^\top \big) + \tfrac{1}{2|\mathcal{S}|} \mathrm{Tr}\big( YY^\top \big) \\
&= \tfrac{1}{2} \mathrm{Tr}\big( (W - \Lambda_{yx})(W - \Lambda_{yx})^\top \big) - \tfrac{1}{2} \mathrm{Tr}\big( \Lambda_{yx}\Lambda_{yx}^\top \big) + \tfrac{1}{2|\mathcal{S}|} \mathrm{Tr}\big( YY^\top \big) \\
&= \tfrac{1}{2} \| W - \Lambda_{yx} \|_{Frobenius}^2 - \tfrac{1}{2} \mathrm{Tr}\big( \Lambda_{yx}\Lambda_{yx}^\top \big) + \tfrac{1}{2|\mathcal{S}|} \mathrm{Tr}\big( YY^\top \big).
\end{aligned}
$$

$c := -\tfrac{1}{2} \mathrm{Tr}(\Lambda_{yx}\Lambda_{yx}^\top) + \tfrac{1}{2|\mathcal{S}|} \mathrm{Tr}(YY^\top)$ does not depend on $W$, so we arrive at the simplified form:

$$
\phi(W) = \tfrac{1}{2} \| W - \Lambda_{yx} \|_{Frobenius}^2 + c.
$$

## H   Convergence with Unbalanced Initialization

In Section 5 we translated an analysis of gradient flow over deep linear neural networks — Proposition 3 — into a convergence guarantee for gradient descent — Theorem 4. In order to leverage known results concerning gradient flow over deep linear neural networks, Proposition 3 assumed that initialization is balanced (*i.e.* meets Equation (15)), which in turn led Theorem 4 to assume the same. We noted (Remark 1), however, that the generic tool used for the translation — Theorem 3 — allows for gradient flow and gradient descent to be initialized differently, thus it is possible to extend Theorem 4 so that it accounts for unbalanced initialization (*i.e.* for initialization which satisfies Equation (15) only approximately). The current appendix presents such an extension.

Consider the setting of Section 5 — depth $n$ fully connected neural network as defined in Equation (8) (and surrounding text), with linear activation ($\sigma(z) = z$) and output dimension $d_n = 1$, learned via minimization of square loss over whitened and normalized data, *i.e.* via minimization of the training loss $f(\cdot)$ presented in Equation (16) (and surrounding text). For simplicity, we assume that the network's hidden widths are all equal to its input dimension, *i.e.* $d_0 = d_1 = \cdots = d_{n-1}$.[32] Deviation from balancedness (Equation (15)) will be quantified per the following definition.

**Definition 1.** The *unbalancedness magnitude* of a weight setting $\boldsymbol{\theta} \in \mathbb{R}^d$ is defined to be:

$$
\max_{j \in \{1,2,...,n-1\}} \| W_{j+1}^\top W_{j+1} - W_j W_j^\top \|_{nuclear}, \tag{33}
$$

where $W_1, W_2, ..., W_n$ denote the weight matrices constituting $\boldsymbol{\theta}$.

By Lemma 8 below, small unbalancedness magnitude implies proximity to perfect balancedness.

**Lemma 8.** *For any weight setting $\boldsymbol{\theta} \in \mathbb{R}^d$ with unbalancedness magnitude (Definition 1) equal to $\hat{\epsilon} \geq 0$, there exists a weight setting $\hat{\boldsymbol{\theta}} \in \mathbb{R}^d$ which is balanced (has unbalancedness magnitude zero) and meets $\| \boldsymbol{\theta} - \hat{\boldsymbol{\theta}} \|_2 \leq n^{1.5} \sqrt{\hat{\epsilon}}$.*

---

[32] Lemma 8 is the only part of the analysis henceforth which relies on this assumption — generalizing the lemma to account for arbitrary hidden widths will accordingly generalize the entire analysis.

*Proof sketch (for complete proof see Subappendix J.20).* By Lemma 1 in [46], an analogous result holds in the case where all weight matrices are square (*i.e.* $d_0 = d_1 = \cdots = d_n$). The proof is based on a reduction to this case, attained by replacing $W_n$ with $\sqrt{W_n^\top W_n}$. $\qquad\qquad\square$

Including Lemma 8 in the translation of Proposition 3 via Theorem 3 yields Theorem 5 below — an extension of Theorem 4 that allows for unbalanced initialization.

**Theorem 5.** *Consider minimization of the training loss $f(\cdot)$ in Equation (16) via gradient descent (Equation (6)). Denote by $\boldsymbol{\theta}_0, \boldsymbol{\theta}_1, \boldsymbol{\theta}_2, \ldots$ the iterates of gradient descent, and by $W_{n:1,0}$ the end-to-end matrix (Equation (10)) of the initial point $\boldsymbol{\theta}_0$. Assume that $\|W_{n:1,0}\|_{Frobenius} \in (0, 0.1]$ (initialization is small but non-zero), and that $W_{n:1,0}$ is not antiparallel to $\Lambda_{yx}$, meaning:*

$$\nu := \mathrm{Tr}(\Lambda_{yx}^\top W_{n:1,0}) \big/ \big(\|\Lambda_{yx}\|_{Frobenius} \|W_{n:1,0}\|_{Frobenius}\big) \neq -1.$$

*Let $\tilde{\epsilon} > 0$. Then, if the unbalancedness magnitude (Definition 1) of $\boldsymbol{\theta}_0$ is no greater than:*

$$\hat{\epsilon} := \frac{\|W_{n:1,0}\|_{Frobenius}^8 \min\{1, \tilde{\epsilon}^2\}}{n^{15} e^{12n+6} \left(\max\left\{3, \frac{3-\nu}{1+\nu}\right\}\right)^{9n-5}} \left(\ln\left(\frac{23n \max\left\{3, \frac{3-\nu}{1+\nu}\right\}}{\|W_{n:1,0}\|_{Frobenius} \min\{1, \tilde{\epsilon}\}}\right)\right)^{-2} \in \tilde{\Omega}\left(\frac{\|W_{n:1,0}\|_{Frobenius}^8 \tilde{\epsilon}^2}{n^{15} \left(poly\left(\frac{3-\nu}{1+\nu}\right)\right)^n}\right), \quad (34)$$

*and if the step size $\eta$ meets:*

$$\eta \leq \frac{\|W_{n:1,0}\|_{Frobenius}^5 \min\{1, \tilde{\epsilon}\}}{n^{17/2} e^{7n+10} \left(\max\left\{3, \frac{3-\nu}{1+\nu}\right\}\right)^{(11n-5)/2}} \left(\ln\left(\frac{23n \max\left\{3, \frac{3-\nu}{1+\nu}\right\}}{\|W_{n:1,0}\|_{Frobenius} \min\{1, \tilde{\epsilon}\}}\right)\right)^{-2} \in \tilde{\Omega}\left(\frac{\|W_{n:1,0}\|_{Frobenius}^5 \tilde{\epsilon}}{n^{17/2} \left(poly\left(\frac{3-\nu}{1+\nu}\right)\right)^n}\right), \quad (35)$$

*it holds that $f(\boldsymbol{\theta}_k) - \min_{\mathbf{q} \in \mathbb{R}^d} f(\mathbf{q}) \leq \tilde{\epsilon}$ for some $k \in \mathbb{N}$ satisfying:*[33]

$$k \leq \frac{3n \left(\frac{3}{2}\max\left\{3, \frac{3-\nu}{1+\nu}\right\}\right)^n}{\|W_{n:1,0}\|_{Frobenius} \eta} \ln\left(\frac{23n \max\left\{3, \frac{3-\nu}{1+\nu}\right\}}{\|W_{n:1,0}\|_{Frobenius} \min\{1, \tilde{\epsilon}\}}\right) + 1 \in \tilde{\mathcal{O}}\left(\frac{n \left(poly\left(\frac{3-\nu}{1+\nu}\right)\right)^n \ln\left(\frac{1}{\tilde{\epsilon}}\right)}{\|W_{n:1,0}\|_{Frobenius} \eta}\right). \quad (36)$$

*Proof sketch (for complete proof see Subappendix J.21).* The proof begins by invoking Lemma 8 for obtaining a weight setting $\hat{\boldsymbol{\theta}}_0$ which is balanced and meets $\|\boldsymbol{\theta}_0 - \hat{\boldsymbol{\theta}}_0\|_2 \leq n^{1.5}\sqrt{\hat{\epsilon}}$. It is then shown that as an initial point for gradient flow, $\hat{\boldsymbol{\theta}}_0$ satisfies the conditions of Proposition 3 (namely, in addition to being balanced, its end-to-end matrix has Frobenius norm in $(0, 0.2]$ and is not antiparallel to $\Lambda_{yx}$). From this point on, the proof is similar to that of Theorem 4 — it confirms that $f(\boldsymbol{\theta}_k) - \min_{\mathbf{q} \in \mathbb{R}^d} f(\mathbf{q}) \leq \tilde{\epsilon}$ by invoking Theorem 3 to establish that gradient descent approximates gradient flow sufficiently well until gradient flow is sufficiently close to global minimum. Throughout this process, the only deviation from the proof of Theorem 4 is that gradient descent and gradient flow are initialized differently — the former starts at $\boldsymbol{\theta}_0$, whereas the latter sets off from the nearby point $\hat{\boldsymbol{\theta}}_0$. Such discrepancy between initializations is permitted by Theorem 3. $\qquad\square$

# I   Further Experiments and Implementation Details

## I.1   Further Experiments

Figure 3 supplements Figure 1 from Section 6 by reporting results obtained on convolutional neural networks.

## I.2   Implementation Details

Below are implementation details omitted from our experimental reports (Section 6 and Subappendix I.1). Source code for reproducing the results, based on the PyTorch framework ([44]), can be found in `https://github.com/elkabzo/cont_disc_opt_dnn`.

---

[33] In addition to an upper bound (Equation (36)), the theorem's proof (Subappendix J.21) also establishes an exact expression for $k$ (Equation (92)). This expression includes terms that depend on $\hat{\boldsymbol{\theta}}_0$ — balanced weight setting near $\boldsymbol{\theta}_0$ whose existence is guaranteed by Lemma 8. Means for computing $\hat{\boldsymbol{\theta}}_0$ based on $\boldsymbol{\theta}_0$ are not provided by the lemma's statement, but are brought forth by its proof (Subappendix J.20) — a constructive reduction to Lemma 1 in [46], which itself is proven constructively.

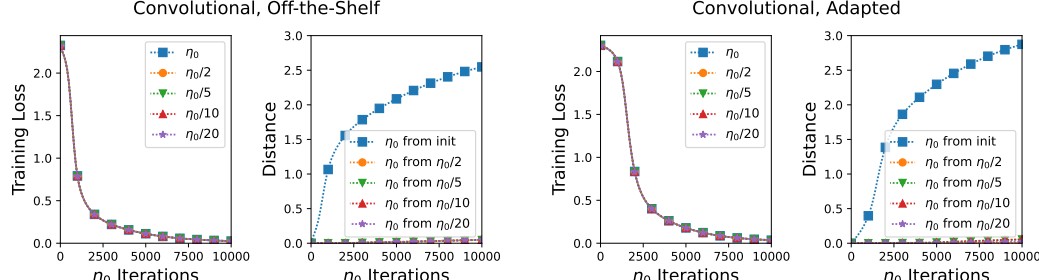

Figure 3: Over deep convolutional neural networks, trajectories of gradient descent with conventional step size barely change when step size is reduced, suggesting they are close to the continuous limit, *i.e.* to trajectories of gradient flow. This figure is identical to Figure 1, except that the results it reports were obtained on convolutional (rather than fully connected) neural networks. Specifically, left pair of plots reports results obtained on a network taken from the online tutorial "Deep Learning with PyTorch: A 60 Minute Blitz" (it comprises two convolutional layers followed by three linear layers, with rectified linear activation in each hidden layer, and max pooling in each convolutional layer),[34] while right pair corresponds to the same network slightly adapted (namely, with no biases in convolutional and linear layers, and with max pooling replaced by regular subsampling, *i.e.* by summarizing each pooling window through its top-left entry) so that it is captured by our theory (*cf.* Subsection 4.2). For further details see caption of Figure 1, as well as Subappendix I.2.

As customary, MNIST images were normalized before being used — we computed mean and standard deviation across all pixels in the dataset, and used those to shift and scale each pixel so as to ensure zero mean and unit standard deviation. To reduce run-time, rather than applying gradient descent to the full MNIST training set (60,000 labeled images), a subset of 1,000 labeled images (chosen once, uniformly at random) was used (altering the size of this subset did not yield a noticeable change in terms of final results). The Xavier distribution employed for initializing neural network weights was of type "uniform" (implemented by calling PyTorch `torch.nn.init.xavier_uniform_()` method with default parameters). Experiments ran on an internal Intel Xeon server with eight NVIDIA GeForce RTX 2080 Ti graphical processing units.

## J  Deferred Proofs

### J.1  Notations

We introduce notations to be used throughout the appendix. Beginning with matrix norms, we use $\|\cdot\|_F$ for Frobenius norm, $\|\cdot\|_n$ for nuclear norm and $\|\cdot\|_s$ for spectral norm. We extend the notation established in Lemma 1 by regarding Hessians not only as matrices and quadratic forms, but also as bilinear forms. Namely, for any $\boldsymbol{\theta} \in \mathbb{R}^d$, we regard $\nabla^2 f(\boldsymbol{\theta})$ not only as a (symmetric) matrix in $\mathbb{R}^{d,d}$ and a quadratic form $\nabla^2 f(\boldsymbol{\theta})[\cdot] : \mathbb{R}^{d_1,d_0} \times \mathbb{R}^{d_2,d_1} \times \cdots \times \mathbb{R}^{d_n,d_{n-1}} \to \mathbb{R}$, but also as a bilinear form $\nabla^2 f(\boldsymbol{\theta})[\cdot,\cdot]$ that intakes two tuples $(\Delta W_1, \Delta W_2,...,\Delta W_n), (\Delta W_1', \Delta W_2',...,\Delta W_n') \in \mathbb{R}^{d_1,d_0} \times \mathbb{R}^{d_2,d_1} \times \cdots \times \mathbb{R}^{d_n,d_{n-1}}$ as its first and second arguments (respectively), arranges them as (respective) vectors $\Delta\boldsymbol{\theta}, \Delta\boldsymbol{\theta}' \in \mathbb{R}^d$ (in correspondence with how weight matrices $W_1, W_2,...,W_n$ are arranged to create $\boldsymbol{\theta}$), and returns $\Delta\boldsymbol{\theta}^\top \nabla^2 f(\boldsymbol{\theta}) \Delta\boldsymbol{\theta}' \in \mathbb{R}$. Additionally, for any $W \in \mathbb{R}^{d_n,d_0}$, we extend the view of $\nabla^2 \phi(W)$ as a quadratic form, and also see it as a bilinear form $\nabla^2 \phi(W)[\cdot,\cdot]$ that intakes two matrices in $\mathbb{R}^{d_n,d_0}$ and returns a scalar. We similarly extend the notation of Lemma 4, regarding the matrix $\nabla^2 \ell_i \in \mathbb{R}^{d_n,d_n}$, for any $i \in \{1,2,...,|\mathcal{S}|\}$, as a bilinear form (in addition to its view as a quadratic form) $\nabla^2 \ell_i[\cdot,\cdot] : \mathbb{R}^{d_n} \times \mathbb{R}^{d_n} \to \mathbb{R}$ defined by $\nabla^2 \ell_i[\mathbf{v}, \mathbf{u}] = \mathbf{v}^\top \nabla^2 \ell_i \mathbf{u}$. Finally, for any $j \in \mathbb{N}$ we denote $[j] := \{1,2,...,j\}$.

### J.2  Proof of Theorem 3

Let $\bar{\boldsymbol{\theta}}(\cdot)$ be the continuous polygonal curve corresponding to the iterates of gradient descent:

$$\bar{\boldsymbol{\theta}} : [0,\infty) \to \mathbb{R}^d \quad, \quad \bar{\boldsymbol{\theta}}(0) = \boldsymbol{\theta}_0 \quad, \quad \tfrac{d}{dt}\bar{\boldsymbol{\theta}}(t) = -\nabla f(\boldsymbol{\theta}_k) \ \text{ for } t \in (k\eta, (k+1)\eta) \, , \, k = 0,1,2,... \, .$$

---

[34]For exact specification of network see `https://pytorch.org/tutorials/beginner/blitz/neural_networks_tutorial.html#sphx-glr-beginner-blitz-neural-networks-tutorial-py`. Note that zero padding (two pixels wide, on each side) was applied to MNIST images for compliance with specified input size (32-by-32).

If $\|\bar{\boldsymbol{\theta}}(t) - \boldsymbol{\theta}(t)\|_2 \leq \epsilon$ for all $t \in [0, \tilde{t}]$ then we are done. Assume by contradiction that this is not the case, and define $t_\epsilon := \inf\{t \in [0, \tilde{t}] : \|\bar{\boldsymbol{\theta}}(t) - \boldsymbol{\theta}(t)\|_2 > \epsilon\}$. It necessarily holds that $\|\bar{\boldsymbol{\theta}}(0) - \boldsymbol{\theta}(0)\|_2 < \epsilon$ (otherwise the expression on the right-hand side of Equation (7) becomes negative as $t \searrow 0$, in contradiction to it being greater than $\eta > 0$ for all $t \in (0, \tilde{t}]$). By continuity, this implies $t_\epsilon > 0$ and $\|\bar{\boldsymbol{\theta}}(t_\epsilon) - \boldsymbol{\theta}(t_\epsilon)\|_2 = \epsilon$. The trajectory of $\bar{\boldsymbol{\theta}}(\cdot)$ between times $0$ and $t_\epsilon$, i.e. $\bar{\boldsymbol{\theta}}([0, t_\epsilon]) := \{\bar{\boldsymbol{\theta}}(t) : t \in [0, t_\epsilon]\}$, is contained in $\mathcal{D}_{\tilde{t}, \epsilon}$. For any $t \in [0, t_\epsilon]$, the line segment (in $\mathbb{R}^d$) between $\bar{\boldsymbol{\theta}}(\lfloor t/\eta \rfloor \eta)$ and $\bar{\boldsymbol{\theta}}(t)$ is a subset of $\bar{\boldsymbol{\theta}}([0, t_\epsilon])$, thus is contained in $\mathcal{D}_{\tilde{t}, \epsilon}$ as well. We therefore have, for any $t \in [0, t_\epsilon]$:

$$
\begin{aligned}
\|\tfrac{d}{dt}\bar{\boldsymbol{\theta}}(t^+) - (-\nabla f(\bar{\boldsymbol{\theta}}(t)))\|_2 &= \|-\nabla f(\bar{\boldsymbol{\theta}}(\lfloor t/\eta \rfloor \eta)) - (-\nabla f(\bar{\boldsymbol{\theta}}(t)))\|_2 \\
&\leq \beta_{\tilde{t}, \epsilon} \|\bar{\boldsymbol{\theta}}(t) - \bar{\boldsymbol{\theta}}(\lfloor t/\eta \rfloor \eta)\|_2 \\
&= \beta_{\tilde{t}, \epsilon} \|\nabla f(\bar{\boldsymbol{\theta}}(\lfloor t/\eta \rfloor \eta))\|_2 (t - \lfloor t/\eta \rfloor \eta) \\
&\leq \beta_{\tilde{t}, \epsilon} \gamma_{\tilde{t}, \epsilon} \eta,
\end{aligned}
$$

where $\frac{d}{dt}\bar{\boldsymbol{\theta}}(t^+)$ represents the right derivative of $\bar{\boldsymbol{\theta}}(\cdot)$ at time $t$. The Fundamental Theorem (Theorem 2) may thus be applied with $\delta(t) = \beta_{\tilde{t}, \epsilon} \gamma_{\tilde{t}, \epsilon} \eta$ for all $t \in [0, t_\epsilon]$, yielding:

$$
\|\boldsymbol{\theta}(t_\epsilon) - \bar{\boldsymbol{\theta}}(t_\epsilon)\|_2 \leq e^{\int_0^{t_\epsilon} m(t') dt'} \|\boldsymbol{\theta}(0) - \bar{\boldsymbol{\theta}}(0)\|_2 + \beta_{\tilde{t}, \epsilon} \gamma_{\tilde{t}, \epsilon} \eta \int_0^{t_\epsilon} e^{\int_{t'}^{t_\epsilon} m(t'') dt''} dt'.
$$

By our assumption on the step size (Equation (7)):

$$
\eta < \frac{\epsilon - e^{\int_0^{t_\epsilon} m(t') dt'} \|\boldsymbol{\theta}_0 - \boldsymbol{\theta}(0)\|_2}{\beta_{\tilde{t}, \epsilon} \gamma_{\tilde{t}, \epsilon} \int_0^{t_\epsilon} e^{\int_{t'}^{t_\epsilon} m(t'') dt''} dt'}.
$$

Combining the latter two inequalities, we obtain $\|\boldsymbol{\theta}(t_\epsilon) - \bar{\boldsymbol{\theta}}(t_\epsilon)\|_2 < \epsilon$. Since it was previously noted that $\|\bar{\boldsymbol{\theta}}(t_\epsilon) - \boldsymbol{\theta}(t_\epsilon)\|_2 = \epsilon$, our proof by contradiction is complete. $\quad\square$

### J.3 Proof of Corollary 1

Non-negativity and $\beta$-smoothness of $f(\cdot)$ imply $\|\nabla f(\mathbf{q})\|_2 \leq \sqrt{2\beta f(\mathbf{q})}$ for all $\mathbf{q} \in \mathbb{R}^d$. Using this inequality, along with the fact that $f(\cdot)$ is non-increasing during gradient flow, we have:

$$
\sup_{t \in [0, t_e)} \|\nabla f(\boldsymbol{\theta}(t))\|_2 \leq \sup_{t \in [0, t_e)} \sqrt{2\beta f(\boldsymbol{\theta}(t))} \leq \sqrt{2\beta f(\boldsymbol{\theta}(0))}.
$$

If $\mathbf{q} \in \mathbb{R}^d$ lies no more than $\epsilon$-away from $\boldsymbol{\theta}(\cdot)$, i.e. $\exists t \in [0, t_e) : \|\mathbf{q} - \boldsymbol{\theta}(t)\|_2 \leq \epsilon$, then $\beta$-smoothness implies $\|\nabla f(\mathbf{q})\|_2 \leq \|\nabla f(\boldsymbol{\theta}(t))\|_2 + \beta\epsilon$, which in turn means $\|\nabla f(\mathbf{q})\|_2 \leq \sqrt{2\beta f(\boldsymbol{\theta}(0))} + \beta\epsilon$. We may therefore call Theorem 3 with $\gamma_{\tilde{t}, \epsilon} = \sqrt{2\beta f(\boldsymbol{\theta}(0))} + \beta\epsilon$, alongside $\beta_{\tilde{t}, \epsilon} = \beta$ and $m(\cdot) \equiv m$. Simplifying the resulting bound on the step size (Equation (7)) then completes the proof. $\quad\square$

### J.4 Proof of Lemma 1

#### J.4.1 Sketch

With $\Delta\boldsymbol{\theta}$ an arbitrary vector in $\mathbb{R}^d$, and $(\Delta W_1, \Delta W_2, ..., \Delta W_n)$ its corresponding matrix tuple, we expand:
$$
f(\boldsymbol{\theta} + \Delta\boldsymbol{\theta}) = \phi\big((W_n + \Delta W_n)(W_{n-1} + \Delta W_{n-1})\cdots(W_1 + \Delta W_1)\big),
$$
and extract $\nabla^2 f(\boldsymbol{\theta})$ from the second order terms.

#### J.4.2 Complete Proof

Recall that $\boldsymbol{\theta} \in \mathbb{R}^d$ is an arrangement of $(W_1, W_2, ..., W_n) \in \mathbb{R}^{d_1, d_0} \times \mathbb{R}^{d_2, d_1} \times \cdots \times \mathbb{R}^{d_n, d_{n-1}}$ as a vector. Let $(\Delta W_1, \Delta W_2, ..., \Delta W_n) \in \mathbb{R}^{d_1, d_0} \times \mathbb{R}^{d_2, d_1} \times \cdots \times \mathbb{R}^{d_n, d_{n-1}}$, and denote by $\Delta\boldsymbol{\theta} \in \mathbb{R}^d$ its arrangement as a vector in corresponding order. Denote:

$$
\begin{aligned}
\Delta^{(1)} &:= \textstyle\sum_{j=1}^n W_{n:j+1}(\Delta W_j) W_{j-1:1}, \\
\Delta^{(2)} &:= \textstyle\sum_{1 \leq j < j' \leq n} W_{n:j'+1}(\Delta W_{j'}) W_{j'-1:j+1}(\Delta W_j) W_{j-1:1}, \\
\Delta^{(3:n)} &:= (W_n + \Delta W_n)\cdots(W_1 + \Delta W_1) - W_{n:1} - \Delta^{(1)} - \Delta^{(2)}.
\end{aligned}
\tag{37}
$$

We now develop a second-order Taylor expansion of $f(\boldsymbol{\theta})$. Since the matrix tuple corresponding to $(\boldsymbol{\theta}+\Delta\boldsymbol{\theta})$ is $\big((W_1+\Delta W_1),...,(W_n+\Delta W_n)\big)$, and $f(\boldsymbol{\theta})=\phi(W_{n:1})$ (see beginning of Subsubsection 4.1.1) on an open region containing $\boldsymbol{\theta}$, for sufficiently small $\Delta\boldsymbol{\theta}$ we obtain:

$$f(\boldsymbol{\theta}+\Delta\boldsymbol{\theta})=\phi\Big((W_n+\Delta W_n)...(W_1+\Delta W_1)\Big)=\phi\Big(W_{n:1}+\Delta^{(1)}+\Delta^{(2)}+\Delta^{(3:n)}\Big). \qquad (38)$$

Let $\Delta W\in\mathbb{R}^{d_n,d_0}$, the second-order Taylor expansion of the twice continuously differentiable $\phi(\cdot)$ at the point $W_{n:1}$ is given by:

$$\phi(W_{n:1}+\Delta W)=\phi(W_{n:1})+\big\langle\nabla\phi(W_{n:1}),\Delta W\big\rangle+\tfrac{1}{2}\nabla^2\phi(W_{n:1})[\Delta W]+\mathcal{O}(\|\Delta W\|_F^2), \qquad (39)$$

where the $\mathcal{O}(\cdot)$ notation refers to some expression satisfying $\lim_{a\to 0}\big(\mathcal{O}(a)/a\big)=0$. We continue to develop Equation (38) using Equation (39):

$$f(\boldsymbol{\theta}+\Delta\boldsymbol{\theta})=\phi\Big(W_{n:1}+(\Delta^{(1)}+\Delta^{(2)}+\Delta^{(3:n)})\Big)$$

$$=\phi(W_{n:1})+\big\langle\nabla\phi(W_{n:1}),\Delta^{(1)}+\Delta^{(2)}+\Delta^{(3:n)}\big\rangle+$$
$$\frac{1}{2}\nabla^2\phi(W_{n:1})\Big[\Delta^{(1)}+\Delta^{(2)}+\Delta^{(3:n)}\Big]+\mathcal{O}(\big\|\Delta^{(1)}+\Delta^{(2)}+\Delta^{(3:n)}\big\|_F^2)$$

$$=\phi(W_{n:1})+\big\langle\nabla\phi(W_{n:1}),\Delta^{(1)}\big\rangle+\big\langle\nabla\phi(W_{n:1}),\Delta^{(2)}\big\rangle+\big\langle\nabla\phi(W_{n:1}),\Delta^{(3:n)}\big\rangle+$$
$$\frac{1}{2}\nabla^2\phi(W_{n:1})\Big[\Delta^{(1)}\Big]+\frac{1}{2}\nabla^2\phi(W_{n:1})\Big[\Delta^{(2)}+\Delta^{(3:n)}\Big]+$$
$$2\cdot\frac{1}{2}\nabla^2\phi(W_{n:1})\Big[\Delta^{(1)},\Delta^{(2)}+\Delta^{(3:n)}\Big]+\mathcal{O}(\big\|\Delta^{(1)}+\Delta^{(2)}+\Delta^{(3:n)}\big\|_F^2),$$

where in the last transition we view $\nabla^2\phi$ as both a quadratic and a bilinear form (see Subappendix J.1). Notice that the following terms $\big\langle\nabla\phi(W_{n:1}),\Delta^{(3:n)}\big\rangle$, $\nabla^2\phi(W_{n:1})\big[\Delta^{(2)}+\Delta^{(3:n)}\big]$, $\nabla^2\phi(W_{n:1})\big[\Delta^{(1)},\Delta^{(2)}+\Delta^{(3:n)}\big]$ and $\mathcal{O}(\big\|\Delta^{(1)}+\Delta^{(2)}+\Delta^{(3:n)}\big\|_F^2)$ are all $\mathcal{O}(\|\Delta\boldsymbol{\theta}\|_F^2)$, thus:

$$f(\boldsymbol{\theta}+\Delta\boldsymbol{\theta})$$
$$=\phi(W_{n:1})+\big\langle\nabla\phi(W_{n:1}),\Delta^{(1)}\big\rangle+\big\langle\nabla\phi(W_{n:1}),\Delta^{(2)}\big\rangle+\frac{1}{2}\nabla^2\phi(W_{n:1})\Big[\Delta^{(1)}\Big]+o\big(\|\Delta\boldsymbol{\theta}\|_F^2\big).$$

This is a Taylor expansion of $f(\cdot)$ at $\boldsymbol{\theta}$ with a constant term $\phi(W_{n:1})$, a linear term $\big\langle\nabla\phi(W_{n:1}),\Delta^{(1)}\big\rangle$, a quadratic term $\big\langle\nabla\phi(W_{n:1}),\Delta^{(2)}\big\rangle+\frac{1}{2}\nabla^2\phi(W_{n:1})\big[\Delta^{(1)}\big]$, and a remainder term of $\mathcal{O}(\|\Delta\boldsymbol{\theta}\|_F^2)$. From uniqueness of the Taylor expansion it follows that the quadratic term is equal to $\frac{1}{2}\nabla^2 f(\boldsymbol{\theta})[\Delta W_1,...,\Delta W_n]$. This implies:

$$\nabla^2 f(\boldsymbol{\theta})[\Delta W_1,...,\Delta W_n]=\nabla^2\phi(W_{n:1})\Big[\Delta^{(1)}\Big]+2\big\langle\nabla\phi(W_{n:1}),\Delta^{(2)}\big\rangle$$
$$=\nabla^2\phi(W_{n:1})\Big[\sum_{j=1}^n W_{n:j+1}(\Delta W_j)W_{j-1:1}\Big]+$$
$$2\mathrm{Tr}\Big(\nabla\phi(W_{n:1})^\top\sum_{1\le j<j'\le n}W_{n:j'+1}(\Delta W_{j'})W_{j'-1:j+1}(\Delta W_j)W_{j-1:1}\Big),$$

where the last transition follows from plugging in the definitions of $\Delta^{(1)}$ and $\Delta^{(2)}$ (see Equation (37)). $\square$

## J.5 Proof of Proposition 1

### J.5.1 Sketch

The proof is constructive — with $c>0$ arbitrary, we define a point $\boldsymbol{\theta}\in\mathbb{R}^d$, and a non-zero translation vector $\Delta\boldsymbol{\theta}\in\mathbb{R}^d\setminus\{\mathbf{0}\}$, such that $\Delta\boldsymbol{\theta}^\top\nabla^2 f(\boldsymbol{\theta})\Delta\boldsymbol{\theta}=-c\|\Delta\boldsymbol{\theta}\|_2^2$.

### J.5.2 Complete Proof

Since $\nabla\phi(0)\ne 0$, there exists $(\Delta W_1', \Delta W_2')\in\mathbb{R}^{d_1,d_0}\times\mathbb{R}^{d_2,d_1}$ and $(W_3', ..., W_n')\in\mathbb{R}^{d_3,d_2}\times\cdots\times\mathbb{R}^{d_n,d_{n-1}}$ such that $\big\langle\nabla\phi(0),W_n'\cdots W_3'\Delta W_2'\Delta W_1'\big\rangle>0$. Notice that none of the following

matrices $\Delta W_1', \Delta W_2', W_3', ..., W_n'$ are equal to zero. Define (while recalling the assumption of $n \geq 3$):

$$\Delta W_1 := \Delta W_1' \in \mathbb{R}^{d_1, d_0},$$
$$\Delta W_2 := \Delta W_2' \in \mathbb{R}^{d_2, d_1},$$
$$\Delta W_3 := 0 \in \mathbb{R}^{d_3, d_2},$$
$$\Delta W_j := 0 \in \mathbb{R}^{d_j, d_{j-1}} \text{ for } j \in \{1, 2, ..., n\}/\{1, 2, 3\}.$$

For some arbitrary $c > 0$, we define:

$$W_1 := 0 \in \mathbb{R}^{d_1, d_0},$$
$$W_2 := 0 \in \mathbb{R}^{d_2, d_1},$$
$$W_3 := W_3' \frac{-c \cdot \sum_{1 \leq j \leq n} \left\| \Delta W_j \right\|_F^2}{2 \langle \nabla \phi(0), W_n' \cdots W_3' \Delta W_2' \Delta W_1' \rangle} \in \mathbb{R}^{d_3, d_2},$$
$$W_j := W_j' \in \mathbb{R}^{d_j, d_{j-1}} \text{ for } j \in \{1, 2, ..., n\}/\{1, 2, 3\}.$$

Recall that we denote by $\boldsymbol{\theta} \in \mathbb{R}^d$ the arrangement of $(W_1, W_2, ..., W_n)$ as a vector. As shown in Lemma 1:

$$\nabla^2 f(\boldsymbol{\theta})[\Delta W_1, ..., \Delta W_n] = \nabla^2 \phi(W_{n:1}) \left[ \sum_{j=1}^n W_{n:j+1} (\Delta W_j) W_{j-1:1} \right]$$
$$+ 2 \text{Tr} \left( \nabla \phi(W_{n:1})^\top \sum_{1 \leq j < j' \leq n} W_{n:j'+1} (\Delta W_{j'}) W_{j'-1:j+1} (\Delta W_j) W_{j-1:1} \right). \tag{40}$$

Notice the first summand in the right-hand side of Equation (40) is equal to zero:

$$\nabla^2 \phi(W_{n:1}) \left[ \sum_{j=1}^n W_{n:j+1} (\Delta W_j) W_{j-1:1} \right] = \nabla^2 \phi(W_{n:1}) \left[ \Sigma_{j=1}^n 0 \right] = 0. \tag{41}$$

We develop the expression of the second summand in the right-hand side of Equation (40):

$$2 \text{Tr} \left( \nabla \phi(W_{n:1})^\top \sum_{1 \leq j < j' \leq n} W_{n:j'+1} (\Delta W_{j'}) W_{j'-1:j+1} (\Delta W_j) W_{j-1:1} \right)$$
$$= 2 \langle \nabla \phi(W_{n:1}), \sum_{1 \leq j < j' \leq n} W_{n:j'+1} (\Delta W_{j'}) W_{j'-1:j+1} (\Delta W_j) W_{j-1:1} \rangle$$
$$= 2 \langle \nabla \phi(0), W_n \cdots W_3 \Delta W_2 \Delta W_1 \rangle$$
$$= -c \cdot \sum_{1 \leq j \leq n} \left\| \Delta W_j \right\|_F^2, \tag{42}$$

where the last transition follows by plugging in the definitions of $\Delta W_1$, $\Delta W_2$ and $W_j$ for $j \in [n]/\{1, 2\}$. Plugging in Equations (41) and (42) in Equation (40), we obtain:

$$\nabla^2 f(\boldsymbol{\theta})[\Delta W_1, ..., \Delta W_n] = -c \cdot \sum_{1 \leq j \leq n} \left\| \Delta W_j \right\|_F^2. \tag{43}$$

Noticing that $\sum_{1 \leq j \leq n} \|\Delta W_j\|_F^2 \neq 0$, Equation (43) implies $\lambda_{\min}(\nabla^2 f(\boldsymbol{\theta})) \leq -c$. This bound holds for every $c > 0$, thus yielding the desired result (i.e. $\inf_{\boldsymbol{\theta} \in \mathbb{R}^d} \lambda_{\min}(\nabla^2 f(\boldsymbol{\theta})) = -\infty$). $\quad \square$

## J.6 Proof of Lemma 2

### J.6.1 Sketch

Appealing to Lemma 1, we lower bound the right-hand side of Equation (12). Convexity of $\phi(\cdot)$ implies that the first summand is non-negative. For the second summand, we use known matrix inequalities to establish a lower bound of $c \sum_{j=1}^n \|\Delta W_j\|_{Frobenius}^2$, with $c$ being the expression on the right-hand side of Equation (13).

### J.6.2 Complete Proof

Recall that $\boldsymbol{\theta} \in \mathbb{R}^d$ is an arrangement of $(W_1, W_2, ..., W_n) \in \mathbb{R}^{d_1, d_0} \times \mathbb{R}^{d_2, d_1} \times \cdots \times \mathbb{R}^{d_n, d_{n-1}}$ as a vector. Let $(\Delta W_1, \Delta W_2, ..., \Delta W_n) \in \mathbb{R}^{d_1, d_0} \times \mathbb{R}^{d_2, d_1} \times \cdots \times \mathbb{R}^{d_n, d_{n-1}}$, and denote by $\Delta \boldsymbol{\theta} \in \mathbb{R}^d$ its arrangement as a vector in corresponding order. As shown in Lemma 1:

$$\nabla^2 f(\boldsymbol{\theta})[\Delta W_1, ..., \Delta W_n] = \nabla^2 \phi(W_{n:1}) \left[ \sum_{j=1}^n W_{n:j+1} (\Delta W_j) W_{j-1:1} \right]$$
$$+ 2 \text{Tr} \left( \nabla \phi(W_{n:1})^\top \sum_{1 \leq j < j' \leq n} W_{n:j'+1} (\Delta W_{j'}) W_{j'-1:j+1} (\Delta W_j) W_{j-1:1} \right).$$

Convexity of $\phi(\cdot)$ implies that $\nabla^2\phi(W_{n:1})$ is positive semi-definite, thus:

$$\nabla^2 f(\boldsymbol{\theta})[\Delta W_1,...,\Delta W_n]\geq 2\mathrm{Tr}\Big(\nabla\phi(W_{n:1})^\top\sum_{1\leq j<j'\leq n}W_{n:j'+1}(\Delta W_{j'})W_{j'-1:j+1}(\Delta W_j)W_{j-1:1}\Big).$$

Using a simple corollary of Von-Neumann's trace inequality (see [41]):

$$\begin{aligned}
&\nabla^2 f(\boldsymbol{\theta})[\Delta W_1,...,\Delta W_n]\\
&\geq -2\|\nabla\phi(W_{n:1})\|_n\cdot\Big\|\sum_{1\leq j<j'\leq n}W_{n:j'+1}(\Delta W_{j'})W_{j'-1:j+1}(\Delta W_j)W_{j-1:1}\Big\|_s.
\end{aligned} \tag{44}$$

Upper bound the nuclear norm:

$$\|\nabla\phi(W_{n:1})\|_n\leq\sqrt{\min\{d_0,d_n\}}\|\nabla\phi(W_{n:1})\|_F. \tag{45}$$

The following bound holds:

$$\begin{aligned}
&\Big\|\sum_{1\leq j<j'\leq n}W_{n:j'+1}(\Delta W_{j'})W_{j'-1:j+1}(\Delta W_j)W_{j-1:1}\Big\|_s\\
&\leq\sum_{1\leq j<j'\leq n}\|W_{n:j'+1}(\Delta W_{j'})W_{j'-1:j+1}(\Delta W_j)W_{j-1:1}\|_s\\
&\leq\sum_{1\leq j<j'\leq n}\|\Delta W_{j'}\|_s\|\Delta W_j\|_s\cdot\prod_{k\in[n]/\{j,j'\}}\|W_k\|_s\\
&\leq\max_{\substack{\mathcal{J}\subseteq[n]\\|\mathcal{J}|=n-2}}\prod_{j\in\mathcal{J}}\|W_j\|_s\cdot\sum_{1\leq j<j'\leq n}\|\Delta W_{j'}\|_s\|\Delta W_j\|_s,
\end{aligned} \tag{46}$$

where the first transition follows from triangle inequalities, the second inequality follows from sub-multiplicativity of the spectral norm, and the last inequality follows from maximizing the term $\prod_{k\in[n]/\{j,j'\}}\|W_k\|_s$ over $j,j'$. Plugging Equations (45) and (46) into Equation (44), we have:

$$\begin{aligned}
&\nabla^2 f(\boldsymbol{\theta})[\Delta W_1,...,\Delta W_n]\\
&\geq -2\sqrt{\min\{d_0,d_n\}}\|\nabla\phi(W_{n:1})\|_F\max_{\substack{\mathcal{J}\subseteq[n]\\|\mathcal{J}|=n-2}}\prod_{j\in\mathcal{J}}\|W_j\|_s\cdot\sum_{1\leq j<j'\leq n}\|\Delta W_{j'}\|_s\|\Delta W_j\|_s.
\end{aligned}$$

It holds that:

$$\begin{aligned}
&\sum_{1\leq j<j'\leq n}\|\Delta W_{j'}\|_s\|\Delta W_j\|_s\\
&\leq\sum_{1\leq j<j'\leq n}\|\Delta W_{j'}\|_F\|\Delta W_j\|_F\\
&=\tfrac{1}{2}\Big(\sum_{j=1}^n\|\Delta W_j\|_F\Big)^2-\tfrac{1}{2}\sum_{j=1}^n\|\Delta W_j\|_F^2\\
&\leq\tfrac{n}{2}\sum_{j=1}^n\|\Delta W_j\|_F^2-\tfrac{1}{2}\sum_{j=1}^n\|\Delta W_j\|_F^2\\
&=\tfrac{n-1}{2}\sum_{j=1}^n\|\Delta W_j\|_F^2,
\end{aligned}$$

where the last inequality follows from the fact that the one-norm of a vector in $\mathbb{R}^n$ is never greater than $\sqrt{n}$ times its euclidean-norm. This leads us to:

$$\begin{aligned}
&\nabla^2 f(\boldsymbol{\theta})[\Delta W_1,...,\Delta W_n]\\
&\geq -(n-1)\sqrt{\min\{d_0,d_n\}}\|\nabla\phi(W_{n:1})\|_F\max_{\substack{\mathcal{J}\subseteq[n]\\|\mathcal{J}|=n-2}}\prod_{j\in\mathcal{J}}\|W_j\|_s\cdot\sum_{j=1}^n\|\Delta W_j\|_F^2.
\end{aligned}$$

The desired result readily follows:

$$\lambda_{\min}(\nabla^2 f(\boldsymbol{\theta}))\geq -(n-1)\sqrt{\min\{d_0,d_n\}}\|\nabla\phi(W_{n:1})\|_F\max_{\substack{\mathcal{J}\subseteq[n]\\|\mathcal{J}|=n-2}}\prod_{j\in\mathcal{J}}\|W_j\|_s.$$

$\square$

## J.7 Proof of Proposition 2

### J.7.1 Sketch

By the analysis of [18], the quantities $W_{j+1}^\top W_{j+1}-W_j W_j^\top$, $j=1,2,...,n-1$, are invariant (constant) along a gradient flow trajectory, and therefore small if initialization is such. This implies that along a trajectory emanating from near-zero initialization, for every $j=1,2,...,n-1$, the singular values of $W_j$ are similar to those of $W_{j+1}$, and the left singular vectors of $W_j$ match the right ones of $W_{j+1}$. Products of adjacent weight matrices thus simplify, and we obtain $\|W_j\|_{spectral}\approx\|W_{n:1}\|_{spectral}^{1/n}$ for $j=1,2,...,n$. Plugging this into Equation (13) yields the desired result (Equation (14)).

### J.7.2 Complete Proof

Denote by $\boldsymbol{\theta}(t)$ the time dependent gradient flow trajectory starting at $\boldsymbol{\theta}_s$ (*i.e.* $\boldsymbol{\theta}(0)=\boldsymbol{\theta}_s$) and by $W_1(t),...,W_n(t)$ the corresponding time dependent curves of weight matrices induced by the flow. From the assumption $\|\boldsymbol{\theta}_s\|_2 \leq \epsilon$ we can infer $\|W_j(0)\|_F \leq \epsilon$ for all $j \in \{1,2,...,n\}$. For $j \in \{1,2,...,n-1\}$:

$$
\begin{aligned}
&\|W_{j+1}^\top(0)W_{j+1}(0)-W_j(0)W_j^\top(0)\|_s \\
&\leq \|W_{j+1}^\top(0)W_{j+1}(0)\|_s + \|W_j(0)W_j^\top(0)\|_s \\
&= \|W_{j+1}(0)\|_s^2 + \|W_j(0)\|_s^2 \\
&\leq \|W_{j+1}(0)\|_F^2 + \|W_j(0)\|_F^2 \leq 2\epsilon^2 \leq (2\epsilon)^2 .
\end{aligned}
$$

Theorem 2.2 from [18] states that $\frac{\partial}{\partial t}\left(W_j(t)W_j^\top(t)-W_{j+1}^\top(t)W_{j+1}(t)\right)=0$ for all $j \in \{1,2,...,n-1\}$ and $t \geq 0$, thus:

$$
\|W_{j+1}^\top(t)W_{j+1}(t)-W_j(t)W_j^\top(t)\|_s = \|W_{j+1}^\top(0)W_{j+1}(0)-W_j(0)W_j^\top(0)\|_s \leq (2\epsilon)^2 .
$$

We can rely on this condition in order to apply Lemma 9 below and get that for all $t \geq 0$:

$$
\max_{j\in\{1,...,n\}}\|W_j(t)\|^n \leq \|W_{n:1}(t)\|_s + 4n\epsilon\cdot\max\left(1,\{\|W_j(t)\|_s\}_{j\in[n]}\right)^{2n} .
$$

Combining the latter inequality together with the result of Lemma 2 (Equation (13)), we get:

$$
\begin{aligned}
&\lambda_{min}(\nabla^2 f(\boldsymbol{\theta}(t))) \\
&\geq -(n-1)\sqrt{\min\{d_0,d_n\}}\|\nabla\phi(W_{n:1}(t))\|_F \max_{\substack{\mathcal{J}\subseteq[n] \\ |\mathcal{J}|=n-2}} \prod_{j\in\mathcal{J}}\|W_j(t)\|_s \\
&\geq -(n-1)\sqrt{\min\{d_0,d_n\}}\|\nabla\phi(W_{n:1}(t))\|_F \max_{j\in[n]}\|W_j(t)\|_s^{n-2} \\
&= -(n-1)\sqrt{\min\{d_0,d_n\}}\|\nabla\phi(W_{n:1}(t))\|_F \left(\max_{j\in[n]}\|W_j(t)\|_s^n\right)^{\frac{n-2}{n}} \\
&\geq -(n-1)\sqrt{\min\{d_0,d_n\}}\|\nabla\phi(W_{n:1}(t))\|_F \left(\|W_{n:1}(t)\|_s + 4n\epsilon\max\left(1,\{\|W_j(t)\|_s\}_{j\in[n]}\right)^{2n}\right)^{\frac{n-2}{n}} \\
&\geq -(n-1)\sqrt{\min\{d_0,d_n\}}\|\nabla\phi(W_{n:1}(t))\|_F \|W_{n:1}(t)\|_s^{\frac{n-2}{2}} \\
&\qquad -(n-1)\sqrt{\min\{d_0,d_n\}}\|\nabla\phi(W_{n:1}(t))\|_F \left(4n\epsilon\max\left(1,\{\|W_j(t)\|_s\}_{j\in[n]}\right)^{2n}\right)^{\frac{n-2}{n}} ,
\end{aligned}
$$

where the last inequality follows from sub-additivity of any power between zero and one. Rewriting the inequality such that we remove the time notation as to be consistent with the proposition statement, we obtain:

$$
\begin{aligned}
\lambda_{\min}(\nabla^2 f(\boldsymbol{\theta})) &\geq -(n-1)\sqrt{\min\{d_0,d_n\}}\|\nabla\phi(W_{n:1})\|_F \|W_{n:1}\|_s^{1-2/n} \\
&\quad -(n-1)\sqrt{\min\{d_0,d_n\}}\|\nabla\phi(W_{n:1})\|_F (4n)^{\frac{n-2}{n}}\max\left(1,\{\|W_j(t)\|_s\}_{j\in[n]}\right)^{2(n-2)}\epsilon^{\frac{n-2}{n}} .
\end{aligned}
$$

$\square$

**Lemma 9.** *Let $A_i \in \mathbb{R}^{d_i,d_{i-1}}$ for $i\in[n]$. Denote $\Delta_i := A_{i+1}^\top A_{i+1}-A_iA_i^\top$ for $i\in[n-1]$. Assume that $\|\Delta_i\|_s \leq \frac{1}{2n}$ for $i\in[n-1]$. It holds that:*

$$
max_{i\in[n]}\|A_i\|_s^n \leq \|A_{n:1}\|_s + 2n\sqrt{\max_{i\in[n-1]}\|\Delta_i\|_s}\cdot\max_{A\in\{I,A_1,...,A_n\}}\|A\|_s^{2n} ,
$$

*where we denote $A_{j:i}$ as $A_j\cdots A_{i+1}A_i$ for $1 \leq i < j \leq n$ and as an identity matrix (with size to be inferred by context) otherwise.*

*Proof.* Define $A_{\max} := \max_{A\in\{I,A_1,...,A_n\}}\|A\|_s$ and $\Delta_{\max} := \max_{i\in[n-1]}\|\Delta_i\|_s$. Let $\boldsymbol{v} \in \mathbb{R}^{d_0}$ such that $\boldsymbol{v}\in\text{argmax}_{\|\boldsymbol{u}\|=1}\|A_1\boldsymbol{u}\|_2$. Define $a_i := \boldsymbol{v}^\top A_{n-i:1}^\top(A_{n-(i-1)}^\top A_{n-(i-1)})^i A_{n-i:1}\boldsymbol{v}$ for $i\in[n]$. For

$i \in [n-1]$ we have:

$$a_i - a_{i+1}$$
$$= \boldsymbol{v}^\top A_{n-i:1}^\top (A_{n-(i-1)}^\top A_{n-(i-1)})^i A_{n-i:1} \boldsymbol{v} - \boldsymbol{v}^\top A_{n-(i+1):1}^\top (A_{n-i}^\top A_{n-i})^{i+1} A_{n-(i+1):1} \boldsymbol{v}$$
$$= \boldsymbol{v}^\top A_{n-i:1}^\top (A_{n-(i-1)}^\top A_{n-(i-1)})^i A_{n-i:1} \boldsymbol{v} - \boldsymbol{v}^\top A_{n-(i+1):1}^\top A_{n-i}^\top (A_{n-i} A_{n-i}^\top)^i A_{n-i} A_{n-(i+1):1} \boldsymbol{v}$$
$$= \boldsymbol{v}^\top A_{n-i:1}^\top (A_{n-(i-1)}^\top A_{n-(i-1)})^i A_{n-i:1} \boldsymbol{v} - \boldsymbol{v}^\top A_{n-i:1}^\top (A_{n-i} A_{n-i}^\top)^i A_{n-i:1} \boldsymbol{v}$$
$$= \boldsymbol{v}^\top A_{n-i:1}^\top (A_{n-i} A_{n-i}^\top + \Delta_{n-i})^i A_{n-i:1} \boldsymbol{v} - \boldsymbol{v}^\top A_{n-i:1}^\top (A_{n-i} A_{n-i}^\top)^i A_{n-i:1} \boldsymbol{v}$$
$$= \boldsymbol{v}^\top A_{n-i:1}^\top \big( (A_{n-i} A_{n-i}^\top + \Delta_{n-i})^i - (A_{n-i} A_{n-i}^\top)^i \big) A_{n-i:1} \boldsymbol{v}$$
$$= \boldsymbol{v}^\top A_{n-i:1}^\top \Big( \sum_{(b_1,\ldots,b_i) \in \{0,1\}^i} \prod_{b \in \{b_1,\ldots,b_i\}} \big( b A_{n-i} A_{n-i}^\top + (1-b) \Delta_{n-i} \big) - (A_{n-i} A_{n-i}^\top)^i \Big) A_{n-i:1} \boldsymbol{v}$$
$$= \boldsymbol{v}^\top A_{n-i:1}^\top \Big( \sum_{(b_1,\ldots,b_i) \in \{0,1\}^i \setminus (1,\ldots,1)} \prod_{b \in \{b_1,\ldots,b_i\}} \big( b A_{n-i} A_{n-i}^\top + (1-b) \Delta_{n-i} \big) \Big) A_{n-i:1} \boldsymbol{v} \,,$$

where the fourth transition follows from the definition of $\Delta_{n-i}$ and the second to last transition follows from unrolling $(A_{n-i} A_{n-i}^\top + \Delta_{n-i})^i$. Taking absolute value on $a_i - a_{i+1}$ we obtain:

$$|a_i - a_{i+1}|$$
$$= \Big| \boldsymbol{v}^\top A_{n-i:1}^\top \Big( \sum_{(b_1,\ldots,b_i) \in \{0,1\}^i \setminus (1,\ldots,1)} \prod_{b \in \{b_1,\ldots,b_i\}} \big( b A_{n-i} A_{n-i}^\top + (1-b) \Delta_{n-i} \big) \Big) A_{n-i:1} \boldsymbol{v} \Big|$$
$$\leq \sum_{(b_1,\ldots,b_i) \in \{0,1\}^i \setminus (1,\ldots,1)} \Big| \boldsymbol{v}^\top A_{n-i:1}^\top \Big( \prod_{b \in \{b_1,\ldots,b_i\}} \big( b A_{n-i} A_{n-i}^\top + (1-b) \Delta_{n-i} \big) \Big) A_{n-i:1} \boldsymbol{v} \Big|$$
$$\leq \sum_{(b_1,\ldots,b_i) \in \{0,1\}^i \setminus (1,\ldots,1)} \| A_{n-i:1} \boldsymbol{v} \|_2 \Big\| \prod_{b \in \{b_1,\ldots,b_i\}} \big( b A_{n-i} A_{n-i}^\top + (1-b) \Delta_{n-i} \big) \Big\|_s \| A_{n-i:1} \boldsymbol{v} \|_2$$
$$\leq \sum_{(b_1,\ldots,b_i) \in \{0,1\}^i \setminus (1,\ldots,1)} \| A_{n-i:1} \|_s \Big( \prod_{b \in \{b_1,\ldots,b_i\}} \big\| b A_{n-i} A_{n-i}^\top + (1-b) \Delta_{n-i} \big\|_s \Big) \| A_{n-i:1} \|_s$$
$$\leq \sum_{(b_1,\ldots,b_i) \in \{0,1\}^i \setminus (1,\ldots,1)} \| A_{n-i:1} \|_s^2 \prod_{b \in \{b_1,\ldots,b_i\}} \big( b \| A_{n-i} A_{n-i}^\top \|_s + (1-b) \| \Delta_{n-i} \|_s \big)$$
$$\leq \sum_{(b_1,\ldots,b_i) \in \{0,1\}^i \setminus (1,\ldots,1)} A_{\max}^{2n} \prod_{b \in \{b_1,\ldots,b_i\}} \big( b A_{\max}^2 + (1-b) \Delta_{\max} \big) A_{\max}^n$$
$$= A_{\max}^{2n} \cdot \Big( \big( A_{\max}^2 + \Delta_{\max} \big)^i - A_{\max}^{2i} \Big) \,,$$

where the second transition follows from the triangle inequality, the third from Cauchy–Schwarz and the definition of the spectral norm, the fourth from sub-multiplicativity of the spectral norm, the fifth from sub-additivity of the spectral norm and the sixth from the definitions of $A_{\max}$ and $\Delta_{\max}$. We continue by unrolling $\big( A_{\max}^2 + \Delta_{\max} \big)^i$:

$$|a_i - a_{i+1}|$$
$$\leq A_{\max}^{2n} \cdot \Big( \sum_{k=0}^i \binom{i}{k} A_{\max}^{2(i-k)} \Delta_{\max}^k - A_{\max}^{2i} \Big)$$
$$= A_{\max}^{2n} \cdot \Big( \sum_{k=1}^i \binom{i}{k} A_{\max}^{2(i-k)} \Delta_{\max}^k \Big)$$
$$\leq A_{\max}^{2n} \cdot \Big( \sum_{k=1}^i n^k A_{\max}^{2n} \Delta_{\max}^k \Big)$$
$$= A_{\max}^{4n} \cdot \Big( \sum_{k=1}^i \big( n \Delta_{\max} \big)^k \Big)$$
$$\leq A_{\max}^{4n} \cdot \Big( \sum_{k=1}^\infty \big( n \Delta_{\max} \big)^k \Big)$$
$$= A_{\max}^{4n} \cdot \frac{n \Delta_{\max}}{1 - n \Delta_{\max}}$$
$$\leq A_{\max}^{4n} \cdot 2n \Delta_{\max} \,,$$

where the two last transitions follow from geometric series formula and the assumption $\Delta_{\max} \leq \frac{1}{2n}$. Overall we have that for $i \in [n-1]$:

$$|a_i - a_{i+1}| \leq 2n A_{\max}^{4n} \cdot \Delta_{\max} \,. \tag{47}$$

The following bound holds:
$$\|A_{n:1}\|_s^2 \geq \|A_{n:1}\boldsymbol{v}\|_2^2$$
$$= \boldsymbol{v}^\top A_{n:1}^\top A_{n:1}\boldsymbol{v}$$
$$= \boldsymbol{v}^\top A_{n-1:1}^\top (A_n^\top A_n)^1 A_{n-1:1}\boldsymbol{v}$$
$$= a_1$$
$$\geq a_2 - |a_2 - a_1|$$
$$\geq a_3 - |a_3 - a_2| - |a_2 - a_1|$$
$$\vdots$$
$$\geq a_n - \sum_{i=1}^{n-1}|a_{i+1}-a_i|$$
$$\geq a_n - \sum_{i=1}^{n-1} 2n A_{\max}^{4n}\cdot\Delta_{\max}$$
$$\geq a_n - 2n^2 A_{\max}^{4n}\cdot\Delta_{\max}$$
$$= \boldsymbol{v}^\top (A_1^\top A_1)^n \boldsymbol{v} - 2n^2 A_{\max}^{4n}\cdot\Delta_{\max}$$
$$= \|A_1\|_s^{2n} - 2n^2 A_{\max}^{4n}\cdot\Delta_{\max}\,,$$
where the second to last inequality follows from Equation (47). Overall we have:
$$\|A_1\|_s^{2n} \leq \|A_{n:1}\|_s^2 + 2n^2 A_{\max}^{4n}\cdot\Delta_{\max}\,. \tag{48}$$
For all $i\in[n-1]$:
$$\|A_i\|_s^2 = \|A_i A_i^\top\|_s$$
$$= \|A_{i+1}^\top A_{i+1} - \Delta_i\|_s$$
$$\geq \|A_{i+1}^\top A_{i+1}\|_s - \|\Delta_i\|_s$$
$$\geq \|A_{i+1}\|_s^2 - \Delta_{\max}\,.$$
It follows that for $i\in[n-1]$:
$$\|A_{i+1}\|_s^{2n} \leq \left(\|A_i\|_s^2 + \Delta_{\max}\right)^n$$
$$= \sum_{k=0}^n \binom{n}{k}\|A_i\|^{2(n-k)}\Delta_{\max}^k$$
$$= \|A_i\|_s^{2n} + \sum_{k=1}^n \binom{n}{k}\|A_i\|^{2(n-k)}\Delta_{\max}^k$$
$$\leq \|A_i\|_s^{2n} + A_{\max}^{2n}\sum_{k=1}^\infty \left(n\Delta_{\max}\right)^k$$
$$= \|A_i\|_s^{2n} + A_{\max}^{2n}\cdot\frac{n\Delta_{\max}}{1-n\Delta_{\max}}$$
$$\leq \|A_i\|_s^{2n} + 2n A_{\max}^{2n}\cdot\Delta_{\max}\,,$$
where the two last transitions follow from geometric series formula and the assumption $\Delta_{\max}\leq\frac{1}{2n}$.
Using the above result repeatedly, we get that for $i\in[n-1]$:
$$\|A_{i+1}\|_s^{2n} \leq \|A_i\|_s^{2n} + 2n A_{\max}^{2n}\cdot\Delta_{\max}$$
$$\vdots$$
$$\leq \|A_1\|_s^{2n} + i\cdot 2n A_{\max}^{2n}\cdot\Delta_{\max}$$
$$\leq \|A_1\|_s^{2n} + 2n^2 A_{\max}^{2n}\cdot\Delta_{\max}\,.$$
Overall we have that for $i\in[n]$:
$$\|A_i\|_s^{2n} \leq \|A_1\|_s^{2n} + 2n^2 A_{\max}^{2n}\cdot\Delta_{\max}\,. \tag{49}$$
Combining Equations (49) and (48) we get for $i\in[n]$:
$$\|A_i\|_s^{2n} \leq \|A_1\|_s^{2n} + 2n^2 A_{\max}^{2n}\cdot\Delta_{\max} \leq \|A_{n:1}\|_s^2 + 4n^2 A_{\max}^{4n}\cdot\Delta_{\max}\,.$$
This leads us to:
$$\max_{i\in[n]}\|A_i\|_s^n \leq \sqrt{\|A_{n:1}\|_s^2 + 4n^2 A_{\max}^{4n}\cdot\Delta_{\max}}$$
$$\leq \sqrt{\|A_{n:1}\|_s^2} + \sqrt{4n^2 A_{\max}^{4n}\cdot\Delta_{\max}}$$
$$= \|A_{n:1}\|_s + 2n\sqrt{\max_{i\in[n-1]}\|\Delta_i\|_s}\cdot\max_{A\in\{I,A_1,\dots,A_n\}}\|A\|_s^{2n}\,,$$

where the second transition follows from sub-additivity of square root and the last transition follows from the definitions of $A_{\max}$ and $\Delta_{\max}$. □

## J.8 Proof of Proposition 3

Subsubappendix J.8.1 below provides a brief proof sketch. The complete proof is delivered by the subsequent subsubappendixes, organized as follows. Subsubappendix J.8.2 establishes preliminaries. Subsubappendix J.8.3 proves that the trajectory of gradient flow is defined over infinite time. Subsubappendix J.8.4 defines a reparameterization of the gradient flow trajectory, to be used as a technical tool. Subsubappendix J.8.5 lower bounds the minimal distance of the reparameterized trajectory from the origin. Subsubappendix J.8.6 confirms that the reparameterized trajectory escapes the origin. Subsubappendix J.8.7 establishes subsequent convergence, during which the reparameterized trajectory approaches global minimum exponentially fast. Subsubappendix J.8.8 shows that at time $\bar{t}$ (defined in Equation (17)) the (original) gradient flow trajectory reaches $\bar{\epsilon}$-optimality. Subsubappendix J.8.9 analyzes the geometry of the optimization landscape around the gradient flow trajectory, namely, it confirms validity of the smoothness and Lipschitz constants $\beta_{t,\epsilon}$ and $\gamma_{t,\epsilon}$ (given in statement of Proposition 3) respectively, and bounds the integral of the minimal eigenvalue of the Hessian in accordance with Equation (18). Finally, Subsubappendix J.8.10 concludes.

### J.8.1 Sketch

By result of [4], gradient flow induces on the end-to-end matrix the following dynamics:

$$\frac{d}{dt} W_{n:1}(t) = -\nabla\phi\big(W_{n:1}(t)\big)\Big(\|W_{n:1}(t)\|_{Frobenius}^{2-2/n} I_{d_0} + (n-1)\big[W_{n:1}^{\top}(t) W_{n:1}(t)\big]^{1-1/n}\Big),$$

where $I_{d_0} \in \mathbb{R}^{d_0, d_0}$ represents identity, and $[\cdot]^c$, $c \geq 0$, stands for a power operator defined over positive semi-definite matrices (with $c = 0$ yielding identity by definition). Carefully analyzing these dynamics, we characterize $W_{n:1}(\cdot)$ — trajectory of end-to-end matrix — and show that, with $\bar{t}$ given by Equation (17), $\frac{1}{2}\|W_{n:1}(\bar{t}) - \Lambda_{yx}\|_{Frobenius}^2 \leq \bar{\epsilon}$ as required. For establishing Equation (18), we use the characterization of $W_{n:1}(\cdot)$, along with a lower bound on the minimal eigenvalue of the Hessian provided in Subsubsection 4.1.1. The expressions for $\beta_{t,\epsilon}$ and $\gamma_{t,\epsilon}$ are also derived using the characterization of $W_{n:1}(\cdot)$ and geometric bounds (bounds on Hessian eigenvalues and gradient norm, respectively), but they involve much coarser computations.

### J.8.2 Preliminaries

We assume $\bar{\epsilon} \leq \frac{1}{2}$ without loss of generality (a proof that is valid for $\bar{\epsilon} = \frac{1}{2}$ automatically accounts for $\bar{\epsilon} > \frac{1}{2}$ as well). Throughout the proof we identify matrices in $\mathbb{R}^{1,d_0}$ with vectors in $\mathbb{R}^{d_0}$. For example, we identify the end-to-end matrix $W_{n:1} \in \mathbb{R}^{1,d_0}$ (Equation (10)) with the vector $\boldsymbol{w}_{n:1} \in \mathbb{R}^{d_0}$, and the empirical (uncentered) cross-covariance matrix between training labels and inputs, $\Lambda_{yx} \in \mathbb{R}^{1,d_0}$, with the vector $\boldsymbol{\lambda}_{yx} \in \mathbb{R}^{d_0}$. Accordingly, we overload notation by regarding the function $\phi(\cdot)$ (defined in Equation (11)) not only as a mapping from $\mathbb{R}^{1,d_0}$ to $\mathbb{R}$, but also as one from $\mathbb{R}^{d_0}$ to $\mathbb{R}$. Under the latter view, $\phi(\cdot)$ is defined by $\phi(\boldsymbol{w}) = \frac{1}{2}\|\boldsymbol{w} - \boldsymbol{\lambda}_{yx}\|_2^2 + \min_{\boldsymbol{q} \in \mathbb{R}^d} f(\boldsymbol{q})$. For $t \geq 0$, we denote by $W_1(t) \in \mathbb{R}^{d_1,d_0}, W_2(t) \in \mathbb{R}^{d_2,d_1}, \ldots, W_{n-1}(t) \in \mathbb{R}^{d_{n-1},d_{n-2}}, W_n(t) \in \mathbb{R}^{1,d_{n-1}}$ the weight matrices constituting $\boldsymbol{\theta}(t) \in \mathbb{R}^d$ (gradient flow trajectory at time $t$), and by $W_{n:1}(t) \in \mathbb{R}^{1,d_0}$ (or $\mathbf{w}_{n:1}(t) \in \mathbb{R}^{d_0}$) the corresponding end-to-end matrix (i.e. $W_{n:1}(t) := W_n(t) W_{n-1}(t) \cdots W_1(t)$).

**Definition 2.** Define $\boldsymbol{h} : \mathbb{R}^{d_0} \to \mathbb{R}^{d_0}$ by:

$$\boldsymbol{h}(\boldsymbol{w}) := \Big(\|\boldsymbol{w}\|_2^{2-\frac{2}{n}} I_{d_0} + (n-1)\big[\boldsymbol{w}\boldsymbol{w}^{\top}\big]^{1-\frac{1}{n}}\Big)\nabla\phi(\boldsymbol{w}),$$

where $I_{d_0} \in \mathbb{R}^{d_0,d_0}$ represents identity, and $[\cdot]^c$, $c \geq 0$, stands for a power operator defined over positive semi-definite matrices (with $c = 0$ yielding identity by definition).

The importance of the vector field $\boldsymbol{h}(\cdot)$ lies in the fact that it characterizes the dynamics of the end-to-end matrix — a result proven in [4], stated hereafter for completeness.

**Lemma 10.** $\boldsymbol{w}_{n:1}(t)$ *is a solution to the following initial value problem:*

$$\boldsymbol{w}_{n:1}(0) = \boldsymbol{w}_{n:1,s} \quad , \quad \frac{d}{dt}\boldsymbol{w}_{n:1}(t) = -\boldsymbol{h}\big(\boldsymbol{w}_{n:1}(t)\big).$$

*Proof.* The lemma follows directly from Theorem 1 in [4]. □

The following lemma will be used throughout the proof.

**Lemma 11.** *Let $t \in [0, \infty) \cup \{\infty\}$. Let $q, \bar{q} : [0, t) \to \mathbb{R}$ be differentiable functions, and let $g : [0,t) \times \mathbb{R} \to \mathbb{R}$ be some locally Lipschitz function. Assume that:*

$$(i) \qquad q(0) \leq \bar{q}(0) \,;$$

$$(ii) \qquad \frac{d}{dt} q(t') \leq g\big(t', q(t')\big) \text{ for all } t' \in [0,t) \,; \text{ and}$$

$$(iii) \qquad \frac{d}{dt} \bar{q}(t') \geq g\big(t', \bar{q}(t')\big) \text{ for all } t' \in [0,t) \,.$$

*Then $q(t') \leq \bar{q}(t')$ for all $t' \in [0,t)$.*

*Proof.* The lemma is a direct consequence of Theorem 10.3 in [27]. □

### J.8.3 Infinite Time

One of the assertions of Proposition 3 is that the gradient flow trajectory is defined over infinite time. This is confirmed by the following lemma.

**Lemma 12.** *The trajectory of gradient flow is defined over infinite time.*

*Proof.* by Theorem 1 we may denote the gradient flow trajectory by $\boldsymbol{\theta} : [0, t_e) \to \mathbb{R}^d$, where either: *(i)* $t_e = \infty$; or *(ii)* $t_e < \infty$ and $\lim_{t \nearrow t_e} \|\boldsymbol{\theta}(t)\|_2 = \infty$. Our objective is to show that $t_e = \infty$, thus it suffices to establish that $\boldsymbol{\theta}(\cdot)$ is bounded, *i.e.* there exists a constant larger than $\|\boldsymbol{\theta}(t)\|$ for all $t \in [0, t_e)$. Recall that $\boldsymbol{\theta}_s$ meets the balancedness condition (Equation (15)). Theorem 2.2 in [18] implies that the balancedness condition is preserved along the gradient flow trajectory, *i.e.* for any $j \in [n-1]$ and $t \in [0, t_e)$, it holds that:

$$W_{j+1}^\top(t) W_{j+1}(t) = W_j(t) W_j^\top(t). \tag{50}$$

Using this relation repeatedly, we obtain:

$$\begin{aligned}
\|\boldsymbol{w}_{n:1}(t)\|_2^2 &= \boldsymbol{w}_{n:1}^\top(t) \boldsymbol{w}_{n:1}(t) \\
&= W_{n:1}(t) W_{n:1}^\top(t) \\
&= W_{n:2}(t) W_1(t) W_1^\top(t) W_{n:2}^\top(t) \\
&= W_{n:2}(t) W_2^\top(t) W_2(t) W_{n:2}^\top(t) \\
&= W_{n:3}(t) W_2(t) W_2^\top(t) W_2(t) W_2^\top(t) W_{n:3}^\top(t) \\
&= W_{n:3}(t) W_3^\top(t) W_3(t) W_3^\top(t) W_3(t) W_{n:3}^\top(t) \\
&\quad \vdots \\
&= \big(W_n(t) W_n^\top(t)\big)^n \\
&= \|W_n(t)\|_F^{2n}.
\end{aligned}$$

Since the balancedness condition implies that $\|W_j(t)\|_F = \|W_{j+1}(t)\|_F$ for any $t \in [0, t_e)$ and $j \in [n-1]$ (to see this, simply apply trace to both sides of Equation (50)), we may conclude $\|W_j(t)\|_F^2 = \|\boldsymbol{w}_{n:1}(t)\|_2^{2/n}$ for any $j \in [n]$. Gradient flow monotonically non-increases the objective it optimizes, *i.e.* $f\big(\boldsymbol{\theta}(t)\big)$ is non-increasing. In particular it holds that $f\big(\boldsymbol{\theta}(t)\big) \leq f\big(\boldsymbol{\theta}(0)\big)$ for all $t \in [0, t_e)$. Relying on Equation (16), we obtain $\|\boldsymbol{w}_{n:1}(t) - \boldsymbol{\lambda}_{yx}\|_2 \leq \|\boldsymbol{w}_{n:1}(0) - \boldsymbol{\lambda}_{yx}\|_2$ for all $t \in [0, t_e)$. By the triangle inequality we have that $\|\boldsymbol{w}_{n:1}(t)\|_2 \leq \|\boldsymbol{w}_{n:1}(0)\|_2 + 2\|\boldsymbol{\lambda}_{yx}\|_2$. Thus, for all $t \in [0, t_e)$:

$$\|\boldsymbol{\theta}(t)\|_2^2 = \sum_{j=1}^n \|W_j(t)\|_F^2 = n\|\boldsymbol{w}_{n:1}(t)\|_2^{2/n} \leq n\big(\|\boldsymbol{w}_{n:1}(0)\|_2 + \|\boldsymbol{\lambda}_{yx}\|_2\big)^{2/n}.$$

This completes the proof. □

### J.8.4 Reparameterization

Consider the initial value problem:

$$\boldsymbol{u}(0) = \boldsymbol{w}_{n:1}(0) \quad , \quad \tfrac{d}{dt}\boldsymbol{u}(t) = -\|\boldsymbol{u}(t)\|_2\big(n\boldsymbol{u}(t) - \boldsymbol{\lambda}_{yx}\big) + (n-1)\|\boldsymbol{u}(t)\|_2^{-1}\boldsymbol{u}(t)\boldsymbol{u}(t)^\top\boldsymbol{\lambda}_{yx} . \quad (51)$$

Lemma 13 below establishes existence of a unique solution to this problem.

**Lemma 13.** *The initial value problem in Equation* (51) *admits a solution* $\boldsymbol{u} : [0,t_e) \to \mathbb{R}^{d_0}\backslash\{\boldsymbol{0}\}$*, where either:* (i) $t_e = \infty$*; or* (ii) $t_e < \infty$ *and* $\lim_{t \nearrow t_e} \|\boldsymbol{u}(t)\|_2 \in \{0,\infty\}$*. Moreover, the solution is unique in the sense that any other solution* $\boldsymbol{u}' : [0,t_e') \to \mathbb{R}^{d_0}\backslash\{\boldsymbol{0}\}$ *must satisfy* $t_e' \le t_e$ *and* $\forall t \in [0,t_e') : \boldsymbol{u}'(t) = \boldsymbol{u}(t)$.

*Proof.* Define $\boldsymbol{g} : [0,\infty) \times \mathbb{R}^{d_0}\backslash\{\boldsymbol{0}\} \to \mathbb{R}^{d_0}$ by:

$$\boldsymbol{g}(t,\boldsymbol{w}) := -\|\boldsymbol{w}\|_2\big(n\boldsymbol{w} - \boldsymbol{\lambda}_{yx}\big) + (n-1)\|\boldsymbol{w}\|_2^{-1}\boldsymbol{w}\boldsymbol{w}^\top\boldsymbol{\lambda}_{yx} .$$

The dynamics in Equation (51) can be written as $\frac{d}{dt}\boldsymbol{u}(t) = \boldsymbol{g}\big(t,\boldsymbol{u}(t)\big)$. Since $g(\cdot)$ is locally Lipschitz continuous, the lemma follows directly from the results in Section 1.5 of [25].[5] $\qquad\square$

Hereafter, we denote by $\boldsymbol{u} : [0,t_e) \to \mathbb{R}^{d_0}\backslash\{\boldsymbol{0}\}$ the (unique) solution to Equation (51). In the remainder of the current subsubappendix we will show that $\boldsymbol{u}(\cdot)$ is a reparameterization of the gradient flow trajectory, or more precisely, of $\boldsymbol{w}_{n:1}(\cdot)$.

The following definition overloads notation by extending the scalar $\nu$ (defined in the statement of Proposition 3) to a function.

**Definition 3.** Define $\nu : [0,t_e) \to [-1,1]$ by $\nu(t) = \frac{\boldsymbol{\lambda}_{yx}^\top\boldsymbol{u}(t)}{\|\boldsymbol{\lambda}_{yx}\|_2\|\boldsymbol{u}(t)\|_2}$.

Notice that $\nu(0)$ coincides with the original (scalar) definition of $\nu$. Lemma 14 below makes use of $\nu(\cdot)$ for characterizing the dynamics of the norm of $\boldsymbol{u}(\cdot)$.

**Lemma 14.** *For all* $t \in [0,t_e)$*:*

$$\tfrac{d}{dt}\|\boldsymbol{u}(t)\|_2 = n\|\boldsymbol{u}(t)\|_2\Big(\nu(t) - \|\boldsymbol{u}(t)\|_2\Big) .$$

*Proof.* Recall that $\|\boldsymbol{\lambda}_{yx}\| = 1$. For all $t \in [0,t_e)$, it holds that:

$$\begin{aligned}
\tfrac{d}{dt}\|\boldsymbol{u}(t)\|_2 &= \tfrac{\boldsymbol{u}(t)^\top}{\|\boldsymbol{u}(t)\|_2}\tfrac{d}{dt}\boldsymbol{u}(t) \\
&= \tfrac{\boldsymbol{u}(t)^\top}{\|\boldsymbol{u}(t)\|_2}\Big(-\|\boldsymbol{u}(t)\|_2\big(n\boldsymbol{u}(t) - \boldsymbol{\lambda}_{yx}\big) + (n-1)\|\boldsymbol{u}(t)\|_2^{-1}\boldsymbol{u}(t)\boldsymbol{u}(t)^\top\boldsymbol{\lambda}_{yx}\Big) \\
&= -n\|\boldsymbol{u}(t)\|_2^2 + \|\boldsymbol{u}(t)\|_2\nu(t) + (n-1)\|\boldsymbol{u}(t)\|_2\nu(t) \\
&= n\|\boldsymbol{u}(t)\|_2\Big(\nu(t) - \|\boldsymbol{u}(t)\|_2\Big),
\end{aligned}$$

where the first transition follows from the chain rule and derivative of a (non-zero) vector norm, and the second transition follows from $u(\cdot)$ being a solution to Equation (51). $\qquad\square$

Relying on Lemma 14, Lemma 15 below derives upper and lower bounds for the norm of $\boldsymbol{u}(\cdot)$.

**Lemma 15.** *For all* $t \in [0,t_e)$*:*

$$\|\boldsymbol{u}(0)\|_2 e^{-2nt} \le \|\boldsymbol{u}(t)\|_2 \le \|\boldsymbol{u}(0)\|_2 e^{nt} .$$

*Proof.* We start by proving the upper bound. Recall that by Lemma 14 we have that $\frac{d}{dt}\|\boldsymbol{u}(t)\|_2 = n\|\boldsymbol{u}(t)\|_2\big(\nu(t) - \|\boldsymbol{u}(t)\|_2\big)$. It holds that $\frac{d}{dt}\|\boldsymbol{u}(t)\|_2 \le n\|\boldsymbol{u}(t)\|_2$, as $\nu(t) \le 1$ (by Definition 3). Integrating over time:

$$\ln(\|\boldsymbol{u}(t)\|_2) - \ln(\|\boldsymbol{u}(0)\|_2) = \int_0^t \tfrac{1}{\|\boldsymbol{u}(t')\|_2}\tfrac{d}{dt}\|\boldsymbol{u}(t')\|_2 dt' \le \int_0^t n\,dt' = nt.$$

It follows that $\|\boldsymbol{u}(t)\|_2 \le \|\boldsymbol{u}(0)\|_2\, e^{nt}$.

Moving on to the lower bound, define $g : [0, t_e) \times \mathbb{R} \to \mathbb{R}$ by:

$$g(t, z) := \begin{cases} -nz(1+z) & z \geq 1 \\ -2nz & z < 1 \end{cases}.$$

Note that $g(\cdot)$ is locally Lipschitz continuous. For all $t \in [0, t_e)$, it holds that:

$$\|\boldsymbol{u}(0)\|_2 e^{-2nt} \leq \|\boldsymbol{u}(0)\|_2 = \|\boldsymbol{w}_{n:1}(0)\|_2 < 1,$$

where the equality follows from $u(\cdot)$ being a solution to Equation (51), and the last inequality follows from an assumption made in Proposition 3. Using this fact, the following holds for all $t \in [0, t_e)$:

$$\tfrac{d}{dt} \big( \|\boldsymbol{u}(0)\|_2 e^{-2nt} \big) = -2n \|\boldsymbol{u}(0)\|_2 e^{-2nt} = g\big(t, \|\boldsymbol{u}(0)\|_2 e^{-2nt}\big).$$

On the other hand, recalling that $\nu(t) \geq -1$ (by Definition 3) for all $t \in [0, t_e)$, it holds (for both cases $\|\boldsymbol{u}(t)\| < 1$ and $\|\boldsymbol{u}(t)\| \geq 1$) that:

$$\tfrac{d}{dt} \|\boldsymbol{u}(t)\|_2 = n \|\boldsymbol{u}(t)\|_2 \big( \nu(t) - \|\boldsymbol{u}(t)\|_2 \big) \geq g\big(t, \|\boldsymbol{u}(t)\|_2\big).$$

We may now use Lemma 11 to conclude $\|\boldsymbol{u}(0)\|_2 \, e^{-2nt} \leq \|\boldsymbol{u}(t)\|_2$ for all $t \in [0, t_e)$. $\qquad\square$

Taken together, Lemmas 13 and 15 imply that $\boldsymbol{u}(\cdot)$ is defined over infinite time. We formalize this in Lemma 16 below.

**Lemma 16.** *It holds that $t_e = \infty$, i.e. we may write $\boldsymbol{u} : [0, \infty) \to \mathbb{R}^{d_0} \setminus \{\boldsymbol{0}\}$.*

*Proof.* Assume by contradiction that $t_e < \infty$. Lemma 13 implies $\lim_{t \nearrow t_e} \|\boldsymbol{u}(t)\|_2 \in \{0, \infty\}$. On the other hand, by Lemma 15 we have that $\liminf_{t \nearrow t_e} \geq \|\boldsymbol{u}(0)\|_2 e^{-2nt_e}$ and $\limsup_{t \nearrow t_e} \leq \|\boldsymbol{u}(0)\| e^{nt_e}$, which is a contradiction. Hence it must be that $t_e = \infty$. $\qquad\square$

Finally, we are in a position to prove that $\boldsymbol{u}(\cdot)$ is indeed a (monotonic) reparameterization of $\boldsymbol{w}_{n:1}(\cdot)$.

**Lemma 17.** *For all $t \geq 0$:*

$$\boldsymbol{w}_{n:1}\big(\xi(t)\big) = \boldsymbol{u}(t),$$

*where $\xi : [0, \infty) \to \mathbb{R}_{\geq 0}$ is defined by $\xi(t) := \int_0^t \|\boldsymbol{u}(t')\|_2^{-(1-2/n)} dt'$.*

*Proof.* Define $\boldsymbol{g} : [0, \infty) \times \mathbb{R}^{d_0} / \{\boldsymbol{0}\} \to \mathbb{R}^{d_0}$ by $\boldsymbol{g}(t, \boldsymbol{w}) := -\boldsymbol{h}(\boldsymbol{w}) / \|\boldsymbol{u}(t)\|_2^{1-2/n}$. Note that $\boldsymbol{g}(\cdot)$ is locally Lipschitz continuous. Define the following initial value problem:

$$\boldsymbol{q}(0) = \boldsymbol{w}_{n:1}(0) \quad , \quad \tfrac{d}{dt} \boldsymbol{q}(t) = \boldsymbol{g}\big(t, \boldsymbol{q}(t)\big). \tag{52}$$

We will show both $\boldsymbol{u}(\cdot)$ and $\boldsymbol{w}_{n:1}(\xi(\cdot))$ are solutions to Equation (52), which by uniqueness implies $\boldsymbol{u}(\cdot) = \boldsymbol{w}_{n:1}(\xi(\cdot))$ for all $t \geq 0$, as required. By the definition of $\boldsymbol{u}(\cdot)$ (solution to Equation (51)) it holds that $\boldsymbol{u}(0) = \boldsymbol{w}_{n:1}(0) = \boldsymbol{w}_{n:1}(\xi(0))$. With the help of Lemma 10 we establish the following for $t \geq 0$:

$$\tfrac{d}{dt} \big(\boldsymbol{w}_{n:1}(\xi(t))\big) = \tfrac{d}{dt} \boldsymbol{w}_{n:1}\big(\xi(t)\big) \cdot \tfrac{d\xi}{dt}(t) = -\boldsymbol{h}\big(\boldsymbol{w}_{n:1}(\xi(t))\big) / \|\boldsymbol{u}(t)\|_2^{1-2/n} = \boldsymbol{g}\big(t, \boldsymbol{w}_{n:1}(\xi(t))\big).$$

Recall that $\boldsymbol{u}(\cdot)$ is a solution to Equation (51). For all $t \geq 0$, it holds that:

$$\tfrac{d}{dt} \boldsymbol{u}(t) = -\|\boldsymbol{u}(t)\|_2 \big(n\boldsymbol{u}(t) - \boldsymbol{\lambda}_{yx}\big) + (n-1)\|\boldsymbol{u}(t)\|_2^{-1} \boldsymbol{u}(t)\boldsymbol{u}(t)^\top \boldsymbol{\lambda}_{yx}$$

$$= -\|\boldsymbol{u}(t)\|_2 \big(\boldsymbol{u}(t) - \boldsymbol{\lambda}_{yx}\big) - (n-1)\|\boldsymbol{u}(t)\|_2 \boldsymbol{u}(t) + (n-1)\|\boldsymbol{u}(t)\|_2^{-1} \boldsymbol{u}(t)\boldsymbol{u}(t)^\top \boldsymbol{\lambda}_{yx}$$

$$= -\|\boldsymbol{u}(t)\|_2 \big(\boldsymbol{u}(t) - \boldsymbol{\lambda}_{yx}\big) - (n-1)\big[\boldsymbol{u}(t)\boldsymbol{u}(t)^\top\big]^{\frac{1}{2}} \boldsymbol{u}(t) + (n-1)\big[\boldsymbol{u}(t)\boldsymbol{u}(t)^\top\big]^{\frac{1}{2}} \boldsymbol{\lambda}_{yx}$$

$$= -\|\boldsymbol{u}(t)\|_2 \big(\boldsymbol{u}(t) - \boldsymbol{\lambda}_{yx}\big) - (n-1)\big[\boldsymbol{u}(t)\boldsymbol{u}(t)^\top\big]^{\frac{1}{2}} \big(\boldsymbol{u}(t) - \boldsymbol{\lambda}_{yx}\big)$$

$$= -\|\boldsymbol{u}(t)\|_2 \big(\boldsymbol{u}(t) - \boldsymbol{\lambda}_{yx}\big) - (n-1)\big[\boldsymbol{u}(t)\boldsymbol{u}(t)^\top\big]^{\frac{1}{2}} \nabla\phi\big(\boldsymbol{u}(t)\big)$$

$$= -\Big(\|\boldsymbol{u}(t)\|_2^{2-2/n} \big(\boldsymbol{u}(t) - \boldsymbol{\lambda}_{yx}\big) + (n-1)\big[\boldsymbol{u}(t)\boldsymbol{u}(t)^\top\big]^{1-1/n} \nabla\phi\big(\boldsymbol{u}(t)\big)\Big) \Big/ \|\boldsymbol{u}(t)\|_2^{1-2/n}$$

$$= -\boldsymbol{h}\big(\boldsymbol{u}(t)\big) / \|\boldsymbol{u}(t)\|_2^{1-2/n}$$

$$= \boldsymbol{g}\big(t, \boldsymbol{u}(t)\big).$$

The above confirms that $\boldsymbol{u}(\cdot)$ and $\boldsymbol{w}_{n:1}(\xi(\cdot))$ are both solutions to Equation (52), thereby completing the proof. $\qquad\square$

### J.8.5 Minimal Distance From Origin

In this subsubappendix we derive a lower bound on the minimal distance of $\boldsymbol{u}(\cdot)$ — solution to Equation (51), which by Lemma 17 is a reparameterization of $\boldsymbol{w}_{n:1}(\cdot)$ — from the origin. We denote this minimal distance by $u_{\min}$, *i.e.* we let $u_{\min} := \inf_{t \geq 0} \|\boldsymbol{u}(t)\|_2$.

Recall the function $\nu(\cdot)$ from Definition 3. Lemma 18 below establishes several properties of this function.

**Lemma 18.** *For all $t \geq 0$, the following hold:*

$$(i) \quad \nu(t) \in (-1, 1] \, ;$$

$$(ii) \quad \tfrac{d}{dt} \nu(t) = 1 - \nu(t)^2 \, ;$$

$$(iii) \quad \nu(t) = 1 - 2 \cdot \tfrac{1 - \nu(0)}{1 + \nu(0)} \Big/ \Big( \tfrac{1 - \nu(0)}{1 + \nu(0)} + e^{2t} \Big) \, ; \text{ and}$$

$$(iv) \quad \lim_{t \nearrow \infty} \nu(t) = 1 \, .$$

*Proof.* Recall $\|\boldsymbol{\lambda}_{yx}\|_2 = 1$. It holds that:

$$\tfrac{d}{dt} \nu(t) = \boldsymbol{\lambda}_{yx}^\top \tfrac{d}{dt} \left( \tfrac{\boldsymbol{u}(t)}{\|\boldsymbol{u}(t)\|_2} \right)$$

$$= \boldsymbol{\lambda}_{yx}^\top \tfrac{\tfrac{d}{dt} \boldsymbol{u}(t) \|\boldsymbol{u}(t)\|_2 - \boldsymbol{u}(t) \tfrac{d}{dt} \|\boldsymbol{u}(t)\|_2}{\|\boldsymbol{u}(t)\|_2^2}$$

$$= \tfrac{\boldsymbol{\lambda}_{yx}^\top}{\|\boldsymbol{u}(t)\|_2} \tfrac{\tfrac{d}{dt} \boldsymbol{u}(t) \|\boldsymbol{u}(t)\|_2 - \boldsymbol{u}(t) \tfrac{d}{dt} \|\boldsymbol{u}(t)\|_2}{\|\boldsymbol{u}(t)\|_2} \, .$$

Plugging in the expression for $\tfrac{d}{dt} \boldsymbol{u}(t)$ from Equation (51) and the one of $\tfrac{d}{dt} \|\boldsymbol{u}(t)\|_2$ from Lemma 14 (while dividing by $\|\boldsymbol{u}(t)\|_2$) affirms property *(ii)*:

$$\tfrac{d}{dt} \nu(t) = \tfrac{\boldsymbol{\lambda}_{yx}^\top}{\|\boldsymbol{u}(t)\|_2} \left( \Big( (n-1) \tfrac{\boldsymbol{u}(t) \boldsymbol{u}(t)^\top}{\|\boldsymbol{u}(t)\|_2} \boldsymbol{\lambda}_{yx} - \|\boldsymbol{u}(t)\|_2 \big( n\boldsymbol{u}(t) - \boldsymbol{\lambda}_{yx} \big) \Big) - \Big( \boldsymbol{u}(t) n \big( \nu(t) - \|\boldsymbol{u}(t)\|_2 \big) \Big) \right)$$

$$= \Big( (n-1) \nu(t)^2 - \|\boldsymbol{u}(t)\|_2 \big( n\nu(t) - 1/\|\boldsymbol{u}(t)\|_2 \big) \Big) - \Big( \nu(t) n \big( \nu(t) - \|\boldsymbol{u}(t)\|_2 \big) \Big)$$

$$= (n-1) \nu(t)^2 - n\nu(t) \|\boldsymbol{u}(t)\|_2 + 1 - n\nu(t)^2 + n\nu(t) \|\boldsymbol{u}(t)\|_2$$

$$= 1 - \nu(t)^2 \, .$$

By Theorem 1, the initial value problem which $\nu(t)$ solves (*i.e.* $\nu(0) = 0$ and $\tfrac{d}{dt} \nu(t) = 1 - \nu(t)^2$) admits a unique solution. Since $t \mapsto 1 - 2 \cdot \big( \tfrac{1-\nu(0)}{1+\nu(0)} \big) \big/ \big( \tfrac{1-\nu(0)}{1+\nu(0)} + e^{2t} \big)$ is a solution to this problem, it must be that $\nu(t) = 1 - 2 \cdot \big( \tfrac{1-\nu(0)}{1+\nu(0)} \big) \big/ \big( \tfrac{1-\nu(0)}{1+\nu(0)} + e^{2t} \big)$. This confirms property *(iii)*. Properties *(i)* and *(iv)* immediately follow. $\square$

Below we define a point in time that will turn out to be one at which the distance of $\boldsymbol{u}(\cdot)$ from the origin is minimal (*i.e.* is equal to $u_{\min}$).

**Definition 4.** Let $t_m := \inf \{ t \geq 0 : \nu(t) \geq \|\boldsymbol{u}(t)\|_2 \}$, where by convention $t_m = \infty$ if $\nu(t) < \|\boldsymbol{u}(t)\|_2$ for all $t \geq 0$.

Lemma 19 below establishes that $t_m$ is finite, that the norm of $\boldsymbol{u}(\cdot)$ is monotonically decreasing until $t_m$ and monotonically non-decreasing thereafter, and that this norm remains smaller than one.

**Lemma 19.** *It holds that:*

$$(i) \quad t_m < \infty \, ;$$

$$(ii) \quad \tfrac{d}{dt} \|\boldsymbol{u}(t)\|_2 < 0 \text{ for all } t \in [0, t_m) \, ;$$

$$(iii) \quad \tfrac{d}{dt} \|\boldsymbol{u}(t)\|_2 \geq 0 \text{ for all } t \in [t_m, \infty) \, ; \text{ and}$$

$$(iv) \quad \|\boldsymbol{u}(t)\|_2 < 1 \text{ for all } t \geq 0 \, .$$

*Proof.* We start by treating the special case where $\nu(0)=1$. Recall that by assumption $\|\boldsymbol{w}_{n:1}(0)\|<1$. Together with Equation (51) this implies $\|\boldsymbol{u}(0)\|_2=\|\boldsymbol{w}_{n:1}(0)\|_2<1=\nu(0)$. Thus, by definition $t_m=0$. We trivially obtain properties *(i)* and *(ii)*. By Lemma 14 together with property *(iii)* from Lemma 18 it holds that $\frac{d}{dt}\|\boldsymbol{u}(t)\|_2=n\|\boldsymbol{u}(t)\|_2\big(1-\|\boldsymbol{u}(t)\|_2\big)$. These dynamics, along with the initial value $\|\boldsymbol{u}(0)\|_2$, induce an initial value problem whose unique solution is $\|\boldsymbol{u}(t)\|_2=e^{nt}/\big(e^{nt}+\|\boldsymbol{u}(0)\|^{-1}-1\big)$. This confirms properties *(iii)* and *(iv)*. From this point onward we assume $\nu(0)\neq 1$.

By definition of $t_m$, it holds that $\|\boldsymbol{u}(t)\|_2>\nu(t)$ for $t\in[0,t_m)$. Together with Lemma 14 this implies property *(ii)*.

Relying on property *(ii)*, we have $\|\boldsymbol{u}(t)\|_2\leq\|\boldsymbol{u}(0)\|_2<1$ for $t\in[0,t_m)$, where we used the fact that $\|\boldsymbol{u}(0)\|_2=\|\boldsymbol{w}_{n:1}(0)\|_2<1$. Assume by contradiction that property *(i)* does not hold, *i.e.* $t_m=\infty$. This means that $\nu(t)<\|\boldsymbol{u}(t)\|_2\leq\|\boldsymbol{u}(0)\|_2<1$ for all $t\geq 0$. On the other hand, by Lemma 18, $\lim_{t\nearrow\infty}\nu(t)=1$ — a contradiction. Thus, property *(i)* must hold.

From the definition of $t_m$, together with continuity of $\nu(t)$ and $\|\boldsymbol{u}(t)\|_2$, it must be that $\nu(t_m)\geq\|\boldsymbol{u}(t_m)\|_2$. Define $\bar{t}_m:=\inf\{t\geq t_m:\nu(t)<\|\boldsymbol{u}(t)\|_2\}$, where by convention the infimum of the empty set is equal to infinity. Property *(iii)* of Lemma 18 together with $\nu(0)<1$ imply that $\nu(t)<1$ for all $t\geq 0$ (recall that we are treating the case $\nu(0)\neq 1$). We have previously shown (in the proof of this lemma) that $\|\boldsymbol{u}(t)\|<1$ for $t\in[0,t_m)$. By definition of $\bar{t}_m$, it holds that $\|\boldsymbol{u}(t)\|_2\leq\nu(t)<1$ for all $t\in[t_m,\bar{t}_m)$. Relying on this inequality together with Lemma 14, we have that $\frac{d}{dt}\|\boldsymbol{u}(t)\|\geq 0$ for $t\in[t_m,\bar{t}_m)$. If $\bar{t}_m=\infty$, then we obtain properties *(iii)* and *(iv)*, thereby finishing the proof. Assume by contradiction that this is not the case, *i.e.* $\bar{t}_m<\infty$. From continuity $\|\boldsymbol{u}(\bar{t}_m)\|=\nu(\bar{t}_m)$. By Lemmas 14 and 18:

$$\frac{d}{dt}\nu(t)\big|_{t=\bar{t}_m}=1-\nu(\bar{t}_m)^2>0=n\|\boldsymbol{u}(\bar{t}_m)\|_2\Big(\nu(\bar{t}_m)-\|\boldsymbol{u}(\bar{t}_m)\|_2\Big)=\frac{d}{dt}\|\boldsymbol{u}(t)\|_2\big|_{t=\bar{t}_m},$$

implying existence of a right neighborhood of $\bar{t}_m$ which contradicts its definition. $\square$

As a direct consequence of Lemma 19, we obtain that the distance of $\boldsymbol{u}(\cdot)$ from the origin is indeed minimal at time $t_m$. This is formalized in Lemma 20 below.

**Lemma 20.** *It holds that* $\|\boldsymbol{u}(t_m)\|_2=u_{\min}$. *Moreover, if* $\nu(0)\leq\|\boldsymbol{u}(0)\|_2$ *then* $\nu(t_m)=u_{\min}$.

*Proof.* $\|\boldsymbol{u}(t_m)\|_2=u_{\min}$ directly follows from properties *(ii)* and *(iii)* of Lemma 19. In the case where $\nu(0)\leq\|\boldsymbol{u}(0)\|_2$, from continuity of $\nu(t)$ and $\|\boldsymbol{u}(t)\|_2$, and from the definition of $t_m$, it must be that $\|\boldsymbol{u}(t_m)\|_2=\nu(t_m)$. Together with $\|\boldsymbol{u}(t_m)\|_2=u_{\min}$, this concludes the proof. $\square$

Finally, we are ready to establish a lower bound for $u_{\min}$.

**Lemma 21.** *It holds that:*

$$u_{\min}\geq\big\|\boldsymbol{w}_{n:1}(0)\big\|_2\min\Big\{1,\Big(\tfrac{2}{3}\cdot\tfrac{1+\nu(0)}{1-\nu(0)}\Big)^n\Big\},$$

*where in the case* $\nu(0)=1$ *the fraction* $(1+\nu(0))/(1-\nu(0))$ *is to be interpreted as equal to infinity, leading to* $u_{\min}\geq\|\boldsymbol{w}_{n:1}(0)\|_2$.

*Proof.* We split the proof into two possible cases: *(i)* $\|\boldsymbol{u}(0)\|_2\leq\nu(0)$; and *(ii)* $\|\boldsymbol{u}(0)\|_2>\nu(0)$.

In case *(i)*, by definition $t_m=0$. By taking Lemma 20 together with Equation (51), we obtain $u_{\min}=\|\boldsymbol{u}(t_m)\|_2=\|\boldsymbol{u}(0)\|_2=\|\boldsymbol{w}_{n:1}(0)\|_2$.

Moving on to case *(ii)*, recall that by assumption $\|\boldsymbol{w}_{n:1}(0)\|_2\leq 0.2$ and $\nu(0)\neq -1$. By Equation (51), we have that $\|\boldsymbol{u}(0)\|_2=\|\boldsymbol{w}_{n:1}(0)\|_2$. Define $t_b:=\frac{1}{2}\ln\big(\frac{1+\|\boldsymbol{u}(0)\|_2}{1-\|\boldsymbol{u}(0)\|_2}\cdot\frac{1-\nu(0)}{1+\nu(0)}\big)$, which we will show upper bounds $t_m$. Plugging $t=t_b$ into the explicit expression for $\nu(t)$ given in property *(iii)* of Lemma 18, we have that $\nu(t_b)=\|\boldsymbol{u}(0)\|_2$. Taking this together with Lemma 20, we obtain $\nu(t_b)=\|\boldsymbol{u}(0)\|_2\geq u_{\min}=\nu(t_m)$. Note that $\nu(0)<\|\boldsymbol{u}(0)\|_2<1$, and property *(iii)* of Lemma 18, together imply that $\nu(t)$ is (strictly) monotonically increasing. Thus, it must be that $t_b\geq t_m$. Combining this observation with Lemma 15 yields:

$$\begin{aligned}
u_{\min}&=\|\boldsymbol{u}(t_m)\|_2\\
&\geq\|\boldsymbol{u}(0)\|_2\exp\big(-2nt_m\big)\\
&\geq\|\boldsymbol{u}(0)\|_2\exp\big(-2nt_b\big)\\
&=\|\boldsymbol{u}(0)\|_2\big(\tfrac{1+\|\boldsymbol{u}(0)\|_2}{1-\|\boldsymbol{u}(0)\|_2}\cdot\tfrac{1-\nu(0)}{1+\nu(0)}\big)^{-n}.
\end{aligned}$$

Recalling that $\|\boldsymbol{u}(0)\|_2 = \|\boldsymbol{w}_{n:1}(0)\|_2 \le 0.2$ enables us to conclude the proof for this case. $\qquad\square$

### J.8.6 Escape From Origin

Recall that $\boldsymbol{u}(\cdot)$ is the (unique) solution to Equation (51), which by Lemma 17 is a reparameterization of $\boldsymbol{w}_{n:1}(\cdot)$. Recall also the function $\nu(\cdot)$ from Definition 3, which quantifies the alignment between $\boldsymbol{u}(\cdot)$ and $\boldsymbol{\lambda}_{yx}$. The current subsubappendix defines a certain point in time (Definition 5), establishes that after this point $\boldsymbol{u}(\cdot)$ and $\boldsymbol{\lambda}_{yx}$ are highly aligned (Lemma 22), and shows that this alignment is accompanied by an escape of $\boldsymbol{u}(\cdot)$ from the origin (Lemma 23).

**Definition 5.** Define $t_a := \frac{1}{2}\ln\big(\max\big\{5 \cdot \frac{1-\nu(0)}{1+\nu(0)}, 1\big\}\big)$.

**Lemma 22.** *For all $t \ge t_a$: $\nu(t) \ge \frac{2}{3}$.*

*Proof.* We split the proof into two possible cases: *(i)* $\nu(0) \ge \frac{2}{3}$; and *(ii)* $\nu(0) < \frac{2}{3}$.

For case *(i)*, it holds that $t_a = 0$. We conclude the proof for this case by relying on Lemma 18, which implies that $\nu(t)$ is monotonically non-decreasing.

Moving on to case *(ii)*, it holds that $t_a = \frac{1}{2}\ln\big(5 \cdot \frac{1-\nu(0)}{1+\nu(0)}\big)$. Plugging $t = t_a$ into the explicit expression for $\nu(t)$ given in Lemma 18, we obtain $\nu(t_a) = \frac{2}{3}$. We conclude the proof for this case by once again relying on the fact that $\nu(t)$ is monotonically non-decreasing. $\qquad\square$

**Lemma 23.** *For all $t \ge 0$:*

$$\big\|\boldsymbol{u}(t_a + t)\big\|_2 \ge \frac{2}{3} \cdot \frac{\exp(\frac{2}{3}nt)}{\exp(\frac{2}{3}nt) + \frac{2}{3}u_{\min}^{-1} - 1} \, ,$$

*where (as defined in Subsubappendix J.8.5) $u_{\min} := \inf_{t \ge 0}\|\boldsymbol{u}(t)\|_2$.*

*Proof.* Define $g : [0, \infty) \to \mathbb{R}$ by $g(z) := nz(\frac{2}{3} - z)$. Notice that $g(\cdot)$ is locally Lipschitz continuous. Define $\bar{u} : [0, \infty) \to \mathbb{R}$ by $\bar{u}(t) := \frac{2}{3}\exp\big(\frac{2}{3}nt\big)\big/\big(\exp\big(\frac{2}{3}nt\big) + \frac{2}{3}u_{\min}^{-1} - 1\big)$. It holds that $\|\boldsymbol{u}(t_a)\|_2 \ge u_{\min} = \bar{u}(0)$. By Lemmas 14 and 22, $\frac{d}{dt}\|\boldsymbol{u}(t)\|_2 \ge g\big(\|\boldsymbol{u}(t)\|_2\big)$ for all $t \ge t_a$. Furthermore, notice that $\frac{d}{dt}\bar{u}(t) = g\big(\bar{u}(t)\big)$. We may now use Lemma 11 to obtain $\|\boldsymbol{u}(t_a + t)\|_2 \ge \bar{u}(t)$ for all $t \ge 0$. $\qquad\square$

### J.8.7 Convergence

Recall that $\boldsymbol{u}(\cdot)$ is the (unique) solution to Equation (51), and (by Lemma 17) a reparameterization of $\boldsymbol{w}_{n:1}(\cdot)$. Recall also the (alignment guaranteeing) time $t_a$ from Definition 5, the notation $u_{\min} := \inf_{t \ge 0}\|\boldsymbol{u}(t)\|_2$, and the fact that (by Lemma 21) $u_{min} > 0$. In this subsubappendix we define a certain time duration (Definition 6), and show that after it elapses from $t_a$: *(i)* the norm of $\boldsymbol{u}(\cdot)$ is on the order of one, which is the norm of the target solution $\boldsymbol{\lambda}_{yx}$ (Lemma 24); and *(ii)* $\boldsymbol{u}(\cdot)$ converges to $\boldsymbol{\lambda}_{yx}$ exponentially fast (Lemma 25).

**Definition 6.** Define $t_c := \frac{3}{2n}\ln\big(\frac{2n}{3u_{\min}}\big)$.[35]

**Lemma 24.** *For all $t \ge 0$: $\|\boldsymbol{u}(t_a + t_c + t)\|_2 \ge \frac{2n}{3(n+1)}$.*

*Proof.* Let $t \ge 0$. From Lemma 23:

$$\|\boldsymbol{u}(t_a + t_c + t)\|_2 \ge \frac{2}{3} \cdot \frac{\exp\big(\frac{2}{3}n(t_c + t)\big)}{\exp\big(\frac{2}{3}n(t_c + t)\big) + \frac{2}{3}u_{\min}^{-1} - 1} \ge \frac{2}{3} \cdot \frac{\exp\big(\frac{2}{3}nt_c\big)}{\exp\big(\frac{2}{3}nt_c\big) + \frac{2}{3}u_{\min}^{-1}} = \frac{2}{3} \cdot \frac{n\big(\frac{2}{3}u_{\min}^{-1}\big)}{n\big(\frac{2}{3}u_{\min}^{-1}\big) + \big(\frac{2}{3}u_{\min}^{-1}\big)} = \frac{2n}{3(n+1)}.$$

$\qquad\square$

**Lemma 25.** *For all $t \ge 0$: $\|\boldsymbol{\lambda}_{yx} - \boldsymbol{u}(t_a + t_c + t)\|_2 \le \frac{6}{5}\exp\big(-\frac{2n}{3(n+1)}t\big)$.*

---

[35]Note that, since $n \ge 2$ and $u_{\min} \le \|\boldsymbol{u}(0)\|_2 = \|\boldsymbol{w}_{n:1}(0)\|_2 \le 0.2$, the time duration $t_c$ is necessarily positive.

*Proof.* Property *(iv)* from Lemma 19 together with $\|\boldsymbol{\lambda}_{yx}\|_2 = 1$ imply that $\|\boldsymbol{\lambda}_{yx} - \boldsymbol{u}(t)\|_2 \neq 0$ for all $t \geq 0$. Relying on Equation (51), while recalling the function $\nu(\cdot)$ from Definition 3, we obtain:

$$
\begin{aligned}
\frac{d}{dt}\|\boldsymbol{\lambda}_{yx} - \boldsymbol{u}(t)\|_2 &= \frac{\left(\boldsymbol{\lambda}_{yx} - \boldsymbol{u}(t)\right)^\top}{\|\boldsymbol{\lambda}_{yx} - \boldsymbol{u}(t)\|_2} \frac{d}{dt}\left(\boldsymbol{\lambda}_{yx} - \boldsymbol{u}(t)\right) \\
&= \frac{\left(\boldsymbol{\lambda}_{yx} - \boldsymbol{u}(t)\right)^\top}{\|\boldsymbol{\lambda}_{yx} - \boldsymbol{u}(t)\|_2}\left(\|\boldsymbol{u}(t)\|_2\left(n\boldsymbol{u}(t) - \boldsymbol{\lambda}_{yx}\right) - (n-1)\|\boldsymbol{u}(t)\|_2^{-1}\boldsymbol{u}(t)\boldsymbol{u}(t)^\top\boldsymbol{\lambda}_{yx}\right) \\
&= \frac{\left(\boldsymbol{\lambda}_{yx} - \boldsymbol{u}(t)\right)^\top}{\|\boldsymbol{\lambda}_{yx} - \boldsymbol{u}(t)\|_2}\left(\|\boldsymbol{u}(t)\|_2\left(\boldsymbol{u}(t) - \boldsymbol{\lambda}_{yx}\right) + (n-1)\|\boldsymbol{u}(t)\|_2\boldsymbol{u}(t) - (n-1)\nu(t)\boldsymbol{u}(t)\right) \\
&= \frac{\left(\boldsymbol{\lambda}_{yx} - \boldsymbol{u}(t)\right)^\top}{\|\boldsymbol{\lambda}_{yx} - \boldsymbol{u}(t)\|_2}\left(\|\boldsymbol{u}(t)\|_2\left(\boldsymbol{u}(t) - \boldsymbol{\lambda}_{yx}\right) + (n-1)\left(\|\boldsymbol{u}(t)\|_2 - \nu(t)\right)\boldsymbol{u}(t)\right) \\
&= -\|\boldsymbol{u}(t)\|_2\|\boldsymbol{\lambda}_{yx} - \boldsymbol{u}(t)\|_2 + (n-1)\left(\|\boldsymbol{u}(t)\|_2 - \nu(t)\right)\frac{\nu(t)\|\boldsymbol{u}(t)\|_2 - \|\boldsymbol{u}(t)\|_2^2}{\|\boldsymbol{\lambda}_{yx} - \boldsymbol{u}(t)\|_2} \\
&= -\|\boldsymbol{u}(t)\|_2\|\boldsymbol{\lambda}_{yx} - \boldsymbol{u}(t)\|_2 - \frac{(n-1)\|\boldsymbol{u}(t)\|_2}{\|\boldsymbol{\lambda}_{yx} - \boldsymbol{u}(t)\|_2}\left(\|\boldsymbol{u}(t)\|_2 - \nu(t)\right)^2,
\end{aligned}
$$

for all $t \geq 0$. By Lemma 24, we may bound $\frac{d}{dt}\|\boldsymbol{\lambda}_{yx} - \boldsymbol{u}(t)\|_2 \leq -\frac{2n}{3(n+1)}\|\boldsymbol{\lambda}_{yx} - \boldsymbol{u}(t)\|_2$ for all $t \geq t_a + t_c$. Let $t' \geq 0$. We integrate $\frac{d}{dt}\|\boldsymbol{\lambda}_{yx} - \boldsymbol{u}(t)\|_2 / \|\boldsymbol{\lambda}_{yx} - \boldsymbol{u}(t)\|_2$ from $t = t_a + t_c$ to $t = t_a + t_c + t'$ in order to obtain $\|\boldsymbol{\lambda}_{yx} - \boldsymbol{u}(t_a + t_c + t')\|_2 \leq \|\boldsymbol{\lambda}_{yx} - \boldsymbol{u}(t_a + t_c)\|_2 \exp\left(-\frac{2n}{3(n+1)}t'\right)$. Recall that by assumption $\|\boldsymbol{w}_{n:1}(0)\| \leq 0.2$. By Equation (51) we have that $\|\boldsymbol{u}(0)\|_2 = \|\boldsymbol{w}_{n:1}(0)\|_2$. We conclude the proof by noting that $\|\boldsymbol{\lambda}_{yx} - \boldsymbol{u}(0)\|_2 \leq \|\boldsymbol{\lambda}_{yx}\|_2 + \|\boldsymbol{u}(0)\|_2 \leq \frac{6}{5}$. $\qquad\square$

### J.8.8 Time to Convergence

Recall that $\boldsymbol{u}(\cdot)$ is the (unique) solution to Equation (51), and that Lemma 17 presents a monotonically increasing function $\xi(\cdot)$ satisfying $\boldsymbol{w}_{n:1}\left(\xi(t)\right) = \boldsymbol{u}(t)$ for all $t \geq 0$. Recall also the function $\nu(\cdot)$ from Definition 3, quantifying the alignment between $\boldsymbol{u}(\cdot)$ and $\boldsymbol{\lambda}_{yx}$. Finally, recall the times $t_a$ and $t_c$ from Definitions 5 and 6, which guarantee alignment and initiation of exponential convergence, respectively. The current subsubappendix makes use of the above to establish that at the time $\bar{t}$ defined in Equation (17), $\boldsymbol{w}_{n:1}(\cdot)$ is $\bar{\epsilon}$-optimal, *i.e.* $\frac{1}{2}\|\boldsymbol{w}_{n:1}(\bar{t}) - \boldsymbol{\lambda}_{yx}\|_2^2 \leq \bar{\epsilon}$.

We begin by defining a certain time duration (Definition 7), and showing that it elapsing from $t_a + t_c$ ensures that $\boldsymbol{u}(\cdot)$ is $\bar{\epsilon}$-optimal (Lemma 26).

**Definition 7.** Define $t_{\bar{\epsilon}} := \frac{3(n+1)}{2n}\ln\left(\frac{6}{5\sqrt{2\bar{\epsilon}}}\right)$.[36]

**Lemma 26.** *It holds that* $\frac{1}{2}\|\boldsymbol{u}(t_a + t_c + t_{\bar{\epsilon}}) - \boldsymbol{\lambda}_{yx}\|_2^2 \leq \bar{\epsilon}$.

*Proof.* The proof follows from plugging $t = t_{\bar{\epsilon}}$ into the result of Lemma 25. $\qquad\square$

Moving from the reparameterized to the original gradient flow trajectory, *i.e.* from $\boldsymbol{u}(\cdot)$ to $\boldsymbol{w}_{n:1}(\cdot)$, we immediately obtain $\bar{\epsilon}$-optimality of $\boldsymbol{w}_{n:1}(\cdot)$ at time $\xi(t_a + t_c + t_{\bar{\epsilon}})$.

**Lemma 27.** *It holds that* $\frac{1}{2}\|\boldsymbol{w}_{n:1}(\xi(t_a + t_c + t_{\bar{\epsilon}})) - \boldsymbol{\lambda}_{yx}\|_2^2 \leq \bar{\epsilon}$.

*Proof.* The proof immediately follows from Lemmas 17 and 26. $\qquad\square$

Lemma 28 below shows that the time $\bar{t}$ defined in Equation (17) (recall that the scalar $\nu$ there, defined in the preceding text, coincides with the value taken by the function $\nu(\cdot)$ at zero) is greater than or equal to $\xi(t_a + t_c + t_{\bar{\epsilon}})$.

**Lemma 28.** *It holds that* $\bar{t} \geq \xi(t_a + t_c + t_{\bar{\epsilon}})$.

*Proof.* By Lemma 19 we have that $\|\boldsymbol{u}(t)\|_2 < 1$ for all $t \geq 0$. Recall the notation $u_{\min} := \inf_{t \geq 0}\|\boldsymbol{u}(t)\|_2$, and the fact that (by Lemma 21) $u_{\min} \geq \|\boldsymbol{w}_{n:1}(0)\|_2 \min\left\{1, \left(\frac{2}{3} \cdot \frac{1 + \nu(0)}{1 - \nu(0)}\right)^n\right\}$. For all $t \geq 0$:

$$
\xi(t) = \int_0^t \|\boldsymbol{u}(t')\|_2^{-(1 - 2/n)}\, dt' \leq \int_0^t u_{\min}^{-1}\, dt' = t u_{\min}^{-1}. \tag{53}
$$

---

[36]Note that $t_{\bar{\epsilon}} > 0$, since we assume $\bar{\epsilon} \leq 1/2$ without loss of generality (*cf.* Subsubappendix J.8.2).

Recall from Subsubappendix J.8.2 that we assume (without loss of generality) $\bar{\epsilon} \leq \frac{1}{2}$. The following holds:

$$
\begin{aligned}
t_a + t_c + t_{\bar{\epsilon}} &= \tfrac{1}{2}\ln\big(\max\big\{5\cdot\tfrac{1-\nu(0)}{1+\nu(0)},1\big\}\big) + \tfrac{3}{2n}\ln\big(\tfrac{2n}{3u_{\min}}\big) + \tfrac{3(n+1)}{2n}\ln\big(\tfrac{6}{5\sqrt{2\bar{\epsilon}}}\big) \\
&\leq \tfrac{1}{2}\ln\big(5\max\big\{\tfrac{1-\nu(0)}{1+\nu(0)},1\big\}\big) + \ln\big(\tfrac{2n}{3u_{\min}}\big) + 4\ln\big(\tfrac{1}{\sqrt{\bar{\epsilon}}}\big) \\
&\leq \ln\Big(\sqrt{5}\max\big\{\tfrac{1-\nu(0)}{1+\nu(0)},1\big\}\cdot\tfrac{2n}{3u_{\min}}\cdot(1/\bar{\epsilon})^2\Big) \\
&\leq \ln\Big(5n\max\big\{\tfrac{1-\nu(0)}{1+\nu(0)},1\big\}\cdot(1/\bar{\epsilon})^2\cdot u_{\min}^{-1}\Big).
\end{aligned}
\tag{54}
$$

Using Equations (53) and (54), we conclude the proof:

$$
\begin{aligned}
&\xi(t_a + t_c + t_{\bar{\epsilon}}) \\
&\leq (t_a + t_c + t_{\bar{\epsilon}})u_{\min}^{-1} \\
&\leq \ln\Big(5n\max\big\{\tfrac{1-\nu(0)}{1+\nu(0)},1\big\}\cdot(1/\bar{\epsilon})^2\cdot u_{\min}^{-1}\Big)\cdot u_{\min}^{-1} \\
&\leq \ln\Big(5n(1/\bar{\epsilon})^2\big\|\boldsymbol{w}_{n:1}(0)\big\|_2^{-1}\max\big\{1,\big(\tfrac{3}{2}\cdot\tfrac{1-\nu(0)}{1+\nu(0)}\big)^{n+1}\big\}\Big)\cdot\big\|\boldsymbol{w}_{n:1}(0)\big\|_2^{-1}\max\big\{1,\big(\tfrac{3}{2}\cdot\tfrac{1-\nu(0)}{1+\nu(0)}\big)^{n}\big\} \\
&\leq \ln\Big(15n(1/2\bar{\epsilon})\big\|\boldsymbol{w}_{n:1}(0)\big\|_2^{-1}\max\big\{1,\tfrac{1-\nu(0)}{1+\nu(0)}\big\}\Big)\cdot 2n\big\|\boldsymbol{w}_{n:1}(0)\big\|_2^{-1}\max\big\{1,\big(\tfrac{3}{2}\cdot\tfrac{1-\nu(0)}{1+\nu(0)}\big)^{n}\big\} \\
&= \bar{t}.
\end{aligned}
$$

$\square$

Combining Lemmas 27 and 28 with the fact that, in general, gradient flow monotonically non-increases the objective it optimizes, we obtain the result which the current subsubappendix set out to prove — $\bar{\epsilon}$-optimality of $\boldsymbol{w}_{n:1}(\cdot)$ at time $\bar{t}$.

**Lemma 29.** *It holds that $f\big(\boldsymbol{\theta}(\bar{t})\big) - \min_{\boldsymbol{q}\in\mathbb{R}^d}f(\boldsymbol{q}) = \frac{1}{2}\|\boldsymbol{w}_{n:1}(\bar{t}) - \boldsymbol{\lambda}_{yx}\|_2^2 \leq \bar{\epsilon}.$*

*Proof.* The proof follows directly from Equation (16), Lemmas 27 and 28, and the fact that $f(\boldsymbol{\theta}(\cdot))$ is monotonically non-increasing. $\square$

### J.8.9 Geometric Analysis

The current subsubappendix analyzes the geometry of the optimization landscape around the gradient flow trajectory. Namely, under the notations of Theorem 3, for $t > 0$, $\epsilon \in \big(0,\frac{1}{2n}\big]$ and corresponding $\mathcal{D}_{t,\epsilon}$ ($\epsilon$-neighborhood of gradient flow trajectory up to time $t$), it establishes a smoothness constant $\beta_{t,\epsilon} = 16n$, a Lipschitz constant $\gamma_{t,\epsilon} = 6\sqrt{n}$, and the (upper) bound on the integral of (minus) the minimal eigenvalue of the Hessian given in Equation (18) (with the function $m(\cdot)$ there being non-negative).

Recall (from Lemma 19) that the (Euclidean) norm of $\boldsymbol{u}(\cdot)$ — the (unique) solution to Equation (51) — is upper bounded by one. Since (by Lemma 17) $\boldsymbol{u}(\cdot)$ is a reparameterization of $\boldsymbol{w}_{n:1}(\cdot)$, the norm of $\boldsymbol{w}_{n:1}(\cdot)$ is upper bounded by one as well. This allows proving the following result.

**Lemma 30.** *For all $t' \geq 0$:*

$$
\big(\|\boldsymbol{w}_{n:1}(t')\|_2^{1/n} + \epsilon\big)^n \leq \|\boldsymbol{w}_{n:1}(t')\|_2 + 2n\epsilon.
$$

*Proof.* For all $t' \geq 0$:

$$
\begin{aligned}
\big(\|\boldsymbol{w}_{n:1}(t')\|_2^{1/n} + \epsilon\big)^n &= \sum_{j=0}^{n}\binom{n}{j}\|\boldsymbol{w}_{n:1}(t')\|_2^{(n-j)/n}\epsilon^j \\
&\leq \sum_{j=0}^{n}n^j\|\boldsymbol{w}_{n:1}(t')\|_2^{(n-j)/n}\epsilon^j \\
&= \|\boldsymbol{w}_{n:1}(t')\|_2 + \sum_{j=1}^{n}n^j\|\boldsymbol{w}_{n:1}(t')\|_2^{(n-j)/n}\epsilon^j.
\end{aligned}
$$

By Lemma 17 we have that $\boldsymbol{w}_{n:1}\big(\xi(t')\big) = \boldsymbol{u}(t')$, where $\xi(t') := \int_0^{t'}\|\boldsymbol{u}(t'')\|_2^{-(1-2/n)}\,dt''$. $\xi(\cdot)$ is unbounded since $\|\boldsymbol{u}(\cdot)\|_2$ is bounded by property *(iv)* of Lemma 19. It follows that:

$$
\begin{aligned}
\big(\|\boldsymbol{w}_{n:1}(t')\|_2^{1/n} + \epsilon\big)^n &\leq \|\boldsymbol{w}_{n:1}(t')\|_2 + \sum_{j=1}^{\infty}\big(n\epsilon\big)^j \\
&= \|\boldsymbol{w}_{n:1}(t')\|_2 + \frac{n\epsilon}{1-n\epsilon} \\
&\leq \|\boldsymbol{w}_{n:1}(t')\|_2 + 2n\epsilon,
\end{aligned}
$$

where the second transition follows from the formula for geometric sum (notice that $n\epsilon < 1$ since by assumption $\epsilon \leq 1/2n$), and the last transition follows from the assumption $\epsilon \leq 1/2n$. $\qquad\square$

Building on Lemma 30, and the fact that (by assumption) the gradient flow trajectory $\boldsymbol{\theta}(\cdot)$ emanates from a balanced initialization (*i.e.* an initialization whose weight matrices satisfy the condition in Equation (15)) — which by [18] implies that $\boldsymbol{\theta}(t')$ is balanced for any $t' \geq 0$ — the lemma below establishes different properties for weight settings lying $\epsilon$-away from the trajectory.

**Lemma 31.** *Let $t' \geq 0$ and let $\boldsymbol{\theta}_\epsilon \in \mathbb{R}^d$ be a weight setting satisfying $\|\boldsymbol{\theta}_\epsilon - \boldsymbol{\theta}(t')\|_2 \leq \epsilon$. Denote by $W_{1,\epsilon} \in \mathbb{R}^{d_1,d_0}, W_{2,\epsilon} \in \mathbb{R}^{d_2,d_1}, ..., W_{n-1,\epsilon} \in \mathbb{R}^{d_{n-1},d_{n-2}}, W_{n,\epsilon} \in \mathbb{R}^{1,d_{n-1}}$ the weight matrices constituting $\boldsymbol{\theta}_\epsilon$, and by $\boldsymbol{w}_{n:1,\epsilon} \in \mathbb{R}^{d_0}$ the corresponding end-to-end matrix $W_{n,\epsilon}W_{n-1,\epsilon}\cdots W_{1,\epsilon}$ (in vectorized form). Then, the following hold:*

$(i) \qquad \|\boldsymbol{w}_{n:1,\epsilon} - \boldsymbol{w}_{n:1}(t')\|_2 \leq \left(\|\boldsymbol{w}_{n:1}(t')\|_2^{1/n} + \epsilon\right)^n - \|\boldsymbol{w}_{n:1}(t')\|_2 \,;$

$(ii) \qquad \|\nabla\phi(\boldsymbol{w}_{n:1,\epsilon})\|_2 \leq \|\nabla\phi(\boldsymbol{w}_{n:1}(t'))\|_2 + 2n\epsilon \,;$ *and*

$(iii) \qquad$ *for any $\mathcal{J} \subseteq [n]\backslash\emptyset$, $\prod_{j\in\mathcal{J}}\|W_{j,\epsilon}\|_F \leq \left(\|\boldsymbol{w}_{n:1}(t')\|_2 + 2n\epsilon\right)^{\frac{|\mathcal{J}|}{n}} \,.$*

*Proof.* For brevity, throughout this proof we omit the time $t'$ from our notation, *i.e.* we denote $\boldsymbol{\theta}(t')$, $\boldsymbol{w}_{n:1}(t')$, $W_{n:1}(t')$ and $W_1(t'),...,W_n(t')$ by $\boldsymbol{\theta}$, $\boldsymbol{w}_{n:1}$, $W_{n:1}$ and $W_1,...,W_n$ respectively.

Starting with property *(i)*, we have that:

$$\left\|\boldsymbol{w}_{n:1,\epsilon} - \boldsymbol{w}_{n:1}\right\|_2$$
$$= \left\|W_{n:1,\epsilon} - W_{n:1}\right\|_F$$
$$= \left\|(W_n + W_{n,\epsilon} - W_n)\text{-}(W_1 + W_{1,\epsilon} - W_1) - W_{n:1}\right\|_F$$
$$= \left\|\sum_{(b_1,..,b_n)\in\{0,1\}^n}\left(b_n W_n + (1-b_n)(W_{n,\epsilon} - W_n)\right)\text{-}\left(b_1 W_1 + (1-b_1)(W_{1,\epsilon} - W_1)\right) - W_{n:1}\right\|_F$$
$$= \left\|\sum_{(b_1,..,b_n)\in\{0,1\}^n\backslash\{1\}^n}\left(b_n W_n + (1-b_n)(W_{\epsilon,n} - W_n)\right)\text{-}\left(b_1 W_1 + (1-b_1)(W_{\epsilon,1} - W_1)\right)\right\|_F$$
$$\leq \sum_{(b_1,..,b_n)\in\{0,1\}^n\backslash\{1\}^n}\left\|\left(b_n W_n + (1-b_n)(W_{\epsilon,n} - W_n)\right)\text{-}\left(b_1 W_1 + (1-b_1)(W_{\epsilon,1} - W_1)\right)\right\|_F$$
$$\leq \sum_{(b_1,..,b_n)\in\{0,1\}^n\backslash\{1\}^n}\left\|b_n W_n + (1-b_n)(W_{\epsilon,n} - W_n)\right\|_F\text{-}\left\|b_1 W_1 + (1-b_1)(W_{\epsilon,1} - W_1)\right\|_F$$
$$\leq \sum_{(b_1,..,b_n)\in\{0,1\}^n\backslash\{1\}^n}\left(b_n\|W_n\|_F + (1-b_n)\|W_{n,\epsilon} - W_n\|_F\right)\text{-}\left(b_1\|W_1\|_F + (1-b_1)\|W_{1,\epsilon} - W_1\|_F\right)$$
$$\leq \sum_{(b_1,..,b_n)\in\{0,1\}^n\backslash\{1\}^n}\left(b_n\|W_n\|_F + (1-b_n)\epsilon\right)\text{-}\left(b_1\|W_1\|_F + (1-b_1)\epsilon\right),$$

(55)

where the inequalities follow from sub-multiplicativity and sub-additivity of Frobenius norm, as well as the assumption $\|\boldsymbol{\theta}_\epsilon - \boldsymbol{\theta}\|_2 \leq \epsilon$. Recall that $\boldsymbol{\theta}_s$ meets the balancedness condition (Equation (15)). Theorem 2.2 from [18] implies that the balancedness condition holds along the gradient flow trajectory. Therefore, $\boldsymbol{\theta}$ is balanced, *i.e.* for any $j \in [n-1]$ it holds that $W_{j+1}^\top W_{j+1} = W_j W_j^\top$. Using this relation repeatedly, we obtain:

$$\|\boldsymbol{w}_{n:1}\|_2^2 = \boldsymbol{w}_{n:1}^\top \boldsymbol{w}_{n:1}$$
$$= W_{n:1}W_{n:1}^\top$$
$$= W_{n:2}W_1 W_1^\top W_{n:2}^\top$$
$$= W_{n:2}W_2^\top W_2 W_{n:2}^\top$$
$$= W_{n:3}W_2 W_2^\top W_2 W_2^\top W_{n:3}^\top$$
$$= W_{n:3}W_3^\top W_3 W_3^\top W_3 W_{n:3}^\top$$
$$\vdots$$
$$= \left(W_n W_n^\top\right)^n$$
$$= \|W_n\|_F^{2n}.$$

(56)

Since the balancedness condition implies that $\|W_j\|_F = \|W_{j+1}\|_F$ for any $j \in [n-1]$, we may conclude $\|W_j\|_F = \|\boldsymbol{w}_{n:1}\|_2^{1/n}$ for any $j \in [n]$. This, along with Equation (55), establishes property *(i)*:

$$
\begin{aligned}
&\left\|\boldsymbol{w}_{n:1,\epsilon} - \boldsymbol{w}_{n:1}\right\|_2 \\
&\leq \textstyle\sum_{(b_1,..,b_n)\in\{0,1\}^n\setminus\{1\}^n} \left(b_n\|\boldsymbol{w}_{n:1}\|_2^{1/n} + (1-b_n)\epsilon\right)\cdot\left(b_1\|\boldsymbol{w}_{n:1}\|_2^{1/n} + (1-b_1)\epsilon\right) \\
&= \textstyle\sum_{(b_1,..,b_n)\in\{0,1\}^n} \left(b_n\|\boldsymbol{w}_{n:1}\|_2^{1/n} + (1-b_n)\epsilon\right)\cdot\left(b_1\|\boldsymbol{w}_{n:1}\|_2^{1/n} + (1-b_1)\epsilon\right) - \|\boldsymbol{w}_{n:1}\|_2 \\
&= \left(\|\boldsymbol{w}_{n:1}\|_2^{1/n} + \epsilon\right)^n - \|\boldsymbol{w}_{n:1}\|_2 .
\end{aligned}
$$

Moving to property *(ii)*, we have that:

$$
\begin{aligned}
\|\nabla\phi(\boldsymbol{w}_{n:1,\epsilon})\|_2 = \|\boldsymbol{w}_{n:1,\epsilon} - \boldsymbol{\lambda}_{yx}\|_2 \\
= \|\boldsymbol{w}_{n:1} - \boldsymbol{\lambda}_{yx} + \boldsymbol{w}_{n:1,\epsilon} - \boldsymbol{w}_{n:1}\|_2 \\
\leq \|\boldsymbol{w}_{n:1} - \boldsymbol{\lambda}_{yx}\|_2 + \|\boldsymbol{w}_{n:1,\epsilon} - \boldsymbol{w}_{n:1}\|_2 \\
= \|\nabla\phi(\boldsymbol{w}_{n:1})\|_2 + \|\boldsymbol{w}_{n:1,\epsilon} - \boldsymbol{w}_{n:1}\|_2.
\end{aligned}
$$

applying property *(i)*, together with Lemma 30, we obtain property *(ii)*:

$$
\|\nabla\phi(\boldsymbol{w}_{n:1,\epsilon})\|_2 \leq \|\nabla\phi(\boldsymbol{w}_{n:1})\|_2 + \left(\|\boldsymbol{w}_{n:1}\|_2^{1/n} + \epsilon\right)^n - \|\boldsymbol{w}_{n:1}\|_2 \leq \|\nabla\phi(\boldsymbol{w}_{n:1})\|_2 + 2n\epsilon.
$$

Regarding property *(iii)*, for any $\mathcal{J} \subseteq [n]$ we have that:

$$
\begin{aligned}
\textstyle\prod_{j\in\mathcal{J}} \|W_{j,\epsilon}\|_F &= \textstyle\prod_{j\in\mathcal{J}} \|W_j + W_{j,\epsilon} - W_j\|_F \\
&\leq \textstyle\prod_{j\in\mathcal{J}} \left(\|W_j\|_F + \|W_{j,\epsilon} - W_j\|_F\right) \\
&\leq \textstyle\prod_{j\in\mathcal{J}} \left(\|W_j\|_F + \epsilon\right) \\
&= \textstyle\prod_{j\in\mathcal{J}} \left(\|\boldsymbol{w}_{n:1}\|_2^{1/n} + \epsilon\right) \\
&= \left(\|\boldsymbol{w}_{n:1}\|_2^{1/n} + \epsilon\right)^{|\mathcal{J}|} \\
&= \left(\left(\|\boldsymbol{w}_{n:1}\|_2^{1/n} + \epsilon\right)^n\right)^{\frac{|\mathcal{J}|}{n}},
\end{aligned}
$$

where the third transition follows from the assumption $\|\boldsymbol{\theta}_\epsilon - \boldsymbol{\theta}\|_2 \leq \epsilon$, and the fourth from Equation (56). Applying Lemma 30 concludes the proof of property *(iii)*, and the entire lemma. $\square$

The following lemma analyzes the Hessian and gradient of the training loss $f(\cdot)$, bounding their spectral and Euclidean norms respectively.

**Lemma 32.** *For any weight setting* $\boldsymbol{\theta} \in \mathbb{R}^d$ *with corresponding weight matrices* $W_1 \in \mathbb{R}^{d_1,d_0}, W_2 \in \mathbb{R}^{d_2,d_1},..., W_{n-1} \in \mathbb{R}^{d_{n-1},d_{n-2}}, W_n \in \mathbb{R}^{1,d_{n-1}}$, *the following hold:*[13]

$$
(i) \quad \|\nabla^2 f(\boldsymbol{\theta})\|_s \leq n \max_{\substack{\mathcal{J}\subseteq[n] \\ |\mathcal{J}|=n-1}} \prod_{j\in\mathcal{J}} \|W_j\|_F^2 + 2n\|\nabla\phi(\boldsymbol{w}_{n:1})\|_2 \max_{\substack{\mathcal{J}\subseteq[n] \\ |\mathcal{J}|=n-2}} \prod_{j\in\mathcal{J}} \|W_j\|_F \; ; \text{ and}
$$

$$
(ii) \quad \|\nabla f(\boldsymbol{\theta})\|_2 \leq \sqrt{n}\|\nabla\phi(\boldsymbol{w}_{n:1})\|_2 \max_{\substack{\mathcal{J}\subseteq[n] \\ |\mathcal{J}|=n-1}} \prod_{j\in\mathcal{J}} \|W_j\|_F .
$$

*Proof.* Let $\Delta W_1 \in \mathbb{R}^{d_1,d_0}, \Delta W_2 \in \mathbb{R}^{d_2,d_1},..., \Delta W_{n-1} \in \mathbb{R}^{d_{n-1},d_{n-2}}, \Delta W_n \in \mathbb{R}^{1,d_{n-1}}$.

We begin with property *(i)*. By Lemma 1 we have that:

$$
\begin{aligned}
\nabla^2 f(\boldsymbol{\theta})[\Delta W_1,...,\Delta W_n] = \nabla^2\phi(W_{n:1})\Big[\textstyle\sum_{j=1}^n W_{n:j+1}(\Delta W_j)W_{j-1:1}\Big] \\
+ 2\mathrm{Tr}\Big(\nabla\phi(W_{n:1})^\top \textstyle\sum_{1\leq j<j'\leq n} W_{n:j'+1}(\Delta W_{j'})W_{j'-1:j+1}(\Delta W_j)W_{j-1:1}\Big),
\end{aligned} \tag{57}
$$

where $W_{j':j}$, for any $j,j' \in \{1,2,...,n\}$, is defined as $W_{j'}W_{j'-1}\cdots W_j$ if $j \leq j'$, and as an identity matrix (with size to be inferred by context) otherwise. We will upper bound each of the two summands on the right-hand side of Equation (57). We bound the first summand as follows:

$$
\begin{aligned}
&\nabla^2 \phi(W_{n:1})\Big[\sum_{j=1}^n W_{n:j+1}(\Delta W_j)W_{j-1:1}\Big] \\
&= \Big\|\sum_{j=1}^n W_{n:j+1}(\Delta W_j)W_{j-1:1}\Big\|_F^2 \\
&\leq \Big(\sum_{j=1}^n \big\|W_{n:j+1}(\Delta W_j)W_{j-1:1}\big\|_F\Big)^2 \\
&\leq n\sum_{j=1}^n \big\|W_{n:j+1}(\Delta W_j)W_{j-1:1}\big\|_F^2 \\
&\leq n\sum_{j=1}^n \big\|W_n\big\|_F^2\cdots\big\|W_{j+1}\big\|_F^2\big\|\Delta W_j\big\|_F^2\big\|W_{j-1}\big\|_F^2\cdots\big\|W_1\big\|_F^2 \\
&\leq n \max_{\substack{\mathcal{J}\subseteq[n]\\|\mathcal{J}|=n-1}} \prod_{j\in\mathcal{J}}\|W_j\|_F^2 \cdot \sum_{j=1}^n\big\|\Delta W_j\big\|_F^2,
\end{aligned}
\tag{58}
$$

where the first transition follows from the fact that the Hessian of $\phi(\cdot)$ is an identity (since $\phi(W) = \frac{1}{2}\|W - \Lambda_{yx}\|_F^2 + c$), the second trasition follows from the triangle inequality, the third trasition follows from the one-norm of a vector in $\mathbb{R}^n$ being no greater than $\sqrt{n}$ times its Euclidean norm, and the fourth transition follows from sub-multiplicativity of Frobenius norm. Moving on to bounding the second summand on the right-hand side of Equation (57):

$$
\begin{aligned}
&2\mathrm{Tr}\Big(\nabla\phi(W_{n:1})^\top \sum_{1\leq j<j'\leq n} W_{n:j'+1}(\Delta W_{j'})W_{j'-1:j+1}(\Delta W_j)W_{j-1:1}\Big) \\
&\leq 2\|\nabla\phi(W_{n:1})\|_F \Big\|\sum_{1\leq j<j'\leq n} W_{n:j'+1}(\Delta W_{j'})W_{j'-1:j+1}(\Delta W_j)W_{j-1:1}\Big\|_F \\
&\leq 2\|\nabla\phi(W_{n:1})\|_F \sum_{1\leq j<j'\leq n} \big\|W_{n:j'+1}(\Delta W_{j'})W_{j'-1:j+1}(\Delta W_j)W_{j-1:1}\big\|_F \\
&\leq 2\|\nabla\phi(W_{n:1})\|_F \sum_{1\leq j<j'\leq n} \big\|\Delta W_{j'}\big\|_F\big\|\Delta W_j\big\|_F \cdot \prod_{j''\in[n]/\{j,j'\}}\big\|W_{j''}\big\|_F \\
&\leq 2\|\nabla\phi(W_{n:1})\|_F \max_{\substack{\mathcal{J}\subseteq[n]\\|\mathcal{J}|=n-2}} \prod_{j\in\mathcal{J}}\big\|W_j\big\|_F \cdot \sum_{1\leq j<j'\leq n}\big\|\Delta W_{j'}\big\|_F\big\|\Delta W_j\big\|_F,
\end{aligned}
$$

where the first transition follows from Cauchy-Schwartz inequality, the second and third from sub-additivity and sub-multiplicativity of Frobenius norm respectively. It holds that:

$$
\sum_{1\leq j<j'\leq n}\|\Delta W_{j'}\|_F\|\Delta W_j\|_F \leq \Big(\sum_{j=1}^n\|\Delta W_j\|_F\Big)^2 \leq n\sum_{j=1}^n\|\Delta W_j\|_F^2,
$$

where the last transition follows from the fact that the one-norm of a vector in $\mathbb{R}^n$ is never greater than $\sqrt{n}$ times its Euclidean norm. This leads us to:

$$
\begin{aligned}
&2\mathrm{Tr}\Big(\nabla\phi(W_{n:1})^\top \sum_{1\leq j<j'\leq n} W_{n:j'+1}(\Delta W_{j'})W_{j'-1:j+1}(\Delta W_j)W_{j-1:1}\Big) \\
&\leq 2n\|\nabla\phi(W_{n:1})\|_F \max_{\substack{\mathcal{J}\subseteq[n]\\|\mathcal{J}|=n-2}} \prod_{j\in\mathcal{J}}\big\|W_j\big\|_F \cdot \sum_{j=1}^n\|\Delta W_j\|_F^2.
\end{aligned}
\tag{59}
$$

Plugging Equations (58) and (59) into Equation (57), we obtain:

$$
\nabla^2 f(\boldsymbol{\theta})[\Delta W_1,..,\Delta W_n] \leq
$$
$$
\Big(n \max_{\substack{\mathcal{J}\subseteq[n]\\|\mathcal{J}|=n-1}} \prod_{j\in\mathcal{J}}\|W_j\|_F^2 + 2n\|\nabla\phi(W_{n:1})\|_F \max_{\substack{\mathcal{J}\subseteq[n]\\|\mathcal{J}|=n-2}} \prod_{j\in\mathcal{J}}\|W_j\|_F\Big)\sum_{j=1}^n\|\Delta W_j\|_F^2.
$$

This proves property *(i)*.

Moving on to property *(ii)*, we overload notation by allowing the function $f(\cdot)$ to intake the tuple $(W_1, W_2, ..., W_n)$ (in which case $W_1, ..., W_n$ are arranged as $\boldsymbol{\theta}$, and the value $f(\boldsymbol{\theta})$ is returned). In Appendix A of [4] it is shown that:

$$
\nabla f(W_1,...,W_n) =
$$
$$
\Big((W_{n:2})^\top \nabla\phi(W_{n:1}),..,(W_{n:j+1})^\top \nabla\phi(W_{n:1})(W_{j-1:1})^\top,..,\nabla\phi(W_{n:1})(W_{n-1:1})^\top\Big).
$$

It follows that:

$$\|\nabla f(\boldsymbol{\theta})\|_2^2 = \|\nabla f(W_1,...,W_n)\|_{Frobenius}^2$$
$$= \sum_{j=1}^{n} \left\|(W_{n:j+1})^\top \nabla \phi(W_{n:1})(W_{j-1:1})^\top\right\|_F^2$$
$$\leq \sum_{j=1}^{n} \left\|\nabla \phi(W_{n:1})\right\|_F^2 \prod_{i \in [n]/\{j\}} \|W_j\|_F^2$$
$$\leq n \left\|\nabla \phi(W_{n:1})\right\|_F^2 \max_{\substack{\mathcal{J} \subseteq [n] \\ |\mathcal{J}|=n-1}} \prod_{j \in \mathcal{J}} \|W_j\|_F^2 \,,$$

where the second transition follows from sub-multiplicativity of Frobenius norm. Taking square root of both sides of the inequality concludes the proof of property *(ii)*, and the entire lemma. $\square$

Combining Lemmas 31 and 32, Lemma 33 below establishes the smoothness and Lipschitz constants $\beta_{t,\epsilon} = 16n$ and $\gamma_{t,\epsilon} = 6\sqrt{n}$ respectively.

**Lemma 33.** *It holds that* $\sup_{\boldsymbol{q} \in \mathcal{D}_{t,\epsilon}} \|\nabla^2 f(\boldsymbol{q})\|_s \leq 16n$ *and* $\sup_{\boldsymbol{q} \in \mathcal{D}_{t,\epsilon}} \|\nabla f(\boldsymbol{q})\|_2 \leq 6\sqrt{n}.$

*Proof.* Under the conditions and notations of Lemma 31, for any $\mathcal{J} \subseteq [n]$:

$$\prod_{j \in \mathcal{J}} \|W_{j,\epsilon}\|_F \leq \left(\|\boldsymbol{w}_{n:1}(t')\|_2 + 2n\epsilon\right)^{\frac{|\mathcal{J}|}{n}}. \tag{60}$$

By Lemma 17 we have that $\boldsymbol{w}_{n:1}\big(\xi(t')\big) = \boldsymbol{u}(t')$, where $\xi(t') := \int_0^{t'} \|\boldsymbol{u}(t'')\|_2^{-(1-2/n)} \, dt''$. $\xi(\cdot)$ is unbounded since $\|\boldsymbol{u}(\cdot)\|_2 < 1$ by property *(iv)* of Lemma 19. This implies $\|\boldsymbol{w}_{n:1}(t')\|_2 < 1$, which together with the fact that by definition $\epsilon \leq 1/2n$, means:

$$\prod_{j \in \mathcal{J}} \|W_{j,\epsilon}\|_F \leq \left(1+1\right)^{\frac{|\mathcal{J}|}{n}} \leq 2. \tag{61}$$

It holds that:

$$\|\nabla \phi(\boldsymbol{w}_{n:1,\epsilon})\|_2 \leq$$
$$\|\nabla \phi(\boldsymbol{w}_{n:1}(t'))\|_2 + 2n\epsilon = \|\boldsymbol{w}_{n:1}(t') - \boldsymbol{\lambda}_{yx}\|_2 + 2n\epsilon \leq \|\boldsymbol{w}_{n:1}(t')\|_2 + \|\boldsymbol{\lambda}_{yx}\|_2 + 2n\epsilon \leq 3, \tag{62}$$

where the first transition follows from Lemma 31, and the last from $\|\boldsymbol{w}_{n:1}(t')\|_2 < 1$, $\|\boldsymbol{\lambda}_{yx}\|_2 = 1$ and $\epsilon \leq 1/2n$. We conclude the proof by plugging Equations (61) and (62) into the results of Lemma 32, while noticing that arbitrary $t' \geq 0$ and $\boldsymbol{\theta}_\epsilon$ account for all $\boldsymbol{q} \in \mathcal{D}_{t,\epsilon}$. $\square$

Lemma 34 below employs Lemma 2 from our analysis in Section 4, along with Lemma 31 above, for deriving a lower bound on the minimal eigenvalue of the Hessian (of the training loss $f(\cdot)$) in the vicinity of a point along the gradient flow trajectory.

**Lemma 34.** *For all* $t' \geq 0$:

$$\inf_{\substack{\mathbf{q} \in \mathbb{R}^d \\ \|\mathbf{q}-\boldsymbol{\theta}(t')\|_2 \leq \epsilon}} \lambda_{min}(\nabla^2 f(\mathbf{q})) \geq -(n-1)\big(\|\nabla \phi(\boldsymbol{w}_{n:1}(t'))\|_2 + 2n\epsilon\big)\big(\|\boldsymbol{w}_{n:1}(t')\|_2 + 2n\epsilon\big)^{1-\frac{2}{n}},$$

*where* $\lambda_{min}(\nabla^2 f(\mathbf{q}))$ *stands for the minimal eigenvalue of* $\nabla^2 f(\mathbf{q})$.

*Proof.* Let $\boldsymbol{\theta}_\epsilon \in \mathbb{R}^d$ be a weight setting satisfying $\|\boldsymbol{\theta}_\epsilon - \boldsymbol{\theta}(t')\|_2 \leq \epsilon$. Denote by $W_{1,\epsilon} \in \mathbb{R}^{d_1,d_0}, W_{2,\epsilon} \in \mathbb{R}^{d_2,d_1},...,W_{n-1,\epsilon} \in \mathbb{R}^{d_{n-1},d_{n-2}}, W_{n,\epsilon} \in \mathbb{R}^{1,d_{n-1}}$ the weight matrices constituting $\boldsymbol{\theta}_\epsilon$, and by $\boldsymbol{w}_{n:1,\epsilon} \in \mathbb{R}^{d_0}$ the corresponding end-to-end matrix $W_{n,\epsilon} W_{n-1,\epsilon} \cdots W_{1,\epsilon}$ (in vectorized form). Lemma 2 ensures:

$$\lambda_{\min}(\nabla^2 f(\boldsymbol{\theta}_\epsilon)) \geq -(n-1)\|\nabla \phi(\boldsymbol{w}_{n:1,\epsilon})\|_2 \max_{\substack{\mathcal{J} \subseteq [n] \\ |\mathcal{J}|=n-2}} \prod_{j \in \mathcal{J}} \|W_{j,\epsilon}\|_s \,.$$

We conclude the proof by bounding spectral norms with Frobenius norms, and applying properties *(ii)* and *(iii)* from Lemma 31. $\square$

Lemma 34 implies that, under the notations of Theorem 3, we may choose the function $m(\cdot)$ to be as follows:

$$m:[0,t]\to\mathbb{R}\ ,\ m(t')=(n-1)(\|\nabla\phi(\boldsymbol{w}_{n:1}(t'))\|_2+2n\epsilon)(\|\boldsymbol{w}_{n:1}(t')\|_2+2n\epsilon)^{1-\frac{2}{n}}.\qquad(63)$$

Lemma 35 below bounds the integral of this choice of $m(\cdot)$ in accordance with Equation (18) (recall that the scalar $\nu$ there, defined in the preceding text, coincides with the value taken by the function $\nu(\cdot)$ from Definition 3 at zero). For doing so, it makes use of the reparameterized trajectory $\boldsymbol{u}(\cdot)$, and splits the reparameterized integral into two parts corresponding to two time intervals: before exponentially fast convergence is guaranteed to have commenced (*i.e.* until time $t_a+t_c$ — see Subsubappendix J.8.7), and afterwards.

**Lemma 35.** *With the function $m(\cdot)$ defined by Equation* (63)*, Equation* (18) *is satisfied.*

*Proof.* We apply a change of variable using the (continuously differentiable and strictly increasing) function $\xi(\cdot)$ defined in Lemma 17:

$$\int_0^t m(t')dt'=\int_{\xi^{-1}(0)}^{\xi^{-1}(t)}m\big(\xi(t')\big)\tfrac{d}{dt'}\xi(t')dt'.$$

Notice that $\xi(0)=0$ and $\frac{d}{dt'}\xi(t')=\|\boldsymbol{u}(t')\|^{-(1-2/n)}$. Plugging this and the definition of m(.) (Equation (63)) into the above leads to:

$$\int_0^t m(t')dt'=\int_0^{\xi^{-1}(t)}(n-1)\Big(\big\|\nabla\phi\big(\boldsymbol{w}_{n:1}\big(\xi(t')\big)\big)\big\|_2+2n\epsilon\Big)\Big(\big\|\boldsymbol{w}_{n:1}\big(\xi(t')\big)\big\|_2+2n\epsilon\Big)^{1-\frac{2}{n}}\big\|\boldsymbol{u}(t')\big\|^{\frac{2}{n}-1}dt'.$$

Since (by Lemma 17) $\boldsymbol{w}_{n:1}(\xi(t))=\boldsymbol{u}(t)$, we have that:

$$\int_0^t m(t')dt'=\int_0^{\xi^{-1}(t)}(n-1)\Big(\big\|\nabla\phi\big(\boldsymbol{u}(t')\big)\big\|_2+2n\epsilon\Big)\Big(\big\|\boldsymbol{u}(t')\big\|_2+2n\epsilon\Big)^{1-\frac{2}{n}}\big\|\boldsymbol{u}(t')\big\|^{\frac{2}{n}-1}dt'.$$

Recall the notation $u_{\min}:=\inf_{t\geq 0}\|\boldsymbol{u}(t)\|_2$ and that, by Lemma 21, $u_{\min}>0$. It holds that:

$$\int_0^t m(t')dt'=\int_0^{\xi^{-1}(t)}(n-1)\Big(\big\|\nabla\phi\big(\boldsymbol{u}(t')\big)\big\|_2+2n\epsilon\Big)\Big(1+2n\epsilon\|\boldsymbol{u}(t')\|^{-1}\Big)^{1-\frac{2}{n}}dt'$$

$$\leq\int_0^{\xi^{-1}(t)}(n-1)\Big(\big\|\nabla\phi\big(\boldsymbol{u}(t')\big)\big\|_2+2n\epsilon\Big)\Big(1+2n\epsilon\|\boldsymbol{u}(t')\|^{-1}\Big)dt'$$

$$\leq\int_0^{\xi^{-1}(t)}(n-1)\Big(\big\|\nabla\phi\big(\boldsymbol{u}(t')\big)\big\|_2+2n\epsilon\Big)\Big(1+2n\epsilon u_{\min}^{-1}\Big)dt'$$

$$=(n-1)\Big(1+2n\epsilon u_{\min}^{-1}\Big)\Big(\textstyle\int_0^{\xi^{-1}(t)}\big\|\nabla\phi\big(\boldsymbol{u}(t')\big)\big\|_2 dt'+2n\epsilon\xi^{-1}(t)\Big).$$

Per Lemma 19 we know that $\|\boldsymbol{u}(t')\|_2<1$ for all $t'\geq 0$. Thus, by the definition of $\xi(\cdot)$, for all $t'\geq 0$ it holds that $\xi(t')\geq t'$, which (since $\xi(\cdot)$ is strictly increasing) implies $\xi^{-1}(t)\leq t$. This leads to:

$$\int_0^t m(t')dt'$$

$$\leq(n-1)\Big(1+2n\epsilon u_{\min}^{-1}\Big)\Big(\textstyle\int_0^t\big\|\nabla\phi\big(\boldsymbol{u}(t')\big)\big\|_2 dt'+2n\epsilon t\Big)$$

$$=(n-1)\textstyle\int_0^t\big\|\nabla\phi\big(\boldsymbol{u}(t')\big)\big\|_2 dt'+(n-1)2n\epsilon t+(n-1)2n\epsilon u_{\min}^{-1}\Big(\textstyle\int_0^t\big\|\nabla\phi\big(\boldsymbol{u}(t')\big)\big\|_2 dt'+2n\epsilon t\Big)$$

$$\leq(n-1)\textstyle\int_0^t\big\|\nabla\phi\big(\boldsymbol{u}(t')\big)\big\|_2 dt'+2n^2\epsilon t+4n^3\epsilon^2 u_{\min}^{-1}t+2n^2\epsilon u_{\min}^{-1}\textstyle\int_0^t\big\|\nabla\phi\big(\boldsymbol{u}(t')\big)\big\|_2 dt'.$$

It holds that $\|\nabla\phi\big(\boldsymbol{u}(t')\big)\|_2=\|\boldsymbol{u}(t')-\boldsymbol{\lambda}_{yx}\|_2\leq\|\boldsymbol{u}(t')\|_2+\|\boldsymbol{\lambda}_{yx}\|_2\leq 2$ for all $t'\geq 0$ (recall that $\|\boldsymbol{\lambda}_{yx}\|_2=1$ by assumption). Thus:

$$\int_0^t m(t')dt'\leq(n-1)\textstyle\int_0^t\big\|\nabla\phi\big(\boldsymbol{u}(t')\big)\big\|_2 dt'+2n^2\epsilon t+4n^3\epsilon^2 u_{\min}^{-1}t+2n^2\epsilon u_{\min}^{-1}\cdot 2t$$

$$\leq(n-1)\textstyle\int_0^t\big\|\nabla\phi\big(\boldsymbol{u}(t')\big)\big\|_2 dt'+3\cdot\max\Big\{2n^2\epsilon t,4n^3\epsilon^2 u_{\min}^{-1}t,4n^2\epsilon u_{\min}^{-1}t\Big\}\qquad(64)$$

$$\leq(n-1)\textstyle\int_0^t\big\|\nabla\phi\big(\boldsymbol{u}(t')\big)\big\|_2 dt'+3\cdot 4n^3\epsilon u_{\min}^{-1}t.$$

We may bound the latter integral as follows:

$$\int_0^t \left\|\nabla\phi\big(\boldsymbol{u}(t')\big)\right\|_2 dt' \le \int_0^\infty \left\|\nabla\phi\big(\boldsymbol{u}(t')\big)\right\|_2 dt'$$
$$= \int_0^{t_a+t_c} \left\|\nabla\phi\big(\boldsymbol{u}(t')\big)\right\|_2 dt' + \int_{t_a+t_c}^\infty \left\|\nabla\phi\big(\boldsymbol{u}(t')\big)\right\|_2 dt'$$
$$= \int_0^{t_a+t_c} \left\|\boldsymbol{u}(t')-\boldsymbol{\lambda}_{yx}\right\|_2 dt' + \int_0^\infty \left\|\boldsymbol{u}(t_a+t_c+t')-\boldsymbol{\lambda}_{yx}\right\|_2 dt',$$

where $t_a$ and $t_c$ are given by Definitions 5 and 6 respectively. Notice that $\left\|\boldsymbol{u}(t')-\boldsymbol{\lambda}_{yx}\right\|_2$ is monotonically non-increasing (since $\boldsymbol{u}(\cdot)$ is a monotonic reparameterization of $\boldsymbol{w}_{n:1}(\cdot)$, and gradient flow monotonically non-increases the objective it optimizes). Applying this fact, as well as Lemma 25, we obtain:

$$\int_0^t \left\|\nabla\phi\big(\boldsymbol{u}(t')\big)\right\|_2 dt' \le \int_0^{t_a+t_c} \left\|\boldsymbol{u}(0)-\boldsymbol{\lambda}_{yx}\right\|_2 dt' + \tfrac{6}{5}\int_0^\infty \exp\big(-\tfrac{2n}{3(n+1)}t'\big)dt'$$
$$= \left\|\boldsymbol{u}(0)-\boldsymbol{\lambda}_{yx}\right\|_2 \big(t_a+t_c\big) + \tfrac{6}{5}\cdot\tfrac{3(n+1)}{2n}$$
$$\le \tfrac{6}{5}\big(t_a+t_c\big)+3,$$

where the last transition follows from the assumptions $\|\boldsymbol{w}_{n:1}(0)\|_2 \le 0.2$ and $\|\boldsymbol{\lambda}_{yx}\|_2 = 1$. Plug in the definitions of $t_a$ and $t_c$ (Definitions 5 and 6 respectively):

$$\int_0^t \left\|\nabla\phi\big(\boldsymbol{u}(t')\big)\right\|_2 dt'$$
$$\le \tfrac{6}{5}\Big(\tfrac{1}{2}\ln\big(\max\big\{5\cdot\tfrac{1-\nu(0)}{1+\nu(0)},1\big\}\big)+\tfrac{3}{2n}\ln\big(\tfrac{2n}{3u_{\min}}\big)\Big)+3$$
$$= \tfrac{3}{5}\ln\big(\max\big\{5\cdot\tfrac{1-\nu(0)}{1+\nu(0)},1\big\}\big)+\tfrac{9}{5n}\ln\big(\tfrac{2n}{3u_{\min}}\big)+3 \tag{65}$$
$$= \tfrac{3}{5n}\ln\Big(\max\big(\big\{5\cdot\tfrac{1-\nu(0)}{1+\nu(0)},1\big\}\big)^n\cdot\big(\tfrac{2n}{3u_{\min}}\big)^3\cdot e^{5n}\Big)$$
$$\le \tfrac{3}{5n}\ln\Big(5^n\max\big(\big\{\tfrac{1-\nu(0)}{1+\nu(0)},1\big\}\big)^n\cdot\big(\tfrac{2n}{3}\big)^3\max\big(\big\{\tfrac{3}{2}\cdot\tfrac{1-\nu(0)}{1+\nu(0)},1\big\}\big)^{3n}\|\boldsymbol{w}_{n:1}(0)\|_2^{-3}\cdot e^{5n}\Big)$$
$$\le \tfrac{3}{5n}\ln\Big(n^3\|\boldsymbol{w}_{n:1}(0)\|_2^{-3}e^{8n}\max\big(\big\{\tfrac{1-\nu(0)}{1+\nu(0)},1\big\}\big)^{4n}\Big),$$

where the fourth transition follows from Lemma 21. Plug Equation (65) into Equation (64):

$$\int_0^t m(t')dt'$$
$$\le \tfrac{3(n-1)}{5n}\ln\Big(n^3\|\boldsymbol{w}_{n:1}(0)\|_2^{-3}e^{8n}\max\big(\big\{\tfrac{1-\nu(0)}{1+\nu(0)},1\big\}\big)^{4n}\Big)+12n^3\epsilon u_{\min}^{-1}t$$
$$\le \ln\Big(n^2\|\boldsymbol{w}_{n:1}(0)\|_2^{-2}e^{5(n-1)}\max\big(\big\{\tfrac{1-\nu(0)}{1+\nu(0)},1\big\}\big)^{\frac{5}{2}(n-1)}\Big)+15n^3 u_{\min}^{-1}\epsilon t.$$

We conclude the proof with the help of Lemma 21:

$$\int_0^t m(t')dt' \le \frac{15n^3\max\big(\big\{\tfrac{3}{2}\cdot\tfrac{1-\nu(0)}{1+\nu(0)},1\big\}\big)^n t\epsilon}{\|\boldsymbol{w}_{n:1}(0)\|_2}+\ln\Big(\frac{n^2 e^{5(n-1)}\max\big(\big\{\tfrac{1-\nu(0)}{1+\nu(0)},1\big\}\big)^{\frac{5}{2}(n-1)}}{\|\boldsymbol{w}_{n:1}(0)\|_2^2}\Big).$$

$\square$

### J.8.10 Conclusion

Lemmas 12, 29, 33 and 35, along with the fact that by the definition of $m(\cdot)$ (Equation (63)) it is non-negative, together form a complete proof for Proposition 3. $\square$

## J.9 Proof of Theorem 4

### J.9.1 Sketch

*Proof sketch (for complete proof see Subappendix J.9).* The proof calls Proposition 3 with $\bar{\epsilon}$ and $\epsilon$ small enough such that for any $t > 0$ and $\mathbf{q}' \in \mathbb{R}^d$, if gradient flow at time $t$ is $\bar{\epsilon}$-optimal (meaning $f(\boldsymbol{\theta}(t))-\min_{\mathbf{q}\in\mathbb{R}^d}f(\mathbf{q}) \le \bar{\epsilon}$) and is $\epsilon$-approximated by $\mathbf{q}'$ (*i.e.* $\|\mathbf{q}'-\boldsymbol{\theta}(t)\|_2 \le \epsilon$), then $\mathbf{q}'$ is $\tilde{\epsilon}$-optimal ($f(\mathbf{q}')-\min_{\mathbf{q}\in\mathbb{R}^d}f(\mathbf{q}) \le \tilde{\epsilon}$). The proposition implies that gradient flow is $\bar{\epsilon}$-optimal at the time $\bar{t}$ given in Equation (17). Since gradient flow monotonically non-increases $f(\cdot)$, it is $\bar{\epsilon}$-optimal at any time after $\bar{t}$

as well. With $\eta$ and $k$ adhering to Equations (19) and (20) respectively, we have $k\eta \geq \bar{t}$, so it suffices to show that when its step size is $\eta$, the first $k$ iterates of gradient descent $\epsilon$-approximate the trajectory of gradient flow up to time $k\eta$. This follows directly from delivering to Theorem 3 the geometric results of Proposition 3 (bound on integral of minimal eigenvalue of the Hessian, as well as smoothness and Lipschitz constants) corresponding to $\mathcal{D}_{k\eta,\epsilon}$ — $\epsilon$-neighborhood of gradient flow trajectory up to time $k\eta$. $\qquad\square$

### J.9.2  Complete Proof

Let $\tilde{\epsilon} > 0$, and consider $\eta > 0$ and $k \in \mathbb{N}$ adhering to Equations (19) and (20) respectively. We would like to show that with step size $\eta$, iterate $k$ of gradient descent is $\tilde{\epsilon}$-optimal, *i.e.* $f(\boldsymbol{\theta}_k) - \min_{\mathbf{q} \in \mathbb{R}^d} f(\mathbf{q}) \leq \tilde{\epsilon}$. Without loss of generality, we may assume $\tilde{\epsilon} \leq 1$ (a proof that is valid for $\tilde{\epsilon} = 1$ automatically accounts for $\tilde{\epsilon} > 1$ as well). Define:

$$\bar{\epsilon} := \tilde{\epsilon}/2 \ , \ \epsilon := \frac{\|W_{n:1,0}\|_F \tilde{\epsilon}}{15 n^3 \left( \max\left\{ 1, \frac{3}{2} \cdot \frac{1-\nu}{1+\nu} \right\} \right)^n k\eta} \ . \tag{66}$$

Invoking Proposition 3 with initial point $\boldsymbol{\theta}_s = \boldsymbol{\theta}_0$, time $t = k\eta$ and $\bar{\epsilon}$, $\epsilon$ as defined above (note that $\epsilon \in (0, 1/(2n)]$), we obtain that the gradient flow trajectory emanating from $\boldsymbol{\theta}_0$ is defined over infinite time, and with $\boldsymbol{\theta} : [0, \infty) \to \mathbb{R}^d$ representing this trajectory, the following time $\bar{t}$ satisfies $f(\boldsymbol{\theta}(\bar{t})) - \min_{\mathbf{q} \in \mathbb{R}^d} f(\mathbf{q}) \leq \bar{\epsilon}$:

$$\bar{t} = \frac{2n \left( \max\left\{ 1, \frac{3}{2} \cdot \frac{1-\nu}{1+\nu} \right\} \right)^n}{\|W_{n:1,0}\|_F} \ln\left( \frac{15 n \max\left\{ 1, \frac{1-\nu}{1+\nu} \right\}}{\|W_{n:1,0}\|_F \min\{1, 2\bar{\epsilon}\}} \right) \ . \tag{67}$$

Moreover, we obtain that under the notations of Theorem 3, in correspondence with $\mathcal{D}_{k\eta,\epsilon}$ ($\epsilon$-neighborhood of gradient flow trajectory up to time $k\eta$) are the smoothness and Lipschitz constants $\beta_{k\eta,\epsilon} = 16n$ and $\gamma_{k\eta,\epsilon} = 6\sqrt{n}$ respectively, and the following (upper) bound on the integral of (minus) the minimal eigenvalue of the Hessian:

$$\int_0^{k\eta} m(t') dt' \leq \frac{15 n^3 \left( \max\left\{ 1, \frac{3}{2} \cdot \frac{1-\nu}{1+\nu} \right\} \right)^n k\eta\epsilon}{\|W_{n:1,0}\|_F} + \ln\left( \frac{n^2 \left( e^2 \max\left\{ 1, \frac{1-\nu}{1+\nu} \right\} \right)^{5(n-1)/2}}{\|W_{n:1,0}\|_F^2} \right) , \tag{68}$$

where the function $m : [0, k\eta] \to \mathbb{R}$ is non-negative.

Notice that $k = \lfloor \bar{t}/\eta + 1 \rfloor$ and therefore $k\eta \geq \bar{t}$. Combining this with the fact that the gradient flow trajectory is $\bar{\epsilon}$-optimal at time $\bar{t}$, and that in general gradient flow monotonically non-increases the objective it optimizes, we infer $\bar{\epsilon}$-optimality of the gradient flow trajectory at time $k\eta$, *i.e.* $\boldsymbol{\theta}(k\eta) - \min_{\boldsymbol{q} \in \mathbb{R}^d} f(\boldsymbol{q}) \leq \bar{\epsilon}$. We will invoke Theorem 3 for showing that, in addition to being $\bar{\epsilon}$-optimal, the gradient flow trajectory at time $k\eta$ is also $\epsilon$-approximated by iterate $k$ of gradient descent, *i.e.* $\|\boldsymbol{\theta}_k - \boldsymbol{\theta}(k\eta)\|_2 \leq \epsilon$. This, along with $f(\cdot)$ being $6\sqrt{n}$-Lipschitz across $\mathcal{D}_{k\eta,\epsilon}$ ($\epsilon$-neighborhood of gradient flow trajectory up to time $k\eta$), yields the desired result — $\tilde{\epsilon}$-optimality for iterate $k$ of gradient descent:

$$\begin{aligned}
&f(\boldsymbol{\theta}_k) - \min_{\boldsymbol{q} \in \mathbb{R}^d} f(\boldsymbol{q}) \\
&= \left( f(\boldsymbol{\theta}_k) - f(\boldsymbol{\theta}(k\eta)) \right) + \left( f(\boldsymbol{\theta}(k\eta)) - \min_{\boldsymbol{q} \in \mathbb{R}^d} f(\boldsymbol{q}) \right) \\
&\leq \left( 6\sqrt{n} \|\boldsymbol{\theta}_k - \boldsymbol{\theta}(k\eta)\|_2 \right) + \left( f(\boldsymbol{\theta}(k\eta)) - \min_{\boldsymbol{q} \in \mathbb{R}^d} f(\boldsymbol{q}) \right) \\
&\leq 6\sqrt{n} \cdot \epsilon + \bar{\epsilon} \\
&\leq \tilde{\epsilon} \ ,
\end{aligned}$$

where the last transition follows from the definitions of $\epsilon$ and $\bar{\epsilon}$ (Equation (66)).

We conclude the proof by showing that indeed $\|\boldsymbol{\theta}_k - \boldsymbol{\theta}(k\eta)\|_2 \leq \epsilon$. Equation (68), the definition of $\epsilon$ (Equation (66)) and the condition $\tilde{\epsilon} \leq 1$ together imply:

$$
\begin{aligned}
\int_0^{k\eta} m(t')dt' &\leq \frac{15n^3\left(\max\left\{1, \frac{3}{2}\cdot\frac{1-\nu}{1+\nu}\right\}\right)^n k\eta\epsilon}{\|W_{n:1,0}\|_F} + \ln\left(\frac{n^2\left(e^2\max\left\{1, \frac{1-\nu}{1+\nu}\right\}\right)^{5(n-1)/2}}{\|W_{n:1,0}\|_F^2}\right) \quad (69) \\
&= \tilde{\epsilon} + \ln\left(\frac{n^2\left(e^2\max\left\{1, \frac{1-\nu}{1+\nu}\right\}\right)^{5(n-1)/2}}{\|W_{n:1,0}\|_F^2}\right) \\
&\leq 1 + \ln\left(\frac{n^2\left(e^2\max\left\{1, \frac{1-\nu}{1+\nu}\right\}\right)^{5(n-1)/2}}{\|W_{n:1,0}\|_F^2}\right) \\
&< \ln\left(\frac{3n^2\left(e^2\max\left\{1, \frac{1-\nu}{1+\nu}\right\}\right)^{5(n-1)/2}}{\|W_{n:1,0}\|_F^2}\right).
\end{aligned}
$$

Recalling the fact that $k = \lfloor \bar{t}/\eta + 1\rfloor$, the expression for $\bar{t}$ (Equation (67)), and the definition of $\bar{\epsilon}$ (Equation (66)), we have:

$$
\begin{aligned}
k\eta = \lfloor \bar{t}/\eta + 1\rfloor \eta &\leq \bar{t} + \eta = \frac{2n\left(\max\left\{1, \frac{3}{2}\cdot\frac{1-\nu}{1+\nu}\right\}\right)^n}{\|W_{n:1,0}\|_F}\ln\left(\frac{15n\max\left\{1, \frac{1-\nu}{1+\nu}\right\}}{\|W_{n:1,0}\|_F\tilde{\epsilon}}\right) + \eta \quad (70) \\
&< \frac{3n\left(\max\left\{1, \frac{3}{2}\cdot\frac{1-\nu}{1+\nu}\right\}\right)^n}{\|W_{n:1,0}\|_F}\ln\left(\frac{15n\max\left\{1, \frac{1-\nu}{1+\nu}\right\}}{\|W_{n:1,0}\|_F\tilde{\epsilon}}\right),
\end{aligned}
$$

where the last transition makes use of the upper bound on $\eta$ given in Equation (19). It holds that:

$$
\epsilon^{-1}\beta_{k\eta,\epsilon}\gamma_{k\eta,\epsilon}k\eta e^{\int_0^{k\eta}m(t')dt'}
$$

$$
< \frac{15n^3\left(\max\left\{1, \frac{3}{2}\cdot\frac{1-\nu}{1+\nu}\right\}\right)^n k\eta}{\|W_{n:1,0}\|_F\tilde{\epsilon}}\cdot 16n\cdot 6\sqrt{n}\cdot k\eta\cdot\frac{3n^2\left(e^2\max\left\{1, \frac{1-\nu}{1+\nu}\right\}\right)^{5(n-1)/2}}{\|W_{n:1,0}\|_F^2}
$$

$$
< \frac{4500n^{13/2}e^{6n-5}\left(\max\left\{1, \frac{1-\nu}{1+\nu}\right\}\right)^{(7n-5)/2}}{\|W_{n:1,0}\|_F^3\tilde{\epsilon}}(k\eta)^2
$$

$$
< \frac{4500n^{13/2}e^{6n-5}\left(\max\left\{1, \frac{1-\nu}{1+\nu}\right\}\right)^{(7n-5)/2}}{\|W_{n:1,0}\|_F^3\tilde{\epsilon}}\cdot\frac{9n^2\left(\max\left\{1, \frac{3}{2}\cdot\frac{1-\nu}{1+\nu}\right\}\right)^{2n}}{\|W_{n:1,0}\|_F^2}\left(\ln\left(\frac{15n\max\left\{1, \frac{1-\nu}{1+\nu}\right\}}{\|W_{n:1,0}\|_F\tilde{\epsilon}}\right)\right)^2
$$

$$
< \frac{n^{17/2}e^{7n+6}\left(\max\left\{1, \frac{1-\nu}{1+\nu}\right\}\right)^{(11n-5)/2}}{\|W_{n:1,0}\|_F^5\tilde{\epsilon}}\left(\ln\left(\frac{15n\max\left\{1, \frac{1-\nu}{1+\nu}\right\}}{\|W_{n:1,0}\|_F\tilde{\epsilon}}\right)\right)^2
$$

$$
\leq 1/\eta,
$$

where the first transition follows from Equation (69) and the definition of $\epsilon$ (Equation (66)); the third makes use of Equation (70); and the last is due to the upper bound on $\eta$ given in Equation (19). Rearrange the derived inequality:

$$
\eta < \frac{\epsilon}{\beta_{k\eta,\epsilon}\gamma_{k\eta,\epsilon}k\eta e^{\int_0^{k\eta}m(t')dt'}}.
$$

Since $m(\cdot)$ is non-negative, it holds that:

$$
\frac{\epsilon}{\beta_{k\eta,\epsilon}\gamma_{k\eta,\epsilon}k\eta e^{\int_0^{k\eta}m(t')dt'}} \leq \inf_{t\in(0,k\eta]}\frac{\epsilon}{\beta_{k\eta,\epsilon}\gamma_{k\eta,\epsilon}\int_0^t e^{\int_{t'}^t m(t'')dt''}dt'},
$$

and therefore:

$$
\eta < \inf_{t\in(0,k\eta]}\frac{\epsilon}{\beta_{k\eta,\epsilon}\gamma_{k\eta,\epsilon}\int_0^t e^{\int_{t'}^t m(t'')dt''}dt'}. \quad (71)
$$

We now invoke Theorem 3 with $\epsilon$ as we have defined (Equation (66)), time $\tilde{t} = k\eta$, and $\beta_{k\eta,\epsilon}$, $\gamma_{k\eta,\epsilon}$ and $m(\cdot)$ as produced by Proposition 3. Recalling that in our context gradient flow and gradient descent are initialized identically, i.e. $\boldsymbol{\theta}(0) = \boldsymbol{\theta}_0$, we conclude from Equation (71) that the first $\lfloor k\eta/\eta\rfloor = k$ iterates of gradient descent $\epsilon$-approximate the gradient flow trajectory up to time $k\eta$, i.e. $\|\boldsymbol{\theta}_{k'} - \boldsymbol{\theta}(k'\eta)\|_2 \leq \epsilon$ for all $k' \in \{1, 2, ..., k\}$. In particular $\|\boldsymbol{\theta}_k - \boldsymbol{\theta}(k\eta)\|_2 \leq \epsilon$, as required. $\qquad\square$

## J.10 Proof of Corollary 2

It suffices to show that the conditions of Theorem 4 are almost surely satisfied. Initialization is balanced by construction, and since $W_{n:1,0}$ (initial end-to-end matrix) follows the distribution $\mathcal{P}$, it almost surely has Frobenius norm no greater than $0.2$. Moreover, since $\mathcal{P}$ is continuous, and the line in $\mathbb{R}^{1,d_0}$ passing through the origin and $\Lambda_{yx}$ has (Lebesgue) measure zero, $W_{n:1,0}$ is almost surely not equal to zero and not antiparallel to $\Lambda_{yx}$. This completes the proof. □

## J.11 Proof of Proposition 4

The proof is organized as follows. Subsubappendix J.11.1 establishes preliminaries. Subsubappendixes J.11.2, J.11.3 and J.11.4 respectively analyze the trajectories of gradient flow and gradient descent in three different regions of the objective function: *(i)* "anisotropic" region where curvatures in first and second coordinates differ; *(ii)* transition region between the previous and the next; and *(iii)* "isotropic" region where curvatures in first and second coordinates are identical. Subsubappendix J.11.5 shows that the location of gradient flow at time $\tilde{t}$ is not $\epsilon$-approximated by different portions of the gradient descent trajectory. Finally, Subsubappendix J.11.6 concludes.

### J.11.1 Preliminaries

Consider an arbitrary time

$$\tilde{t} \in \left[\tfrac{2}{a}\ln\left(\tfrac{2-3\bar{\rho}/2}{\theta_{s,2}-(\bar{\rho}/2-1)}\right) + \tfrac{1}{a}\ln\left(\tfrac{2}{1-\bar{\rho}}\right), \tfrac{2}{a}\ln\left(\tfrac{2-3\bar{\rho}/2}{\theta_{s,2}-(\bar{\rho}/2-1)}\right) + \tfrac{1}{a}\ln\left(\tfrac{1+\bar{\rho}/4}{1-3\bar{\rho}/4}\right) + \tfrac{1}{a}\ln(b)\right],$$

and suppose the step size $\eta$ is greater than or equal to $\frac{10^{14}}{a}e^{-a\tilde{t}}\epsilon$. We aim to prove $\|\boldsymbol{\theta}_k - \boldsymbol{\theta}(\tilde{t})\|_2 > \epsilon$ for all $k \in \mathbb{N} \cup \{0\}$.

Since the objective function $f(\cdot)$ (defined in Equation (23)) is additively separable (can be expressed as a sum of terms, each depending on a single input variable), the dynamics in $\mathbb{R}^d$ induced by gradient flow and gradient descent can be analyzed separately for different coordinates. Lemma 36 below analyzes the dynamics in the third coordinate, establishing the sought after result for the case where $\eta$ is greater than $\frac{1}{6a}$.

**Lemma 36.** *Assume $\eta > \frac{1}{6a}$. Then $\|\boldsymbol{\theta}_k - \boldsymbol{\theta}(\tilde{t})\|_2 > \epsilon$ for all $k \in \mathbb{N} \cup \{0\}$.*

*Proof.* Denote by $\hat{\theta}(\cdot)$ the third coordinate of the gradient flow trajectory $\boldsymbol{\theta}(\cdot)$. Similarly, for any $k \in \mathbb{N} \cup \{0\}$, denote by $\hat{\theta}_k$ the third coordinate of the gradient descent iterate $\boldsymbol{\theta}_k$. For any $k \in \mathbb{N} \cup \{0\}$, it holds that:

$$|\hat{\theta}_{k+1}| = |\hat{\theta}_k - \eta\tfrac{\partial f}{\partial q_3}(\boldsymbol{\theta}_k)| = |\hat{\theta}_k - 12a\eta\hat{\theta}_k| = |\hat{\theta}_k| \cdot |1 - 12a\eta| > |\hat{\theta}_k|.$$

Thus, we may conclude $|\hat{\theta}_k| > |\hat{\theta}_0|$ for any $k \in \mathbb{N}$. The solution to the gradient flow equation of the third coordinate (*i.e.* $\frac{d}{dt}\hat{\theta}(t) = -12a\hat{\theta}(t)$) is $\hat{\theta}(t) = \hat{\theta}(0)e^{-12at}$. Recall that $\hat{\theta}(0) = \hat{\theta}_0 > 2$ and notice that $\tilde{t} \geq \frac{\ln(2)}{12a}$. For any $k \in \mathbb{N} \cup \{0\}$ we have:

$$\|\boldsymbol{\theta}(\tilde{t}) - \boldsymbol{\theta}_k\|_2 \geq |\hat{\theta}(\tilde{t}) - \hat{\theta}_k| \geq |\hat{\theta}_k| - |\hat{\theta}(\tilde{t})| \geq |\hat{\theta}_0| - |\hat{\theta}\left(\tfrac{\ln(2)}{12a}\right)| = |\hat{\theta}_0| - \tfrac{1}{2}|\hat{\theta}_0| > 1 > \epsilon.$$

□

It remains to treat the case where $\eta$ is no greater than $\frac{1}{6a}$. In the remainder of the proof we restrict our attention to this case, *i.e.* we assume $\eta \in \left[\frac{10^{14}}{a}e^{-a\tilde{t}}\epsilon, \frac{1}{6a}\right]$. Special focus will be devoted to the dynamics in the first two coordinates. Denote by $\theta(\cdot)$ and $\bar{\theta}(\cdot)$ the first and second coordinates, respectively, of the gradient flow trajectory $\boldsymbol{\theta}(\cdot)$. Similarly, for $k \in \mathbb{N} \cup \{0\}$, denote by $\theta_k$ and $\bar{\theta}_k$ the first and second coordinates, respectively, of the gradient descent iterate $\boldsymbol{\theta}_k$. The following lemma shows that in the first two coordinates, the trajectories of gradient flow and gradient descent are monotonically non-decreasing.

**Lemma 37.** *The functions $\theta(\cdot)$ and $\bar{\theta}(\cdot)$, and the series $(\theta_k)_{k=0}^{\infty}$ and $(\bar{\theta}_k)_{k=0}^{\infty}$, are all monotonically non-decreasing.*

*Proof.* The results follows from the fact that the derivative of $\varphi(\cdot)$ over $[0, \infty)$, and that of $\bar{\varphi}(\cdot)$ over $[\frac{\bar{\rho}}{2} - 1, \infty)$, are both non-positive. □

With Lemma 37 at hand, we consider three regions (in $\mathbb{R}^d$) which may be traversed by the trajectories of gradient flow and gradient descent: *(i) "anisotropic" region* $[0,z_c] \times [\bar{\rho}/2 - 1, 1 - \bar{\rho}] \times \mathbb{R}^{d-2}$, where the curvatures of $f(\cdot)$ in the first and second coordinates differ (namely, they equal $-a$ and $-a/2$ respectively); *(ii) transition region* $[0,z_c] \times [1-\bar{\rho},1) \times \mathbb{R}^{d-2}$; and *(iii) "isotropic" region* $[0,z_c] \times [1,\bar{z}_c) \times \mathbb{R}^{d-2}$, where the curvatures of $f(\cdot)$ in the first and second coordinates are identical (namely, they both equal $-a$). As we now show, throughout the above regions, the trajectories of gradient flow and gradient descent admit simple characterizations for their first coordinate.

**Lemma 38.** *It holds that* $\theta(t) = \theta(0)e^{at}$ *for all* $t \in \left[0, a^{-1}\ln\left(z_c/\theta(0)\right)\right]$, *and* $\theta_k = \theta_0(1+a\eta)^k$ *for all* $k \in \left\{0,1,...,\lceil \ln(z_c/\theta_0)/\ln(1+a\eta)\rceil\right\}$.

*Proof.* Notice that $\theta(0) \in (0,z_c)$. For any $t \in [0,\infty)$ such that $\theta(t) \in (0,z_c)$, we obtain:
$$\tfrac{d\theta}{dt}(t) = -\tfrac{d\varphi}{dz}\big(\theta(t)\big) = a\theta(t).$$

The function $t \mapsto \theta(0)e^{at}$ is a solution to this initial value problem valid through $t \in \left(0, \ln\left(\tfrac{z_c}{\theta_0}\right)/a\right)$, and from uniqueness of the solution together with continuity of $\theta(\cdot)$, we conclude that $\theta(t) = \theta(0)e^{at}$ for $t \in \left[0, \ln\left(\tfrac{z_c}{\theta_0}\right)/a\right]$.

Moving on to gradient descent. Notice that $\theta_0 \in (0,z_c)$, and for any $k \in \mathbb{N}$ such that $\theta_{k-1} \in (0,z_c)$ we have:
$$\theta_k = \theta_{k-1} - \eta \tfrac{d\varphi}{dz}(\theta_{k-1}) = \theta_{k-1} + a\eta\theta_{k-1} = \theta_{k-1}(1+a\eta).$$

It follows that $\theta_k = \theta_0(1+a\eta)^k$ for any $k \in \left\{0,1,..,\lceil \ln(\tfrac{z_c}{\theta_0})/\ln(1+a\eta)\rceil\right\}$, where by plugging in $k = \lceil \ln(\tfrac{z_c}{\theta_0})/\ln(1+a\eta)\rceil$ we obtain $\theta_{k-1} < z_c$. $\qquad\square$

Compared to the first coordinate, in the second coordinate the trajectories of gradient flow and gradient descent are more involved — analyses for the anisotropic, transition and isotropic regions are conducted in Subsubappendixes J.11.2, J.11.3 and J.11.4 respectively.

### J.11.2 Anisotropic Region

The current subsubappendix analyzes the second coordinate of the gradient flow and gradient descent trajectories throughout the anisotropic region, or more specifically, when the second coordinate is in the range $[\bar{\rho}/2 - 1, 1 - \bar{\rho}]$. Beginning with gradient flow, we recall that (by Lemma 37) the second coordinate of the trajectory is monotonically non-decreasing, and consider the time at which it exits the range $[\bar{\rho}/2 - 1, 1 - \bar{\rho}]$.

**Definition 8.** Define $t_{1-\bar{\rho}} := \inf\{t \geq 0 : \bar{\theta}(t) \geq 1 - \bar{\rho}\}$.[37]

Lemma 39 below provides an explicit expression for $t_{1-\bar{\rho}}$, and for the second coordinate of the gradient flow trajectory until this time.

**Lemma 39.** *The following hold:*

$$(i) \qquad t_{1-\bar{\rho}} = \tfrac{2}{a}\ln\big((4-3\bar{\rho})/(2\bar{\theta}(0)+2-\bar{\rho})\big) \text{ ; and}$$

$$(ii) \qquad \bar{\theta}(t) = \big(\bar{\theta}(0)-(\bar{\rho}/2-1)\big)e^{at/2}+(\bar{\rho}/2-1) \text{ for all } t \in [0,t_{1-\bar{\rho}}].$$

*Proof.* Notice that $\bar{\theta}(0) \in (\tfrac{\bar{\rho}}{2}-1, 1-\bar{\rho})$. For any $t \in [0,\infty)$ such that $\bar{\theta}(t) \in (\tfrac{\bar{\rho}}{2}-1, 1-\bar{\rho})$, we obtain:

$$\tfrac{d\bar{\theta}}{dt}(t) = -\tfrac{d\bar{\varphi}}{dz}\big(\bar{\theta}(t)\big) = \tfrac{a}{2}\big(\bar{\theta}(t)-(\tfrac{\bar{\rho}}{2}-1)\big).$$

The function $t \mapsto \big(\bar{\theta}(0) - (\tfrac{1}{2}\bar{\rho}-1)\big)e^{at/2} + \big(\tfrac{1}{2}\bar{\rho}-1\big)$ is a solution to this initial value problem valid through $t \in \big(0, \tfrac{2}{a}\ln((4-3\bar{\rho})/(2\bar{\theta}(0)+2-\bar{\rho}))\big)$, and from uniqueness of the solution together with continuity of $\bar{\theta}(\cdot)$, we conclude that $\bar{\theta}(t) = \big(\bar{\theta}(0) - (\tfrac{1}{2}\bar{\rho}-1)\big)e^{at/2} + \big(\tfrac{1}{2}\bar{\rho}-1\big)$ for $t \in \big[0, \tfrac{2}{a}\ln((4-3\bar{\rho})/(2\bar{\theta}(0)+2-\bar{\rho}))\big]$. This, along with the definition of $t_{1-\bar{\rho}}$, implies that $t_{1-\bar{\rho}} = \tfrac{2}{a}\ln((4-3\bar{\rho})/(2\bar{\theta}(0)+2-\bar{\rho}))$. $\qquad\square$

---

[37] Note that by convention, the infimum of the empty set is equal to infinity.

Moving on to gradient descent, we provide a treatment analogous to that of gradient flow. Namely, we recall that (by Lemma 37) the second coordinate of the trajectory is monotonically non-decreasing, consider the iteration at which it exits the range $[\bar{\rho}/2-1, 1-\bar{\rho})$, and present an explicit expression for the index of this iteration as well as the second coordinate of the gradient descent trajectory until the iteration is reached.

**Definition 9.** Define $k_{1-\bar{\rho}} := \inf\{k \in \mathbb{N} \cup \{0\} : \bar{\theta}_k \geq 1-\bar{\rho}\}$.[37]

**Lemma 40.** *The following hold:*

$$(i) \qquad k_{1-\bar{\rho}} = \left\lceil \left( \ln(2-3\bar{\rho}/2) - \ln(\bar{\theta}_0 + 1 - \bar{\rho}/2) \right) / \ln(1 + a\eta/2) \right\rceil ; \text{ and}$$

$$(ii) \qquad \bar{\theta}_k = \left( \bar{\theta}_0 - (\bar{\rho}/2 - 1) \right)(1 + a\eta/2)^k + (\bar{\rho}/2 - 1) \text{ for all } k \in \{0, 1, ..., k_{1-\bar{\rho}}\}.$$

*Proof.* Notice that $\bar{\theta}_0 \in (\frac{\bar{\rho}}{2} - 1, 1 - \bar{\rho})$. For any $k \in \mathbb{N}$ such that $\bar{\theta}_{k-1} \in (\frac{\bar{\rho}}{2} - 1, 1 - \bar{\rho})$, we obtain:

$$\bar{\theta}_k = \bar{\theta}_{k-1} - \eta \frac{d\bar{\varphi}}{dz}(\bar{\theta}_{k-1}) = \bar{\theta}_{k-1} + \frac{a}{2}\eta\left(\bar{\theta}_{k-1} - (\frac{\bar{\rho}}{2} - 1)\right).$$

Subtract $(\frac{\bar{\rho}}{2} - 1)$ from both sides of the equation:

$$\left(\bar{\theta}_k - (\frac{\bar{\rho}}{2} - 1)\right) = \left(\bar{\theta}_{k-1} - (\frac{\bar{\rho}}{2} - 1)\right) + \frac{a}{2}\eta\left(\bar{\theta}_{k-1} - (\frac{\bar{\rho}}{2} - 1)\right).$$

This leads us to:

$$\left(\bar{\theta}_k - (\frac{\bar{\rho}}{2} - 1)\right) = \left(\bar{\theta}_{k-1} - (\frac{\bar{\rho}}{2} - 1)\right)(1 + \frac{a}{2}\eta).$$

It follows that $\left(\bar{\theta}_k - (\frac{\bar{\rho}}{2} - 1)\right) = \left(\bar{\theta}_0 - (\frac{\bar{\rho}}{2} - 1)\right)(1 + \frac{a}{2}\eta)^k$ for any $k \in \{0, 1, ..., k_{1-\bar{\rho}}\}$. From the definition of $k_{1-\bar{\rho}}$ it must be equal to $\left\lceil \left( \ln(2 - 3\bar{\rho}/2) - \ln(\bar{\theta}_0 + 1 - \bar{\rho}/2) \right) / \ln(1 + a\eta/2) \right\rceil$, as if it is smaller we get $\bar{\theta}_{k_{1-\bar{\rho}}} < 1 - \bar{\rho}$, and if it is larger we get $\bar{\theta}_{k_{1-\bar{\rho}}-1} \geq 1 - \bar{\rho}$, both contradicting the definition of $k_{1-\bar{\rho}}$. $\qquad\square$

We conclude this subsubappendix by combining its results with Lemma 38, thereby showing that the gradient flow trajectory between initialization and time $t_{1-\bar{\rho}}$, and the gradient descent trajectory between initialization and iteration $k_{1-\bar{\rho}}$, both lie in the anisotropic region.

**Lemma 41.** *It holds that $\left(\theta(t), \bar{\theta}(t)\right) \in [0, z_c) \times [\bar{\rho}/2 - 1, 1 - \bar{\rho})$ for all $t \in [0, t_{1-\bar{\rho}})$, and $(\theta_k, \bar{\theta}_k) \in [0, z_c) \times [\bar{\rho}/2 - 1, 1 - \bar{\rho})$ for all $k \in \{0, 1, ..., k_{1-\bar{\rho}} - 1\}$.*

*Proof.* We start by proving the result for gradient flow. By assumption it holds that $\bar{\theta}(0) \in [\bar{\rho}/2 - 1, 1 - \bar{\rho})$. From monotonicity of $\bar{\theta}(\cdot)$ (Lemma 37) together with the definition of $t_{1-\bar{\rho}}$, we have that $\bar{\theta}(t) \in [\bar{\rho}/2 - 1, 1 - \bar{\rho})$ for all $t \in [0, t_{1-\bar{\rho}})$. Recall that we assume $\theta(0) \in (0.5, 1)$. By Lemma 38 we have that $\theta(t) \in [\theta(0), z_c)$ for all $t \in [0, \frac{1}{a}\ln(z_c/\theta(0)))$. By Lemma 39 $t_{1-\bar{\rho}} = \frac{2}{a}\ln\left((4 - 3\bar{\rho})/(2\bar{\theta}(0) + 2 - \bar{\rho})\right)$. Since $\frac{2}{a}\ln\left((4 - 3\bar{\rho})/(2\bar{\theta}(0) + 2 - \bar{\rho})\right) \leq \frac{1}{a}\ln(z_c/\theta(0))$ (can be verified by recalling the assumptions on $a, z_c, \bar{\rho}, \theta(0)$ and $\bar{\theta}(0)$), it follows that $\theta(t) \in [0, z_c)$ for all $t \in [0, t_{1-\bar{\rho}})$. In conclusion, we have shown that $\left(\theta(t), \bar{\theta}(t)\right) \in [0, z_c) \times [\bar{\rho}/2 - 1, 1 - \bar{\rho})$ for all $t \in [0, t_{1-\bar{\rho}})$.

Moving on to gradient descent, by assumption it holds that $\bar{\theta}_0 \in [\bar{\rho}/2 - 1, 1 - \bar{\rho})$. From monotonicity of $(\bar{\theta}_k)_{k=0}^{\infty}$ (Lemma 37) together with the definition of $k_{1-\bar{\rho}}$, we have that $\bar{\theta}_k \in [\bar{\rho}/2 - 1, 1 - \bar{\rho})$ for all $k \in \{0, 1, ..., k_{1-\bar{\rho}} - 1\}$. Recall that we assume $\theta_0 \in (0.5, 1)$. By Lemma 38 we have that $\theta_k \in [\theta(0), z_c)$ for all $k \in \left\{0, 1, ..., \left\lceil \ln(z_c/\theta_0)/\ln(1 + a\eta) \right\rceil - 1\right\}$. By Lemma 40 $k_{1-\bar{\rho}} = \left\lceil \left(\ln(2 - 3\bar{\rho}/2) - \ln(\bar{\theta}_0 + 1 - \bar{\rho}/2)\right)/\ln(1 + a\eta/2) \right\rceil$. Since $k_{1-\bar{\rho}} \leq \left\lceil \ln(z_c/\theta_0)/\ln(1 + a\eta) \right\rceil$ (can be verified by recalling the assumptions on $a, z_c, \bar{\rho}, \theta_0$ and $\bar{\theta}_0$), it follows that $\theta_k \in [0, z_c)$ for all $k \in \{0, 1, ..., k_{1-\bar{\rho}} - 1\}$. In conclusion, we have shown that $(\theta_k, \bar{\theta}_k) \in [0, z_c) \times [\bar{\rho}/2 - 1, 1 - \bar{\rho})$ for all $k \in \{0, 1, ..., k_{1-\bar{\rho}} - 1\}$. $\qquad\square$

### J.11.3 Transition Region

The current subsubappendix analyzes the second coordinate of the gradient flow and gradient descent trajectories throughout the transition region, or more specifically, when the second coordinate is in the range $[1 - \bar{\rho}, 1)$. Beginning with gradient flow, we recall that (by Lemma 37) the second coordinate of the trajectory is monotonically non-decreasing, and consider the time at which it exits the range $[1 - \bar{\rho}, 1)$.

**Definition 10.** Define $t_1 := \inf\{t \geq 0 : \bar{\theta}(t) \geq 1\}$.[37]

Using $t_{1-\bar{\rho}}$ from Definition 8, Lemma 42 below provides lower and upper bounds for $t_1$, and an upper bound for the ratio between the second coordinate of the gradient flow trajectory at time $t_1$, and its first coordinate at the same time.

**Lemma 42.** *The following hold:*

$$(i) \qquad t_{1-\bar{\rho}}+a^{-1}\ln\big((4+\bar{\rho})/(4-3\bar{\rho})\big)\leq t_1\leq t_{1-\bar{\rho}}+a^{-1}\ln\big(1/(1-\bar{\rho})\big)\text{ ; and}$$

$$(ii) \qquad \bar{\theta}(t_1)/\theta(t_1)\leq\big((\bar{\theta}(0)+1-\bar{\rho}/2)/(2-3\bar{\rho}/2)\big)^2/\theta(0).$$

*Proof.* We start by proving property *(i)*. Lemma 39 implies $\bar{\theta}(t_{1-\bar{\rho}})=1-\bar{\rho}$. Recall that by Lemma 37, $\bar{\theta}(\cdot)$ is monotonically non-decreasing. For any $t\in[0,\infty)$ such that $\bar{\theta}(t)\in[1-\bar{\rho},1]$, we have:

$$\tfrac{d\bar{\theta}}{dt}(t)=-\tfrac{d\bar{\varphi}}{dz}\big(\bar{\theta}(t)\big)=a\bar{\theta}(t)+\tfrac{a}{4\bar{\rho}}\big(\bar{\theta}(t)-1\big)^2. \tag{72}$$

By lower bounding Equation (72) we get $\tfrac{d\bar{\theta}}{dt}(t)\geq a\bar{\theta}(t)$ (which implies $t_1<\infty$). Dividing both sides of this inequality by $\bar{\theta}(t)$ and integrating over time from $t_{1-\bar{\rho}}$ until $t_1$ we have that $\bar{\theta}(t_1)\geq(1-\bar{\rho})e^{a(t_1-t_{1-\bar{\rho}})}$. From continuity of $\bar{\theta}(\cdot)$, the definition of $t_1$ and the fact that $t_1<\infty$, we have that $\boldsymbol{\theta}(t_1)=1$. This implies $(1-\bar{\rho})e^{a(t_1-t_{1-\bar{\rho}})}\leq1$. We may conclude $t_1\leq t_{1-\bar{\rho}}+a^{-1}\ln\big(1/(1-\bar{\rho})\big)$. We now turn to upper bound Equation (72). For any $t\in[0,\infty)$ such that $\bar{\theta}(t)\in[1-\bar{\rho},1]$:

$$\tfrac{d\bar{\theta}}{dt}(t)\leq a\bar{\theta}(t)+\tfrac{a}{4\bar{\rho}}\big(\bar{\rho}\big)^2=a\bar{\theta}(t)+\tfrac{a\bar{\rho}}{4}.$$

Dividing both sides of this inequality by $\bar{\theta}(t)$ and integrating over time from $t_{1-\bar{\rho}}$ until $t_1$, we have that $\bar{\theta}(t_1)\leq\big(1-\tfrac{3}{4}\bar{\rho}\big)e^{a(t_1-t_{1-\bar{\rho}})}-\tfrac{\bar{\rho}}{4}$. Since $\boldsymbol{\theta}(t_1)=1$, this implies $\big(1-\tfrac{3}{4}\bar{\rho}\big)e^{a(t_1-t_{1-\bar{\rho}})}-\tfrac{\bar{\rho}}{4}\geq1$. We may conclude $t_1\geq t_{1-\bar{\rho}}+a^{-1}\ln\big((4+\bar{\rho})/(4-3\bar{\rho})\big)$.

Moving on to property *(ii)*, by property *(i)* we know that $t_1<\infty$. Recall that $\bar{\theta}(t_1)=1$. Lemma 39 showed that $t_{1-\bar{\rho}}=\tfrac{2}{a}\ln\big((4-3\bar{\rho})/(2\bar{\theta}(0)+2-\bar{\rho})\big)$. Lemma 38 ensures $\theta(t)=\theta_0e^{at}$ for $t\in\big[0,\ln(\tfrac{z_c}{\theta(0)})/a\big]$. Notice that $t_1\leq t_{1-\bar{\rho}}+a^{-1}\ln\big(1/(1-\bar{\rho})\big)\leq\ln(\tfrac{z_c}{\theta(0)})/a$, where the first inequality follows from property *(i)*. It holds that:

$$\bar{\theta}(t_1)/\theta(t_1)=1/\big(\theta(0)e^{at_1}\big)\leq1/\big(\theta(0)e^{at_{1-\bar{\rho}}}\big)\leq\big(\tfrac{2\bar{\theta}(0)+2-\bar{\rho}}{4-3\bar{\rho}}\big)^2/\theta(0).$$

$\square$

Moving on to gradient descent, we recall that here too the second coordinate of the trajectory is monotonically non-decreasing (see Lemma 37), and consider the iteration at which this second coordinate exits the range $[1-\bar{\rho},1)$.

**Definition 11.** Define $k_1:=\inf\big\{k\in\mathbb{N}\cup\{0\}:\bar{\theta}_k\geq1\big\}$.[37]

Using $k_{1-\bar{\rho}}$ from Definition 9, Lemma 43 below provides an upper bound for $k_1$, and a lower bound for the ratio between the second coordinate of the gradient descent trajectory at iteration $k_1$, and its first coordinate at the same iteration.

**Lemma 43.** *The following hold:*

$$(i) \qquad k_1\leq k_{1-\bar{\rho}}+\big\lceil\max\big\{0,-\ln(\bar{\theta}_{k_{1-\bar{\rho}}})/\ln(1+a\eta)\big\}\big\rceil\text{ ; and}$$

$$(ii) \qquad \bar{\theta}_{k_1}/\theta_{k_1}\geq\big((\bar{\theta}_0+1-\bar{\rho}/2)/(2-3\bar{\rho}/2)\big)^2/\big(\theta_0(1-a\eta/10)\big).$$

*Proof.* We start by proving property *(i)*. By the definition of $\bar{\theta}_{k_{1-\bar{\rho}}}$ (and from the fact that it is finite from Lemma 40), we know that $\bar{\theta}_{k_{1-\bar{\rho}}}\geq1-\bar{\rho}$. Recall that by Lemma 37 $(\bar{\theta}_k)_{k=0}^{\infty}$ is monotonically non-decreasing. Notice that it is possible for $k_1$ to be equal to $k_{1-\bar{\rho}}$; this will be the case if $\bar{\theta}_{k_{1-\bar{\rho}}}\geq1$. For any $k\in\mathbb{N}$ such that $\bar{\theta}_{k-1}\in[1-\bar{\rho},1]$, we obtain:

$$\bar{\theta}_k=\bar{\theta}_{k-1}-\eta\tfrac{d\bar{\varphi}}{dz}(\bar{\theta}_{k-1})=\bar{\theta}_{k-1}+\eta\big(a\bar{\theta}_{k-1}+\tfrac{a}{4\bar{\rho}}(\bar{\theta}_{k-1}-1)^2\big)\geq\bar{\theta}_{k-1}+a\eta\bar{\theta}_{k-1}=\bar{\theta}_{k-1}(1+a\eta).$$

It follows that $\bar{\theta}_k\geq\bar{\theta}_{k_{1-\bar{\rho}}}(1+a\eta)^{k-k_{1-\bar{\rho}}}$ for any $k\in\{k_{1-\bar{\rho}},k_{1-\bar{\rho}}+1,...,k_1\}$. Plugging $k=k_{1-\bar{\rho}}+\big\lceil\max\{0,-\ln(\bar{\theta}_{k_{1-\bar{\rho}}})/\ln(1+a\eta)\}\big\rceil$ yields $\bar{\theta}_{k_{1-\bar{\rho}}}(1+a\eta)^{k-k_{1-\bar{\rho}}}\geq1$. From monotonicity

of $(\bar{\theta}_k)_{k=0}^{\infty}$, we may conclude that $k_1 \leq k_{1-\bar{\rho}} + \lceil \max\{0, -\ln(\bar{\theta}_{k_{1-\bar{\rho}}})/\ln(1+a\eta)\} \rceil$, thereby finishing the proof of property *(i)*.

Moving on to property *(ii)*, with the help of Lemma 38, Lemma 40 and property *(i)*, we obtain:

$$\frac{\theta_{k_1}}{\bar{\theta}_{k_1}} \leq \theta_{k_1}$$

$$= \theta_0(1+a\eta)^{k_1}$$

$$= \theta_0 \exp\left(\ln(1+a\eta)k_1\right)$$

$$\leq \theta_0 \exp\left(\ln(1+a\eta)\left(\left\lceil \frac{\ln(2-3\bar{\rho}/2)-\ln(\bar{\theta}_0+1-\bar{\rho}/2)}{\ln(1+a\eta/2)}\right\rceil + \left\lceil \frac{-\ln(\bar{\theta}_{k_{1-\bar{\rho}}})}{\ln(1+a\eta)}\right\rceil\right)\right)$$

$$\leq \theta_0 \exp\left(\ln(1+a\eta)\left(2 + \frac{\ln(2-3\bar{\rho}/2)-\ln(\bar{\theta}_0+1-\bar{\rho}/2)}{\ln(1+a\eta/2)} - \frac{\ln(\bar{\theta}_{k_{1-\bar{\rho}}})}{\ln(1+a\eta)}\right)\right)$$

$$= \theta_0 \left(\frac{2-3\bar{\rho}/2}{\bar{\theta}_0+1-\bar{\rho}/2}\right)^{\frac{\ln(1+a\eta)}{\ln(1+a\eta/2)}} \frac{1}{\bar{\theta}_{k_{1-\bar{\rho}}}}(1+a\eta)^2$$

$$\leq \frac{\theta_0}{1-\bar{\rho}}\left(\frac{2-3\bar{\rho}/2}{\bar{\theta}_0+1-\bar{\rho}/2}\right)^{\frac{\ln(1+a\eta)}{\ln(1+a\eta/2)}}(1+a\eta)^2.$$

Equation (3) in [56] states that $\frac{2z}{2+z} \leq \ln(1+z) \leq \frac{z}{2} \cdot \frac{2+z}{1+z}$ for all $z \geq 0$. This, along with the fact that $\eta \leq \frac{1}{6a}$ (see Subsubappendix J.11.1), leads us to $\frac{\ln(1+a\eta)}{\ln(1+a\eta/2)} \leq \left(\frac{a\eta}{2} \cdot \frac{2+a\eta}{1+a\eta}\right)/\left(\frac{a\eta}{2+a\eta/2}\right) = \frac{4+3a\eta+(a\eta)^2/2}{2+2a\eta} = 2 + \frac{(a\eta)^2/2-a\eta}{2+2a\eta} \leq 2-a\eta/3$. Thus:

$$\frac{\theta_{k_1}}{\bar{\theta}_{k_1}} \leq \frac{\theta_0}{1-\bar{\rho}}\left(\frac{2-3\bar{\rho}/2}{\bar{\theta}_0+1-\bar{\rho}/2}\right)^{2-a\eta/3}(1+a\eta)^2 = \frac{\theta_0}{1-\bar{\rho}}\left(\frac{2-3\bar{\rho}/2}{\bar{\theta}_0+1-\bar{\rho}/2}\right)^2\left(\frac{\bar{\theta}_0+1-\bar{\rho}/2}{2-3\bar{\rho}/2}\right)^{a\eta/3}(1+a\eta)^2.$$

By assumption on $\bar{\theta}_0$ and $\bar{\rho}$, namely $\bar{\theta}_0 \leq e^{-12}-1$ and $\bar{\rho} \in [0, e^{-12}/2]$, we may bound as follows:

$$\frac{\theta_{k_1}}{\bar{\theta}_{k_1}} \leq \frac{\theta_0}{1-\bar{\rho}}\left(\frac{2-3\bar{\rho}/2}{\bar{\theta}_0+1-\bar{\rho}/2}\right)^2\left(e^{-12}\right)^{a\eta/3}(1+a\eta)^2 = \frac{\theta_0}{1-\bar{\rho}}\left(\frac{2-3\bar{\rho}/2}{\bar{\theta}_0+1-\bar{\rho}/2}\right)^2 e^{-4a\eta}(1+a\eta)^2.$$

Since $1+z \leq e^z$ for all $z \geq 0$, we have that:

$$\frac{\theta_{k_1}}{\bar{\theta}_{k_1}} \leq \frac{\theta_0}{1-\bar{\rho}}\left(\frac{2-3\bar{\rho}/2}{\bar{\theta}_0+1-\bar{\rho}/2}\right)^2 \frac{1}{1+4a\eta}(1+a\eta)^2 = \frac{\theta_0}{1-\bar{\rho}}\left(\frac{2-3\bar{\rho}/2}{\bar{\theta}_0+1-\bar{\rho}/2}\right)^2\left(1-\frac{4}{1+4a\eta}a\eta\right)\left(1+2a\eta+(a\eta)^2\right).$$

Once again relying on the fact that $\eta \leq \frac{1}{6a}$, we obtain:

$$\frac{\theta_{k_1}}{\bar{\theta}_{k_1}} \leq \frac{\theta_0}{1-\bar{\rho}}\left(\frac{2-3\bar{\rho}/2}{\bar{\theta}_0+1-\bar{\rho}/2}\right)^2\left(1-\frac{12}{5}a\eta\right)\left(1+\frac{11}{5}a\eta\right) \leq \frac{\theta_0}{1-\bar{\rho}}\left(\frac{2-3\bar{\rho}/2}{\bar{\theta}_0+1-\bar{\rho}/2}\right)^2(1-a\eta/5).$$

We will show that $(1-a\eta/5)/(1-\bar{\rho}) \leq 1-a\eta/10$, thereby finishing the proof, as this leads to $\bar{\theta}_{k_1}/\theta_{k_1} \geq \left((\bar{\theta}_0+1-\bar{\rho}/2)/(2-3\bar{\rho}/2)\right)^2/\left(\theta_0(1-a\eta/10)\right)$. It holds that:

$$a\tilde{t} \leq 2\ln\left(\frac{2-3\bar{\rho}/2}{\bar{\theta}_0-(\bar{\rho}/2-1)}\right) + \ln\left(\frac{1+\bar{\rho}/4}{1-3\bar{\rho}/4}\right) + \ln(b) \leq 2\ln\left(\frac{2}{e^{-12}/4}\right) + \ln(e) + \ln(b) \leq 30 + \ln(b), \qquad (73)$$

where the first transition follows from the upper bound for $\tilde{t}$; and the second follows from the definition $\bar{\rho} := \min\{e^{-12}/2, \epsilon/2b\}$ together with the assumption $\bar{\theta}_0 \in (e^{-12}/2-1, e^{-12}-1)$. The following holds:

$$\bar{\rho} \leq \frac{\epsilon}{2b} = 10^{13}\epsilon \cdot \frac{1}{10^{13} \cdot 2b} \leq 10^{13}\epsilon \cdot \frac{1}{e^{30}b} = 10^{13}\epsilon \cdot e^{-(30+\ln(b))} \leq 10^{13}\epsilon \cdot e^{-a\tilde{t}} \leq a\eta/10,$$

where the first transition follows from the definition of $\bar{\rho}$; the fifth from Equation (73); and the last from the assumption $\eta \geq e^{-a\tilde{t}} \cdot 10^{14}\epsilon/a$. It follows that:

$$\frac{1-a\eta/5}{1-\bar{\rho}} \leq \frac{1-a\eta/5}{1-a\eta/10} = 1 - \frac{a\eta/10}{1-a\eta/10} \leq 1-a\eta/10.$$

$\square$

We conclude this subsubappendix by combining its results with Lemma 38, thereby showing that the gradient flow trajectory between times $t_{1-\bar{\rho}}$ and $t_1$, and the gradient descent trajectory between iterations $k_{1-\bar{\rho}}$ and $k_1$, both lie in the transition region.

**Lemma 44.** *It holds that* $\big(\theta(t), \bar{\theta}(t)\big) \in [0, z_c) \times [1 - \bar{\rho}, 1)$ *for all* $t \in [t_{1-\bar{\rho}}, t_1)$, *and* $(\theta_k, \bar{\theta}_k) \in [0, z_c) \times [1-\bar{\rho}, 1)$ *for all* $k \in \{k_{1-\bar{\rho}}, k_{1-\bar{\rho}}+1, ..., k_1 - 1\}$.

*Proof.* We start by proving the result for gradient flow. Lemma 39 implies $t_{1-\bar{\rho}} < \infty$, and similarly Lemma 42 implies $t_1 < \infty$. From continuity of $\bar{\theta}(\cdot)$ together with the definitions of $t_{1-\bar{\rho}}$ and $t_1$, we have that $\bar{\theta}(t_{1-\bar{\rho}}) = 1 - \bar{\rho}$ and $\bar{\theta}(t_1) = 1$. By monotonicity of $\bar{\theta}(\cdot)$ (Lemma 37) we conclude $\theta(t) \in [1 - \bar{\rho}, 1)$ for all $t \in [t_{1-\bar{\rho}}, t_1)$. Recall that we assume $\theta(0) \in (0.5, 1)$. Lemma 38 implies $\theta(t) \in [\theta(0), z_c)$ for all $t \in [0, a^{-1}\ln(z_c/\theta(0)))$. By Lemma 42 $t_1 \leq t_{1-\bar{\rho}} + a^{-1}\ln(1/(1-\bar{\rho}))$. By recalling the explicit expression for $t_{1-\bar{\rho}}$ from Lemma 39, and all the assumptions on $a$, $z_c$, $\bar{\rho}$, $\theta(0)$ and $\bar{\theta}(0)$, it can be seen that $t_{1-\bar{\rho}} + a^{-1}\ln(1/(1-\bar{\rho})) \leq a^{-1}\ln(z_c/\theta(0))$, which implies $t_1 \leq a^{-1}\ln(z_c/\theta(0))$. It follows that $\theta(t) \in [0, z_c)$ for all $t \in [t_{1-\bar{\rho}}, t_1)$. Overall we proved that $\big(\theta(t), \bar{\theta}(t)\big) \in [0, z_c) \times [1-\bar{\rho}, 1)$ for all $t \in [t_{1-\bar{\rho}}, t_1)$, as required.

Moving on to gradient descent, Lemma 40 implies $k_{1-\bar{\rho}} < \infty$, and similarly Lemma 43 implies $k_1 < \infty$. By definition of $k_{1-\bar{\rho}}$ we have that $\bar{\theta}_{k_{1-\bar{\rho}}} \geq 1 - \bar{\rho}$, and by definition of $k_1$ it holds that $\bar{\theta}_{k_1 - 1} < 1$. From monotonicity of $(\bar{\theta}_k)_{k=0}^{\infty}$ (Lemma 37) and the definitions of $k_{1-\bar{\rho}}$ and $k_1$ (from Definitions 9 and 11 respectively), we know that $\bar{\theta}_k \in [1 - \bar{\rho}, 1)$ for all $k \in \{k_{1-\bar{\rho}}, k_{1-\bar{\rho}} + 1, ..., k_1 - 1\}$. Recall that we assume $\theta_0 \in (0.5, 1)$. Lemma 38 implies $\theta_k \in [\theta_0, z_c)$ for all $k \in \{0, 1, ..., \lceil \ln(z_c/\theta_0)/\ln(1 + a\eta) \rceil - 1\}$. By Lemma 43, $k_1 \leq k_{1-\bar{\rho}} + \lceil \max\{0, -\ln(\bar{\theta}_{k_{1-\bar{\rho}}})/\ln(1+a\eta)\} \rceil$. By recalling the definition of $k_{1-\bar{\rho}}$ (Definition 9) and also its explicit expression from Lemma 40, while also recalling the assumptions on $a$, $z_c$, $\bar{\rho}$, $\theta_0$ and $\bar{\theta}_0$, it can be seen that $k_{1-\bar{\rho}} + \lceil \max\{0, -\ln(\bar{\theta}_{k_{1-\bar{\rho}}})/\ln(1+a\eta)\} \rceil \leq \lceil \ln(z_c/\theta_0)/\ln(1+a\eta) \rceil$, which implies $k_1 \leq \lceil \ln(z_c/\theta_0)/\ln(1+a\eta) \rceil$. It follows that $\theta_k \in [0, z_c)$ for all $k \in \{k_{1-\bar{\rho}}, k_{1-\bar{\rho}}+1, ..., k_1 - 1\}$. Overall we proved that $(\theta_k, \bar{\theta}_k) \in [0, z_c) \times [1-\bar{\rho}, 1)$ for all $k \in \{k_{1-\bar{\rho}}, k_{1-\bar{\rho}}+1, ..., k_1 - 1\}$. $\qquad\square$

### J.11.4 Isotropic Region

The current subsubappendix analyzes the second coordinate of the gradient flow and gradient descent trajectories throughout the isotropic region, or more specifically, when the second coordinate is in the range $[1, \bar{z}_c)$. Beginning with gradient flow, we recall that (by Lemma 37) the second coordinate of the trajectory is monotonically non-decreasing, and consider the time at which it exits the range $[1, \bar{z}_c)$.

**Definition 12.** Define $t_{\bar{z}_c} := \inf\{t \geq 0 : \bar{\theta}(t) \geq \bar{z}_c\}$.[37]

Using $t_1$ from Definition 10, Lemma 45 below provides an expression for $t_{\bar{z}_c}$, and for the second coordinate of the gradient flow trajectory between times $t_1$ and $t_{\bar{z}_c}$, *i.e.* between the time it enters the range $[1, \bar{z}_c)$ and that at which it exits.

**Lemma 45.** *The following hold:*

$$(i) \qquad t_{\bar{z}_c} = t_1 + a^{-1}\ln(\bar{z}_c); \text{ and}$$

$$(ii) \qquad \bar{\theta}(t) = e^{a(t-t_1)} \text{ for all } t \in [t_1, t_{\bar{z}_c}].$$

*Proof.* For any $t \in [0, \infty)$ such that $\bar{\theta}(t) \in [1, \bar{z}_c)$, we obtain:

$$\tfrac{d\bar{\theta}}{dt}(t) = -\tfrac{d\bar{\varphi}}{dz}\big(\bar{\theta}(t)\big) = a\bar{\theta}(t).$$

Lemma 42, the definition of $t_1$ and continuity of $\bar{\theta}(\cdot)$ together imply that $\bar{\theta}(t_1) = 1$. The function $t \mapsto e^{a(t-t_1)}$ is a solution to this initial value problem (starting at $t_1$), valid through $t \in [t_1, t_1 + a^{-1}\ln(\bar{z}_c))$, and from uniqueness of the solution together with continuity of $\bar{\theta}(\cdot)$, we conclude that $\bar{\theta}(t) = e^{a(t-t_1)}$ for $t \in [t_1, t_1 + a^{-1}\ln(\bar{z}_c)]$. This, along with the definition of $t_{\bar{z}_c}$, implies that $t_{\bar{z}_c} = t_1 + a^{-1}\ln(\bar{z}_c)$. $\quad\square$

Moving on to gradient descent, we recall that here too the second coordinate of the trajectory is monotonically non-decreasing (see Lemma 37), and consider the iteration at which this second coordinate exits the range $[1, \bar{z}_c)$.

**Definition 13.** Define $k_{\bar{z}_c} := \inf\{k \in \mathbb{N} \cup \{0\} : \bar{\theta}_k \geq \bar{z}_c\}$.[37]

Using $k_1$ from Definition 11, Lemma 46 below provides an expression for $k_{\bar{z}_c}$, and for the second coordinate of the gradient descent trajectory between iterations $k_1$ and $k_{\bar{z}_c}$, *i.e.* between the iteration where it enters the range $[1,\bar{z}_c)$ and that at which it exits.

**Lemma 46.** *It holds that:*

$$(i) \qquad k_{\bar{z}_c} = k_1 + \left\lceil \ln(\bar{z}_c/\bar{\theta}_{k_1})/\ln(1+a\eta) \right\rceil \text{; and}$$

$$(ii) \qquad \bar{\theta}_k = \bar{\theta}_{k_1}(1+a\eta)^{k-k_1} \text{ for all } k \in \{k_1, k_1+1, ..., k_{\bar{z}_c}\}.$$

*Proof.* Lemma 43 and the definition of $k_1$ imply that $\bar{\theta}_{k_1} \geq 1$. Recall that by Lemma 37 $(\bar{\theta}_k)_{k=0}^{\infty}$ is monotonically non-decreasing. For any $k \in \mathbb{N}$ such that $\bar{\theta}_{k-1} \in [1, \bar{z}_c)$, we obtain:

$$\bar{\theta}_k = \bar{\theta}_{k-1} - \eta \frac{d\bar{\varphi}}{dz}(\bar{\theta}_{k-1}) = \bar{\theta}_{k-1} + a\eta\bar{\theta}_{k-1}.$$

The solution of this recursive equation is $\bar{\theta}_k = \bar{\theta}_{k_1}(1+a\eta)^{k-k_1}$ for all $k \geq k_1$ such that $\bar{\theta}_{k-1} \in [1, \bar{z}_c)$ *i.e.* for all $k \in \{k_1, k_1+1, ..., k_1 + \left\lceil \ln(\bar{z}_c/\bar{\theta}_{k_1})/\ln(1+a\eta) \right\rceil\}$. From the definition of $k_{\bar{z}_c}$ it must be equal to $k_1 + \left\lceil \ln(\bar{z}_c/\bar{\theta}_{k_1})/\ln(1+a\eta) \right\rceil$, as if it is smaller we get $\bar{\theta}_{k_{\bar{z}_c}} < \bar{z}_c$, and if it is larger we get $\bar{\theta}_{k_{\bar{z}_c}-1} \geq \bar{z}_c$, both contradicting the definition of $k_{\bar{z}_c}$. $\qquad \square$

We conclude this subsubappendix by combining its results with Lemma 38, thereby showing that the gradient flow trajectory between times $t_1$ and $t_{\bar{z}_c}$, and the gradient descent trajectory between iterations $k_1$ and $k_{\bar{z}_c}$, both lie in the isotropic region.

**Lemma 47.** *It holds that $\left(\theta(t), \bar{\theta}(t)\right) \in [0, z_c) \times [1, \bar{z}_c)$ for all $t \in [t_1, t_{\bar{z}_c})$, and $(\theta_k, \bar{\theta}_k) \in [0, z_c) \times [1, \bar{z}_c)$ for all $k \in \{k_1, k_1+1, ..., k_{\bar{z}_c}-1\}$.*

*Proof.* We start by proving the result for gradient flow. Lemma 42 implies $t_1 < \infty$, and similarly Lemma 45 implies $t_{\bar{z}_c} < \infty$. From continuity of $\bar{\theta}(\cdot)$ and the definition of $t_1$ we have that $\bar{\theta}(t_1) = 1$, and similarly by definition of $t_{\bar{z}_c}$ we have $\bar{\theta}(t_{\bar{z}_c}) = \bar{z}_c$. By monotonicity of $\bar{\theta}(\cdot)$ (Lemma 37) we conclude $\bar{\theta}(t) \in [1, \bar{z}_c)$ for all $t \in [t_1, t_{\bar{z}_c})$. Recall that we assume $\theta(0) \in (0.5, 1)$. By Lemma 38 we have that $\theta(t) \in [\theta(0), z_c)$ for all $t \in [0, a^{-1}\ln(z_c/\theta(0)))$. Recall the explicit expression for $t_{1-\bar{\rho}}$ from Lemma 39. By Lemma 42 $t_1 \leq t_{1-\bar{\rho}} + a^{-1}\ln(1/(1-\bar{\rho}))$, and by Lemma 45 we have that $t_{\bar{z}_c} = t_1 + a^{-1}\ln(\bar{z}_c)$. Overall, we obtain $t_{\bar{z}_c} \leq \frac{2}{a}\ln\left((4-3\bar{\rho})/(2\bar{\theta}(0)+2-\bar{\rho})\right) + a^{-1}\ln(1/(1-\bar{\rho})) + a^{-1}\ln(\bar{z}_c)$. By recalling the assumptions on $a$, $z_c$, $\bar{\rho}$, $\theta(0)$ and $\bar{\theta}(0)$, it can be shown that $t_{\bar{z}_c} \leq a^{-1}\ln(z_c/\theta(0))$. It follows that $\theta(t) \in [\theta(0), z_c)$ for all $t \in [t_1, t_{\bar{z}_c})$. Overall we proved that $\left(\theta(t), \bar{\theta}(t)\right) \in [0, z_c) \times [1, \bar{z}_c)$ for all $t \in [t_1, t_{\bar{z}_c})$, as required.

Moving on to gradient descent, Lemma 43 implies $k_1 < \infty$, and similarly Lemma 46 implies $k_{\bar{z}_c} < \infty$. By definition of $k_1$ we have that $\bar{\theta}_{k_1} \geq 1$, and by definition of $k_{\bar{z}_c}$ we have $\bar{\theta}_{k_{\bar{z}_c}-1} < \bar{z}_c$. From monotonicity of $(\bar{\theta}_k)_{k=0}^{\infty}$ (Lemma 37) we conclude $\bar{\theta}_k \in [1, \bar{z}_c)$ for all $k \in \{k_1, k_1+1, ..., k_{\bar{z}_c}-1\}$. Recall that we assume $\theta_0 \in (0.5, 1)$. By Lemma 38 $\theta_k \in [\theta_0, z_c)$ for all $k \in \{0, 1, ..., \left\lceil \ln(z_c/\theta_0)/\ln(1+a\eta) \right\rceil - 1\}$. Recall the definition of $k_{1-\bar{\rho}}$ and also its explicit expression from Lemma 40. By Lemma 43 we have that $k_1 \leq k_{1-\bar{\rho}} + \left\lceil \max\{0, -\ln(\bar{\theta}_{k_{1-\bar{\rho}}})/\ln(1+a\eta)\} \right\rceil$. By Lemma 46 $k_{\bar{z}_c} = k_1 + \left\lceil \ln(\bar{z}_c/\bar{\theta}_{k_1})/\ln(1+a\eta) \right\rceil$. Overall, we obtain $k_{\bar{z}_c} \leq \left\lceil \left(\ln(2-3\bar{\rho}/2) - \ln(\bar{\theta}_0+1-\bar{\rho}/2)\right)/\ln(1+a\eta/2) \right\rceil + \left\lceil \max\{0, -\ln(\bar{\theta}_{k_{1-\bar{\rho}}})/\ln(1+a\eta)\} \right\rceil + \left\lceil \ln(\bar{z}_c/\bar{\theta}_{k_1})/\ln(1+a\eta) \right\rceil$. By recalling the assumptions on $a$, $z_c$, $\bar{\rho}$, $\theta_0$ and $\bar{\theta}_0$, it can be shown that $k_{\bar{z}_c} \leq \left\lceil \ln(z_c/\theta_0)/\ln(1+a\eta) \right\rceil$. It follows that $\theta_k \in [\theta_0, z_c)$ for all $k \in \{k_1, k_1+1, ..., k_{\bar{z}_c}-1\}$. Overall we proved that $(\theta_k, \bar{\theta}_k) \in [0, z_c) \times [1, \bar{z}_c)$ for all $k \in \{k_1, k_1+1, ..., k_{\bar{z}_c}-1\}$, as required. $\qquad \square$

### J.11.5 Inapproximation

The current subsubappendix shows that the location of gradient flow at time $\tilde{t}$ is not $\epsilon$-approximated by different portions of the gradient descent trajectory. Key to the derived results is the following lemma, which establishes that in the isotropic region, for both gradient flow and gradient descent trajectories, the first two coordinates proceed in a straight line at an exponential pace.

**Lemma 48.** *It holds that $\left(\theta(t), \bar{\theta}(t)\right) = \left(\theta(t_1), \bar{\theta}(t_1)\right) \cdot e^{a(t-t_1)}$ for all $t \in [t_1, t_{\bar{z}_c})$, and $(\theta_k, \bar{\theta}_k) = (\theta_{k_1}, \bar{\theta}_{k_1}) \cdot (1+a\eta)^{k-k_1}$ for all $k \in \{k_1, k_1+1, ..., k_{\bar{z}_c}-1\}$, where $t_1$, $t_{\bar{z}_c}$, $k_1$ and $k_{\bar{z}_c}$ are given by Definitions 10, 12, 11 and 13 respectively.*

*Proof.* We start by proving the result for gradient flow. Lemma 45 implies $\bar{\theta}(t) = \bar{\theta}(t_1)e^{a(t-t_1)}$ for all $t \in [t_1, t_{\bar{z}_c}]$. By Lemma 38 we have that $\theta(t) = \theta(0)e^{at}$ for all $t \in \left[0, a^{-1}\ln(z_c/\theta(0))\right)$. Recall the explicit expression for $t_{1-\bar{\rho}}$ from Lemma 39. By Lemma 42 $t_1 \leq t_{1-\bar{\rho}} + a^{-1}\ln\left(1/(1-\bar{\rho})\right)$, and by Lemma 45 we have that $t_{\bar{z}_c} = t_1 + a^{-1}\ln(\bar{z}_c)$. Overall, we obtain $t_{\bar{z}_c} \leq \frac{2}{a}\ln\left((4-3\bar{\rho})/(2\bar{\theta}(0)+2-\bar{\rho})\right) + a^{-1}\ln\left(1/(1-\bar{\rho})\right) + a^{-1}\ln(\bar{z}_c)$. By recalling the assumptions on $a$, $z_c$, $\bar{\rho}$, $\theta(0)$ and $\bar{\theta}(0)$, it can be shown that $t_{\bar{z}_c} \leq a^{-1}\ln\left(z_c/\theta(0)\right)$. It follows that $\theta(t) = \theta(0)e^{at} = \theta(t_1)e^{a(t-t_1)}$ for all $t \in \left[t_1, t_{\bar{z}_c}\right)$.

Moving on to gradient descent, Lemma 46 implies $\bar{\theta}_k = \bar{\theta}_{k_1}(1 + a\eta)^{k-k_1}$ for all $k \in \{k_1, k_1 + 1, \ldots, k_{\bar{z}_c} - 1\}$. By Lemma 38 we have that $\theta_k = \theta_0(1 + a\eta)^k$ for all $k \in \left\{0, 1, \ldots, \lceil \ln(z_c/\theta_0)/\ln(1+a\eta) \rceil \right\}$. Recall the definition of $k_{1-\bar{\rho}}$ and its explicit expression from Lemma 40. Lemma 43 ensures $k_1 \leq k_{1-\bar{\rho}} + \lceil \max\{0, -\ln(\bar{\theta}_{k_{1-\bar{\rho}}})/\ln(1+a\eta)\} \rceil$. By Lemma 46 $k_{\bar{z}_c} = k_1 + \lceil \ln(\bar{z}_c/\bar{\theta}_{k_1})/\ln(1+a\eta) \rceil$. Putting it all together, we obtain $k_{\bar{z}_c} \leq \lceil \left(\ln(2-3\bar{\rho}/2) - \ln(\bar{\theta}_0 + 1 - \bar{\rho}/2)\right)/\ln(1+a\eta/2) \rceil + \lceil \max\{0, -\ln(\bar{\theta}_{k_{1-\bar{\rho}}})/\ln(1+a\eta)\} \rceil + \lceil \ln(\bar{z}_c/\bar{\theta}_{k_1})/\ln(1+a\eta) \rceil$. By recalling the assumptions on $a$, $z_c$, $\bar{\rho}$, $\theta_0$ and $\bar{\theta}_0$, it can be shown that $k_{\bar{z}_c} \leq \lceil \ln(z_c/\theta_0)/\ln(1+a\eta) \rceil$. It follows that $\theta_k = \theta_0(1+a\eta)^k = \theta_{k_1}(1+a\eta)^{k-k_1}$ for all $k \in \{k_1, k_1+1, \ldots, k_{\bar{z}_c} - 1\}$. $\qquad\square$

**Lemma 49.** *It holds that $\|\boldsymbol{\theta}_k - \boldsymbol{\theta}(\tilde{t})\| > \epsilon$ for all $k \in \{0, 1, \ldots, k_1 - 1\}$, where $k_1$ is given by Definition 11.*

*Proof.* Recall that $\tilde{t} \in \left[\frac{2}{a}\ln\left(\frac{2-3\bar{\rho}/2}{\bar{\theta}(0)-(\bar{\rho}/2-1)}\right) + \frac{1}{a}\ln\left(\frac{2}{1-\bar{\rho}}\right), \frac{2}{a}\ln\left(\frac{2-3\bar{\rho}/2}{\bar{\theta}(0)-(\bar{\rho}/2-1)}\right) + \frac{1}{a}\ln\left(\frac{1+\bar{\rho}/4}{1-3\bar{\rho}/4}\right) + \frac{1}{a}\ln(b)\right]$ and $\epsilon < 1$. Lemmas 39 and 42 imply $\tilde{t} \in [t_1 + \frac{1}{a}\ln(2), t_1 + \frac{1}{a}\ln(b)]$. Notice that $t_1 \leq t_1 + \frac{1}{a}\ln(2) \leq \tilde{t} \leq t_1 + \frac{1}{a}\ln(b) \leq t_1 + \frac{1}{a}\ln(b+1) = t_{\bar{z}_c}$. Thus, by Lemma 47, gradient flow is in the isotropic region at time $\tilde{t}$. We may use Lemma 45 together with monotonicity of $\bar{\theta}(\cdot)$ (Lemma 37) to obtain $\bar{\theta}(\tilde{t}) \geq \bar{\theta}\left(t_1 + \frac{1}{a}\ln(2)\right) = e^{a\ln(2)/a} = 2$. From the definition of $k_1$ (and from the fact that it is finite from Lemma 43) we know that $|\bar{\theta}_{k_1-1}| \leq 1$. For all $k \in \{0, 1, \ldots, k_1 - 1\}$, using monotonicity of $(\bar{\theta}_k)_{k=0}^{\infty}$ (Lemma 37), we may conclude:

$$\|\boldsymbol{\theta}_k - \boldsymbol{\theta}(\tilde{t})\|_2 \geq |\bar{\theta}_k - \bar{\theta}(\tilde{t})| \geq |\bar{\theta}(\tilde{t})| - |\bar{\theta}_k| \geq |\bar{\theta}(\tilde{t})| - |\bar{\theta}_{k_1-1}| \geq 2 - 1 = 1 > \epsilon.$$

$\square$

**Lemma 50.** *It holds that $\|\boldsymbol{\theta}_k - \boldsymbol{\theta}(\tilde{t})\| > \epsilon$ for all $k \in \{k_1, k_1+1, \ldots, k_{\bar{z}_c} - 1\}$, where $k_1$ and $k_{\bar{z}_c}$ are given by Definitions 11 and 13 respectively.*

*Proof.* Recall that $\tilde{t} \in \left[\frac{2}{a}\ln\left(\frac{2-3\bar{\rho}/2}{\bar{\theta}(0)-(\bar{\rho}/2-1)}\right) + \frac{1}{a}\ln\left(\frac{2}{1-\bar{\rho}}\right), \frac{2}{a}\ln\left(\frac{2-3\bar{\rho}/2}{\bar{\theta}(0)-(\bar{\rho}/2-1)}\right) + \frac{1}{a}\ln\left(\frac{1+\bar{\rho}/4}{1-3\bar{\rho}/4}\right) + \frac{1}{a}\ln(b)\right]$ and $\epsilon < 1$. Lemmas 39 and 42 imply $\tilde{t} \in [t_1 + \frac{1}{a}\ln(2), t_1 + \frac{1}{a}\ln(b)]$. Notice that $t_1 \leq t_1 + \frac{1}{a}\ln(2) \leq \tilde{t} \leq t_1 + \frac{1}{a}\ln(b) \leq t_1 + \frac{1}{a}\ln(b+1) = t_{\bar{z}_c}$. Thus, by Lemma 47, gradient flow is in the isotropic region at time $\tilde{t}$. By Lemma 43 we know that $k_1 < \infty$, and by Lemma 47 it holds that $\bar{\theta}_{k_1} \geq 1$. By monotonicity of $(\theta_k)_{k=0}^{\infty}$ (Lemma 37) and since $\theta_0 > 0$, we know that $\theta_k > 0$ for all $k \in \mathbb{N}$. For all $k \in \{k_1, k_1+1, \ldots, k_{\bar{z}_c} - 1\}$, Lemma 48 ensures $\theta_k/\bar{\theta}_k = \theta_{k_1}/\bar{\theta}_{k_1}$, thus $(\theta_k, \bar{\theta}_k) \in \left\{(q, \bar{q}) : q, \bar{q} \in (0, \infty) \text{ s.t. } q/\bar{q} = \theta_{k_1}/\bar{\theta}_{k_1}\right\} = \left\{c(\theta_{k_1}, \bar{\theta}_{k_1}) : c > 0\right\}$. This leads us to:

$$\left\|\boldsymbol{\theta}_k - \boldsymbol{\theta}(\tilde{t})\right\|_2 \geq \left\|(\theta_k, \bar{\theta}_k) - (\theta(\tilde{t}), \bar{\theta}(\tilde{t}))\right\|_2 \geq \inf_{c>0}\left\|c(\theta_{k_1}, \bar{\theta}_{k_1}) - (\theta(\tilde{t}), \bar{\theta}(\tilde{t}))\right\|_2.$$

Minimizing over $c > 0$ yields $c_{\min} = \left\langle (\theta(\tilde{t}), \bar{\theta}(\tilde{t})), (\theta_{k_1}, \bar{\theta}_{k_1}) \right\rangle / \|(\theta_{k_1}, \bar{\theta}_{k_1})\|_2^2$. Note that since gradient flow is in the isotropic region at time $\tilde{t}$, then $\|(\theta(\tilde{t}), \bar{\theta}(\tilde{t}))\|_2 \neq 0$. We obtain:

$$\|\boldsymbol{\theta}_k - \boldsymbol{\theta}(\tilde{t})\|_2 \geq \left\|(\theta_{k_1}, \bar{\theta}_{k_1})\left\langle (\theta(\tilde{t}), \bar{\theta}(\tilde{t})), (\theta_{k_1}, \bar{\theta}_{k_1}) \right\rangle / \|(\theta_{k_1}, \bar{\theta}_{k_1})\|_2^2 - (\theta(\tilde{t}), \bar{\theta}(\tilde{t}))\right\|_2$$

$$= \sqrt{\|(\theta(\tilde{t}), \bar{\theta}(\tilde{t}))\|_2^2 - \left\langle (\theta(\tilde{t}), \bar{\theta}(\tilde{t})), \frac{(\theta_{k_1}, \bar{\theta}_{k_1})}{\|(\theta_{k_1}, \bar{\theta}_{k_1})\|_2} \right\rangle^2}$$

$$= \|(\theta(\tilde{t}), \bar{\theta}(\tilde{t}))\|_2 \sqrt{1 - \left\langle \frac{(\theta(\tilde{t}), \bar{\theta}(\tilde{t}))}{\|(\theta(\tilde{t}), \bar{\theta}(\tilde{t}))\|_2}, \frac{(\theta_{k_1}, \bar{\theta}_{k_1})}{\|(\theta_{k_1}, \bar{\theta}_{k_1})\|_2} \right\rangle^2}$$

$$\geq |\theta(\tilde{t})| \sqrt{1 - \left\langle \frac{(\theta(\tilde{t}), \bar{\theta}(\tilde{t}))}{\|(\theta(\tilde{t}), \bar{\theta}(\tilde{t}))\|_2}, \frac{(\theta_{k_1}, \bar{\theta}_{k_1})}{\|(\theta_{k_1}, \bar{\theta}_{k_1})\|_2} \right\rangle^2}.$$

By Lemma 38 we have that $\theta(t) = \theta(0)e^{at}$ for all $t \in \left[0, a^{-1}\ln\left(z_c/\theta(0)\right)\right)$. Recall the explicit expression for $t_{1-\bar{\rho}}$ from Lemma 39. By Lemma 42 $t_1 \leq t_{1-\bar{\rho}} + a^{-1}\ln\left(1/(1-\bar{\rho})\right)$, and by Lemma 45 we have that $t_{\bar{z}_c} = t_1 + a^{-1}\ln(\bar{z}_c)$. Since $\tilde{t} \leq t_{\bar{z}_c}$, overall we obtain $\tilde{t} \leq \frac{2}{a}\ln\left((4-3\bar{\rho})/(2\bar{\theta}(0)+2-\bar{\rho})\right) + a^{-1}\ln\left(1/(1-\bar{\rho})\right) + a^{-1}\ln(\bar{z}_c)$. By recalling the assumptions on $a, z_c, \bar{\rho}, \theta(0)$ and $\bar{\theta}(0)$, it can be shown that $\tilde{t} \leq a^{-1}\ln\left(z_c/\theta(0)\right)$. It follows that:

$$\|\boldsymbol{\theta}_k - \boldsymbol{\theta}(\tilde{t})\|_2 \geq \theta(0)e^{a\tilde{t}}\sqrt{1 - \left\langle \frac{(\theta(\tilde{t}),\bar{\theta}(\tilde{t}))}{\left\|(\theta(\tilde{t}),\bar{\theta}(\tilde{t}))\right\|_2}, \frac{(\theta_{k_1},\bar{\theta}_{k_1})}{\left\|(\theta_{k_1},\bar{\theta}_{k_1})\right\|_2} \right\rangle^2}.$$

Recall that $\tilde{t} \in [t_1, t_{\bar{z}_c}]$. Lemma 47 implies that both $\bar{\theta}(\tilde{t})$ and $\bar{\theta}(t_1)$ are greater or equal to one. Since Lemma 48 implies $\theta(\tilde{t})/\bar{\theta}(\tilde{t}) = \theta(t_1)/\bar{\theta}(t_1)$, it follows that:

$$\|\boldsymbol{\theta}_k - \boldsymbol{\theta}(\tilde{t})\|_2 \geq \theta(0)e^{a\tilde{t}}\sqrt{1 - \left\langle \frac{(\theta(t_1),\bar{\theta}(t_1))}{\left\|(\theta(t_1),\bar{\theta}(t_1))\right\|_2}, \frac{(\theta_{k_1},\bar{\theta}_{k_1})}{\left\|(\theta_{k_1},\bar{\theta}_{k_1})\right\|_2} \right\rangle^2}$$

$$= \theta(0)e^{a\tilde{t}}\sqrt{1 - \left\langle \frac{(\theta(t_1)/\bar{\theta}(t_1),1)}{\left\|(\theta(t_1)/\bar{\theta}(t_1),1)\right\|_2}, \frac{(\theta_{k_1}/\bar{\theta}_{k_1},1)}{\left\|(\theta_{k_1}/\bar{\theta}_{k_1},1)\right\|_2} \right\rangle^2}$$

Note that the latter inner product is between positively correlated unit vectors, and that it is squared. We use Lemmas 42 and 43, ensuring that $\bar{\theta}(t_1)/\theta(t_1) \leq \left((\bar{\theta}(0) + 1 - \bar{\rho}/2)/(2 - 3\bar{\rho}/2)\right)^2/\theta(0)$, and $\bar{\theta}_{k_1}/\theta_{k_1} \geq \left((\bar{\theta}_0 + 1 - \bar{\rho}/2)/(2 - 3\bar{\rho}/2)\right)^2/\left(\theta_0(1 - a\eta/10)\right)$ respectively. For brevity, denote $\alpha := \theta(0)\left((4 - 3\bar{\rho})/(2\bar{\theta}(0) + 2 - \bar{\rho})\right)^2$ and $\beta := (1 - a\eta/10)$. Notice that $\beta \in (0,1)$. Recall that by definition $\theta(0) = \theta_0$ and $\bar{\theta}(0) = \bar{\theta}_0$. Thus, $\theta_{t_1}/\bar{\theta}_{t_1} \leq \alpha\beta < \alpha \leq \theta(t_1)/\bar{\theta}(t_1)$. Replacing $\theta(t_1)/\bar{\theta}(t_1)$ with $\alpha$, and $\theta_{k_1}/\bar{\theta}_{k_1}$ with $\alpha\beta$, decreases the angle between the unit vectors, thereby increasing their inner product. We thus have that:

$$\|\boldsymbol{\theta}_k - \boldsymbol{\theta}(\tilde{t})\|_2 \geq \theta(0)e^{a\tilde{t}}\sqrt{1 - \left\langle \frac{(\alpha,1)}{\|(\alpha,1)\|_2}, \frac{(\alpha\beta,1)}{\|(\alpha\beta,1)\|_2} \right\rangle^2}$$

$$= \theta(0)e^{a\tilde{t}}\sqrt{1 - \left(\frac{\alpha^2\beta + 1}{\sqrt{\alpha^2 + 1}\cdot\sqrt{\alpha^2\beta^2 + 1}}\right)^2}$$

$$= \theta(0)e^{a\tilde{t}}\sqrt{\frac{\alpha^4\beta^2 + \alpha^2\beta^2 + \alpha^2 + 1}{\alpha^4\beta^2 + \alpha^2\beta^2 + \alpha^2 + 1} - \left(\frac{\alpha^4\beta^2 + 2\alpha^2\beta + 1}{\alpha^4\beta^2 + \alpha^2\beta^2 + \alpha^2 + 1}\right)}$$

$$= \theta(0)e^{a\tilde{t}}\sqrt{\frac{\alpha^2(1-\beta)^2}{\alpha^4\beta^2 + \alpha^2\beta^2 + \alpha^2 + 1}}$$

Since $\beta \in (0,1)$ and $\alpha \geq 1$ (can be verified by recalling the assumptions on $a, \eta, \bar{\rho}, \theta(0)$ and $\bar{\theta}(0)$), we obtain:

$$\|\boldsymbol{\theta}_k - \boldsymbol{\theta}(\tilde{t})\|_2 \geq \theta(0)e^{a\tilde{t}}\left(\frac{1-\beta}{2\alpha}\right).$$

Plugging in $\alpha$ and $\beta$, we obtain:

$$\|\boldsymbol{\theta}_k - \boldsymbol{\theta}(\tilde{t})\|_2 \geq \theta(0)e^{a\tilde{t}\cdot\frac{a\eta}{10}}\cdot\frac{1}{2\theta(0)}\left(\frac{2\bar{\theta}(0)+2-\bar{\rho}}{4-3\bar{\rho}}\right)^2 = \frac{1}{20}\left(\frac{2\bar{\theta}(0)+2-\bar{\rho}}{4-3\bar{\rho}}\right)^2 a\eta e^{a\tilde{t}}.$$

The definition of $\bar{\rho}$, and the assumptions on $\theta(0)$ and $\bar{\theta}(0)$ lead us to:

$$\|\boldsymbol{\theta}_k - \boldsymbol{\theta}(\tilde{t})\|_2 \geq \frac{1}{20}\left(\frac{0.5e^{-12}}{4}\right)^2 a\eta e^{a\tilde{t}} > 10^{-14}a\eta e^{a\tilde{t}}.$$

Since $\eta \geq \epsilon 10^{14}e^{-a\tilde{t}}/a$, we have $\|\boldsymbol{\theta}_k - \boldsymbol{\theta}(\tilde{t})\|_2 > \epsilon$, and this holds for all $k \in \{k_1, k_1+1, ..., k_{\bar{z}_c} - 1\}$. $\quad\square$

**Lemma 51.** *It holds that $\|\boldsymbol{\theta}_k - \boldsymbol{\theta}(\tilde{t})\| > \epsilon$ for all $k \in \{k_{\bar{z}_c}, k_{\bar{z}_c}+1, k_{\bar{z}_c}+2, ...\}$, where $k_{\bar{z}_c}$ is given by Definition 13.*

*Proof.* Recall that $\tilde{t} \in \left[\frac{2}{a}\ln\left(\frac{2-3\bar{\rho}/2}{\bar{\theta}(0)-(\bar{\rho}/2-1)}\right) + \frac{1}{a}\ln\left(\frac{2}{1-\bar{\rho}}\right), \frac{2}{a}\ln\left(\frac{2-3\bar{\rho}/2}{\bar{\theta}(0)-(\bar{\rho}/2-1)}\right) + \frac{1}{a}\ln\left(\frac{1+\bar{\rho}/4}{1-3\bar{\rho}/4}\right) + \frac{1}{a}\ln(b)\right]$ and $\epsilon < 1$. Lemmas 39 and 42 imply $\tilde{t} \in [t_1 + \frac{1}{a}\ln(2), t_1 + \frac{1}{a}\ln(b)]$. Notice that $t_1 \leq t_1 + \frac{1}{a}\ln(2) \leq \tilde{t} \leq t_1 + \frac{1}{a}\ln(b) \leq t_1 + \frac{1}{a}\ln(b+1) = t_{\bar{z}_c}$. Thus, by Lemma 47, gradient flow is in the isotropic region at time $\tilde{t}$. We may use Lemma 45 together with monotonicity of $\bar{\theta}(\cdot)$

(Lemma 37) to obtain $\bar{\theta}(\tilde{t}) \leq \bar{\theta}\left(t_1 + \frac{1}{a}\ln(b)\right) = e^{a\ln(b)/a} = b$. Lemma 37 further ensures monotonicity of $(\bar{\theta}_k)_{k=0}^{\infty}$. By the definition of $\bar{\theta}_{k_{\bar{z}_c}}$ (and from the fact that it is finite from Lemma 46) we know that $\bar{\theta}_{k_{\bar{z}_c}} \geq \bar{z}_c = b+1$. This implies, for all $k \in \{k_{\bar{z}_c}, k_{\bar{z}_c}+1, k_{\bar{z}_c}+2, ...\}$:

$$\|\boldsymbol{\theta}_k - \boldsymbol{\theta}(\tilde{t})\|_2 \geq |\bar{\theta}_k - \bar{\theta}(\tilde{t})| \geq |\bar{\theta}_k| - |\bar{\theta}(\tilde{t})| \geq |\bar{\theta}_{k_{\bar{z}_c}}| - |\bar{\theta}(\tilde{t})| \geq (b+1) - b = 1 > \epsilon.$$

$\square$

### J.11.6  Conclusion

Taken together, Lemmas 49, 50 and 51 form a proof for Proposition 4 in the case where the step size $\eta$ is no greater than $\frac{1}{6a}$. The complementary case $\eta > \frac{1}{6a}$ is accounted for by Lemma 36.

### J.12  Proof of Lemma 4

This proof is very similar to that of Lemma 1 (see Subappendix J.4). We repeat all details for completeness. Recall that $\boldsymbol{\theta} \in \mathbb{R}^d$ is an arrangement of $(W_1, W_2, ..., W_n) \in \mathbb{R}^{d_1, d_0} \times \mathbb{R}^{d_2, d_1} \times \cdots \times \mathbb{R}^{d_n, d_{n-1}}$ as a vector. Let $(\Delta W_1, \Delta W_2, ..., \Delta W_n) \in \mathbb{R}^{d_1, d_0} \times \mathbb{R}^{d_2, d_1} \times \cdots \times \mathbb{R}^{d_n, d_{n-1}}$, and denote by $\Delta\boldsymbol{\theta} \in \mathbb{R}^d$ its arrangement as a vector in corresponding order. Denote the following for $i \in \{1, ..., |\mathcal{S}|\}$:

$$\Delta_i^{(1)} := \sum_{j=1}^n (D'_{i,*}W_*)_{n:j+1} D'_{i,j}(\Delta W_j)(D'_{i,*}W_*)_{j\text{-}1:1}, \tag{74}$$

$$\Delta_i^{(2)} := \sum_{1 \leq j < j' \leq n} (D'_{i,*}W_*)_{n:j'+1} D'_{i,j'}(\Delta W_{j'})(D'_{i,*}W_*)_{j'\text{-}1:j+1} D'_{i,j}(\Delta W_j)(D'_{i,*}W_*)_{j\text{-}1:1}, \tag{75}$$

$$\Delta_i^{(3:n)} := D'_{i,n}(W_n + \Delta W_n)\cdots D'_{i,1}(W_1 + \Delta W_1) - (D'_{i,*}W_*)_{n:1} - \Delta_i^{(1)} - \Delta_i^{(2)}. \tag{76}$$

We now develop a second-order Taylor expansion of $f(\boldsymbol{\theta})$. Since the matrix tuple corresponding to $(\boldsymbol{\theta} + \Delta\boldsymbol{\theta})$ is $\left((W_1 + \Delta W_1), ..., (W_n + \Delta W_n)\right)$, and the function $f(\cdot)$ coincides with the function given in Equation (24) on an open region containing $\boldsymbol{\theta}$, for sufficiently small $\Delta\boldsymbol{\theta}$ we obtain:

$$
\begin{aligned}
&f(\boldsymbol{\theta} + \Delta\boldsymbol{\theta}) \\
&= \frac{1}{|\mathcal{S}|} \sum_{i=1}^{|\mathcal{S}|} \ell\left(D'_{i,n}(W_n + \Delta W_n)...D'_{i,1}(W_1 + \Delta W_1)\mathbf{x}_i, y_i\right) \\
&= \frac{1}{|\mathcal{S}|} \sum_{i=1}^{|\mathcal{S}|} \ell\left(\left((D'_{i,*}W_*)_{n:1} + \Delta_i^{(1)} + \Delta_i^{(2)} + \Delta_i^{(3:n)}\right)\mathbf{x}_i, y_i\right) \\
&= \frac{1}{|\mathcal{S}|} \sum_{i=1}^{|\mathcal{S}|} \ell\left((D'_{i,*}W_*)_{n:1}\mathbf{x}_i + \left(\Delta_i^{(1)} + \Delta_i^{(2)} + \Delta_i^{(3:n)}\right)\mathbf{x}_i, y_i\right),
\end{aligned}
\tag{77}
$$

where the second transition follows from the definition of $\Delta_i^{(3:n)}$ (Equation (76)). Let $\Delta\boldsymbol{v} \in \mathbb{R}^{d_n}$. For every $i \in \{1, ..., |\mathcal{S}|\}$, the second-order Taylor expansion of $\ell(\cdot)$ with respect to its first argument at the point $\left((D'_{i,*}W_*)_{n:1}\mathbf{x}_i, y_i\right)$ is given by:

$$\ell\left((D'_{i,*}W_*)_{n:1}\mathbf{x}_i + \Delta\boldsymbol{v}, y_i\right) = \ell\left((D'_{i,*}W_*)_{n:1}\mathbf{x}_i, y_i\right) + \langle\nabla\ell_i, \Delta\boldsymbol{v}\rangle + \tfrac{1}{2}\nabla^2\ell_i[\Delta\boldsymbol{v}] + \mathcal{O}\left(\|\Delta\boldsymbol{v}\|_2^2\right), \tag{78}$$

where the $\mathcal{o}(\cdot)$ notation refers to some expression satisfying $\lim_{a\to 0}\big(\mathcal{o}(a)/a\big)=0$. We continue to develop Equation (77) using Equation (78):

$$f(\boldsymbol{\theta}+\Delta\boldsymbol{\theta})$$

$$=\frac{1}{|\mathcal{S}|}\sum_{i=1}^{|\mathcal{S}|}\Big(\ell\big((D_{i,*}'W_*)_{n:1}\mathbf{x}_i,y_i\big)+\big\langle\nabla\ell_i,(\Delta_i^{(1)}+\Delta_i^{(2)}+\Delta_i^{(3:n)})\mathbf{x}_i\big\rangle+$$

$$\tfrac{1}{2}\nabla^2\ell_i\big[(\Delta_i^{(1)}+\Delta_i^{(2)}+\Delta_i^{(3:n)})\mathbf{x}_i\big]+\mathcal{o}\big(\big\|(\Delta_i^{(1)}+\Delta_i^{(2)}+\Delta_i^{(3:n)})\mathbf{x}_i\big\|_2^2\big)\Big)$$

$$=\frac{1}{|\mathcal{S}|}\sum_{i=1}^{|\mathcal{S}|}\ell\big((D_{i,*}'W_*)_{n:1}\mathbf{x}_i,y_i\big)+$$

$$\frac{1}{|\mathcal{S}|}\sum_{i=1}^{|\mathcal{S}|}\big\langle\nabla\ell_i,\Delta_i^{(1)}\mathbf{x}_i\big\rangle+\big\langle\nabla\ell_i,\Delta_i^{(2)}\mathbf{x}_i\big\rangle+\big\langle\nabla\ell_i,\Delta_i^{(3:n)}\mathbf{x}_i\big\rangle+$$

$$\frac{1}{|\mathcal{S}|}\sum_{i=1}^{|\mathcal{S}|}\tfrac{1}{2}\nabla^2\ell_i\big[\Delta_i^{(1)}\mathbf{x}_i\big]+\tfrac{1}{2}\nabla^2\ell_i\big[(\Delta_i^{(2)}+\Delta_i^{(3:n)})\mathbf{x}_i\big]+2\cdot\tfrac{1}{2}\nabla^2\ell_i\big[\Delta_i^{(1)}\mathbf{x}_i,(\Delta_i^{(2)}+\Delta_i^{(3:n)})\mathbf{x}_i\big]+$$

$$\frac{1}{|\mathcal{S}|}\sum_{i=1}^{|\mathcal{S}|}\mathcal{o}\big(\big\|(\Delta_i^{(1)}+\Delta_i^{(2)}+\Delta_i^{(3:n)})\mathbf{x}_i\big\|_2^2\big),$$

where in the last transition we view $\nabla^2\ell_i$ as both a quadratic and a bilinear form (see Subappendix J.1). Notice that $\big\langle\nabla\ell_i,\Delta_i^{(3:n)}\mathbf{x}_i\big\rangle$, $\tfrac{1}{2}\nabla^2\ell_i\big[(\Delta_i^{(2)}+\Delta_i^{(3:n)})\mathbf{x}_i\big]$, $\nabla^2\ell_i\big[\Delta_i^{(1)}\mathbf{x}_i,(\Delta_i^{(2)}+\Delta_i^{(3:n)})\mathbf{x}_i\big]$ and $\mathcal{o}\big(\big\|(\Delta_i^{(1)}+\Delta_i^{(2)}+\Delta_i^{(3:n)})\mathbf{x}_i\big\|_2^2\big)$ are all $\mathcal{o}\big(\|\Delta\boldsymbol{\theta}\|_2^2\big)$, thus:

$$f(\boldsymbol{\theta}+\Delta\boldsymbol{\theta})$$

$$=\frac{1}{|\mathcal{S}|}\sum_{i=1}^{|\mathcal{S}|}\ell\big((D_{i,*}'W_*)_{n:1}\mathbf{x}_i,y_i\big)+\big\langle\nabla\ell_i,\Delta_i^{(1)}\mathbf{x}_i\big\rangle+\big\langle\nabla\ell_i,\Delta_i^{(2)}\mathbf{x}_i\big\rangle+\tfrac{1}{2}\nabla^2\ell_i\big[\Delta_i^{(1)}\mathbf{x}_i\big]+\mathcal{o}\big(\|\Delta\boldsymbol{\theta}\|_2^2\big).$$

This is a Taylor expansion of $f(\cdot)$ evaluated at $\boldsymbol{\theta}$ with a constant term $\frac{1}{|\mathcal{S}|}\sum_{i=1}^{|\mathcal{S}|}\ell\big((D_{i,*}'W_*)_{n:1}\mathbf{x}_i,y_i\big)$, a linear term $\frac{1}{|\mathcal{S}|}\sum_{i=1}^{|\mathcal{S}|}\big\langle\nabla\ell_i,\Delta_i^{(1)}\mathbf{x}_i\big\rangle$, a quadtratic term of two summands $\frac{1}{|\mathcal{S}|}\sum_{i=1}^{|\mathcal{S}|}\big\langle\nabla\ell_i,\Delta_i^{(2)}\mathbf{x}_i\big\rangle+\tfrac{1}{2}\nabla^2\ell_i\big[\Delta_i^{(1)}\mathbf{x}_i\big]$, and a remainder term of $\mathcal{o}\big(\|\Delta\boldsymbol{\theta}\|_2^2\big)$. From uniqueness of the Taylor expansion, the quadratic term must be equal to $\frac{1}{2}\nabla^2 f(\boldsymbol{\theta})[\Delta W_1,...,\Delta W_n]$. This implies:

$$\nabla^2 f(\boldsymbol{\theta})[\Delta W_1,...,\Delta W_n]$$

$$=\frac{1}{|\mathcal{S}|}\sum_{i=1}^{|\mathcal{S}|}\Big(\nabla^2\ell_i\big[\Delta_i^{(1)}\mathbf{x}_i\big]+2\big\langle\nabla\ell_i,\Delta_i^{(2)}\mathbf{x}_i\big\rangle\Big)$$

$$=\frac{1}{|\mathcal{S}|}\sum_{i=1}^{|\mathcal{S}|}\bigg(\nabla^2\ell_i\Big[\textstyle\sum_{j=1}^n(D_{i,*}'W_*)_{n:j+1}D_{i,j}'(\Delta W_j)(D_{i,*}'W_*)_{j-1:1}\mathbf{x}_i\Big]+$$

$$2\Big\langle\nabla\ell_i,\textstyle\sum_{1\le j<j'\le n}(D_{i,*}'W_*)_{n:j'+1}D_{i,j'}'(\Delta W_{j'})(D_{i,*}'W_*)_{j'-1:j+1}D_{i,j}'(\Delta W_j)(D_{i,*}'W_*)_{j-1:1}\mathbf{x}_i\Big\rangle\bigg),$$

where the last transition follows from plugging in the definitions of $\Delta^{(1)}$ and $\Delta^{(2)}$ (see Equations (74) and (75)).  $\square$

### J.13   Proof of Proposition 5

From assumption *(ii)* there exists some $\boldsymbol{\theta}\in\mathbb{R}^d$ such that $\sum_{i=1}^{|\mathcal{S}|}\nabla\ell(\mathbf{0},y_i)^\top h_{\boldsymbol{\theta}}(\mathbf{x}_i)\ne 0$. Define $\big(W_1,W_2,...,W_n\big)\in\mathbb{R}^{d_1,d_0}\times\mathbb{R}^{d_2,d_1}\times\cdots\times\mathbb{R}^{d_n,d_{n-1}}$ to be the weight matrices constituting $\boldsymbol{\theta}$. We may assume $\sum_{i=1}^{|\mathcal{S}|}\nabla\ell(\mathbf{0},y_i)^\top h_{\boldsymbol{\theta}}(\mathbf{x}_i)<0$ without loss of generality, as we can negate the vectors $h_{\boldsymbol{\theta}}(\mathbf{x}_i)\in\mathbb{R}^{d_n}$ for all $i\in\{1,2,...,|\mathcal{S}|\}$ by flipping the signs of the entries in $\boldsymbol{\theta}$ corresponding to the last

weight matrix $W_n$ (see Equation (8)). From continuity, there exists a neighborhood $\mathcal{N}$ of $\boldsymbol{\theta}$ such that for all $\tilde{\boldsymbol{\theta}} \in \mathcal{N}$ it holds that $\sum_{i=1}^{|\mathcal{S}|} \nabla \ell(\mathbf{0}, y_i)^\top h_{\tilde{\boldsymbol{\theta}}}(\mathbf{x}_i) < 0$. Moreover, as discussed in Appendix C, for almost all $\boldsymbol{\theta}' \in \mathbb{R}^d$ there exists an open region $\mathcal{D}_{\boldsymbol{\theta}'} \subseteq \mathbb{R}^d$ containing $\boldsymbol{\theta}'$, which is closed under positive rescaling of weight matrices and across which $f(\cdot)$ coincides with a function as given in Equation (24). There must exist some $\boldsymbol{\theta}'$ in the neighborhood $\mathcal{N}$ for which a region of the type $\mathcal{D}_{\boldsymbol{\theta}'}$ exists. We may assume, without loss of generality, that $\boldsymbol{\theta} \in \mathcal{D}_{\boldsymbol{\theta}'}$. Notice that none of the matrices $W_1, W_2, ..., W_n$ are equal to zero (as that would lead to $\sum_{i=1}^{|\mathcal{S}|} \nabla \ell(\mathbf{0}, y_i)^\top h_{\boldsymbol{\theta}}(\mathbf{x}_i) = 0$). Define the following weight matrices parameterized by $a > 0$ (while recalling that $n \geq 3$ by assumption *(i)*):

$$
\begin{aligned}
W_1(a) &:= W_1 \cdot a^{-2} \in \mathbb{R}^{d_1, d_0} , \\
W_2(a) &:= W_2 \cdot a^{-2} \in \mathbb{R}^{d_2, d_1} , \\
W_3(a) &:= W_3 \cdot a \in \mathbb{R}^{d_3, d_2} , \\
W_j(a) &:= W_j \in \mathbb{R}^{d_j, d_{j-1}} \text{ for } j \in \{1, 2, ..., n\}/\{1, 2, 3\} ,
\end{aligned}
$$

and denote by $\boldsymbol{\theta}(a) \in \mathbb{R}^d$ their corresponding weight setting. Since $\mathcal{D}_{\boldsymbol{\theta}'}$ is closed under positive rescaling of weight matrices, it holds that $\{\boldsymbol{\theta}(a) : a > 0\} \subseteq \mathcal{D}_{\boldsymbol{\theta}'}$. Define:

$$
\begin{aligned}
\Delta W_1 &:= W_1 \in \mathbb{R}^{d_1, d_0} , \\
\Delta W_2 &:= W_2 \in \mathbb{R}^{d_2, d_1} , \\
\Delta W_j &:= 0 \in \mathbb{R}^{d_j, d_{j-1}} \text{ for } j \in [n]/\{1, 2\} .
\end{aligned}
$$

For $a > 0$, $i \in \{1, 2, ..., |\mathcal{S}|\}$ and $j, j' \in \{1, 2, ..., n\}$, define $(D'_{i,*} W_*(a))_{j':j}$ to be the matrix $D'_{i,j'} W_{j'}(a) D'_{i,j'-1} W_{j'-1}(a) \cdots D'_{i,j} W_j(a)$ (where by convention $D'_{i,n} \in \mathbb{R}^{d_n, d_n}$ stands for identity) if $j \leq j'$, and an identity matrix (with size to be inferred by context) otherwise. For $i \in \{1, 2, ..., |\mathcal{S}|\}$ and $a > 0$ let $\nabla \ell_i(a) \in \mathbb{R}^{d_n}$ and $\nabla^2 \ell_i(a) \in \mathbb{R}^{d_n, d_n}$ be the gradient and Hessian (respectively) of the loss $\ell(\cdot)$ at the point $\left( (D'_{i,*} W_*(a))_{n:1} \mathbf{x}_i, y_i \right)$ with respect to its first argument. For every $a > 0$, since $\boldsymbol{\theta}(a) \in \mathcal{D}_{\boldsymbol{\theta}'}$ we may apply Lemma 4, obtaining:

$$\nabla^2 f\big(\boldsymbol{\theta}(a)\big)[\Delta W_1, ..., \Delta W_n] =$$

$$
\frac{1}{|\mathcal{S}|} \sum_{i=1}^{|\mathcal{S}|} \nabla^2 \ell_i(a) \left[ \sum_{j=1}^{n} (D'_{i,*} W_*(a))_{n:j+1} D'_{i,j} (\Delta W_j) (D'_{i,*} W_*(a))_{j-1:1} \mathbf{x}_i \right] +
\tag{79}
$$

$$
\frac{2}{|\mathcal{S}|} \sum_{i=1}^{|\mathcal{S}|} \nabla \ell_i(a)^\top \sum_{1 \leq j < j' \leq n} (D'_{i,*} W_*(a))_{n:j'+1} D'_{i,j'} (\Delta W_{j'}) (D'_{i,*} W_*(a))_{j'-1:j+1} D'_{i,j} (\Delta W_j) (D'_{i,*} W_*(a))_{j-1:1} \mathbf{x}_i,
$$

where we regard Hessians as quadratic forms (see Subappendix J.1). Plugging in the definitions of $W_j(a)$ and $\Delta W_j$ for $j \in [n]$ we have:

$$
\begin{aligned}
\nabla^2 f\big(\boldsymbol{\theta}(a)\big)&[\Delta W_1, ..., \Delta W_n] \\
&= \frac{1}{|\mathcal{S}|} \sum_{i=1}^{|\mathcal{S}|} \nabla^2 \ell_i(a) \left[ 2a^{-1} (D'_{i,*} W_*)_{n:1} \mathbf{x}_i \right] + \frac{2}{|\mathcal{S}|} \sum_{i=1}^{|\mathcal{S}|} \nabla \ell_i(a)^\top a (D'_{i,*} W_*)_{n:1} \mathbf{x}_i \\
&= \frac{4}{a^2} \cdot \frac{1}{|\mathcal{S}|} \sum_{i=1}^{|\mathcal{S}|} \nabla^2 \ell_i(a) \left[ (D'_{i,*} W_*)_{n:1} \mathbf{x}_i \right] + a \cdot \frac{2}{|\mathcal{S}|} \sum_{i=1}^{|\mathcal{S}|} \nabla \ell_i(a)^\top h_{\boldsymbol{\theta}}(\mathbf{x}_i) ,
\end{aligned}
\tag{80}
$$

where the second transition follows from pulling $2/a$ out of the quadratic operator and the fact that $h_{\boldsymbol{\theta}}(\mathbf{x}_i) = (D'_{i,*} W_*)_{n:1} \mathbf{x}_i$. Note that $\lim_{a \to \infty} (D'_{i,*} W_*)_{n:1} = 0$. Since $\ell(\cdot)$ is twice continuously differentiable in its first argument, it holds that $\lim_{a \to \infty} \nabla^2 \ell_i(a) = \lim_{a \to \infty} \nabla^2 \ell\big((D'_{i,*} W_*(a))_{n:1} \mathbf{x}_i, y_i\big) = \nabla^2 \ell(\mathbf{0}, y_i)$, and similarly $\lim_{a \to \infty} \nabla \ell_i(a) = \lim_{a \to \infty} \nabla \ell\big((D'_{i,*} W_*(a))_{n:1} \mathbf{x}_i, y_i\big) = \nabla \ell(\mathbf{0}, y_i)$. Therefore,

in the limit $a \to \infty$, Equation (80) becomes:

$$\lim_{a\to\infty}\Big(\nabla^2 f(\boldsymbol{\theta}(a))[\Delta W_1,...,\Delta W_n]\Big)$$

$$=\lim_{a\to\infty}\Big(\frac{4}{a^2}\cdot\frac{4}{|\mathcal{S}|}\sum_{i=1}^{|\mathcal{S}|}\nabla^2\ell_i(a)\Big[(D'_{i,*}W_*)_{n:1}\mathbf{x}_i\Big]\Big)+\lim_{a\to\infty}\Big(a\cdot\frac{2}{|\mathcal{S}|}\sum_{i=1}^{|\mathcal{S}|}\nabla\ell_i(a)^\top h_{\boldsymbol{\theta}}(\mathbf{x}_i)\Big)$$

$$=\lim_{a\to\infty}\Big(\frac{4}{a^2}\Big)\cdot\lim_{a\to\infty}\Big(\frac{4}{|\mathcal{S}|}\sum_{i=1}^{|\mathcal{S}|}\nabla^2\ell_i(a)\Big[(D'_{i,*}W_*)_{n:1}\mathbf{x}_i\Big]\Big)+\lim_{a\to\infty}\Big(a\cdot\lim_{a\to\infty}\Big(\frac{2}{|\mathcal{S}|}\sum_{i=1}^{|\mathcal{S}|}\nabla\ell_i(a)^\top h_{\boldsymbol{\theta}}(\mathbf{x}_i)\Big)\Big)$$

$$=0\cdot\Big(\frac{4}{|\mathcal{S}|}\sum_{i=1}^{|\mathcal{S}|}\nabla^2\ell(\mathbf{0},y_i)\Big[(D'_{i,*}W_*)_{n:1}\mathbf{x}_i\Big]\Big)+\lim_{a\to\infty}\Big(a\cdot\frac{2}{|\mathcal{S}|}\sum_{i=1}^{|\mathcal{S}|}\nabla\ell_i(\mathbf{0},y_i)^\top h_{\boldsymbol{\theta}}(\mathbf{x}_i)\Big)$$

$$=-\infty\,,$$

where the second transition is valid since the multiplied limits are finite and the limit inside a limit is non-zero, and the last transition follows from $\sum_{i=1}^{|\mathcal{S}|}\nabla\ell(\mathbf{0},y_i)^\top h_{\boldsymbol{\theta}}(\mathbf{x}_i)<0$. Notice that the matrices $\Delta W_1,\Delta W_2,...,\Delta W_n$ are independent of $a$, thus it must hold that $\lim_{a\to\infty}\lambda_{\min}\big(\nabla^2 f(\boldsymbol{\theta}(a))\big)=-\infty$. This in particular implies the desired result:

$$\inf_{\boldsymbol{\theta}\in\mathbb{R}^d \text{ s.t. }\nabla^2 f(\boldsymbol{\theta})\text{ exists}}\lambda_{\min}(\nabla^2 f(\boldsymbol{\theta}))=-\infty\,.$$

$\square$

## J.14  Proof of Lemma 5

This proof is very similar to that of Lemma 2 (see Subappendix J.6). We repeat all details for completeness. Recall that $\boldsymbol{\theta}\in\mathbb{R}^d$ is an arrangement of $(W_1,W_2,...,W_n)\in\mathbb{R}^{d_1,d_0}\times\mathbb{R}^{d_2,d_1}\times\cdots\times\mathbb{R}^{d_n,d_{n-1}}$ as a vector. Let $(\Delta W_1,\Delta W_2,...,\Delta W_n)\in\mathbb{R}^{d_1,d_0}\times\mathbb{R}^{d_2,d_1}\times\cdots\times\mathbb{R}^{d_n,d_{n-1}}$, and denote by $\Delta\boldsymbol{\theta}\in\mathbb{R}^d$ its arrangement as a vector in corresponding order. As shown in Lemma 4:

$$\nabla^2 f(\boldsymbol{\theta})[\Delta W_1,...,\Delta W_n]=$$

$$\frac{1}{|\mathcal{S}|}\sum_{i=1}^{|\mathcal{S}|}\nabla^2\ell_i\Big[\sum_{j=1}^n(D'_{i,*}W_*)_{n:j+1}D'_{i,j}(\Delta W_j)(D'_{i,*}W_*)_{j\text{-}1:1}\mathbf{x}_i\Big]+$$

$$\frac{2}{|\mathcal{S}|}\sum_{i=1}^{|\mathcal{S}|}\nabla\ell_i^\top\sum_{1\le j<j'\le n}(D'_{i,*}W_*)_{n:j'+1}D'_{i,j'}(\Delta W_{j'})(D'_{i,*}W_*)_{j'\text{-}1:j+1}D'_{i,j}(\Delta W_j)(D'_{i,*}W_*)_{j\text{-}1:1}\mathbf{x}_i\,,$$

where we regard Hessians as quadratic forms (see Subappendix J.1). Convexity of $\ell(\cdot)$ in its first argument implies that for $i\in\{1,2,...,|\mathcal{S}|\}$, $\nabla^2\ell_i$ is positive semi-definite, thus:

$$\nabla^2 f(\boldsymbol{\theta})[\Delta W_1,...,\Delta W_n]\ge$$

$$\frac{2}{|\mathcal{S}|}\sum_{i=1}^{|\mathcal{S}|}\nabla\ell_i^\top\sum_{1\le j<j'\le n}(D'_{i,*}W_*)_{n:j'+1}D'_{i,j'}(\Delta W_{j'})(D'_{i,*}W_*)_{j'\text{-}1:j+1}D'_{i,j}(\Delta W_j)(D'_{i,*}W_*)_{j\text{-}1:1}\mathbf{x}_i\,.$$

Applying Cauchy-Schwarz and triangle inequalities, we get:

$$\nabla^2 f(\boldsymbol{\theta})[\Delta W_1,...,\Delta W_n]$$

$$\ge-\frac{2}{|\mathcal{S}|}\sum_{i=1}^{|\mathcal{S}|}\big\|\nabla\ell_i\big\|_2\sum_{1\le j<j'\le n}\big\|(D'_{i,*}W_*)_{n:j'+1}D'_{i,j'}(\Delta W_{j'})(D'_{i,*}W_*)_{j'\text{-}1:j+1}D'_{i,j}(\Delta W_j)(D'_{i,*}W_*)_{j\text{-}1:1}\mathbf{x}_i\big\|_2$$

$$\ge-\frac{2}{|\mathcal{S}|}\sum_{i=1}^{|\mathcal{S}|}\big\|\nabla\ell_i\big\|_2\sum_{1\le j<j'\le n}\big\|\Delta W_j\big\|_s\big\|\Delta W_{j'}\big\|_s\prod_{k\in[n]/\{j,j'\}}\big\|W_k\big\|_s\prod_{k=1}^n\big\|D'_{i,k}\big\|_s\cdot\big\|\mathbf{x}_i\big\|_2$$

$$\ge-\frac{2}{|\mathcal{S}|}\sum_{i=1}^{|\mathcal{S}|}\big\|\nabla\ell_i\big\|_2\cdot\Big(\max_{\substack{\mathcal{J}\subseteq[n]\\|\mathcal{J}|=n-2}}\prod_{j\in\mathcal{J}}\big\|W_j\big\|_s\Big)\max\{|\alpha|,|\bar{\alpha}|\}^{n-1}\big\|\mathbf{x}_i\big\|_2\sum_{1\le j<j'\le n}\big\|\Delta W_j\big\|_s\big\|\Delta W_{j'}\big\|_s\,,$$

where the second transition follows from the definition and sub-multiplicativity of spectral norm, and the last transition follows from maximizing $\prod_{k\in[n]/\{j,j'\}}\|W_k\|_s$ over $j,j'$, upper bounding $\|D'_{i,j}\|_s \leq \max\{|\alpha|,|\bar\alpha|\}$ for $j \in [n-1]$ and recalling that $D'_{i,n}$ is an identity matrix, meaning $\|D'_{i,n}\|_s = 1$. It holds that:

$$\sum_{1\leq j<j'\leq n}\|\Delta W_{j'}\|_s\|\Delta W_j\|_s$$

$$\leq\sum_{1\leq j<j'\leq n}\|\Delta W_{j'}\|_F\|\Delta W_j\|_F$$

$$=\tfrac{1}{2}\left(\sum_{j=1}^n\|\Delta W_j\|_F\right)^2-\tfrac{1}{2}\sum_{j=1}^n\|\Delta W_j\|_F^2$$

$$\leq\tfrac{n}{2}\sum_{j=1}^n\|\Delta W_j\|_F^2-\tfrac{1}{2}\sum_{j=1}^n\|\Delta W_j\|_F^2$$

$$=\tfrac{n-1}{2}\sum_{j=1}^n\|\Delta W_j\|_F^2\,,$$

where the last inequality follows from the fact that the one-norm of a vector in $\mathbb{R}^n$ is never greater than $\sqrt{n}$ times its euclidean-norm. This leads us to the following bound:

$$\nabla^2 f(\boldsymbol\theta)[\Delta W_1,...,\Delta W_n]$$

$$\geq-\max\{|\alpha|,|\bar\alpha|\}^{n-1}\left(\max_{\substack{\mathcal{J}\subseteq[n]\\|\mathcal{J}|=n-2}}\prod_{j\in\mathcal{J}}\|W_j\|_s\right)\frac{n-1}{|\mathcal{S}|}\sum_{i=1}^{|\mathcal{S}|}\|\nabla\ell_i\|_2\|\mathbf{x}_i\|_2\cdot\sum_{j=1}^n\|\Delta W_j\|_F^2\,.$$

The desired result readily follows:

$$\lambda_{\min}\left(\nabla^2 f(\boldsymbol\theta)\right)\geq-\max\{|\alpha|,|\bar\alpha|\}^{n-1}\cdot\left(\max_{\substack{\mathcal{J}\subseteq[n]\\|\mathcal{J}|=n-2}}\prod_{j\in\mathcal{J}}\|W_j\|_F\right)\frac{n-1}{|\mathcal{S}|}\sum_{i=1}^{|\mathcal{S}|}\|\nabla\ell_i\|_2\|\mathbf{x}_i\|_2\,.$$

$\square$

## J.15  Proof of Proposition 6

Recall that $(W_1,W_2,...,W_n)\in\mathbb{R}^{d_1,d_0}\times\mathbb{R}^{d_2,d_1}\times\cdots\times\mathbb{R}^{d_n,d_{n-1}}$ are the weight matrices constituting $\boldsymbol\theta\in\mathbb{R}^d$, and denote by $(W_{1,s},W_{2,s},...,W_{n,s})\in\mathbb{R}^{d_1,d_0}\times\mathbb{R}^{d_2,d_1}\times\cdots\times\mathbb{R}^{d_n,d_{n-1}}$ those that constitute $\boldsymbol\theta_s$. For $j,j'\in[n]$:

$$\left|\|W_{j,s}\|_F^2-\|W_{j',s}\|_F^2\right|\leq\max\left\{\|W_{j,s}\|_F^2,\|W_{j',s}\|_F^2\right\}\leq\max_{j\in[n]}\|W_{j,s}\|_F^2\leq\|\boldsymbol\theta_s\|_2^2\leq\epsilon^2\,.$$

Corollary 2.1 from [18] implies that throughout a gradient flow trajectory differences between squared Frobenius norms of weight matrices are constant. Therefore, for $j,j'\in[n]$:

$$\left|\|W_j\|_F^2-\|W_{j'}\|_F^2\right|=\left|\|W_{j,s}\|_F^2-\|W_{j',s}\|_F^2\right|\leq\epsilon^2\,. \tag{81}$$

If the network is shallow (*i.e.* $n=2$), then Equation (27) coincides with Equation (26), thus the desired result follows trivially from Lemma 5. Hereafter we assume that the network is deep (*i.e.* $n\geq 3$). It holds that:

$$\max_{\mathcal{J}\subseteq[n],|\mathcal{J}|=n-2}\prod_{j\in\mathcal{J}}\|W_j\|_F\leq\max_{j\in[n]}\|W_j\|_F^{n-2}$$

$$=\left(\min_{j\in[n]}\|W_j\|_F^2+\max_{j\in[n]}\|W_j\|_F^2-\min_{j\in[n]}\|W_j\|_F^2\right)^{\frac{n-2}{2}}$$

$$\leq\left(\min_{j\in[n]}\|W_j\|_F^2+\epsilon^2\right)^{\frac{n-2}{2}}$$

$$=\left(\sqrt{\min_{j\in[n]}\|W_j\|_F^2+\epsilon^2}\right)^{n-2}$$

$$\leq\left(\min_{j\in[n]}\|W_j\|_F+\epsilon\right)^{n-2}\,,$$

where the third transition follows from Equation (81) and the last transition follows from subadditivity of square root. Combining the latter inequality together with the result of Lemma 5 (Equation (26)), we obtain the desired result:

$$\lambda_{\min}\big(\nabla^2 f(\boldsymbol{\theta})\big) \geq -\max\{|\alpha|,|\bar{\alpha}|\}^{n-1}\frac{n-1}{|\mathcal{S}|}\sum_{i=1}^{|\mathcal{S}|}\big\|\nabla\ell_i\big\|_2\big\|\mathbf{x}_i\big\|_2\Big(\min_{j\in[n]}\|W_j\|_F+\epsilon\Big)^{n-2}.$$

$\square$

## J.16  Proof of Lemma 6

This proof is very similar to that of Lemmas 1 and 4 (see Subappendixes J.4 and J.12 respectively). We repeat all details for completeness. Recall that $\boldsymbol{\theta}\in\mathbb{R}^d$ is a concatenation of $(\mathbf{w}_1,\mathbf{w}_2,...,\mathbf{w}_n)\in\mathbb{R}^{d'_1}\times\mathbb{R}^{d'_2}\times\cdots\times\mathbb{R}^{d'_n}$ as a vector. Let $(\Delta\mathbf{w}_1,\Delta\mathbf{w}_2,...,\Delta\mathbf{w}_n)\in\mathbb{R}^{d'_1}\times\mathbb{R}^{d'_2}\times\cdots\times\mathbb{R}^{d'_n}$, and denote by $\Delta\boldsymbol{\theta}\in\mathbb{R}^d$ its concatenation as a vector in corresponding order. Denote the following for $i\in\{1,...,|\mathcal{S}|\}$:

$$\Delta_i^{(1)}:=\sum_{j=1}^n(D'_{i,*}W_*(\mathbf{w}_*))_{n:j+1}D'_{i,j}(W_j(\Delta\mathbf{w}_j))(D'_{i,*}W_*(\mathbf{w}_*))_{j\text{-}1:1},$$

$$\Delta_i^{(2)}:=\sum_{1\leq j<j'\leq n}(D'_{i,*}W_*(\mathbf{w}_*))_{n:j'+1}D'_{i,j'}(W_{j'}(\Delta\mathbf{w}_{j'}))(D'_{i,*}W_*(\mathbf{w}_*))_{j'\text{-}1:j+1}$$
$$D'_{i,j}(W_j(\Delta\mathbf{w}_j))(D'_{i,*}W_*(\mathbf{w}_*))_{j\text{-}1:1},$$

$$\Delta_i^{(3:n)}:=D'_{i,n}(W_n(\mathbf{w}_n)+W_n(\Delta\mathbf{w}_n))\cdots D'_{i,1}(W_1(\mathbf{w}_1)+W_1(\Delta\mathbf{w}_1))$$
$$-(D'_{i,*}W_*(\mathbf{w}_*))_{n:1}-\Delta_i^{(1)}-\Delta_i^{(2)}.$$

$$(82)$$

We now develop a second-order Taylor expansion of $f(\boldsymbol{\theta})$. Since the vector tuple corresponding to $(\boldsymbol{\theta}+\Delta\boldsymbol{\theta})$ is $\big((\mathbf{w}_1+\Delta\mathbf{w}_1),...,(\mathbf{w}_n+\Delta\mathbf{w}_n)\big)$, and the function $f(\cdot)$ coincides with the function given in Equation (29) on an open region containing $\boldsymbol{\theta}$, for sufficiently small $\Delta\boldsymbol{\theta}$ we obtain:

$$f(\boldsymbol{\theta}+\Delta\boldsymbol{\theta})$$
$$=\frac{1}{|\mathcal{S}|}\sum_{i=1}^{|\mathcal{S}|}\ell\Big(D'_{i,n}\big(W_n(\mathbf{w}_n+\Delta\mathbf{w}_n)\big)\cdots D'_{i,1}\big(W_1(\mathbf{w}_1+\Delta\mathbf{w}_1)\big)\mathbf{x}_i,y_i\Big)$$
$$=\frac{1}{|\mathcal{S}|}\sum_{i=1}^{|\mathcal{S}|}\ell\Big(D'_{i,n}\big(W_n(\mathbf{w}_n)+W_n(\Delta\mathbf{w}_n)\big)\cdots D'_{i,1}\big(W_1(\mathbf{w}_1)+W_1(\Delta\mathbf{w}_1)\big)\mathbf{x}_i,y_i\Big)$$
$$=\frac{1}{|\mathcal{S}|}\sum_{i=1}^{|\mathcal{S}|}\ell\Big(\big((D'_{i,*}W_*(\mathbf{w}_*))_{n:1}+\Delta_i^{(1)}+\Delta_i^{(2)}+\Delta_i^{(3:n)}\big)\mathbf{x}_i,y_i\Big)$$
$$=\frac{1}{|\mathcal{S}|}\sum_{i=1}^{|\mathcal{S}|}\ell\Big((D'_{i,*}W_*(\mathbf{w}_*))_{n:1}\mathbf{x}_i+\big(\Delta_i^{(1)}+\Delta_i^{(2)}+\Delta_i^{(3:n)}\big)\mathbf{x}_i,y_i\Big),$$

$$(83)$$

where the second transition follows from linearity of $W_j(\cdot)$ for $j\in\{1,...,n\}$ and the third transition follows from the definition of $\Delta_i^{(3:n)}$ (Equation (82)). Let $\Delta\boldsymbol{v}\in\mathbb{R}^{d_n}$. For every $i\in\{1,...,|\mathcal{S}|\}$, the second-order Taylor expansion of $\ell(\cdot)$ with respect to its first argument at $\big((D'_{i,*}W_*(\mathbf{w}_*))_{n:1}\mathbf{x}_i,y_i\big)$ is given by:

$$\ell\big((D'_{i,*}W_*(\mathbf{w}_*))_{n:1}\mathbf{x}_i+\Delta\boldsymbol{v},y_i\big)=$$
$$\ell\big((D'_{i,*}W_*(\mathbf{w}_*))_{n:1}\mathbf{x}_i,y_i\big)+\big\langle\nabla\ell_i,\Delta\boldsymbol{v}\big\rangle+\tfrac{1}{2}\nabla^2\ell_i[\Delta\boldsymbol{v}]+\mathcal{O}\big(\|\Delta\boldsymbol{v}\|_2^2\big),$$

$$(84)$$

where the $\mathcal{o}(\cdot)$ notation refers to some expression satisfying $\lim_{a\to0}\big(\mathcal{o}(a)/a\big)=0$. We continue to develop Equation (83) using Equation (84):

$$f(\boldsymbol{\theta}+\Delta\boldsymbol{\theta})$$

$$=\frac{1}{|\mathcal{S}|}\sum_{i=1}^{|\mathcal{S}|}\Big(\ell\big((D'_{i,*}W_*(\mathbf{w}_*))_{n:1}\mathbf{x}_i,y_i\big)+\big\langle\nabla\ell_i,(\Delta_i^{(1)}+\Delta_i^{(2)}+\Delta_i^{(3:n)})\mathbf{x}_i\big\rangle+$$

$$\frac{1}{2}\nabla^2\ell_i\big[(\Delta_i^{(1)}+\Delta_i^{(2)}+\Delta_i^{(3:n)})\mathbf{x}_i\big]+\mathcal{o}\big(\big\|(\Delta_i^{(1)}+\Delta_i^{(2)}+\Delta_i^{(3:n)})\mathbf{x}_i\big\|_2^2\big)\Big)$$

$$=\frac{1}{|\mathcal{S}|}\sum_{i=1}^{|\mathcal{S}|}\ell\big((D'_{i,*}W_*(\mathbf{w}_*))_{n:1}\mathbf{x}_i,y_i\big)+$$

$$\frac{1}{|\mathcal{S}|}\sum_{i=1}^{|\mathcal{S}|}\big\langle\nabla\ell_i,\Delta_i^{(1)}\mathbf{x}_i\big\rangle+\big\langle\nabla\ell_i,\Delta_i^{(2)}\mathbf{x}_i\big\rangle+\big\langle\nabla\ell_i,\Delta_i^{(3:n)}\mathbf{x}_i\big\rangle+$$

$$\frac{1}{|\mathcal{S}|}\sum_{i=1}^{|\mathcal{S}|}\frac{1}{2}\nabla^2\ell_i\big[\Delta_i^{(1)}\mathbf{x}_i\big]+\frac{1}{2}\nabla^2\ell_i\big[(\Delta_i^{(2)}+\Delta_i^{(3:n)})\mathbf{x}_i\big]+2\cdot\frac{1}{2}\nabla^2\ell_i\big[\Delta_i^{(1)}\mathbf{x}_i,(\Delta_i^{(2)}+\Delta_i^{(3:n)})\mathbf{x}_i\big]+$$

$$\frac{1}{|\mathcal{S}|}\sum_{i=1}^{|\mathcal{S}|}\mathcal{o}\big(\big\|(\Delta_i^{(1)}+\Delta_i^{(2)}+\Delta_i^{(3:n)})\mathbf{x}_i\big\|_2^2\big)\,,$$

where in the last transition we view $\nabla^2\ell_i$ as both a quadratic and a bilinear form (see Subappendix J.1). Notice that $\big\langle\nabla\ell_i,\Delta_i^{(3:n)}\mathbf{x}_i\big\rangle$, $\frac{1}{2}\nabla^2\ell_i\big[(\Delta_i^{(2)}+\Delta_i^{(3:n)})\mathbf{x}_i\big]$, $\nabla^2\ell_i\big[\Delta_i^{(1)}\mathbf{x}_i,(\Delta_i^{(2)}+\Delta_i^{(3:n)})\mathbf{x}_i\big]$ and $\mathcal{o}\big(\big\|(\Delta_i^{(1)}+\Delta_i^{(2)}+\Delta_i^{(3:n)})\mathbf{x}_i\big\|_2^2\big)$ are all $\mathcal{o}\big(\|\Delta\boldsymbol{\theta}\|_2^2\big)$, thus:

$$f(\boldsymbol{\theta}+\Delta\boldsymbol{\theta})=$$

$$\frac{1}{|\mathcal{S}|}\sum_{i=1}^{|\mathcal{S}|}\ell\big((D'_{i,*}W_*(\mathbf{w}_*))_{n:1}\mathbf{x}_i,y_i\big)+\big\langle\nabla\ell_i,\Delta_i^{(1)}\mathbf{x}_i\big\rangle+\big\langle\nabla\ell_i,\Delta_i^{(2)}\mathbf{x}_i\big\rangle+\frac{1}{2}\nabla^2\ell_i\big[\Delta_i^{(1)}\mathbf{x}_i\big]+\mathcal{o}\big(\|\Delta\boldsymbol{\theta}\|_2^2\big).$$

This is in fact a Taylor expansion of the function $f(\cdot)$ evaluated at the point $\boldsymbol{\theta}$ with a constant term $\frac{1}{|\mathcal{S}|}\sum_{i=1}^{|\mathcal{S}|}\ell\big((D'_{i,*}W_*(\mathbf{w}_*))_{n:1}\mathbf{x}_i,y_i\big)$, a linear term $\frac{1}{|\mathcal{S}|}\sum_{i=1}^{|\mathcal{S}|}\big\langle\nabla\ell_i,\Delta_i^{(1)}\mathbf{x}_i\big\rangle$, a quadratic term $\frac{1}{|\mathcal{S}|}\sum_{i=1}^{|\mathcal{S}|}\big\langle\nabla\ell_i,\Delta_i^{(2)}\mathbf{x}_i\big\rangle+\frac{1}{2}\nabla^2\ell_i\big[\Delta_i^{(1)}\mathbf{x}_i\big]$, and a remainder term of $\mathcal{o}\big(\|\Delta\boldsymbol{\theta}\|_2^2\big)$. From uniqueness of the Taylor expansion, the quadratic term must be equal to $\frac{1}{2}\nabla^2f(\boldsymbol{\theta})[\Delta\mathbf{w}_1,...,\Delta\mathbf{w}_n]$. This implies:

$$\nabla^2f(\boldsymbol{\theta})[\Delta\mathbf{w}_1,...,\Delta\mathbf{w}_n]$$

$$=\frac{1}{|\mathcal{S}|}\sum_{i=1}^{|\mathcal{S}|}\Big(\nabla^2\ell_i\big[\Delta_i^{(1)}\mathbf{x}_i\big]+2\big\langle\nabla\ell_i,\Delta_i^{(2)}\mathbf{x}_i\big\rangle\Big)$$

$$=\frac{1}{|\mathcal{S}|}\sum_{i=1}^{|\mathcal{S}|}\nabla^2\ell_i\Big[\sum_{j=1}^{n}(D'_{i,*}W_*(\mathbf{w}_*))_{n:j+1}D'_{i,j}(W_j(\Delta\mathbf{w}_j))(D'_{i,*}W_*(\mathbf{w}_*))_{j\text{-}1:1}\mathbf{x}_i\Big]+$$

$$\frac{2}{|\mathcal{S}|}\sum_{i=1}^{|\mathcal{S}|}\nabla\ell_i^\top\sum_{1\le j<j'\le n}(D'_{i,*}W_*(\mathbf{w}_*))_{n:j'+1}D'_{i,j'}(W_{j'}(\Delta\mathbf{w}_{j'}))(D'_{i,*}W_*(\mathbf{w}_*))_{j'\text{-}1:j+1}\cdot$$

$$D'_{i,j}(W_j(\Delta\mathbf{w}_j))(D'_{i,*}W_*(\mathbf{w}_*))_{j\text{-}1:1}\mathbf{x}_i\,,$$

where the last transition follows from plugging in the definitions of $\Delta^{(1)}$ and $\Delta^{(2)}$ (see Equation (82)). $\square$

## J.17    Proof of Proposition 7

This proof is very similar to that of Proposition 5 (see Subappendix J.13). We repeat all details for completeness. From assumption *(ii)* there exists some $\boldsymbol{\theta}\in\mathbb{R}^d$ such that $\sum_{i=1}^{|\mathcal{S}|}\nabla\ell(\mathbf{0},y_i)^\top h_{\boldsymbol{\theta}}(\mathbf{x}_i)\ne0$.

Define $(\mathbf{w}_1, \mathbf{w}_2, ..., \mathbf{w}_n) \in \mathbb{R}^{d'_1} \times \mathbb{R}^{d'_2} \times \cdots \times \mathbb{R}^{d'_n}$ to be the weight vectors constituting $\boldsymbol{\theta}$. We may assume $\sum_{i=1}^{|\mathcal{S}|} \nabla \ell(\mathbf{0}, y_i)^\top h_{\boldsymbol{\theta}}(\mathbf{x}_i) < 0$ without loss of generality, as we can negate the vectors $h_{\boldsymbol{\theta}}(\mathbf{x}_i) \in \mathbb{R}^{d_n}$ for all $i \in \{1, 2, ..., |\mathcal{S}|\}$ by flipping the signs of the entries in $\boldsymbol{\theta}$ corresponding to the last vector $\mathbf{w}_n$ (see Equation (28)). From continuity, there exists a neighborhood $\mathcal{N}$ of $\boldsymbol{\theta}$ such that for all $\tilde{\boldsymbol{\theta}} \in \mathcal{N}$ it holds that $\sum_{i=1}^{|\mathcal{S}|} \nabla \ell(\mathbf{0}, y_i)^\top h_{\tilde{\boldsymbol{\theta}}}(\mathbf{x}_i) < 0$. Moreover, as discussed in Appendix D, for almost all $\boldsymbol{\theta}' \in \mathbb{R}^d$ there exists an open region $\mathcal{D}_{\boldsymbol{\theta}'} \subseteq \mathbb{R}^d$ containing $\boldsymbol{\theta}'$, which is closed under positive rescaling of weight matrices and across which $f(\cdot)$ coincides with a function as given in Equation (29). There must exist some $\boldsymbol{\theta}'$ in the neighborhood $\mathcal{N}$ for which a region of the type $\mathcal{D}_{\boldsymbol{\theta}'}$ exists. We may assume, without loss of generality, that $\boldsymbol{\theta} \in \mathcal{D}_{\boldsymbol{\theta}'}$. Notice that none of the weight vectors $\mathbf{w}_1, \mathbf{w}_2, ..., \mathbf{w}_n$ are equal to zero (as that would lead to $\sum_{i=1}^{|\mathcal{S}|} \nabla \ell(\mathbf{0}, y_i)^\top h_{\boldsymbol{\theta}}(\mathbf{x}_i) = 0$). Define the following weight vectors parameterized by $a > 0$ (while recalling that $n \geq 3$ by assumption *(i)*):

$$\mathbf{w}_1(a) := \mathbf{w}_1 \cdot a^{-2} \in \mathbb{R}^{d'_1},$$
$$\mathbf{w}_2(a) := \mathbf{w}_2 \cdot a^{-2} \in \mathbb{R}^{d'_2},$$
$$\mathbf{w}_3(a) := \mathbf{w}_3 \cdot a \in \mathbb{R}^{d'_3},$$
$$\mathbf{w}_j(a) := \mathbf{w}_j \in \mathbb{R}^{d'_j} \text{ for } j \in \{1, 2, ..., n\}/\{1, 2, 3\},$$

and denote by $\boldsymbol{\theta}(a) \in \mathbb{R}^d$ their corresponding weight setting. Since $\mathcal{D}_{\boldsymbol{\theta}'}$ is closed under positive rescaling of weight vectors, it holds that $\{\boldsymbol{\theta}(a) : a > 0\} \subseteq \mathcal{D}_{\boldsymbol{\theta}'}$. Define:

$$\Delta \mathbf{w}_1 := \mathbf{w}_1 \in \mathbb{R}^{d'_1},$$
$$\Delta \mathbf{w}_2 := \mathbf{w}_2 \in \mathbb{R}^{d'_2},$$
$$\Delta \mathbf{w}_j := \mathbf{0} \in \mathbb{R}^{d'_j} \text{ for } j \in [n]/\{1, 2\}.$$

For $a > 0$, $i \in \{1, 2, ..., |\mathcal{S}|\}$ and $j, j' \in \{1, 2, ..., n\}$, define $(D'_{i,*} W_*(\mathbf{w}_*(a)))_{j':j}$ to be the matrix $D'_{i,j'} W_{j'}(\mathbf{w}_{j'}(a)) D'_{i,j'-1} W_{j'-1}(\mathbf{w}_{j'-1}(a)) \cdots D'_{i,j} W_j(\mathbf{w}_j(a))$ (where by convention $D'_{i,n} \in \mathbb{R}^{d_n, d_n}$ stands for identity) if $j \leq j'$, and an identity matrix (with size to be inferred by context) otherwise. For $i \in \{1, 2, ..., |\mathcal{S}|\}$ and $a > 0$ let $\nabla \ell_i(a) \in \mathbb{R}^{d_n}$ and $\nabla^2 \ell_i(a) \in \mathbb{R}^{d_n, d_n}$ be the gradient and Hessian (respectively) of the loss $\ell(\cdot)$ at the point $\left((D'_{i,*} W_*(\mathbf{w}_*(a)))_{n:1} \mathbf{x}_i, y_i\right)$ with respect to its first argument. For every $a > 0$, since $\boldsymbol{\theta}(a) \in \mathcal{D}_{\boldsymbol{\theta}'}$, we may apply Lemma 6, obtaining:

$$\nabla^2 f\big(\boldsymbol{\theta}(a)\big)[\Delta \mathbf{w}_1, ..., \Delta \mathbf{w}_n] =$$
$$\frac{1}{|\mathcal{S}|} \sum_{i=1}^{|\mathcal{S}|} \nabla^2 \ell_i(a) \Big[ \sum_{j=1}^n (D'_{i,*} W_*(\mathbf{w}_*(a)))_{n:j+1} D'_{i,j} (W_j(\Delta \mathbf{w}_j))(D'_{i,*} W_*(\mathbf{w}_*(a)))_{j-1:1} \mathbf{x}_i \Big] +$$
$$\frac{2}{|\mathcal{S}|} \sum_{i=1}^{|\mathcal{S}|} \nabla \ell_i(a)^\top \sum_{1 \leq j < j' \leq n} (D'_{i,*} W_*(\mathbf{w}_*(a)))_{n:j'+1} D'_{i,j'} (W_{j'}(\Delta \mathbf{w}_{j'}))(D'_{i,*} W_*(\mathbf{w}_*(a)))_{j'-1:j+1}$$
$$D'_{i,j} (W_j(\Delta \mathbf{w}_j))(D'_{i,*} W_*(\mathbf{w}_*(a)))_{j-1:1} \mathbf{x}_i,$$

(85)

where we regard Hessians as quadratic forms (see Subappendix J.1). Plugging in the definitions of $\mathbf{w}_j(a)$ and $\Delta \mathbf{w}_j$ for $j \in [n]$ and relying on linearity of $W_j(\cdot)$ for $j \in [n]$, we have:

$$\nabla^2 f\big(\boldsymbol{\theta}(a)\big)[\Delta \mathbf{w}_1, ..., \Delta \mathbf{w}_n]$$
$$= \frac{1}{|\mathcal{S}|} \sum_{i=1}^{|\mathcal{S}|} \nabla^2 \ell_i(a) \Big[ 2a^{-1} (D'_{i,*} W_*(\mathbf{w}_*))_{n:1} \mathbf{x}_i \Big] + \frac{2}{|\mathcal{S}|} \sum_{i=1}^{|\mathcal{S}|} \nabla \ell_i(a)^\top a (D'_{i,*} W_*(\mathbf{w}_*))_{n:1} \mathbf{x}_i$$
$$= \frac{4}{a^2} \cdot \frac{1}{|\mathcal{S}|} \sum_{i=1}^{|\mathcal{S}|} \nabla^2 \ell_i(a) \Big[ (D'_{i,*} W_*(\mathbf{w}_*))_{n:1} \mathbf{x}_i \Big] + a \cdot \frac{2}{|\mathcal{S}|} \sum_{i=1}^{|\mathcal{S}|} \nabla \ell_i(a)^\top h_{\boldsymbol{\theta}}(\mathbf{x}_i),$$

(86)

where the second transition follows from pulling $2/a$ out of the quadratic operator and the fact that $h_{\boldsymbol{\theta}}(\mathbf{x}_i) = (D'_{i,*} W_*(\mathbf{w}_*))_{n:1} \mathbf{x}_i$. Note that $\lim_{a \to \infty} (D'_{i,*} W_*(\mathbf{w}_*(a)))_{n:1} = 0$. Since the function $\ell(\cdot)$ is twice continuously differentiable in its first argument, it holds that $\lim_{a \to \infty} \nabla^2 \ell_i(a) = \lim_{a \to \infty} \nabla^2 \ell\big((D'_{i,*} W_*(\mathbf{w}_*(a)))_{n:1} \mathbf{x}_i, y_i\big) = \nabla^2 \ell(\mathbf{0}, y_i)$, and similarly we

have that $\lim_{a\to\infty}\nabla\ell_i(a)=\lim_{a\to\infty}\nabla\ell\big((D'_{i,*}W_*(\mathbf{w}_*(a)))_{n:1}\mathbf{x}_i,y_i\big)=\nabla\ell(\mathbf{0},y_i)$. Therefore, in the limit $a\to\infty$, Equation (86) becomes:

$$
\begin{aligned}
&\lim_{a\to\infty}\Big(\nabla^2 f\big(\boldsymbol{\theta}(a)\big)[\Delta\mathbf{w}_1,...,\Delta\mathbf{w}_n]\Big)\\
&=\lim_{a\to\infty}\Big(\frac{4}{a^2}\cdot\frac{4}{|\mathcal{S}|}\sum_{i=1}^{|\mathcal{S}|}\nabla^2\ell_i(a)\Big[(D'_{i,*}W_*(\mathbf{w}_*))_{n:1}\mathbf{x}_i\Big]\Big)+\lim_{a\to\infty}\Big(a\cdot\frac{2}{|\mathcal{S}|}\sum_{i=1}^{|\mathcal{S}|}\nabla\ell_i(a)^\top h_{\boldsymbol{\theta}}(\mathbf{x}_i)\Big)\\
&=\lim_{a\to\infty}\Big(\frac{4}{a^2}\Big)\lim_{a\to\infty}\Big(\frac{4}{|\mathcal{S}|}\sum_{i=1}^{|\mathcal{S}|}\nabla^2\ell_i(a)\Big[(D'_{i,*}W_*(\mathbf{w}_*))_{n:1}\mathbf{x}_i\Big]\Big)+\lim_{a\to\infty}\Big(a\lim_{a\to\infty}\Big(\frac{2}{|\mathcal{S}|}\sum_{i=1}^{|\mathcal{S}|}\nabla\ell_i(a)^\top h_{\boldsymbol{\theta}}(\mathbf{x}_i)\Big)\Big)\\
&=0\cdot\Big(\frac{4}{|\mathcal{S}|}\sum_{i=1}^{|\mathcal{S}|}\nabla^2\ell(\mathbf{0},y_i)\Big[(D'_{i,*}W_*(\mathbf{w}_*))_{n:1}\mathbf{x}_i\Big]\Big)+\lim_{a\to\infty}\Big(a\cdot\frac{2}{|\mathcal{S}|}\sum_{i=1}^{|\mathcal{S}|}\nabla\ell_i(\mathbf{0},y_i)^\top h_{\boldsymbol{\theta}}(\mathbf{x}_i)\Big)\\
&=-\infty\,,
\end{aligned}
$$

where the second transition is valid since the multiplied limits are finite and the limit inside a limit is non-zero, and the last transition follows from $\sum_{i=1}^{|\mathcal{S}|}\nabla\ell(\mathbf{0},y_i)^\top h_{\boldsymbol{\theta}}(\mathbf{x}_i)<0$. Notice that the vectors $\Delta\mathbf{w}_1,\Delta\mathbf{w}_2,...,\Delta\mathbf{w}_n$ are independent of $a$, thus it must hold that $\lim_{a\to\infty}\lambda_{\min}\big(\nabla^2 f\big(\boldsymbol{\theta}(a)\big)\big)=-\infty$. This in particular implies the desired result:

$$
\inf_{\boldsymbol{\theta}\in\mathbb{R}^d\ s.t.\nabla^2 f(\boldsymbol{\theta})\ exists}\lambda_{\min}(\nabla^2 f(\boldsymbol{\theta}))=-\infty\,.
$$

$\square$

## J.18 Proof of Lemma 7

This proof is very similar to that of Lemmas 2 and 5 (see Subappendixes J.6 and J.14 respectively). We repeat all details for completeness. Recall that $\boldsymbol{\theta}\in\mathbb{R}^d$ is a concatenation of $(\mathbf{w}_1,\mathbf{w}_2,...,\mathbf{w}_n)\in\mathbb{R}^{d'_1}\times\mathbb{R}^{d'_2}\times\cdots\times\mathbb{R}^{d'_n}$ as a vector. Let $(\Delta\mathbf{w}_1,\Delta\mathbf{w}_2,...,\Delta\mathbf{w}_n)\in\mathbb{R}^{d'_1}\times\mathbb{R}^{d'_2}\times\cdots\times\mathbb{R}^{d'_n}$, and denote by $\Delta\boldsymbol{\theta}\in\mathbb{R}^d$ its concatenation as a vector in corresponding order. As shown in Lemma 6:

$$
\nabla^2 f(\boldsymbol{\theta})[\Delta\mathbf{w}_1,...,\Delta\mathbf{w}_n]=
$$
$$
\frac{1}{|\mathcal{S}|}\sum_{i=1}^{|\mathcal{S}|}\nabla^2\ell_i\Big[\sum_{j=1}^n(D'_{i,*}W_*(\mathbf{w}_*))_{n:j+1}D'_{i,j}(W_j(\Delta\mathbf{w}_j))(D'_{i,*}W_*(\mathbf{w}_*))_{j\text{-}1:1}\mathbf{x}_i\Big]+
$$
$$
\frac{2}{|\mathcal{S}|}\sum_{i=1}^{|\mathcal{S}|}\nabla\ell_i^\top\sum_{1\le j<j'\le n}(D'_{i,*}W_*(\mathbf{w}_*))_{n:j'+1}D'_{i,j'}(W_{j'}(\Delta\mathbf{w}_{j'}))(D'_{i,*}W_*(\mathbf{w}_*))_{j'\text{-}1:j+1}\cdot
$$
$$
D'_{i,j}(W_j(\Delta\mathbf{w}_j))(D'_{i,*}W_*(\mathbf{w}_*))_{j\text{-}1:1}\mathbf{x}_i\,,
$$

where we regard Hessians as quadratic forms (see Subappendix J.1). Convexity of $\ell(\cdot)$ in its first argument implies that for $i\in\{1,2,...,|\mathcal{S}|\}$, $\nabla^2\ell_i$ is positive semi-definite, thus:

$$
\nabla^2 f(\boldsymbol{\theta})[\Delta\mathbf{w}_1,...,\Delta\mathbf{w}_n]\ge
$$
$$
\frac{2}{|\mathcal{S}|}\sum_{i=1}^{|\mathcal{S}|}\nabla\ell_i^\top\sum_{1\le j<j'\le n}(D'_{i,*}W_*(\mathbf{w}_*))_{n:j'+1}D'_{i,j'}(W_{j'}(\Delta\mathbf{w}_{j'}))(D'_{i,*}W_*(\mathbf{w}_*))_{j'\text{-}1:j+1}\cdot
$$
$$
D'_{i,j}(W_j(\Delta\mathbf{w}_j))(D'_{i,*}W_*(\mathbf{w}_*))_{j\text{-}1:1}\mathbf{x}_i\,.
$$

Applying Cauchy-Schwarz and triangle inequalities, we get:

$$\nabla^2 f(\boldsymbol{\theta})[\Delta\mathbf{w}_1,...,\Delta\mathbf{w}_n]$$

$$\geq -\frac{2}{|\mathcal{S}|}\sum_{i=1}^{|\mathcal{S}|}\|\nabla\ell_i\|_2\sum_{1\leq j<j'\leq n}\left\|(D'_{i,*}W_*(\mathbf{w}_*))_{n:j'+1}D'_{i,j'}(W_{j'}(\Delta\mathbf{w}_{j'}))(D'_{i,*}W_*(\mathbf{w}_*))_{j'-1:j+1}\cdot\right.$$

$$\left.D'_{i,j}(W_j(\Delta\mathbf{w}_j))(D'_{i,*}W_*(\mathbf{w}_*))_{j-1:1}\mathbf{x}_i\right\|_2$$

$$\geq -\frac{2}{|\mathcal{S}|}\sum_{i=1}^{|\mathcal{S}|}\|\nabla\ell_i\|_2\sum_{1\leq j<j'\leq n}\|W_j(\Delta\mathbf{w}_j)\|_s\|W_{j'}(\Delta\mathbf{w}_{j'})\|_s\prod_{k\in[n]/\{j,j'\}}\|W_k(\mathbf{w}_k)\|_s\prod_{k\in[n]}\|D'_{i,k}\|_s\cdot\|\mathbf{x}_i\|_2$$

$$\geq -\frac{2}{|\mathcal{S}|}\sum_{i=1}^{|\mathcal{S}|}\|\nabla\ell_i\|_2\cdot\prod_{j=1}^n\|W_j(\cdot)\|_{op}\sum_{1\leq j<j'\leq n}\|\Delta\mathbf{w}_j\|_2\|\Delta\mathbf{w}_{j'}\|_2\prod_{k\in[n]/\{j,j'\}}\|\mathbf{w}_k\|_2\prod_{k=1}^n\|D'_{i,k}\|_s\cdot\|\mathbf{x}_i\|_2$$

$$\geq -\frac{2}{|\mathcal{S}|}\sum_{i=1}^{|\mathcal{S}|}\|\nabla\ell_i\|_2\prod_{j=1}^n\|W_j(\cdot)\|_{op}\max_{\substack{\mathcal{J}\subseteq[n]\\|\mathcal{J}|=n-2}}\prod_{j\in\mathcal{J}}\|\mathbf{w}_j\|_2\max\{|\alpha|,|\bar{\alpha}|\}^{n-1}\|\mathbf{x}_i\|_2\sum_{1\leq j<j'\leq n}\|\Delta\mathbf{w}_j\|_2\|\Delta\mathbf{w}_{j'}\|_2,$$

where the second transition follows from the definition and sub-multiplicativity of spectral norm, the third transition follows from bounding spectral norms with Frobenius norms and the definition of $\|W_j(\cdot)\|_{op}$, and the last transition follows from maximizing $\prod_{k\in[n]/\{j,j'\}}\|\mathbf{w}_k\|_2$ over $j,j'$, upper bounding $\|D'_{i,j}\|_s\leq\max\{|\alpha|,|\bar{\alpha}|\}$ for $j\in[n-1]$ and recalling that $D'_{i,n}$ is an identity matrix, meaning $\|D'_{i,n}\|_s=1$. It holds that:

$$\sum_{1\leq j<j'\leq n}\|\Delta\mathbf{w}_{j'}\|_2\|\Delta\mathbf{w}_j\|_2$$
$$=\frac{1}{2}\left(\sum_{j=1}^n\|\Delta\mathbf{w}_j\|_2\right)^2-\frac{1}{2}\sum_{j=1}^n\|\Delta\mathbf{w}_j\|_2^2$$
$$\leq\frac{n}{2}\sum_{j=1}^n\|\Delta\mathbf{w}_j\|_2^2-\frac{1}{2}\sum_{j=1}^n\|\Delta\mathbf{w}_j\|_2^2$$
$$=\frac{n-1}{2}\sum_{j=1}^n\|\Delta\mathbf{w}_j\|_2^2,$$

where the last inequality follows from the fact that the one-norm of a vector in $\mathbb{R}^n$ is never greater than $\sqrt{n}$ times its euclidean-norm. This leads us to the following bound:

$$\nabla^2 f(\boldsymbol{\theta})[\Delta\mathbf{w}_1,...,\Delta\mathbf{w}_n]\geq$$

$$-\max\{|\alpha|,|\bar{\alpha}|\}^{n-1}\prod_{j=1}^n\|W_j(\cdot)\|_{op}\max_{\substack{\mathcal{J}\subseteq[n]\\|\mathcal{J}|=n-2}}\prod_{j\in\mathcal{J}}\|\mathbf{w}_j\|_2\frac{n-1}{|\mathcal{S}|}\sum_{i=1}^{|\mathcal{S}|}\|\nabla\ell_i\|_2\|\mathbf{x}_i\|_2\sum_{j=1}^n\|\Delta\mathbf{w}_j\|_2^2.$$

The desired result readily follows:

$$\lambda_{\min}\left(\nabla^2 f(\boldsymbol{\theta})\right)\geq -\max\{|\alpha|,|\bar{\alpha}|\}^{n-1}\frac{n-1}{|\mathcal{S}|}\sum_{i=1}^{|\mathcal{S}|}\|\nabla\ell_i\|_2\|\mathbf{x}_i\|_2\prod_{j=1}^n\|W_j(\cdot)\|_{op}\max_{\substack{\mathcal{J}\subseteq[n]\\|\mathcal{J}|=n-2}}\prod_{j\in\mathcal{J}}\|\mathbf{w}_j\|_2.$$

$\square$

### J.19    Proof of Proposition 8

This proof is very similar to that of Proposition 6 (see Subappendix J.15). Recall that $(\mathbf{w}_1,\mathbf{w}_2,...,\mathbf{w}_n)\in\mathbb{R}^{d'_1}\times\mathbb{R}^{d'_2}\times\cdots\times\mathbb{R}^{d'_n}$ are the weight vectors constituting $\boldsymbol{\theta}\in\mathbb{R}^d$, and denote by $(\mathbf{w}_{1,s},\mathbf{w}_{2,s},...,\mathbf{w}_{n,s})\in\mathbb{R}^{d'_1}\times\mathbb{R}^{d'_2}\times\cdots\times\mathbb{R}^{d'_n}$ those that constitute $\boldsymbol{\theta}_s$. For $j,j'\in[n]$:

$$\left|\|\mathbf{w}_{j,s}\|_2^2-\|\mathbf{w}_{j',s}\|_2^2\right|\leq\max\left\{\|\mathbf{w}_{j,s}\|_2^2,\|\mathbf{w}_{j',s}\|_2^2\right\}\leq\max_{j\in[n]}\|\mathbf{w}_{j,s}\|_2^2\leq\|\boldsymbol{\theta}_s\|_2^2\leq\epsilon^2.$$

Theorem 2.3 from [18] implies that throughout a gradient flow trajectory differences between squared Euclidean norms of weight vectors are constant. Therefore, for $j,j'\in[n]$:

$$\left|\|\mathbf{w}_j\|_2^2-\|\mathbf{w}_{j'}\|_2^2\right|=\left|\|\mathbf{w}_{j,s}\|_2^2-\|\mathbf{w}_{j',s}\|_2^2\right|\leq\epsilon^2. \tag{87}$$

If the network is shallow (*i.e.* $n = 2$), then Equation (32) coincides with Equation (31), thus the desired result follows trivially from Lemma 7. Hereafter we assume that the network is deep (*i.e.* $n \geq 3$). It holds that:

$$
\max_{\mathcal{J} \subseteq [n], |\mathcal{J}| = n-2} \prod_{j \in \mathcal{J}} \|\mathbf{w}_j\|_2 \leq \max_{j \in [n]} \|\mathbf{w}_j\|_2^{n-2}
$$

$$
= \left( \min_{j \in [n]} \|\mathbf{w}_j\|_2^2 + \max_{j \in [n]} \|\mathbf{w}_j\|_2^2 - \min_{j \in [n]} \|\mathbf{w}_j\|_2^2 \right)^{\frac{n-2}{2}}
$$

$$
\leq \left( \min_{j \in [n]} \|\mathbf{w}_j\|_2^2 + \epsilon^2 \right)^{\frac{n-2}{2}}
$$

$$
= \left( \sqrt{\min_{j \in [n]} \|\mathbf{w}_j\|_2^2 + \epsilon^2} \right)^{n-2}
$$

$$
\leq \left( \min_{j \in [n]} \|\mathbf{w}_j\|_2 + \epsilon \right)^{n-2} ,
$$

where the third transition follows from Equation (87) and the last transition follows from subadditivity of square root. Combining the latter inequality together with the result of Lemma 7 (Equation (31)), we obtain the desired result:

$$
\lambda_{\min}\left( \nabla^2 f(\boldsymbol{\theta}) \right) \geq -\max\{|\alpha|, |\bar{\alpha}|\}^{n-1} \frac{n-1}{|\mathcal{S}|} \sum_{i=1}^{|\mathcal{S}|} \|\nabla \ell_i\|_2 \|\mathbf{x}_i\|_2 \prod_{j \in [n]} \|W_j(\cdot)\|_{op} \left( \min_{j \in [n]} \|\mathbf{w}_j\|_2 + \epsilon \right)^{n-2} .
$$

$\square$

## J.20  Proof of Lemma 8

In this proof we overload the definition of unbalancedness magnitude (Definition 1) to account for arbitrary matrix dimensions, namely, for any matrices $A_1, ..., A_n$ such that the product $A_n \cdots A_1$ is defined, we refer to $\max_{j \in [n-1]} \|A_{j+1}^\top A_{j+1} - A_j A_j^\top\|_n$ as their unbalancedness magnitude. Recall that $\boldsymbol{\theta} \in \mathbb{R}^d$ is the arrangement of $W_1, W_2, ..., W_{n-1} \in \mathbb{R}^{d_0, d_0}$ and $W_n \in \mathbb{R}^{d_n, d_0}$ as a vector. Define the matrices $B_1, B_2, ..., B_n \in \mathbb{R}^{d_0, d_0}$ as follows: $B_j := W_j$ for $j \in [n-1]$ and $B_n := \sqrt{W_n^\top W_n}$. Notice that:

$$
B_n^\top B_n = \sqrt{W_n^\top W_n} \sqrt{W_n^\top W_n} = W_n^\top W_n ,
$$

thus the unbalancedness magnitude of $B_1, ..., B_n$ is equal to that of $W_1, ..., W_n$, *i.e.* to $\hat{\epsilon}$. Define the matrices $C_1, C_2, ..., C_n \in \mathbb{R}^{d_0, d_0}$ by transposing and reversing the order of $B_1, ..., B_n$, formally: $C_j := B_{n-j+1}^\top$ for $j \in [n]$. Notice that transposition and order reversal do not change the unbalancedness magnitude. Namely, since for $j \in [n-1]$ we have that $\|C_{j+1}^\top C_{j+1} - C_j C_j^\top\|_n = \|B_{n-j} B_{n-j}^\top - B_{n-j+1}^\top B_{n-j+1}\|_n$, the unbalancedness magnitude of $C_1, ..., C_n$ is equal to that of $B_1, ..., B_n$, *i.e.* to $\hat{\epsilon}$. Applying Lemma 1 from [46] to $C_1, ..., C_n$, we conclude that there exists $\hat{C}_1, ..., \hat{C}_n \in \mathbb{R}^{d_0, d_0}$ which are balanced (*i.e.* have unbalancedness magnitude zero), such that $\|C_j - \hat{C}_j\|_F \leq (j-1)\sqrt{\hat{\epsilon}}$ for $j \in [n]$. Pay special notice to the fact that $\hat{C}_1 = C_1$ (as the Frobenius norm of the discrepancy is zero). Define the matrices $\hat{B}_1, \hat{B}_2, ..., \hat{B}_n \in \mathbb{R}^{d_0, d_0}$ by transposing and reversing the order of $\hat{C}_1, ..., \hat{C}_n$, formally: $\hat{B}_j := \hat{C}_{n-j+1}^\top$ for $j \in [n]$. Relying again on the fact that transposition and order reversal do not change unbalancedness magnitude, we have that $\hat{B}_1, ..., \hat{B}_n$, similarly to $\hat{C}_1, ..., \hat{C}_n$, are balanced. Define the matrices $\hat{W}_1, \hat{W}_2, ..., \hat{W}_{n-1} \in \mathbb{R}^{d_0, d_0}$ and $\hat{W}_n \in \mathbb{R}^{d_n, d_0}$ as follows: $\hat{W}_j := \hat{B}_j$ for $j \in [n-1]$ and $\hat{W}_n := W_n$. Notice that the dimensions of $\hat{W}_1, \hat{W}_2, ..., \hat{W}_n$ correspond to those of $W_1, W_2, ..., W_n$, and in particular that these are valid weight matrices. We denote their corresponding weight setting by $\hat{\boldsymbol{\theta}} \in \mathbb{R}^d$. Notice that:

$$
\hat{W}_n^\top \hat{W}_n = W_n^\top W_n = \sqrt{W_n^\top W_n} \sqrt{W_n^\top W_n} = B_n B_n = C_1 C_1 = \hat{C}_1 \hat{C}_1 = \hat{B}_n \hat{B}_n ,
$$

which means that $\hat{W}_1, \hat{W}_2, ..., \hat{W}_n$ are balanced, as they have the same unbalancedness magnitude of $\hat{B}_1, ..., \hat{B}_n$, *i.e.* zero. Furthermore we have that:

$$
\begin{aligned}
\|\hat{\boldsymbol{\theta}} - \boldsymbol{\theta}\|_2 &= \|(\hat{W}_n, \hat{W}_{n-1}..., \hat{W}_1) - (W_n, W_{n-1}, ..., W_1)\|_F \\
&= \|(W_n, \hat{B}_{n-1}..., \hat{B}_1) - (W_n, B_{n-1}, ..., B_1)\|_F \\
&= \sqrt{\|W_n - W_n\|_F^2 + \|\hat{B}_{n-1} - B_{n-1}\|_F^2 + ... + \|\hat{B}_1 - B_1\|_F^2} \\
&= \sqrt{0 + \|\hat{C}_2^\top - C_2^\top\|_F^2 + ... + \|\hat{C}_n^\top - C_n^\top\|_F^2} \\
&\leq \sqrt{(n-1) \cdot (n-1)^2 \hat{\epsilon}} \\
&\leq n^{1.5} \sqrt{\hat{\epsilon}},
\end{aligned}
$$

where the second transition follows from the definitions of $\hat{W}_1, ..., \hat{W}_{n-1}$ and $B_1, ..., B_{n-1}$, the third from the definition of Frobenius norm, the forth from the definitions of $\hat{B}_1, ..., \hat{B}_{n-1}$ and $C_1, ..., C_{n-1}$ and the fifth transition follows from the conclusion of Lemma 1 from [46] applied to $C_1, ..., C_n$. $\quad\square$

## J.21 Proof of Theorem 5

Without loss of generality, we may assume $\tilde{\epsilon} \leq 1$ (a proof that is valid for $\tilde{\epsilon} = 1$ automatically accounts for $\tilde{\epsilon} > 1$ as well). Given that the unbalancedness magnitude (Definition 1) of $\boldsymbol{\theta}_0$ is no greater than $\hat{\epsilon}$ (defined in Equation (34)), by Lemma 8, there exists a weight setting $\hat{\boldsymbol{\theta}}_0 \in \mathbb{R}^d$ which is balanced (has unbalancedness magnitude zero) and meets $\|\boldsymbol{\theta}_0 - \hat{\boldsymbol{\theta}}_0\|_2 \leq n^{1.5}\sqrt{\hat{\epsilon}}$. Denote by $(\hat{W}_{1,0}, \hat{W}_{2,0}, ..., \hat{W}_{n,0}) \in \mathbb{R}^{d_1, d_0} \times \mathbb{R}^{d_2, d_1} \times \cdots \times \mathbb{R}^{d_n, d_{n-1}}$ the weight matrices corresponding to $\hat{\boldsymbol{\theta}}_0$, and by $\hat{W}_{n:1,0} \in \mathbb{R}^{d_n, d_0}$ its end-to-end matrix (*i.e.* $\hat{W}_{n:1,0} := \hat{W}_{n,0}\hat{W}_{n-1,0}\cdots\hat{W}_{1,0}$). Define $\hat{\nu}$ as $\mathrm{Tr}(\Lambda_{yx}^\top \hat{W}_{n:1,0}) / (\|\Lambda_{yx}\|_F \|\hat{W}_{n:1,0}\|_F)$ if $\|\hat{W}_{n:1}\|_F \neq 0$, and as $0$ otherwise. The following lemma establishes several bounds relating $\hat{W}_{n:1,0}$ and $\hat{\nu}$ to $W_{n:1,0}$ and $\nu$ respectively.

**Lemma 52.** *The following hold:*

$$\|W_{n:1,0} - \hat{W}_{n:1,0}\|_F \leq \tfrac{1}{3}\|W_{n:1,0}\|_F ; \tag{88}$$

$$\hat{\nu} \geq \min\left\{-\tfrac{1}{2}, sign(\nu)\tfrac{|\nu|+1}{2}\right\} ; \tag{89}$$

$$\|\hat{W}_{n:1,0}\|_F^{-1} \leq \tfrac{3}{2}\|W_{n:1,0}\|_F^{-1} ; \text{ and} \tag{90}$$

$$\max\left\{1, \tfrac{1-\hat{\nu}}{1+\hat{\nu}}\right\} \leq \max\left\{3, \tfrac{3-\nu}{1+\nu}\right\}. \tag{91}$$

Proof for Lemma 52 is provided in Subsubappendix J.21.1.

Given $\eta > 0$ adhering to Equation (35), define:

$$k := \left\lfloor \frac{2n\left(\max\left\{1, \tfrac{3}{2} \cdot \tfrac{1-\hat{\nu}}{1+\hat{\nu}}\right\}\right)^n}{\|\hat{W}_{n:1,0}\|_F \eta} \ln\left(\frac{15n\max\left\{1, \tfrac{1-\hat{\nu}}{1+\hat{\nu}}\right\}}{\|\hat{W}_{n:1,0}\|_F \tilde{\epsilon}}\right) + 1 \right\rfloor. \tag{92}$$

Taken together, Equations (90) and (91) imply that $k$ adheres to the upper bound in Equation (36). It thus suffices to show that with step size $\eta$, iterate $k$ of gradient descent is $\tilde{\epsilon}$-optimal, *i.e.* $f(\boldsymbol{\theta}_k) - \min_{\mathbf{q} \in \mathbb{R}^d} f(\mathbf{q}) \leq \tilde{\epsilon}$.

Equations (88) and (89) respectively imply that $\|\hat{W}_{n:1,0}\|_F \leq 0.2$ and $\hat{\nu} \neq -1$. Therefore, as an initial point for gradient flow, the (balanced) weight setting $\hat{\boldsymbol{\theta}}_0$ satisfies the conditions of Proposition 3. Define:

$$\bar{\epsilon} := \tilde{\epsilon}/2 \ , \ \epsilon := \frac{\|\hat{W}_{n:1,0}\|_F \tilde{\epsilon}}{15n^3\left(\max\left\{1, \tfrac{3}{2} \cdot \tfrac{1-\hat{\nu}}{1+\hat{\nu}}\right\}\right)^n k\eta} \ , \tag{93}$$

and invoke Proposition 3 with initial point $\boldsymbol{\theta}_s = \hat{\boldsymbol{\theta}}_0$, time $t = k\eta$ and $\bar{\epsilon}, \epsilon$ as above (note that $\epsilon \in (0, 1/(2n)]$). From the proposition we obtain that the gradient flow trajectory emanating from $\hat{\boldsymbol{\theta}}_0$

is defined over infinite time, and with $\hat{\boldsymbol{\theta}}:[0,\infty)\to\mathbb{R}^d$ representing this trajectory, the following time $\bar{t}$ satisfies $f(\hat{\boldsymbol{\theta}}(\bar{t}))-\min_{\mathbf{q}\in\mathbb{R}^d}f(\mathbf{q})\le\bar{\epsilon}$:

$$\bar{t}=\frac{2n\left(\max\left\{1,\frac{3}{2}\cdot\frac{1-\hat{\nu}}{1+\hat{\nu}}\right\}\right)^n}{\|\hat{W}_{n:1,0}\|_F}\ln\left(\frac{15n\max\left\{1,\frac{1-\hat{\nu}}{1+\hat{\nu}}\right\}}{\|\hat{W}_{n:1,0}\|_F\min\{1,2\bar{\epsilon}\}}\right). \tag{94}$$

Moreover, we obtain that under the notations of Theorem 3, in correspondence with $\mathcal{D}_{k\eta,\epsilon}$ ($\epsilon$-neighborhood of gradient flow trajectory up to time $k\eta$) are the smoothness and Lipschitz constants $\beta_{k\eta,\epsilon}=16n$ and $\gamma_{k\eta,\epsilon}=6\sqrt{n}$ respectively, and the following (upper) bound on the integral of (minus) the minimal eigenvalue of the Hessian:

$$\int_0^{k\eta}m(t')dt'\le\frac{15n^3\left(\max\left\{1,\frac{3}{2}\cdot\frac{1-\hat{\nu}}{1+\hat{\nu}}\right\}\right)^n k\eta\epsilon}{\|\hat{W}_{n:1,0}\|_F}+\ln\left(\frac{n^2\left(e^2\max\left\{1,\frac{1-\hat{\nu}}{1+\hat{\nu}}\right\}\right)^{5(n-1)/2}}{\|\hat{W}_{n:1,0}\|_F^2}\right), \tag{95}$$

where the function $m:[0,k\eta]\to\mathbb{R}$ is non-negative.

Notice that $k=\lfloor\bar{t}/\eta+1\rfloor$ and therefore $k\eta\ge\bar{t}$. Combining this with the fact that the gradient flow trajectory $\hat{\boldsymbol{\theta}}(\cdot)$ is $\bar{\epsilon}$-optimal at time $\bar{t}$, and that in general gradient flow monotonically non-increases the objective it optimizes, we infer $\bar{\epsilon}$-optimality of the gradient flow trajectory at time $k\eta$, *i.e.* $\hat{\boldsymbol{\theta}}(k\eta)-\min_{\boldsymbol{q}\in\mathbb{R}^d}f(\boldsymbol{q})\le\bar{\epsilon}$. We will invoke Theorem 3 for showing that, in addition to being $\bar{\epsilon}$-optimal, the gradient flow trajectory at time $k\eta$ is also $\epsilon$-approximated by iterate $k$ of gradient descent, *i.e.* $\|\boldsymbol{\theta}_k-\hat{\boldsymbol{\theta}}(k\eta)\|_2\le\epsilon$. This, along with $f(\cdot)$ being $6\sqrt{n}$-Lipschitz across $\mathcal{D}_{k\eta,\epsilon}$ ($\epsilon$-neighborhood of gradient flow trajectory up to time $k\eta$), yields the desired result — $\tilde{\epsilon}$-optimality for iterate $k$ of gradient descent:

$$\begin{aligned}
&f(\boldsymbol{\theta}_k)-\min_{\boldsymbol{q}\in\mathbb{R}^d}f(\boldsymbol{q})\\
&=\left(f(\boldsymbol{\theta}_k)-f(\hat{\boldsymbol{\theta}}(k\eta))\right)+\left(f(\hat{\boldsymbol{\theta}}(k\eta))-\min_{\boldsymbol{q}\in\mathbb{R}^d}f(\boldsymbol{q})\right)\\
&\le\left(6\sqrt{n}\left\|\boldsymbol{\theta}_k-\hat{\boldsymbol{\theta}}(k\eta)\right\|_2\right)+\left(f(\hat{\boldsymbol{\theta}}(k\eta))-\min_{\boldsymbol{q}\in\mathbb{R}^d}f(\boldsymbol{q})\right)\\
&\le 6\sqrt{n}\cdot\epsilon+\bar{\epsilon}\\
&\le\tilde{\epsilon},
\end{aligned}$$

where the last transition follows from the definitions of $\epsilon$ and $\bar{\epsilon}$ (Equation (93)).

We conclude the proof by showing that indeed $\|\boldsymbol{\theta}_k-\hat{\boldsymbol{\theta}}(k\eta)\|_2\le\epsilon$. Equation (95), the definition of $\epsilon$ (Equation (93)) and the condition $\tilde{\epsilon}\le1$ together imply:

$$\begin{aligned}
\int_0^{k\eta}m(t')dt' &\le \frac{15n^3\left(\max\left\{1,\frac{3}{2}\cdot\frac{1-\hat{\nu}}{1+\hat{\nu}}\right\}\right)^n k\eta\epsilon}{\|\hat{W}_{n:1,0}\|_F}+\ln\left(\frac{n^2\left(e^2\max\left\{1,\frac{1-\hat{\nu}}{1+\hat{\nu}}\right\}\right)^{5(n-1)/2}}{\|\hat{W}_{n:1,0}\|_F^2}\right)\\
&= \tilde{\epsilon}+\ln\left(\frac{n^2\left(e^2\max\left\{1,\frac{1-\hat{\nu}}{1+\hat{\nu}}\right\}\right)^{5(n-1)/2}}{\|\hat{W}_{n:1,0}\|_F^2}\right)\\
&\le 1+\ln\left(\frac{n^2\left(e^2\max\left\{1,\frac{1-\hat{\nu}}{1+\hat{\nu}}\right\}\right)^{5(n-1)/2}}{\|\hat{W}_{n:1,0}\|_F^2}\right)\\
&< \ln\left(\frac{3n^2\left(e^2\max\left\{1,\frac{1-\hat{\nu}}{1+\hat{\nu}}\right\}\right)^{5(n-1)/2}}{\|\hat{W}_{n:1,0}\|_F^2}\right). \tag{96}
\end{aligned}$$

Recalling the expressions for $k$ and $\bar{t}$ (Equations (92) and (94) respectively), and the definition of $\bar{\epsilon}$ (Equation (93)), we have:

$$k\eta=\lfloor\bar{t}/\eta+1\rfloor\eta\le\bar{t}+\eta=\frac{2n\left(\max\left\{1,\frac{3}{2}\cdot\frac{1-\hat{\nu}}{1+\hat{\nu}}\right\}\right)^n}{\|\hat{W}_{n:1,0}\|_F}\ln\left(\frac{15n\max\left\{1,\frac{1-\hat{\nu}}{1+\hat{\nu}}\right\}}{\|\hat{W}_{n:1,0}\|_F\tilde{\epsilon}}\right)+\eta \tag{97}$$

$$<\frac{3n\left(\max\left\{1,\frac{3}{2}\cdot\frac{1-\hat{\nu}}{1+\hat{\nu}}\right\}\right)^n}{\|\hat{W}_{n:1,0}\|_F}\ln\left(\frac{15n\max\left\{1,\frac{1-\hat{\nu}}{1+\hat{\nu}}\right\}}{\|\hat{W}_{n:1,0}\|_F\tilde{\epsilon}}\right),$$

where the last transition makes use of the upper bound on $\eta$ given in Equation (35). It holds that:

$$4n^3\epsilon^{-2}e^{2\int_0^{k\eta}m(t')dt'}$$

$$< 4n^3\frac{225n^6\left(\max\left\{1,\frac{3}{2}\cdot\frac{1-\hat{\nu}}{1+\hat{\nu}}\right\}\right)^{2n}(k\eta)^2}{\|\hat{W}_{n:1}(0)\|_F^2\tilde{\epsilon}^2}\cdot\frac{9n^4\left(e^2\max\left\{1,\frac{1-\hat{\nu}}{1+\hat{\nu}}\right\}\right)^{5(n-1)}}{\|\hat{W}_{n:1}(0)\|_F^4}$$

$$< \frac{8100n^{13}e^{11n-10}\left(\max\left\{1,\frac{1-\hat{\nu}}{1+\hat{\nu}}\right\}\right)^{7n-5}}{\|\hat{W}_{n:1}(0)\|_F^6\tilde{\epsilon}^2}(k\eta)^2$$

$$< \frac{8100n^{13}e^{11n-10}\left(\max\left\{1,\frac{1-\hat{\nu}}{1+\hat{\nu}}\right\}\right)^{7n-5}}{\|\hat{W}_{n:1}(0)\|_F^6\tilde{\epsilon}^2}\cdot\frac{9n^2\left(\max\left\{1,\frac{3}{2}\cdot\frac{1-\hat{\nu}}{1+\hat{\nu}}\right\}\right)^{2n}}{\|\hat{W}_{n:1}(0)\|_F^2}\left(\ln\left(\frac{15n\max\left\{1,\frac{1-\hat{\nu}}{1+\hat{\nu}}\right\}}{\|\hat{W}_{n:1}(0)\|_F\tilde{\epsilon}}\right)\right)^2$$

$$< \frac{n^{15}e^{12n+2}\left(\max\left\{1,\frac{1-\hat{\nu}}{1+\hat{\nu}}\right\}\right)^{9n-5}}{\|\hat{W}_{n:1}(0)\|_F^8\tilde{\epsilon}^2}\left(\ln\left(\frac{15n\max\left\{1,\frac{1-\hat{\nu}}{1+\hat{\nu}}\right\}}{\|\hat{W}_{n:1}(0)\|_F\tilde{\epsilon}}\right)\right)^2$$

$$\leq \frac{n^{15}e^{12n+2}\left(\max\left\{3,\frac{3-\nu}{1+\nu}\right\}\right)^{9n-5}}{(\frac{2}{3})^8\|W_{n:1}(0)\|_F^8\tilde{\epsilon}^2}\left(\ln\left(\frac{15n\max\left\{3,\frac{3-\nu}{1+\nu}\right\}}{\frac{2}{3}\|W_{n:1}(0)\|_F\tilde{\epsilon}}\right)\right)^2$$

$$\leq \frac{n^{15}e^{12n+6}\left(\max\left\{3,\frac{3-\nu}{1+\nu}\right\}\right)^{9n-5}}{\|W_{n:1}(0)\|_F^8\tilde{\epsilon}^2}\left(\ln\left(\frac{23n\max\left\{3,\frac{3-\nu}{1+\nu}\right\}}{\|W_{n:1}(0)\|_F\tilde{\epsilon}}\right)\right)^2$$

$$= 1/\hat{\epsilon},$$

where the first transition follows from Equation (96) and the definition of $\epsilon$ (Equation (93)); the third makes use of Equation (97); the fifth relies on Equations (90) and (91); and the last is based on the definition of $\hat{\epsilon}$ (Equation (34)) and the condition $\tilde{\epsilon} \leq 1$. Rearranging the derived inequality gives $\sqrt{\hat{\epsilon}} < \frac{1}{2}n^{-1.5}\epsilon e^{-\int_0^{k\eta}m(t')dt'}$. Combining this with the fact that $\|\boldsymbol{\theta}_0-\hat{\boldsymbol{\theta}}(0)\|_2 \leq n^{1.5}\sqrt{\hat{\epsilon}}$, we obtain:

$$\epsilon - e^{\int_0^{k\eta}m(t')dt'}\|\boldsymbol{\theta}_0-\hat{\boldsymbol{\theta}}(0)\|_2 \geq \epsilon - e^{\int_0^{k\eta}m(t')dt'}n^{1.5}\sqrt{\hat{\epsilon}} > \epsilon - \frac{1}{2}\epsilon = \frac{1}{2}\epsilon. \tag{98}$$

We now have:

$$\beta_{k\eta,\epsilon}\gamma_{k\eta,\epsilon}k\eta e^{\int_0^{k\eta}m(t')dt'}\left/\left(\epsilon - e^{\int_0^{k\eta}m(t')d'}\|\boldsymbol{\theta}_0-\hat{\boldsymbol{\theta}}(0)\|_2\right)\right.$$

$$< \beta_{k\eta,\epsilon}\gamma_{k\eta,\epsilon}k\eta e^{\int_0^{k\eta}m(t')dt'}2\epsilon^{-1}$$

$$< (16n)(6\sqrt{n})k\eta\cdot\frac{3n^2\left(e^2\max\left\{1,\frac{1-\hat{\nu}}{1+\hat{\nu}}\right\}\right)^{5(n-1)/2}}{\|\hat{W}_{n:1,0}\|_F^2}\cdot2\frac{15n^3\left(\max\left\{1,\frac{3}{2}\cdot\frac{1-\hat{\nu}}{1+\hat{\nu}}\right\}\right)^nk\eta}{\|\hat{W}_{n:1,0}\|_F\tilde{\epsilon}}$$

$$< \frac{9000n^{13/2}e^{6n-5}\left(\max\left\{1,\frac{1-\hat{\nu}}{1+\hat{\nu}}\right\}\right)^{(7n-5)/2}}{\|\hat{W}_{n:1,0}\|_F^3\tilde{\epsilon}}(k\eta)^2$$

$$< \frac{9000n^{13/2}e^{6n-5}\left(\max\left\{1,\frac{1-\hat{\nu}}{1+\hat{\nu}}\right\}\right)^{(7n-5)/2}}{\|\hat{W}_{n:1,0}\|_F^3\tilde{\epsilon}}\cdot\frac{9n^2\left(\max\left\{1,\frac{3}{2}\cdot\frac{1-\hat{\nu}}{1+\hat{\nu}}\right\}\right)^{2n}}{\|\hat{W}_{n:1,0}\|_F^2}\left(\ln\left(\frac{15n\max\left\{1,\frac{1-\hat{\nu}}{1+\hat{\nu}}\right\}}{\|\hat{W}_{n:1,0}\|_F\tilde{\epsilon}}\right)\right)^2$$

$$< \frac{n^{17/2}e^{7n+7}\left(\max\left\{1,\frac{1-\hat{\nu}}{1+\hat{\nu}}\right\}\right)^{(11n-5)/2}}{\|\hat{W}_{n:1,0}\|_F^5\tilde{\epsilon}}\left(\ln\left(\frac{15n\max\left\{1,\frac{1-\hat{\nu}}{1+\hat{\nu}}\right\}}{\|\hat{W}_{n:1,0}\|_F\tilde{\epsilon}}\right)\right)^2$$

$$\leq \frac{n^{17/2}e^{7n+7}\left(\max\left\{3,\frac{3-\nu}{1+\nu}\right\}\right)^{(11n-5)/2}}{(\frac{2}{3})^5\|W_{n:1,0}\|_F^5\tilde{\epsilon}}\left(\ln\left(\frac{15n\max\left\{3,\frac{3-\nu}{1+\nu}\right\}}{\frac{2}{3}\|W_{n:1,0}\|_F\tilde{\epsilon}}\right)\right)^2$$

$$\leq \frac{n^{17/2}e^{7n+10}\left(\max\left\{3,\frac{3-\nu}{1+\nu}\right\}\right)^{(11n-5)/2}}{\|W_{n:1,0}\|_F^5\tilde{\epsilon}}\left(\ln\left(\frac{23n\max\left\{3,\frac{3-\nu}{1+\nu}\right\}}{\|W_{n:1,0}\|_F\tilde{\epsilon}}\right)\right)^2$$

$$\leq 1/\eta,$$

where the first transition is due to Equation (98); the second makes use of $\beta_{k\eta,\epsilon}=16n$, $\gamma_{k\eta,\epsilon}=6\sqrt{n}$, Equation (96) and the definition of $\epsilon$ (Equation (93)); the fourth relies on Equation (97); the sixth

is an outcome of Equations (90) and (91); and the last follows from the upper bound on $\eta$ given in Equation (35), as well as the condition $\tilde{\epsilon} \leq 1$. Rearrange the inequality above:

$$\eta < \frac{\epsilon - e^{\int_0^{k\eta} m(t')dt'}\|\boldsymbol{\theta}_0 - \hat{\boldsymbol{\theta}}(0)\|_2}{\beta_{k\eta,\epsilon}\gamma_{k\eta,\epsilon}k\eta e^{\int_0^{k\eta}m(t')dt'}}.$$

Since $m(\cdot)$ is non-negative, it holds that:

$$\frac{\epsilon - e^{\int_0^{k\eta}m(t')dt'}\|\boldsymbol{\theta}_0 - \hat{\boldsymbol{\theta}}(0)\|_2}{\beta_{k\eta,\epsilon}\gamma_{k\eta,\epsilon}k\eta e^{\int_0^{k\eta}m(t')dt'}} \leq \inf_{t\in(0,k\eta]} \frac{\epsilon - e^{\int_0^{t}m(t')dt'}\|\boldsymbol{\theta}_0 - \hat{\boldsymbol{\theta}}(0)\|_2}{\beta_{k\eta,\epsilon}\gamma_{k\eta,\epsilon}\int_0^{t}e^{\int_{t'}^{t}m(t'')dt''}dt'},$$

and therefore:

$$\eta < \inf_{t\in(0,k\eta]} \frac{\epsilon - e^{\int_0^{k\eta}m(t')dt'}\|\boldsymbol{\theta}_0 - \hat{\boldsymbol{\theta}}(0)\|_2}{\beta_{k\eta,\epsilon}\gamma_{k\eta,\epsilon}\int_0^{t}e^{\int_{t'}^{t}m(t'')dt''}dt'}. \tag{99}$$

We now invoke Theorem 3 with $\epsilon$ as we have defined (Equation (93)), time $\tilde{t} = k\eta$, and $\beta_{k\eta,\epsilon}$, $\gamma_{k\eta,\epsilon}$ and $m(\cdot)$ as produced by Proposition 3. The theorem implies that, by Equation (99), the first $\lfloor k\eta/\eta \rfloor = k$ iterates of gradient descent $\epsilon$-approximate the gradient flow trajectory up to time $k\eta$, i.e. $\|\boldsymbol{\theta}_{k'} - \hat{\boldsymbol{\theta}}(k'\eta)\|_2 \leq \epsilon$ for all $k' \in \{1, 2, ..., k\}$. In particular $\|\boldsymbol{\theta}_k - \hat{\boldsymbol{\theta}}(k\eta)\|_2 \leq \epsilon$, as required. $\qquad\square$

### J.21.1 Proof of Lemma 52

For conciseness, in the current proof we omit a second subscript "0" from our notation. Namely, we use $W_{n:1}$ and $\hat{W}_{n:1}$ as shorthand for $W_{n:1,0}$ and $\hat{W}_{n:1,0}$ respectively, and for any $j \in [n]$, $W_j$ and $\hat{W}_j$ serve as shorthand for $W_{j,0}$ and $\hat{W}_{j,0}$ respectively.

We start by proving Equation (88). The following matrix $\hat{W}_{j':j}$, for any $j, j' \in [n]$, is defined as $\hat{W}_{j'}\hat{W}_{j'-1}\cdots\hat{W}_j$ if $j \leq j'$, and as an identity matrix (with size to be inferred by context) otherwise. Recall that $\hat{\boldsymbol{\theta}}_0$ meets the balancedness condition, i.e. $\hat{W}_{j+1}^{\top}\hat{W}_{j+1} = \hat{W}_j\hat{W}_j^{\top}$ for all $j \in [n-1]$. Using this relation repeatedly (while recalling that $d_n = 1$), we have:

$$\begin{aligned}
\|\hat{W}_{n:1}\|_F^2 &= \hat{W}_{n:1}\hat{W}_{n:1}^{\top} \\
&= \hat{W}_{n:2}\hat{W}_1\hat{W}_1^{\top}\hat{W}_{n:2}^{\top} \\
&= \hat{W}_{n:2}\hat{W}_2^{\top}\hat{W}_2\hat{W}_{n:2}^{\top} \\
&= \hat{W}_{n:3}\hat{W}_2\hat{W}_2^{\top}\hat{W}_2\hat{W}_2^{\top}\hat{W}_{n:3}^{\top} \\
&= \hat{W}_{n:3}\hat{W}_3^{\top}\hat{W}_3\hat{W}_3^{\top}\hat{W}_3\hat{W}_{n:3}^{\top} \\
&\;\;\vdots \\
&= (\hat{W}_n\hat{W}_n^{\top})^n \\
&= \|\hat{W}_n\|_F^{2n}.
\end{aligned}$$

Since the balancedness condition implies that $\|\hat{W}_j\|_F = \|\hat{W}_{j+1}\|_F$ for any $j \in [n-1]$, we may conclude $\|\hat{W}_j\|_F = \|\hat{W}_{n:1}\|_F^{1/n}$ for any $j \in [n]$. It holds that:

$$\begin{aligned}
&\left\|W_{n:1} - \hat{W}_{n:1}\right\|_F \\
&= \left\|(\hat{W}_n + W_n - \hat{W}_n)\cdots(\hat{W}_1 + W_1 - \hat{W}_1) - \hat{W}_{n:1}\right\|_F \\
&= \left\|\sum_{(b_1,...,b_n)\in\{0,1\}^n}\left(b_n\hat{W}_n + (1-b_n)(W_n - \hat{W}_n)\right)\cdots\left(b_1\hat{W}_1 + (1-b_1)(W_1 - \hat{W}_1)\right) - \hat{W}_{n:1}\right\|_F \\
&= \left\|\sum_{(b_1,...,b_n)\in\{0,1\}^n\setminus\{1\}^n}\left(b_n\hat{W}_n + (1-b_n)(W_n - \hat{W}_n)\right)\cdots\left(b_1\hat{W}_1 + (1-b_1)(W_1 - \hat{W}_1)\right)\right\|_F \\
&\leq \sum_{(b_1,...,b_n)\in\{0,1\}^n\setminus\{1\}^n}\left\|\left(b_n\hat{W}_n + (1-b_n)(W_n - \hat{W}_n)\right)\cdots\left(b_1\hat{W}_1 + (1-b_1)(W_1 - \hat{W}_1)\right)\right\|_F \\
&\leq \sum_{(b_1,...,b_n)\in\{0,1\}^n\setminus\{1\}^n}\left\|b_n\hat{W}_n + (1-b_n)(W_n - \hat{W}_n)\right\|_F\cdots\left\|b_1\hat{W}_1 + (1-b_1)(W_1 - \hat{W}_1)\right\|_F \\
&\leq \sum_{(b_1,...,b_n)\in\{0,1\}^n\setminus\{1\}^n}\left(b_n\|\hat{W}_n\|_F + (1-b_n)\|W_n - \hat{W}_n\|_F\right)\cdots\left(b_1\|\hat{W}_1\|_F + (1-b_1)\|W_1 - \hat{W}_1\|_F\right),
\end{aligned}$$

where the inequalities follow from sub-multiplicativity and sub-additivity of Frobenius norm. Since $\|\boldsymbol{\theta}_0 - \hat{\boldsymbol{\theta}}_0\|_2 \le \sqrt{n^3 \hat{\epsilon}}$ and $\|\hat{W}_j\|_F = \|\hat{W}_{n:1}\|_F^{1/n}$ for any $j \in [n]$, we obtain:

$$\left\|W_{n:1} - \hat{W}_{n:1}\right\|_F$$

$$\le \sum_{(b_1,..,b_n) \in \{0,1\}^n \setminus \{1\}^n} \left(b_n \|\hat{W}_{n:1}\|_F^{1/n} + (1-b_n)\sqrt{n^3\hat{\epsilon}}\right)\cdots\left(b_1 \|\hat{W}_{n:1}\|_F^{1/n} + (1-b_1)\sqrt{n^3\hat{\epsilon}}\right)$$

$$= \left(\|\hat{W}_{n:1}\|_F^{1/n} + \sqrt{n^3\hat{\epsilon}}\right)^n - \|\hat{W}_{n:1}\|_F$$

$$= \sum_{j=0}^n \binom{n}{j} \|\hat{W}_{n:1}\|_F^{(n-j)/n}\left(n^3\hat{\epsilon}\right)^{j/2} - \|\hat{W}_{n:1}\|_F$$

$$= \sum_{j=1}^n \binom{n}{j} \|\hat{W}_{n:1}\|_F^{(n-j)/n}\left(n^3\hat{\epsilon}\right)^{j/2}$$

$$\le \sum_{j=1}^n n^j \max\left\{1, \|\hat{W}_{n:1}\|_F\right\}\left(n^3\hat{\epsilon}\right)^{j/2}$$

$$= \max\left\{1, \|\hat{W}_{n:1}\|_F\right\}\sum_{j=1}^n \left(n^5\hat{\epsilon}\right)^{j/2}$$

$$\le \max\left\{1, \|\hat{W}_{n:1}\|_F\right\}\sum_{j=1}^\infty \left(n^5\hat{\epsilon}\right)^{j/2}.$$

Since $\sqrt{n^5\hat{\epsilon}} < 1$ (relying on the definition of $\hat{\epsilon}$ in Equation (34)), we obtain:

$$\left\|W_{n:1} - \hat{W}_{n:1}\right\|_F \le \max\left\{1, \|\hat{W}_{n:1}\|_F\right\}\frac{\sqrt{n^5\hat{\epsilon}}}{1 - \sqrt{n^5\hat{\epsilon}}}. \tag{100}$$

By the definition of $\hat{\epsilon}$ (Equation (34)), it follows that $\sqrt{n^5\hat{\epsilon}}\big/\left(1 - \sqrt{n^5\hat{\epsilon}}\right) \le \frac{1}{3}\|W_{n:1}\|_F$, thus:

$$\|W_{n:1} - \hat{W}_{n:1}\|_F \le \tfrac{1}{3}\max\left\{1, \|\hat{W}_{n:1}\|_F\right\}\|W_{n:1}\|_F.$$

We conclude the proof of Equation (88) by showing that $\|\hat{W}_{n:1}\|_F \le 1$. Indeed, assuming that this is not the case, *i.e.* $\|\hat{W}_{n:1}\|_F > 1$, while recalling that $\|W_{n:1}\|_F \le 0.1$, leads us to a contradiction:

$$\left\|\hat{W}_{n:1}\right\|_F \le \left\|W_{n:1}\right\|_F + \left\|\hat{W}_{n:1} - W_{n:1}\right\|_F \le 0.1\left\|\hat{W}_{n:1}\right\|_F + \tfrac{1}{3}\left\|W_{n:1}\right\|_F\left\|\hat{W}_{n:1}\right\|_F < \left\|\hat{W}_{n:1}\right\|_F.$$

Note that in addition to Equation (88), from Equation (100) and the fact that $\|\hat{W}_{n:1}\|_F \le 1$, we may also establish the following:

$$\|W_{n:1} - \hat{W}_{n:1}\|_F \le \|\hat{W}_{n:1}\|_F \frac{\sqrt{n^5\hat{\epsilon}}}{1 - \sqrt{n^5\hat{\epsilon}}}. \tag{101}$$

Moving on to the proof of Equation (89), we split the analysis into the following two cases: *(i)* $\nu \in [0,1]$; and *(ii)* $\nu \in (-1,0)$. We start by analyzing case *(i)*. Note that Equation (88) together with the fact that $\|W_{n:1}\|_F \ne 0$ imply $\|\hat{W}_{n:1}\|_F \ne 0$. It holds that:

$$\hat{\nu} = \frac{\langle \Lambda_{yx}, \hat{W}_{n:1}\rangle}{\|\Lambda_{yx}\|_F \|\hat{W}_{n:1}\|_F}$$

$$= \frac{\langle \Lambda_{yx}, W_{n:1} + \hat{W}_{n:1} - W_{n:1}\rangle}{\|\Lambda_{yx}\|_F \|\hat{W}_{n:1}\|_F}$$

$$= \frac{\langle \Lambda_{yx}, W_{n:1}\rangle}{\|\Lambda_{yx}\|_F \|\hat{W}_{n:1}\|_F} + \frac{\langle \Lambda_{yx}, \hat{W}_{n:1} - W_{n:1}\rangle}{\|\Lambda_{yx}\|_F \|\hat{W}_{n:1}\|_F}$$

$$= \nu \cdot \frac{\|W_{n:1}\|_F}{\|\hat{W}_{n:1}\|_F} + \frac{\langle \Lambda_{yx}, \hat{W}_{n:1} - W_{n:1}\rangle}{\|\Lambda_{yx}\|_F \|\hat{W}_{n:1}\|_F}$$

$$\ge 0 + \frac{\langle \Lambda_{yx}, \hat{W}_{n:1} - W_{n:1}\rangle}{\|\Lambda_{yx}\|_F \|\hat{W}_{n:1}\|_F}.$$

Recall that $\|\Lambda_{yx}\|_F = 1$. We may finish the proof for case *(i)* by using Cauchy-Schwartz and triangle inequalities together with Equation (88):

$$\hat{\nu} \ge -\frac{1 \cdot \|\hat{W}_{n:1} - W_{n:1}\|_F}{1 \cdot \|\hat{W}_{n:1}\|_F}$$

$$= -\frac{\|\hat{W}_{n:1} - W_{n:1}\|_F}{\|W_{n:1} + \hat{W}_{n:1} - W_{n:1}\|_F}$$

$$\ge -\frac{\|\hat{W}_{n:1} - W_{n:1}\|_F}{\|W_{n:1}\|_F - \|\hat{W}_{n:1} - W_{n:1}\|_F}$$

$$\ge -\frac{\|W_{n:1}\|_F/3}{2\|W_{n:1}\|_F/3}$$

$$= -\tfrac{1}{2}.$$

Regarding case *(ii)* (*i.e.* $\nu \in (-1, 0)$), we have that:

$$|\hat{\nu}| = \frac{|\langle \Lambda_{yx}, \hat{W}_{n:1} \rangle|}{\|\Lambda_{yx}\|_F \|\hat{W}_{n:1}\|_F}$$

$$= \frac{|\langle \Lambda_{yx}, W_{n:1} + \hat{W}_{n:1} - W_{n:1} \rangle|}{1 \cdot \|W_{n:1} + \hat{W}_{n:1} - W_{n:1}\|_F}$$

$$= \frac{|\langle \Lambda_{yx}, W_{n:1} \rangle + \langle \Lambda_{yx}, \hat{W}_{n:1} - W_{n:1} \rangle|}{\|W_{n:1} + \hat{W}_{n:1} - W_{n:1}\|_F}$$

$$\leq \frac{|\langle \Lambda_{yx}, W_{n:1} \rangle| + |\langle \Lambda_{yx}, \hat{W}_{n:1} - W_{n:1} \rangle|}{(\|W_{n:1}\|_F - \|\hat{W}_{n:1} - W_{n:1}\|_F)}$$

$$\leq \frac{|\langle \Lambda_{yx}, W_{n:1} \rangle| + 1 \cdot \|\hat{W}_{n:1} - W_{n:1}\|_F}{\|W_{n:1}\|_F - \|\hat{W}_{n:1} - W_{n:1}\|_F}$$

$$= |\nu| - |\nu| + \frac{|\langle \Lambda_{yx}, W_{n:1} \rangle| + \|\hat{W}_{n:1} - W_{n:1}\|_F}{\|W_{n:1}\|_F - \|\hat{W}_{n:1} - W_{n:1}\|_F}$$

$$= |\nu| - \frac{|\langle \Lambda_{yx}, W_{n:1} \rangle|}{\|\Lambda_{yx}\|_F \|W_{n:1}\|_F} + \frac{|\langle \Lambda_{yx}, W_{n:1} \rangle| + \|\hat{W}_{n:1} - W_{n:1}\|_F}{\|W_{n:1}\|_F - \|\hat{W}_{n:1} - W_{n:1}\|_F}$$

$$= |\nu| - \frac{|\langle \Lambda_{yx}, W_{n:1} \rangle|}{1 \cdot \|W_{n:1}\|_F} + \frac{|\langle \Lambda_{yx}, W_{n:1} \rangle| + \|\hat{W}_{n:1} - W_{n:1}\|_F}{\|W_{n:1}\|_F - \|\hat{W}_{n:1} - W_{n:1}\|_F}$$

$$= |\nu| + \|\hat{W}_{n:1} - W_{n:1}\|_F \cdot \frac{\|W_{n:1}\|_F^{-1} |\langle \Lambda_{yx}, W_{n:1} \rangle| + 1}{\|W_{n:1}\|_F - \|\hat{W}_{n:1} - W_{n:1}\|_F}$$

$$= |\nu| + \|\hat{W}_{n:1} - W_{n:1}\|_F \cdot \frac{|\nu| + 1}{\|W_{n:1}\|_F - \|\hat{W}_{n:1} - W_{n:1}\|_F} ,$$

where the first transition relies on $\|\hat{W}_{n:1}\|_F \neq 0$; the second uses $\|\Lambda_{yx}\|_F = 1$; the fourth uses triangle inequality, and relies on Equation (88) ensuring positive denominator; the fifth uses Cauchy-Schwartz and $\|\Lambda_{yx}\|_F = 1$; and both the eighth and the last follow from $\|\Lambda_{yx}\|_F = 1$. It holds that:

$$|\hat{\nu}| \leq |\nu| + \|\hat{W}_{n:1} - W_{n:1}\|_F \cdot \frac{3}{2} \frac{|\nu| + 1}{\|W_{n:1}\|_F}$$

$$\leq |\nu| + \|\hat{W}_{n:1} - W_{n:1}\|_F \cdot \frac{3}{\|W_{n:1}\|_F}$$

$$\leq |\nu| + 3 \frac{\sqrt{n^5 \hat{\epsilon}}}{1 - \sqrt{n^5 \hat{\epsilon}}} \cdot \frac{\|\hat{W}_{n:1}\|_F}{\|W_{n:1}\|_F}$$

$$\leq |\nu| + 3 \frac{\sqrt{n^5 \hat{\epsilon}}}{1 - \sqrt{n^5 \hat{\epsilon}}} \cdot \frac{\|W_{n:1}\|_F + \|\hat{W}_{n:1} - W_{n:1}\|_F}{\|W_{n:1}\|_F}$$

$$\leq |\nu| + 4 \frac{\sqrt{n^5 \hat{\epsilon}}}{1 - \sqrt{n^5 \hat{\epsilon}}}$$

$$\leq |\nu| + 4 \frac{(1 + \nu)/16}{1 - 1/2}$$

$$= |\nu| + \frac{1 - |\nu|}{2}$$

$$= \frac{|\nu| + 1}{2} ,$$

where the first transition uses Equation (88); the second relies on $|\nu| \leq 1$; the third uses Equation (101); the fourth follows from triange inequality; the fifth uses Equation (88); the sixth follows from the definition of $\hat{\epsilon}$ (Equation (34)), namely that $\sqrt{n^5 \hat{\epsilon}} \leq (1 + \nu)/16$ and $\sqrt{n^5 \hat{\epsilon}} \leq 1/2$; and the seventh relies on the assumption of case *(ii)* (*i.e.* $\nu < 0$). After proving both cases *(i)* and *(ii)*, we may conclude Equation (89).

Equation (90) follows from triangle inequality and Equation (88):

$$\|\hat{W}_{n:1}\|_F \geq \|W_{n:1}\|_F - \|\hat{W}_{n:1} - W_{n:1}\|_F \geq \|W_{n:1}\|_F - \frac{1}{3} \|W_{n:1}\|_F = \frac{2}{3} \|W_{n:1}\|_F.$$

To prove Equation (91), it suffices to show that $\frac{1 - \hat{\nu}}{1 + \hat{\nu}} \leq 3$ or $\frac{1 - \hat{\nu}}{1 + \hat{\nu}} \leq \frac{3 - \nu}{1 + \nu}$. We prove this separately for the following two cases: $-\frac{1}{2} \leq \text{sign}(\nu) \frac{|\nu| + 1}{2}$ and $-\frac{1}{2} > \text{sign}(\nu) \frac{|\nu| + 1}{2}$. In the case of $-\frac{1}{2} \leq \text{sign}(\nu) \frac{|\nu| + 1}{2}$, Equation (89) implies $\hat{\nu} \geq -\frac{1}{2}$. Thus, we have that $\frac{1 - \hat{\nu}}{1 + \hat{\nu}} \leq \frac{1 - (-0.5)}{1 - 0.5} = 3$, thereby proving that Equation (91) holds for this case. For the other case (*i.e.* $-0.5 > \text{sign}(\nu)(|\nu| + 1)/2$), we have that $\nu < 0$, and Equation (89) implies $\hat{\nu} \geq -\frac{1 - \nu}{2}$. Thus, we have that $\frac{1 - \hat{\nu}}{1 + \hat{\nu}} \leq \frac{1 - (-(1 - \nu)/2)}{1 - (1 - \nu)/2} = \frac{1.5 - \nu/2}{0.5 + \nu/2} = \frac{3 - \nu}{1 + \nu}$, thereby proving that Equation (91) holds for the second (and last) case. $\qquad\square$