# OpenReview forum: "Continuous vs. Discrete Optimization of Deep Neural Networks"
_NeurIPS.cc/2021/Conference — NeurIPS 2021 Spotlight_

### Official Review · Reviewer_kYKg · 2021-07-15

**Rating:** 7
**Confidence:** 3

**Summary:**

This paper studies under which conditions gradient descent (discrete) may approximate gradient flow (continuous) in neural networks. The motivation is that gradient flow is often considered in theoretical studies while gradient descent (with a finite step size) is used in practice. An upper bound for the distance between the two is derived in terms of the eigenvalues of the Hessian and the Lipschitz constant of the loss.  The bound is then applied to fully connected architectures with linear and leaky ReLU activations. A convergence proof of gradient descent for deep linear networks is provided and also some experiments on MNIST to test the theory.


**Ethical Concerns:**

No ethical concerns

**Limitations And Societal Impact:**

Little societal impact

**Main Review:**

Pros:
- First convergence proof of deep (three or more layer) neural network of fixed size.
- Results may inspire future theoretical work.
- Writing is OK

Cons:
- Convergence proof only applies to linear activation functions.
- The upper bound for the distance between continuous vs discrete problem is not novel. The main theoretical contribution is bounding the minimum eigenvalue of the Hessian for deep linear networks and leaky ReLU.
- My main concern is about the practical significance of the work. Currently, the widespread opinion is that learning rate is a crucial parameter for training neural networks, and that (generalization) performance is optimal for a finite value of the learning rate which is far from gradient flow (see for example the recent work of https://arxiv.org/abs/2101.12176). Therefore, gradient flow, while interesting theoretically, is not very useful in practice.

Other comments:
- The statement: “the “more convex” the objective function is along the trajectory of gradient flow, the better the match between that and gradient descent”, is misleading. If the loss is very "sharp" along some directions, then gradient descent will fail to follow gradient flow. This is accounted for by the "smoothness" beta: if the max eigenvalue of the Hessian is very large, then beta is large and the bound is small. I understand that the author’s statement refers to the minimum eigenvalue, not the maximum, but the statement remains misleading.
- Unresolved questionmarks on line 314
- A detailed discussion of the limitations of the work is missing.



**Time Spent Reviewing:**

4

---

> ### Author Response · Authors · 2021-08-09
> **Response to Reviewer kYKg**
>
> Thank you for the feedback!
>
> ***Cons:***
>
> Our theoretical contributions include much more than bounding the minimum eigenvalue of the Hessian along gradient flow trajectories over deep neural networks (Section 4).  In particular, among other contributions (see Subsection 1.1), we provide a new analysis for gradient flow over deep linear neural networks (Proposition 5), and a translation of this analysis to gradient descent (Theorem 4).
>
> With regards to your point on practical significance:
> Indeed, nascent evidence suggests that large step size is often beneficial for generalization, and this phenomenon is not captured by gradient flow.  However, several recent works (e.g. [8], [30] as well as [48] which you referred to) proposed variants of gradient flow aimed at capturing large step size regimes, and we believe our work lays foundations for formally quantifying the discrepancy between gradient descent with large step size and these variants of gradient flow.  This is discussed in our conclusion (Appendix B).
>
> ***Other comments:***
>
> * We will clarify what we mean by “more convex.”  Thank you for drawing our attention to this point!
>
> * We will fix the broken link on line 314.  Thank you for noticing it!
>
> * We have attempted to transparently delineate the scope of our results, but given your feedback, we will consider adding text (either in the conclusion or in a separate appendix) focused exclusively on limitations.  Please let us know if there are specific issues you believe warrant special attention.  Thank you!
>
> ----
> In view of our responses, we would greatly appreciate it if you would be willing to consider raising your score.  Many thanks!

---

> > ### Comment · Reviewer_kYKg · 2021-08-16
> > **reply to authors**
> >
> > Thank you for your answers.
> > I do think that this work could be useful to analyze also the case of large step sizes.
> > Following ref. [48], this would require generalizing the author's work to the case in which there is a regularization term (of the kind derived in ref.[48]).
> > Do you think that is possible?
> > In any case, it sounds like an interesting new direction.
> > I will raise my score from 6 to 7.

---

> > > ### Author Response · Authors · 2021-08-17
> > > **Thank You**
> > >
> > > Thank you for the support!
> > >
> > > Indeed, we believe it is possible to extend our analysis for formally quantifying the discrepancy between gradient descent with large step size and modifications of gradient flow of the type derived in [48] (namely, modifications born from certain regularization terms).  The main challenge towards that is to derive a result analogous to Theorem 3.  We believe this can be achieved by adapting the proof of Theorem 2 (Fundamental Theorem) --- on which Theorem 3 is based --- to the special case at hand.  As discussed in our conclusion (Appendix B), this direction is viewed as a promising avenue for future work.

---

### Official Review · Reviewer_r4iB · 2021-07-16

**Rating:** 6
**Confidence:** 4

**Summary:**

This paper studies the approximation of discrete gradient descent algorithm to the continuous gradient flow for neural network models. Firstly, a general result is cited from previous work on numerical analysis to control the distance between Euler discretization and the ODE solution. Then, the result is translated into a theorem on the distance between gradient descent solution and gradient flow solution. A critical quantity in the bounds is the minimal eigenvalue of the loss function's Hessian. The bigger this smallest eigenvalue, the better the gradient descent approximates the gradient flow. For deep neural networks with linear, ReLU and leaky ReLU activation functions trained from small initializations, lower bounds for this smallest eigenvalues are derived along the GF trajectory. Especially, for linear networks, this lower bound is extended to the neighborhood of the GF trajectory, and together with the distance control between GD and GF solutions, a convergence result for GD is proven.

Numerical experiments show that for fully connected linear and ReLU networks the GD trajectories with sufficiently small learning rate do not differ drastically when the learning rate changes. Hence the GD trajectory should be close to the GF trajectory.

**Limitations And Societal Impact:**

The authors have adequately addressed the limitations of the work.

**Main Review:**

This paper combines existing works from several different sources---the numerical discretization of ODEs, the weight balance along the GD trajectory for neural networks, and the convergence of GF for deep linear networks---and derive new results on the global convergence of GD on linear networks. The convergence result is indeed new. However, the results are not surprising, and the analysis techniques used are also not new.

Detailed comments as follows:
1. In theorem 4, the learning rate should be super small, depending exponentially on the depth. This makes the results less meaningful. By classical numerical analysis, as long as the loss landscape is smooth, the GD trajectory, as a Euler scheme, will always tends to the GF trajectory when the learning rate decreases to zero. Hence, as long as the learning rate is small enough, results like that in theorem 4 is natural and trivial. I did not see any big improvement of Theorem 4 over those given by traditional understanding. Just show that the GF trajectory is bounded, and thus the Hessian of the loss function is also bounded, we can easily get results like Theorem 4 using truncation error analysis of Euler scheme. The statement of the results will be much simpler than Theorem 4 in the paper, while no essential difference. I believe the upper bound for the learning rate will still depend exponentially on the depth.

2. In proposition 1 and 3, the authors show that the smallest eigenvalue of the Hessian has no lower bound if the parameter \theta can be arbitrary. However, this is achieved by pushing the parameters to infinity. This means, as long as some norm of the parameter vector is bounded, the smallest eigenvalue is bounded. Hence, for any dynamics that will not push the parameters to infinity, the closeness between the dynamics and its discretization can always be bounded.

3. In real applications, many phenomena confirm the big difference between GD and GF. For example, the loss curve jumps down when learning rate is decayed; and GD with different learning rate finds solutions with different generalization performance. These phenomena should not happen if GD is just a close approximation of GF.

**Time Spent Reviewing:**

3h

---

> ### Author Response · Authors · 2021-08-09
> **Response to Reviewer r4iB**
>
> Thank you for the feedback.
>
> It is stated that although our convergence result is new, the analysis techniques used to derive it are not.  To our knowledge, in the context of deep learning (non-convex optimization), the current paper is the first to prove discrete convergence via translation of continuous convergence using generic numerical analysis machinery.  Could you please refer us to prior work in the area (deep learning theory) which employs this technique?
>
> ***Detailed comments:***
>
> **(1)** There seems to be a misunderstanding.  If one regards depth as a constant (as we do in this work), e.g. considers depth three or four, the step size (and accordingly the computational complexity) guaranteed by our convergence result is polynomial (with low degree).  On the other hand, deriving a result via classic means like you suggest would lead to a step size exponentially small (thus to a computational complexity exponentially high) in the magnitude of the minimal eigenvalue of the Hessian (multiplied by the time it takes gradient flow to converge).  We discuss this point in lines 136-150 (as well as Appendixes C and E).
>
> With regards to the dependence on depth, we note that (as discussed in Remark 2) [46] has proven, for a setting similar to ours, that exponential dependence is unavoidable.  We believe the same holds in our setting (but defer to future work formal affirmation of this hypothesis).
>
> **(2)** We believe this comment may have originated from the same misunderstanding as above.  If one applies classic arguments like you suggest, the computational complexity of the resulting convergence guarantee would be exponentially high, even for depth as small as two or three.
>
> **(3)** We do not claim that gradient flow represents optimization of deep neural networks in practical large-scale settings.  Such settings typically include factors which are not covered by our analysis, for example stochasticity and momentum.  However, several recent papers (see [8], [30], [48]) proposed variants of gradient flow aimed at capturing such factors (in the large step size regime), and we believe our work lays foundations for formally quantifying the discrepancy between these variants of gradient flow and the practical optimizers they aim to represent.  This is discussed in our conclusion (Appendix B).
>
> ----
> In light of our responses, we would greatly appreciate it if you would be able to consider a raise in your score.  Thank you!

---

> > ### Author Response · Authors · 2021-08-25
> > **Re: Response to Reviewer r4iB**
> >
> > Dear Reviewer r4iB,
> >
> > As the discussion period is approaching its conclusion, we were wondering if our response has addressed your concerns.  Can you please let us know if you have any further comments or questions?
> >
> > Thank you!
> >
> > Authors

---

> > > ### Comment · Reviewer_r4iB · 2021-08-30
> > > **Re: Response**
> > >
> > > Thank you for addressing the questions. This response clarifies the settings concerned in the paper. I've changed the score.

---

> > > > ### Author Response · Authors · 2021-08-30
> > > > **Re: Response**
> > > >
> > > > Thank you!

---

### Official Review · Reviewer_hENG · 2021-07-16

**Rating:** 7
**Confidence:** 1

**Summary:**

DISCLAIMER: I am not a theorist, and have not looked at the proofs. My confidence level for this review is hence low, and is based on the theorem statements only. I am also not familiar with what would count as a solid contribution in a theory paper, so this is a best guess on my part.

This paper uses classical results from numerical analysis to analyze the discrepancy between gradient descent and gradient flow, as the latter is often used in proofs, while the former is applied in practice. They show a bound (based on classical results) for the discrepancy based on the minimum Hessian eigenvalue along the gradient flow path, and then proceed to bound the min eigenvalue for linear, nonlinear (homogeneous activation) fc and conv nets along this path with near-zero initialization (but also show that it can be arbitrarily negative in general).

Using this, they are able to transfer a proof of convergence to global minimum (under appropriate initialization) for a linear deep network from gradient flow to gradient descent.

They also include small experiments to verify that the gradient descent path is indeed close to the gradient flow path by decreasing the step size.

**Limitations And Societal Impact:**

The authors seem to provide precise theorem statements of their results and adequately addressed the shortcomings.

**Main Review:**

Noting the disclaimer above, I think this is a solid contribution. The authors seem to have successfully applied a classical numerical analysis theorem to proving convergence to a global minimum of gradient descent. I vote to accept the paper.

The proofs are quite extensive, with a 95 page appendix. This might distract from the clarity of the paper. I also do think the conclusion and some related work would be a better fit in the main paper, rather than in the appendix. Perhaps some theorem statements could be delegated to the appendix, such as the lower bound of the min eigenvalue for non-linear activations (since this is not used in the final transfer proof from flow -> descent).

One suggestion for experiments:
- Another way I can think of that could help compare ordinary GD to GF could be to do optimization using more advanced ODE time-steppers, such as RK4 or adaptive time-stepping. Of course this could be computationally expensive to do in practice for training real NNs, but for the purposes of seeing how well an approximation Euler is it could be a valuable comparison compared to just decreasing the step size.

**Time Spent Reviewing:**

2 hours

---

> ### Author Response · Authors · 2021-08-09
> **Response to Reviewer hENG**
>
> Thank you for the feedback and support!
>
> As you suggest, we will include the related work and conclusion (Appendixes A and B) in the main body of the paper instead of the results on non-linear neural networks.
>
> We will definitely consider adding to our experiments more advanced numerical solvers such as Runge-Kutta methods.  Thank you for the suggestion!

---

### Official Review · Reviewer_DACB · 2021-07-19

**Rating:** 7
**Confidence:** 2

**Summary:**

The authors propose to use a result from numerical analysis that connects approximate numerical solution to the exact solution of an ODE to analyze the relationship between gradient descent and gradient flow.  In particular, more convex i.e. more positive the smallest eigenvalue of the Hessian, the better the approximation. The authors claim that for a certain class of networks (mainly with homogeneous activations),  starting from a near zero initial point, the curvature is favorable and gradient descent approximates gradient flow well. This leads them to be able to to "lift" an argument about how well such networks can be optimized from gradient flow to gradient descent.


**Main Review:**

The paper is mathematically dense with most of the real meat in 100 pages of supplementary material. I have not checked the math, and so these are more or less high level comments.

First off, I really like the experimental bit (which is only a small part of the paper) that shows that as step size decreases, the trajectory of gradient descent more or less remains the same. This is a nice check.

Second, the main point about using results from numerical analysis to understand the relationship between gradient flow and gradient descent is also a good one, and perhaps even though obvious, is worth saying, and reminding the community.

That said, some high level questions:

(1) I do not have a good sense of how "convex" gradient descent trajectories are (even if started near zero), and would like a reconciliation of what the authors are saying here with the work from Belkin and colleagues recently: https://arxiv.org/abs/2003.00307 Are these papers in agreement or disagreement about the convexity (either overall or near optimum)?

(2) I am unclear on if the authors are claiming the gradient flow convergence to minimum as a new result, or if that was well known, and their contribution is lifting that to the discrete case?

(3) Does gradient flow always converge to the minimum (within the class of networks considered, >= 3 levels, homogeneous) even if they are not over parameterized? (And why >= 3, i.e., why isn't >=2 "deep"?)

---

Originality: Not sure, please see clarifying questions above.

Quality: Unable to evaluate correctness of the main claims.

Clarity: Not very clear. The paper seems to be a hasty condensation of a much longer work.

Significance: Somewhat. Theoretically connecting gradient flow and gradient descent using tools from numerical analysis seems useful and something that folks could build on. The rest I am not sure.

Overall my sense is that this work probably is more appropriate for a journal given its length.

---

Thank you for the response. I raise my score. Sadly my confidence remains low since I cannot intuitively understand or vouch for the new results obtained by the authors. I would encourage the authors to provide more intuition in the final writeup for the main parts, e.g., why is the trajectory of gradient flow mildly non-convex, Proposition 5, etc.

Regarding my question (3) above, I mean "almost always" since that's what I think you are claiming. I am still surprised that something magical happens for 3 layer networks that doesn't for 2. I would imagine that the watershed is the presence of a single hidden layer (i.e. 2 layer networks) rather than the requirement for 2 hidden layers (which is how I interpret 3 layer networks). I may be off by 1 in interpreting your depth numbers (if so, please let me know and clarify in the paper), but if not, can you provide some intuition for this requirement?






**Time Spent Reviewing:**

8

---

> ### Author Response · Authors · 2021-08-09
> **Response to Reviewer DACB**
>
> Thank you for the feedback!
>
> We will remove from the main body of the paper our analysis of non-linear neural networks, and will use the available space for including the related work and conclusion (Appendixes A and B).  We believe this change will make the text much more accessible and easier to read.
>
> ***High-level questions:***
>
> **(1)** There is no contradiction between our work and https://arxiv.org/abs/2003.00307.  The latter claims that with overparameterized deep neural networks the neighborhoods of global minima are non-convex (Hessian has negative eigenvalues), while we argue that along trajectories of gradient flow emanating from near-zero initialization the degree of non-convexity (i.e. the negativity of Hessian eigenvalues) is mild.
>
> **(2)** Both our convergence result for gradient flow (Proposition 5) and its translation to gradient descent (Theorem 4) are novel theoretical contributions.  The key differentiator between these results and existing literature (e.g. [5]) is that they apply almost surely under a random near-zero initialization.
>
> **(3)** In general, gradient flow does not always converge to global minimum (for example, it may be initialized at a saddle point, in which case it stays completely still).  We use the popular nomenclature by which two layer networks are called “shallow” and three or more layer networks are referred to as “deep”.  Please let us know if we misunderstood your question.
>
> ----
> In light of the above, we kindly ask that you consider raising your score.  Thank you very much!

---

> ### Author Response · Authors · 2021-09-02
> **Response to Update by Reviewer DACB**
>
> Thank you for the support!  Per your suggestion, we will take measures to make the text more intuitive.
>
> With regards to high level question (3):
>
> * By "almost always" we mean "with probability one".  There could still be initializations with which gradient flow will not converge to global minimum (e.g. saddle points), but the probability of these being drawn is zero.
>
> * Our convergence guarantee applies not only to three or more layer ("deep") networks, but to two layer ("shallow") networks as well.  We highlight applicability to deep networks since they are arguably more interesting and more difficult to analyze than shallow ones.
>
> We hope the above addresses your question more effectively than our previous response.  Please let us know if further clarification is needed.

---

### Official Review · Reviewer_4TKP · 2021-07-19

**Rating:** 8
**Confidence:** 3

**Summary:**

Using continuous time flows to analyze discrete time descent methods is very appealing, but the soundness and utility of the connection between the two is still being determined by the community.  Existing analyses using gradient flow disregard computational efficiency, and do not obviously imply that discrete time descents will converge, given convergence of flow trajectories.  The authors address these concerns by employing modern tools from numerical integration literature which depend upon curvature bounds local to flow trajectories.  The authors show that these conditions can be expected in deep linear neural networks, and hint towards their applicability in networks with homogeneous nonlinear activations.

**Limitations And Societal Impact:**

Limitations: Technical limitations are explicitly suggested, and the boundaries between well-established results and more speculative claims are made clear.

Societal Impact:  The work is highly theoretical and fundamental, so the ethical considerations are the same as those which apply to science and technology in general.  The authors need not address them specifically in this work.

Hence, Both are addressed adequately in the work.

**Main Review:**

EDIT: Score updated from 6 to 8 in light of author rebuttal.

## Major Feedback

Summary:
- Contributions to the linear setting are good, but empirical support and clarity could be improved.
- Contributions to the nonlinear setting are less convincing and complete.
- Scope is too large for a conference paper, and the standalone completeness of the first 9 pages has been hurt by relegating the related work and conclusion to appendices.

My recommendation for publication comes from the positive aspects of the linear setting.  Similarly, the opportunity for improving my review score lies mostly in improving clarity and empirical support for the linear setting.

### Deep linear setting

The contributions to the deep linear network setting are sufficiently convincing, complete, and novel to warrant publication.  Especially if empirical support is improved.

To my knowledge, the soundness of flow-based analysis in existing literature is grounded in three elementary integrator properties: consistency, convergence, and some flavor of stability (CCS).  Specifically, when a numerical integrator with CCS properties is shown to connect discrete and continuous time, the continuous time flow is used to make claims about the discrete time descent method.  But important questions about descent methods, i.e. convergence conditions and rate over long horizons, cannot be determined based only on CCS.  Especially for the naive Euler-like integrators which correspond to gradient descent methods, because they have only first order consistency (which entails very poor bounds on long-tail behaviour via global truncation error) and poor numerical stability in many cases.  The “Fundamental Theorem” cited in this work provides much more relaxed bounds on step size for similar guarantees of the approximation error between discrete and continuous trajectories.  The Fundamental Theorem is most powerful when the curvature along flow trajectories is well-behaved.  The authors spend considerable effort showing that such conditions can be expected for fully connected and convolutional neural networks with linear activations, culminating in Section 5.  This is a significant and novel contribution.  In my opinion, these results alone are sufficient for publication, and more than enough scope for a conference paper.

The experiments summarized by the left pair of plots in Figures 1 and 2 offer some support for theoretical claims, but only in that “standard choices” are close to the continuous limit.  This is important to establish, but there is room for validation of the theory itself.  Specifically:
- A bound on step size is theoretically provided, but the experiments do not use it.  Rather, they start from a step size $\eta_0$ according to a “standard choice” of 0.001.  If the theory is correct, we should see identical experimental results if $\eta_0$ were chosen at or near the established bound.
- A similar experiment should be run with step size increasing above and beyond the theoretically guaranteed step size bound, so that the severity of approximation error is made clear.  This would help determine the tightness of the bound.
- For a fixed $\bar{t}$, Theorem 3 shows a step size for which the descent and flow will be epsilon close to each other over the interval $(0, \bar{t}]$.  So we have a theoretical prediction for the horizon where approximation might break down.  Similar to the point above, the tightness of the bound could be experimentally indicated if the specific time $\bar{t}$ were indicated on the plots (e.g. as a vertical line) and the optimization process were allowed to continue well beyond $\bar{t}$.

### Deep nonlinear setting

The nonlinear setting analysis is not as complete, and I am hesitant to suggest its publication.

On L240-242, nonlinear, homogeneous activations are excluded from the results of the fundamental theorem.  Due to the lack of theoretical guarantee, claims in this nonlinear setting warrant specific and considerable empirical evaluation.  But there are only two experiments with nonlinear activations, and their discussion is quite minimal.  Further, the authors claim that the linear setting analysis is still useful as a “piecewise characterization” without much discussion on where the failures of such piecewise characterization might lay.  The contributions to nonlinear setting would be much improved by further discussion of the meaning and potential limitations of “piecewise characterization,” as well as experiments which specifically demonstrate them.

However, this work is already 95 pages with appendices, so I do _not_ recommend adding even more content for discussion empirical support.  I am open to discussion on how this might be addressed, but my initial preference is for the nonlinear setting to be completely deferred to future publication.  That way, the finishing touches to the linear setting can be included in the main body, along with related work and concluding remarks.

## Minor Feedback

- The bounds on step size, e.g. equation (23), appear quite cryptic, and they strongly motivate the reader to wonder how large such step sizes might be.  It would be useful to see examples of the magnitudes of step sizes these bounds entail.  (Similar feedback applies to the expressions for tbar, the endpoint of the time interval for which flow and descent are epsilon close.)
- Since \mu(t) appears twice in equation (4), once as a positive exponent, and again as a negative exponent in an integrand, it is not immediately obvious what \mu(t) we should hope for in order to have good correspondence between continuous and discrete time.  In what cases should we expect \mu(t) to be pathological?    i.e. which \mu(t) would induce a poor correspondence?  Perhaps the worst case described in Appendix E would be more edifying if accompanied by some figures to demonstrate this.
- In sections 2 and 3, the difference between discrete and continuous initializations $||\theta_0 - \theta(0)||$ appears several times.  To my knowledge, we are primarily concerned about the case when initialization is identical.  And the expressions would be more clear and interpretable if the difference were to vanish.  Remark 1 suggests that it is to “account for initialization which is not perfectly balanced.”  Is this a significant enough reason to hurt clarity and interpretability of your results?  Since you already defer analysis of this case to Appendix J, perhaps it would be worth simplifying the expressions in the main body to improve clarity and interpretability.  Then their more general forms could be used and proven in Appendices J and L, respectively.
- In Figure 1, the plots showing distance (right of each pair) are reasonably convincing as presented, but it would be helpful to see the same data on a log scale y axis.
- Several times, the phrase “the latter” is used ambiguously.  Please replace these occurrences with the actual noun or noun phrase being referred to.  It is frustrating to parse the same sentence repeatedly, and it hurts overall clarity when the correct reference is ambiguous.

**Time Spent Reviewing:**

12

---

> ### Author Response · Authors · 2021-08-09
> **Response to Reviewer 4TKP**
>
> Thank you for the thoughtful and helpful feedback!
>
> ### ***Deep Linear Setting***
> We appreciate your perspective on the significance and novelty of our theoretical contributions.
>
> With regards to your feedback on the experimentation:
> The theoretical bounds we derive for guaranteeing small discrepancy between gradient descent and gradient flow over linear neural networks are sufficient, and while they may be asymptotically tight (this is an interesting question for future work), we believe there is a significant gap originating from constants.  That is, we believe the discrepancy will be small even if the step size is far greater than that suggested by our theory.  The purpose of our experiments was to demonstrate this point, using a standard choice of step size (1e-3) much larger than the upper bound in Equation (23) (which goes below 1e-20).  We will implement your suggestion, and extend the experiments in order to quantitatively characterize the tightness of our bounds.  Thank you!
>
> ### ***Deep Non-Linear Setting***
> We agree that our analysis of the non-linear setting is not as complete as that of its linear counterpart.  In line with your suggestion, we will remove this analysis from the main body, and use the available space for extending our experimentation and including the related work and conclusion (Appendixes A and B).  Thank you!
>
> ### ***Minor Feedback***
> * As you propose, we will add to the appendixes numerical instantiations of the expressions in Proposition 5 and Theorem 4.
> * $\mu(t)$ is merely a shorthand for the integral of $m(t)$.  Plugging this integral into Equation (4) reveals that the larger $m(t)$ is, the bigger the discrepancy (and vice versa).  In the context of gradient flow and gradient descent, $m(t)$ corresponds to minus the minimal eigenvalue of the Hessian, so the discrepancy grows as this eigenvalue gets smaller.  The worst case scenario portrayed in Appendix E exemplifies the phenomenon.  As suggested, we will add illustrative figures to this appendix.
> * We will treat your suggestions for improving clarity (removal of difference between initializations, scale of Y axis in Figures 1 and 2, use of the term “the latter”).
>
> Thank you for the constructive feedback!
>
> ----
> In light of planned extensions to experimentation and changes to paper organization, we would greatly appreciate it if you would consider raising your score.  Thank you!

---

> > ### Comment · Reviewer_4TKP · 2021-08-10
> > **Response to Authors**
> >
> > The proposed changes completely address my concerns over the limitations of contributions to the linear setting.  Considering the community's ongoing excitement about continuous-time analysis proxies for discrete optimization, there is a significant need to improve our understanding of how the two regimes relate to each other in the context of machine learning problems.  The work addresses this need, so I believe it is a significant contribution, and that we need more like it.  I have commensurately raised my score from 6 to 8.

---

> > > ### Author Response · Authors · 2021-08-10
> > > **Thank You**
> > >
> > > Thank you for the support, and again, for the very helpful feedback!

---

### Decision · Program_Chairs · 2021-09-28

**Decision:**

Accept (Spotlight)

**Comment:**

The paper is an important theoretical contribution to the field of deep neural networks providing one of the first rigorous mathematical analysis of the commonly used discrete gradient-descent methods via the rich literature on continuous gradient flows. All the reviewers agree that the manuscript is a strong contribution to the field. Furthermore, reviewers' comments were addressed by the authors in detail in the rebuttal phase.

**Consistency Experiment:**

NeurIPS has a long history of experimentation. In 2014, NeurIPS ran an experiment in which 10% of submissions were reviewed by two independent committees to quantify the randomness in the review process. This year, we repeated a variant of this experiment to see how the quality of the review process has changed over time.  This paper was part of the experiment and was therefore assigned to two committees (consisting of reviewers, an Area Chair, and a Senior Area Chair) that reached independent decisions.  If both committees made the same recommendation, this recommendation was followed. If a single committee recommended acceptance, the paper was accepted (with the exception of a few cases in which the other committee identified what we considered a fatal flaw, e.g., an error in a key result).

This copy’s committee reached the following decision: **Accept (Spotlight)**

The other committee assigned to the paper recommended **Reject**.  You can find the other set of reviews, along with any follow up discussion with the authors here:
https://openreview.net/forum?id=kLJjmSrRB3S